

# Remote-sensing algorithm for surface evapotranspiration considering landscape and statistical effects on mixed-pixels

**Z. Q. Peng[1,2], X. Z. Xin[1], J. J. Jiao[1,2], T. Zhou[1,2], and Q. H. Liu[1,3]**

[1]State Key Laboratory of Remote Sensing Science, Institute of Remote Sensing and Digital Earth, Chinese Academy of Sciences, Beijing, 100101, China
[2]University of Chinese Academy of Sciences, Beijing, 100049, China
[3]Joint Center for Global Change Studies (JCGCS), Beijing, 100875, China

Received: 12 November 2015 – Accepted: 11 December 2015 – Published: 15 January 2016

Correspondence to: X. Xin (xin_xzh@163.com)

Published by Copernicus Publications on behalf of the European Geosciences Union.

HESSD

doi:10.5194/hess-2015-491

Remote-sensing algorithm for surface evapotranspiration considering landscape

Z. Q. Peng et al.

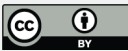

**HESSD**

doi:10.5194/hess-2015-491

**Remote-sensing algorithm for surface evapotranspiration considering landscape**

Z. Q. Peng et al.

## Abstract

Evapotranspiration (ET) plays an important role in surface-atmosphere interactions. Remote sensing has long been identified as a technology that is capable of monitoring ET. However, spatial problems greatly affect the accuracy of ET retrievals by satellite. The objective of this paper is to reduce the spatial-scale uncertainty produced by surface heterogeneity using Chinese HJ-1B data. Two upscaling schemes with area-weighting aggregation for different steps and variables were applied. One scheme is input parameter upscaling (IPUS), which refers to parameter aggregation, and the other is temperature sharpening and flux aggregation (TSFA). Footprint validation results show that TSFA is more accurate and less uncertain than IPUS, and additional analysis shows that TSFA can capture land surface heterogeneities and integrate the effect of overlooked land types in the mixed pixel.

## 1 Introduction

Evapotranspiration (ET) is a major component of the surface energy balance and water budget. Estimating ET accurately is important for the rational utilization and management of water resources (Brutsaert, 1982). Furthermore, ET is an important input parameter in weather forecasting, and it has a significant effect on global climate change (Jung et al., 2010; Shukla and Mintz, 1982). Compared with in situ measurements, remote sensing by satellites has long been identified as a technology that is capable of monitoring ET due to its advantages in regional and long-term simulations.

In recent decades, remotely sensed estimations of evapotranspiration have contributed to models, methods and applications of satellite data. Five types of methods have been developed to estimate evapotranspiration or latent heat flux (LE) via remote sensing. (1) Surface energy balance models calculate LE as a residual term. According to the partitioning of the sources and sinks of the Soil-Plant-Atmosphere Continuum (SPAC), surface energy balance models could be classified as one-source mod-



els (Allen et al., 2007; Bastiaanssen et al., 1998; Long and Singh, 2012a; Su, 2002) or two-source models (Norman et al., 1995; Shuttleworth and Wallace, 1985; Xin and Liu, 2010; Zhu et al., 2013). (2) Penman–Monteith models calculate LE using the Penman–Monteith equation and numerous parameterization schemes of surface resistance that control the diffusion of evaporation from the land surface and transpiration from the plant canopy. These two-source Penman–Monteith models separate soil evaporation from plant transpiration (Chen et al., 2013; Cleugh et al., 2007; Leuning et al., 2008; Mu et al., 2011; Sun et al., 2013). (3) Land surface temperature-vegetation index (LST-VI) space methods assign the dry edge and the wet edge of LST-VI feature space as the minimum and maximum evapotranspiration, respectively. These methods interpolate the mediums combined with the Penman–Monteith or Priestley–Taylor equation to calculate LE and share features with two-source methods and Penman–Monteith models (Fan et al., 2015; Jiang and Islam, 1999, 2001; Long and Singh, 2012b; Sun et al., 2011; Yang and Shang, 2013; Zhang et al., 2005). (4) Priestley–Taylor models expand the range of the Priestley–Taylor coefficient in the Priestley–Taylor equation (Jiang and Islam, 2003; Jin et al., 2011) or combine physiological force factors with the energy component of evapotranspiration (Fisher et al., 2008; Yao et al., 2013). (5) Additional methods include empirical/statistical methods (Wang and Liang, 2008; Yebra et al., 2013), complementary based models (Venturini et al., 2008) and land-process models with data assimilation schemes (Bateni and Liang, 2012; Xu et al., 2015). These ET estimation models are usually developed for simple and homogeneous surface conditions.

However, inhomogeneity is a natural attribute of the Earth's surface. As heat and mass are exchanged in the SPAC, for flux retrieval, inhomogeneity is a relative concept of homogeneity that could be classified in two scenarios: the first is the nonlinear density variation between sub-pixels, while the other is coarse pixels, which include different landscape, such as vegetation mixed with buildings or water. However, in mixed pixels, surface parameters, such as land surface temperature and canopy height, are set as singular to represent the whole pixel area; when these models are applied to

**HESSD**

doi:10.5194/hess-2015-491

**Remote-sensing algorithm for surface evapotranspiration considering landscape**

Z. Q. Peng et al.

calculate the regional ET via remotely sensed satellite data, large errors occur. Inhomogeneity is another key issue of the spatial scale problem, along with the non-linear operational model. However, linear operational models have been slow to develop due to the complexity of mass and heat transfer processes between the atmosphere and the

land surface. The spatial scale effect is usually revealed by a discrepancy between different upscaling schemes: aggregate parameters to large scale then calculate the heat flux, and calculate the heat flux at the small scale then aggregate it to the large scale. Studies have coupled high- and low-resolution satellite data and statistically quantified the inhomogeneity in mixed pixels to correct the scale error in ET estimations (Zhou

et al., 2015), such as temperature downscaling (Cammalleri et al., 2013; Kustas et al., 2003; Norman et al., 2003), the correction-factor method (Chen, 1999; Maayar and Chen, 2006) and the area-weighting method (Xin et al., 2012).

The HJ-1A/B satellites of China were launched on 6 September 2008, and were designed for disaster and environmental monitoring, as well as other applications.

These satellites are on quasi-sun-synchronous orbits at an altitude of 650 km, each with a swath width of 700 km and a revisit period of 4 days. Together, the satellites' revisit period is 48 h. The HJ-1B satellites are equipped with two charge-coupled device (CCD) cameras and one infrared scanner (IRS), whose spatial resolutions are 30 and 300 m, respectively. Compared with high-temporal-resolution satellites, such as

Moderate-Resolution Imaging Spectroradiometer (MODIS), or high-spatial-resolution satellites, such as Landsat 7 or 8, HJ-1B has the advantage of a high spatial-temporal resolution.

Since the satellites were launched, the HJ-1/CCD time series data have been widely used in China for land cover classification with high mapping accuracy (Zhong et al.,

2014a) and for monitoring various environmental disasters (Wang et al., 2010). Land-based parameters, such as leaf area index (LAI), land surface temperature (LST), and downward longwave radiation (DLR), have been retrieved by the HJ-1 satellites using algorithms developed by Chen et al. (2010), H. Li et al. (2010, 2011) and Yu et al. (2013), respectively. These parameters lay the foundation for ET research.

**HESSD**

doi:10.5194/hess-2015-491

**Remote-sensing algorithm for surface evapotranspiration considering landscape**

Z. Q. Peng et al.

Although the HJ-1B satellites provide CCD data with a high spatial resolution of 30 m, the spatial resolution of the thermal infrared band is only 300 m; thus, when estimating the heat flux, spatial scale effects must be considered. The objective of this paper is to reduce the uncertainty produced by surface heterogeneity when estimating evapotranspiration. Surface parameters mainly derived from HJ-1B satellite data were used for this purpose.

## 2 Methodology

### 2.1 Temperature-sharpening method based on statistical relationship

Surface thermal dynamics are a driving force of evapotranspiration. The resolution of thermal infrared (TIR) images is usually not as high as visible near-infrared bands (VNIR) because the energy of VNIR photons is higher than that of thermal photons. The inhomogeneity of thermal infrared images is enhanced, then the uncertainty of the variables calculated in the thermal infrared band, such as the land surface temperature, is unpredictable. Therefore, we would like to derive the land surface parameters with a high spatial resolution, especially the land surface temperature.

The spatial-resolution inconsistency between TIR and VNIR makes it possible to obtain the land surface temperature at the VNIR spatial resolution, called temperature sharpening. Kustas et al. (2003) proposed a brief statistical temperature-sharpening method that could be applied to remotely sensed evapotranspiration models. This method assumes that the negative correlation between parameters calculated by VNIR, such as the Normalized Difference Vegetation Index (NDVI) and LST, is scale invariant. NDVI reflects the vegetation growth and cover, while LST reflects surface thermal dynamics. LST decreases with increasing vegetation cover. The resulting scatter plots would form a feature space that is applicable at different scales if enough pixels exist.

**HESSD**

doi:10.5194/hess-2015-491

**Remote-sensing algorithm for surface evapotranspiration considering landscape**

Z. Q. Peng et al.

HJ-1B satellite images can provide vegetation information at a 30 m resolution and thermal information at a 300 m resolution. However, the 300 m resolution thermal data are not able to discriminate the surface temperatures of small targets within the pixel. This deficiency can be addressed by using the functional relationship between NDVI and LST. The flowchart of temperature sharpening is shown in Fig. 1, and the LST at the NDVI pixel resolution can be derived according to the following steps (Kustas et al., 2003):

1. Select a subset of pixels from the scene where NDVI is most uniform within the 300 m pixel resolution. Calculate the coefficient of variation (CV) by using the original 30-m resolution NDVI data ($NDVI_{30}$), sorting the values from smallest to largest. The CV is expressed as:

$$CV = \frac{STD}{mean} \tag{1}$$

   where STD and mean are the standard deviation and the average value, respectively, among $10 \times 10$ pixels that make up each 300 m NDVI ($NDVI_{300}$) aggregated from $NDVI_{30}$.

2. Divide the $NDVI_{300}$ into several classes ($0 \leq NDVI_{300} < 0.2$, $0.2 \leq NDVI_{300} < 0.5$ and $0.5 \leq NDVI_{300}$). Then, fractions (25 %) of pixels having the lowest CV are selected from each class.

3. Fit a least-squares expression between $NDVI_{300}$ and $T_{300}$ using the selected pixels.

$$\hat{T}_{300}(NDVI_{300}) = a + b \times NDVI_{300} + c \times NDVI_{300}^2 \tag{2}$$

4. For each 30 m pixel within the 300 m pixel, $\hat{T}_{s30}$ can be computed according to Eq. (2):

$$\hat{T}_{30}(NDVI_{30}) = a + b \times NDVI_{30} + c \times NDVI_{30}^2 + \Delta\hat{T}_{300} \tag{3}$$

Discussion Paper | Discussion Paper | Discussion Paper | Discussion Paper | Discussion Paper |

**HESSD**

doi:10.5194/hess-2015-491

**Remote-sensing algorithm for surface evapotranspiration considering landscape**

Z. Q. Peng et al.

where $\Delta\hat{T}_{300} = T_{300} - \hat{T}_{300}$ is the deviation between the regressed temperature and the temperature that was observed by the satellite at 300 m.

## 2.2 Area-weighting method based on landscape information

Coarse pixels are inhomogeneous as various types of land use may be included. Using a dominant type to represent such a large landscape is irrational. When a sharpened temperature is obtained, the spatial details could be provided by surface parameters at a high resolution, and the inhomogeneous problem could be greatly diminished as the landscape is divided into finer pixels.

Combined with a high-resolution classification map, sub-pixel scale parameters can be applied to the ET algorithm, which is more rational than using a dominate-class type only as different landscapes might require different ET algorithms. The surface energy flux can be linearly averaged due to the conservation of energy (Kustas et al., 2003), and a simple average that calculates the arithmetic mean over sub-pixels is the best choice in flux upscaling approaches (Ershadi et al., 2013); thus, the aggregated flux at a low resolution $F(x, y)$ is the arithmetic mean of all of the $n \times n$ sub-pixels fluxes that constitute the contributing flux $F(x_i, y_j)$ at coordinate $(x_i, y_j)$ as follows:

$$F(x,y) = \frac{1}{n \times n} \sum_{i=1}^{n} \sum_{j=1}^{n} F(x_i, y_j). \tag{4}$$

Because the average of all the sub-pixels fluxes is equal to the area-weighted sum of each land-type result, the final coarse result can be derived by the area-weighted sum of each land-type result within the landscape. The main steps of the area-weighting process are shown below (Xin et al., 2012):

1. Geometric correction and registration of the VNIR and TIR input datasets.

2. Count area ratio of different land-cover types within each pixel of a low-spatial-resolution classification image.

**HESSD**

doi:10.5194/hess-2015-491

**Remote-sensing algorithm for surface evapotranspiration considering landscape**

Z. Q. Peng et al.

3. According to the fine-classification data, different parameterization schemes can be used in the ET algorithm to calculate the sub-pixel flux, such as net radiation $R_n$, soil heat flux $G$ and sensible heat flux $H$.

4. To calculate the regional flux by pixel, the flux of the large pixel is calculated by the area-weighting method as follows:

$$F = \sum_{i=1}^{n} w_i \cdot F_i \qquad (5)$$

where $w_i$ is the fractional area contributing flux $F_i$ of class type $i$, and $F$ is the aggregated fluxes at the coarse resolution. The LE is computed as a residual of the surface energy balance in the process of TSFA (Temperature Sharpening and Flux Aggregation, see Sect. 2.3), in which a high-spatial-resolution image is used to reduce the mixed pixels.

## 2.3 Algorithm of pixel evapotranspiration

The surface energy balance describes the energy between the land surface and the atmosphere. The energy budget is commonly expressed as:

$$R_n = \text{LE} + H + G \qquad (6)$$

where $R_n$ is the net radiation, $G$ is the soil heat flux, $H$ is the sensible heat flux, and LE is the latent heat flux absorbed by water vapor when it evaporates from the soil surface and transpires from plants' stomata. The widely used one-source energy balance model considers the SPAC homogeneous medium and omits the inhomogeneity and structure. The LE can be expressed as:

$$\text{LE} = \frac{\rho c_p}{\gamma} \cdot \frac{e_s - e_a}{r_a + r_s} \qquad (7)$$

Discussion Paper | Discussion Paper | Discussion Paper | Discussion Paper | Discussion Paper |

**HESSD**

doi:10.5194/hess-2015-491

**Remote-sensing algorithm for surface evapotranspiration considering landscape**

Z. Q. Peng et al.

where $\gamma$ is the psychometric constant; $e_s$ and $e_a$ are the surface aerodynamic water vapor pressure and atmospheric water vapor pressure, respectively; and $r_a$ and $r_s$ are the water vapor transfer aerodynamic resistance and surface resistance, respectively. Surface resistance includes soil resistance and canopy resistance. The surface resistance is influenced by the vegetation's physiological characteristics and the water supply of roots; thus, surface resistance is difficult to obtain by remote sensing and is highly uncertain, particularly over heterogeneous surfaces. To avoid the errors that may have been introduced by the uncertainty in the surface resistance, LE is computed as a residual of the surface energy balance equation.

Net radiation is the difference between the incoming and outgoing radiation:

$$R_n = S_d(1 - \alpha) + \varepsilon_s L_d - \varepsilon_s \sigma T_{rad}^4 \tag{8}$$

where $S_d$ is the downward shortwave radiation (DSR), $\alpha$ is the surface albedo, $\varepsilon_s$ is the emissivity of the land surface, $L_d$ is the downward atmospheric longwave radiation (DLR), $\sigma = 5.67 \times 10^{-8}\,\mathrm{W\,m^{-2}\,K^{-4}}$ is the Stefan–Boltzmann constant, and $T_{rad}$ is the surface radiation temperature.

The estimation of $G$ is commonly approached through the derivation of empirical equations employing surface parameters, such as $R_n$. The canopy exerts a significant influence on $G$, so the fractional canopy coverage $f_c$ is used to determine the ratio of $G$ to $R_n$:

$$G = R_n \times \left[ \Gamma_c + (1 - f_c) \times (\Gamma_s - \Gamma_c) \right] \tag{9}$$

in which $\Gamma_s$ is equal to 0.315 for bare soil, and $\Gamma_c$ is equal to 0.05 for a full vegetation canopy (Su, 2002).

Sensible heat flux is the turbulence heat transfer between the surface and atmosphere as driven by potentiation and controlled by resistances that depend on the local atmospheric conditions and land cover properties. According to gradient diffusion theory:

Discussion Paper | Discussion Paper | Discussion Paper | Discussion Paper |

**HESSD**

doi:10.5194/hess-2015-491

**Remote-sensing algorithm for surface evapotranspiration considering landscape**

Z. Q. Peng et al.

$$H = \rho c_{\mathrm{p}} \frac{T_{\mathrm{aero}} - T_{\mathrm{a}}}{r_{\mathrm{a}}} \tag{10}$$

where $\rho$ is the density of air; $c_{\mathrm{p}}$ is the specific heat of air at a constant pressure; $T_{\mathrm{aero}}$ is the aerodynamic surface temperature obtained by extrapolating the logarithmic air-temperature profile to the roughness length for heat transport; $T_{\mathrm{a}}$ is the air temper-
ature at a reference height; and $r_{\mathrm{a}}$ is the aerodynamic resistance, which influences the heat transfer between the turbulent heat flux source and the reference height $z$. Aerodynamic resistance was calculated based on the Monin–Obukhov similarity theory (MOST) with a stability correction function (Ambast et al., 2002; Paulson, 1970). The zero-plane displacement height $d$ and roughness length $z_{0\mathrm{m}}$ were parameterized
by the scheme proposed by Choudhury and Monteith (1988).

In this approach, $H$ must be estimated accurately. However, calculating $H$ using Eq. (10) is difficult. Because remote sensing cannot obtain $T_{\mathrm{aero}}$, it is usually replaced by the radiative surface temperature $T_{\mathrm{rad}}$, which is not strictly equal to $T_{\mathrm{aero}}$. The difference between these terms in homogeneous and fully covered vegetation is approxi-
mately 1–2° (Choudhury et al., 1986), or up to 10° in sparsely vegetated areas (Kustas, 1990). The method that corrects the discrepancy adds "excess resistance" $r_{\mathrm{ex}}$ to $r_{\mathrm{a}}$. We used the brief method $r_{\mathrm{ex}} = 4/u_*$ proposed by Chen (1988) to calculate $r_{\mathrm{ex}}$.

Figure 2 illustrates the flowchart for merging the ET retrieval with temperature sharpening based on the HJ-1B satellites.
In this paper, the resolution of the final output result is 300 m. To evaluate the reduced heterogeneity effect of TSFA, another upscaling scheme called the IPUS was used (see Fig. 3). In IPUS, the surface-parameter retrieving algorithms (see Sect. 3.2.1.1) are applied to HJ-1 CCD data. Then, the parameter results are aggregated to the 300 m scale average. These 300 m outputs are used as input parameters in the one-source
energy balance model to obtain the four energy-balance components at 300 m.

**HESSD**

doi:10.5194/hess-2015-491

**Remote-sensing algorithm for surface evapotranspiration considering landscape**

Z. Q. Peng et al.

Discussion Paper | Discussion Paper | Discussion Paper | Discussion Paper |

**HESSD**

doi:10.5194/hess-2015-491

Remote-sensing algorithm for surface evapotranspiration considering landscape

Z. Q. Peng et al.

## 3 Study area and dataset

### 3.1 Study area

Our study was conducted in the middle stream of the Heihe River Basin (HRB), which is located near the city of Zhangye in the arid region of Gansu Province, northwestern China (100.11–100.16° E, 39.10–39.15° N). The middle reach of the HRB has a typical desert-oasis agriculture ecosystem. Maize and wheat are the dominant crops in this area. A large portion of the Gobi Desert and the alpine vegetation of Qilian Mountain are near the study area (see Fig. 4). The artificial oasis is highly heterogeneous, which impacts the thermal-dynamic and hydraulic features. As a result, the efficiency of water use and evapotranspiration are variable.

### 3.2 Dataset

In this paper, the data are mainly derived from the HJ-1B satellite. We combined these data with ancillary data and in situ "Multi-Scale Observation Experiment on Evapotranspiration over heterogeneous land surfaces of The Heihe Watershed Allied Telemetry Experimental Research" (HiWATER-MUSOEXE) data to estimate and validate the HRB land surface parameters and heat fluxes.

#### 3.2.1 Remote sensing data

**HJ-1B satellites data**

The specifications of HJ-1B are shown in Table 1. Because HJ-1 CCDs lack an onboard calibration system, scholars have proposed cross-calibration methods to calibrate the CCD instruments (Zhang et al., 2013; Zhong et al., 2014b). The image quality of HJ-1A/B CCDs is stable, the performances of each band are balanced (Zhang et al., 2013), and the radiometric performance of the HJ-1A/B CCD sensors is close to that of the Landsat-5 TM, ALI, and ASTER sensors; the image quality of HJ-1 CCD data is very

Discussion Paper | Discussion Paper | Discussion Paper | Discussion Paper |

similar to that of the Landsat-5 TM (Jiang et al., 2013). In addition, the accuracy of the thermal infrared band's onboard calibration meets the requirements of the land surface temperature retrieval but not the sea surface temperature retrieval (J. Li et al., 2011). The China Center For Resources Satellite Data and Application (CRESDA) releases the calibration coefficients once per year through its website at http://www.cresda.com. This data is freely available from the CRESDA website (http://218.247.138.121/DSSPlatform/index.html).

We chose HJ-1B satellite data for the HRB region in 2012. Because many parameter-retrieving algorithms needed for the ET calculation were developed in clear-sky conditions, we combined the data-quality information with a visual interpretation to select satellite images without clouds. Considering the time period of the ground observations discussed in Sect. 3.2.2, we obtained data for 11 days: 19 June, 30 June, 8 July, 27 July, 2 August, 15 August, 22 August, 29 August, 2 September, 13 September and 14 September.

The HJ-1B satellite data at the HRB were pre-processed, including geometric correction, radiometric calibration, and atmosphere correction. For Eq. (1) to (10), the following surface parameters are needed: downward shortwave radiation, downward longwave radiation, emissivity, albedo, fractional vegetation coverage (FVC), cloud mask data, meteorological data, LAI and LST. Figure 5 illustrates the flowchart for retrieving these parameters.

1. Surface albedo: according to the algorithm proposed by Liang (2003) and Liu et al. (2012), the surface albedo was obtained from the bidirectional reflectance distribution function (BRDF) with a look-up table based on the regression relationship among POLDER-3/PARASOL BRDF, the HJ-1 satellites' BRDF and the albedo.

2. Normalized Difference Vegetation Index (NDVI), FVC and LAI: NDVI is the central regression of temperature sharpening, and it was used to calculate the FVC. The atmospherically corrected surface reflectance values were used to calculate the NDVI:

**HESSD**

doi:10.5194/hess-2015-491

**Remote-sensing algorithm for surface evapotranspiration considering landscape**

Z. Q. Peng et al.

$$\text{NDVI} = \frac{\rho_{\text{nir}} - \rho_{\text{red}}}{\rho_{\text{nir}} + \rho_{\text{red}}} \qquad (11)$$

and

$$\text{FVC} = \frac{\text{NDVI} - \text{NDVI}_s}{\text{NDVI}_v + \text{NDVI}_s} \qquad (12)$$

where $\rho_{\text{nir}}$ and $\rho_{\text{red}}$ are the reflectances in the near-infrared and red band, respectively. $\text{NDVI}_v$ and $\text{NDVI}_s$ are the fully vegetated and bare soil NDVI values, respectively. As an important input in the parameterization of surface roughness length and aerodynamic resistance, LAI was determined from the following equation (Nilson, 1971):

$$P(\theta) = e^{-G(\theta) \cdot \Omega \cdot \text{LAI} / \cos(\theta)} \qquad (13)$$
$$P(\theta) = 1 - \text{FVC} \qquad (14)$$

where $\theta$ is the zenith angle, $P(\theta)$ is the angular distribution of the canopy gap fraction, $G(\theta)$ is the projection coefficient (0.5), and $\Omega$ is the total foliage clumping index, which can be obtained in the GLC global clumping index database according to the land use type (He et al., 2012).

3. Land surface emissivity (LSE): LSE is needed to calculate the net radiation. It is extremely important in LST retrievals. In this paper, LSE was calculated by FVC as follows (Valor and Caselles, 1996):

$$\varepsilon = \varepsilon_v \cdot \text{FVC} + \varepsilon_g (1 - \text{FVC}) + 4 < d\varepsilon > \cdot \text{FVC} \cdot (1 - \text{FVC}) \qquad (15)$$

where $\varepsilon$ is LSE, and $\varepsilon_v$ and $\varepsilon_g$ are the vegetation and ground emissivity, respectively.

Discussion Paper | Discussion Paper | Discussion Paper | Discussion Paper | Discussion Paper |

**HESSD**

doi:10.5194/hess-2015-491

**Remote-sensing algorithm for surface evapotranspiration considering landscape**

Z. Q. Peng et al.

**HESSD**

doi:10.5194/hess-2015-491

4. Land surface temperature: H. Li et al. (2010a) developed a single-channel parametric model algorithm for retrieving LST based on HJ-1B/IRS thermal infrared data. This model was developed from a parametric model based on MODTRAN4 using NCEP atmospheric profile data.

5. Downward shortwave radiation: the algorithm proposed by L. Li et al. (2010) was applied. MOD05, TOMS, aerosol, and solar angle data were used to estimate the direct light flux and diffuse light flux by 6S LUT. This method considered the influence of complex terrain, and a topographic correction was performed by using products of ASTER DEM.

6. Downward longwave radiation: the top-of-atmosphere (TOA) brightness temperature of the HJ-1B thermal channel was used to express the atmospheric effective temperature. Effective atmospheric emissivity was parameterized as the empirical function of the water vapor content. These values were substituted for atmospheric temperature and atmospheric emissivity to estimate DLR. Because this DLR retrieval method proposed by Yu et al. (2013) was only valid for clear-sky conditions, cloud mask information was used to determine clear skies. When cloud contamination existed in the image, the brightness temperature was relatively low, causing the DLR to be lower than that in the cloudless images.

**Ancillary data**

Ancillary data were used because the satellite's bands could not invert all of the parameters needed for the evapotranspiration retrieval.

1. Atmospheric water vapor data: MODIS provides water vapor data (MOD05), including a 1 km near-infrared product and a 5 km thermal infrared product every day; the 1 km near-infrared water vapor product was used to retrieve DLR in this paper.

Remote-sensing
algorithm for surface
evapotranspiration
considering
landscape

Z. Q. Peng et al.

2. Surface elevation data: we used the 30 m resolution Global Digital Elevation Model (GDEM) based on ASTER, which covers 83° N–83° S, to derive DSR.

3. Atmosphere ozone data: a Total Ozone Mapping Spectrometer (TOMS), which was carried on an Earth Probe satellite, was used to derive DSR. TOMS-EP provided daily global atmosphere ozone data at a resolution of 1° × 1.25° (L. Li et al., 2010).

4. Atmosphere profile data: National Centers for Environmental Prediction (NCEP) global reanalysis data were used to derive LST. The data were generated globally every 6 h (00:00, 06:00, 12:00, 18:00 UTC) and every 1° latitude and longitude (H. Li et al., 2010a).

## HiWATER experiment dataset

The HRB in situ observation data were provided by Heihe Watershed Allied Telemetry Experimental Research (HiWATER). From June to September 2012, HiWATER designed two nested observation matrices over 30 km × 30 km and 5.5 km × 5.5 km within the middle stream oasis in Zhangye to focus on the heterogeneity of the scale effect in the so-called HiWATER-MUSOEXE.

In a larger observation matrix, four eddy covariance (EC) system and one superstation were installed in the oasis–desert ecosystem. Each station was supplemented with an automatic meteorological station (AMS) to record meteorological and soil variables for monitoring the spatial–temporal variation of ET and its impact factors (Li et al., 2013). The station information is shown in Table 2, and their distribution is shown in Fig. 4. Within the artificial oasis, an observation matrix was composed of 17 EC towers and ordinary AMSs, where the superstation was located. The land surface was heterogeneous and dominated by seed maize, maize inter-planted with spring wheat, vegetables, orchards, and residential areas (Li et al., 2013). Because the EC16 and HHZ stations lacked net radiation and soil heat flux observation data, they were excluded from this study.

Discussion Paper | Discussion Paper | Discussion Paper | Discussion Paper | Discussion Paper |

**HESSD**

doi:10.5194/hess-2015-491

**Remote-sensing algorithm for surface evapotranspiration considering landscape**

Z. Q. Peng et al.

The ground observation data included sensible heat flux and latent heat flux. Turbulent heat flux data and reliable methods were applied to ensure data quality. For example, before the main campaign, an intercomparison of all instruments was conducted in the Gobi Desert (Xu et al., 2013). After the basic processing, including spike removal and corrections for density fluctuations (WPL-correction), a four-step procedure was performed to control the EC data quality. The EC output were available every 30 min; for more details, see Liu et al. (2011) and Xu et al. (2013). The soil heat flux was measured by three soil heat plates at a depth of 6 cm at each site, while the surface soil heat flux was calculated by the method proposed by Yang and Wang (2008) based on the soil temperature and moisture above the plates. Surface meteorological parameters, such as wind speed, wind direction, relative humidity and air pressure, were used to interpolate images by the inverse-distance weighted method. Researchers can apply these data from the website of Cold and Arid Regions Science Data Center at LanZhou http://card.westgis.ac.cn/ or the Heihe Plan Data Management Center http://www.heihedata.org/.

An energy imbalance is common in the ground flux observations. The conserving Bowen ratio ($H$/LE) and residual closure technique are often used to force an energy balance. Computing LE as a residual may be a better method for energy balance closure under conditions of large LE (small or negative Bowen ratios due to strong advection) (Kustas et al., 2012). Thus, the residual closure method was applied because the "oasis effect" was distinctly observed in the desert-oasis system on clear days in summer (Liu et al., 2011).

**HESSD**

doi:10.5194/hess-2015-491

**Remote-sensing algorithm for surface evapotranspiration considering landscape**

Z. Q. Peng et al.

# 4   Results and analysis

## 4.1   Evaluation of surface parameters

To control the model input parameters and analyze the error source, the coarse-resolution land surface temperature, downward shortwave radiation, downward long-wave radiation, net radiation and soil heat flux were evaluated using in situ data.

The ground-based land surface temperature $T_s$ was calculated using the Stefan–Boltzman Law from the AMS measurements of the longwave radiation fluxes (Li et al., 2014) as follows:

$$T_s = \left[ \frac{L^\uparrow - (1 - \varepsilon_s) \cdot L^\downarrow}{\varepsilon_s \cdot \sigma} \right]^{\frac{1}{4}} \tag{16}$$

in which $L^\uparrow$ and $L^\downarrow$ are in situ surface upwelling and atmospheric downwelling longwave radiation, respectively. $\varepsilon_s$ is the surface broadband emissivity, which is regarded as the pixel value of the HJ-1B at the AMS. The coefficient of determination $R^2$, mean bias error (MBE) and root mean square error (RMSE) of LST are 0.71, −0.14 K and 3.37 K, respectively. As seen in Table 3, the accuracy of EC4 is low. The main causes of the large errors are as follows: (1) buildings and soil/vegetation are distinct materials, so the land surface emissivity algorithm may be not suitable for buildings, and (2) the EC4 foundation is non-uniform, which is not suitable for validation. After removing EC4's data, the $R^2$, MBE, and RMSE of LST are 0.83, 0.69 K and 2.51 K, respectively. The LST error of SSW and SD are large due to the large error on particular days. For example, on 27 July, it was briefly cloudy above station SSW, but this area was not selected in the process of cloud detection.

The $R^2$, MBE, and RMSE of DSR were 0.81, 13.80 W m$^{-2}$, and 25.35 W m$^{-2}$, respectively. The station validation results are shown in Table 4. The accuracy of SSW is low. Because it was briefly cloudy on 27 July, few ground observations were obtained, and

**HESSD**

doi:10.5194/hess-2015-491

**Remote-sensing algorithm for surface evapotranspiration considering landscape**

Z. Q. Peng et al.

DSR was significantly overestimated. After removing these data, the DSR $R^2$, MBE, and RMSE at SSW were 0.87, 10.90 W m$^{-2}$ and 21.13 W m$^{-2}$.

The $R^2$, MBE, and RMSE of the HRB DLR were 0.73, 0.28 W m$^{-2}$, and 21.24 W m$^{-2}$, respectively. As seen in Table 5, the accuracy of EC3, SD and SSW was low. EC3 and SD's low accuracy may have been caused by (1) the high humidity, which led to low at-nadir brightness temperature and low retrieved DLR, or (2) instrument error, as EC3's ground observations were always higher than those of the other stations in the same period. Considering SSW was located in a desert, the ground-air temperature differ-ence was large. The DLR retrieval may have large error because the models use sur-face temperature in the DLR estimation to approximate or substitute the near-surface temperature (Yu et al., 2013). Our DLR retrieving algorithm corrected error caused by the ground-air temperature difference in non-vegetated areas. The inaccuracy of the SSW LST may influence the DLR result.

The $R^2$, MBE, and RMSE of the HRB net radiation $R_n$ are 0.70, −9.64 W m$^{-2}$, 42.77 W m$^{-2}$, respectively. The station validation results of $R_n$ are shown in Table 6, showing that the accuracy of EC4, EC7, EC17 and SSW was relatively low. Accord-ing to the sensitivity analysis of Eq. (8), DLR and DSR are highly sensitive parameters when calculating $R_n$, while the albedo, LSE and LST are not as sensitive. Although LST was not a sensitive parameter, EC4's LST, MBE and RMSE reached −9.87 and 10.04 K because the 300 m land cover of EC4 was maize. The observation tower was in a built-up area, which may cause errors when estimating $R_n$. The accuracy of EC7's, DSR and DLR was low on several days, and after removing these data, MBE = −43.40 W m$^{-2}$, and RMSE = 50.50 W m$^{-2}$. EC17 was within an orchard, and the signal that was re-ceived by the sensors was impacted by the complex vertical structure of the orchard ecosystem. The information on substrate plants may be ignored, leading to albedo re-trieval errors. Although the albedo was not a sensitive parameter, a 0.03 bias could lead to an approximately 20 W m$^{-2}$ error in $R_n$ when the solar incoming radiation was large. As mentioned, it was briefly cloudy on 27 July, and after removing that data, the $R^2$, MBE, and RMSE of SSW's $R_n$ were 0.72, 8.20 W m$^{-2}$, and 37.60 W m$^{-2}$, respectively.

Discussion Paper | Discussion Paper | Discussion Paper | Discussion Paper |

**HESSD**

doi:10.5194/hess-2015-491

**Remote-sensing algorithm for surface evapotranspiration considering landscape**

Z. Q. Peng et al.

The $R^2$, MBE, and RMSE of the HRB soil heat flux were 0.58, 7.37 W m$^{-2}$, and 28.87 W m$^{-2}$, respectively. The station validation results of $R_n$ are shown in Table 7. For EC5, the soil temperature and moisture were the same at different depths after 19 July, leading to a surface soil heat flux equal to the soil heat flux at a depth of 6 cm. The soil heat flux below the surface is usually less than that at the surface; thus, the validation results of EC5's soil heat flux indicate overestimation. For SSW, the brief cloudiness decreased the observed soil surface temperature, lowering the calculated surface soil heat flux; the remotely sensed $G$ did not reflect this situation. $G$ was overestimated because the net radiation was overestimated. After removing the data on 27 July, the soil heat flux $R^2$, MBE, and RMSE of SSW were 0.17, 19.34 W m$^{-2}$, and 33.30 W m$^{-2}$, respectively.

## 4.2 Validation of TSFA turbulent heat flux results

Figure 6 provides the turbulent heat flux results calculated by TSFA on 13 September 2012. The spatial distribution of the turbulent heat flux is obvious. The sensible heat flux of buildings and uncultivated land, including the Gobi Desert, barren areas and other deserts, was high, along with the latent heat flux of water and agriculture in the oasis. The southern area of the images is uncultivated barren land that borders Qilian Mountain; because the snow melts and moves downward, the groundwater level and soil moisture are high, approximately 30 % according to the in situ soil moisture at a depth of 2 cm. Therefore, the latent heat flux is higher in the south than in the southeast desert, although both areas were classified as uncultivated land.

Study showed that a validation method that considers the source area is more appropriate for evaluating ET models than a traditional validation method based on a single pixel (Jia et al., 2012; Song et al., 2012). In this study, a user-friendly tool presented by Neftel et al. (2008) was used to calculate the footprint function parameters. The tool is based on the Eulerian analytic flux footprint model proposed by Kormann

**HESSD**

doi:10.5194/hess-2015-491

**Remote-sensing algorithm for surface evapotranspiration considering landscape**

Z. Q. Peng et al.

**HESSD**

doi:10.5194/hess-2015-491

**Remote-sensing algorithm for surface evapotranspiration considering landscape**

Z. Q. Peng et al.

and Meixner (2001), and the continuous footprint function was dispersed to the relative weight of the pixels on which the source area falls.

The validation results of the TSFA turbulent heat fluxes are shown in Fig. 7 and Table 8. The $R^2$, MBE, and RMSE of the sensible heat flux are 0.68, 5.01 W m$^{-2}$ and 45.84 W m$^{-2}$, respectively, while the corresponding terms for LE are 0.84, $-18.85$ W m$^{-2}$ and 65.8 W m$^{-2}$. Because LE was calculated as a residual term, it was impacted by the net radiation, surface soil heat flux and sensible heat flux. The error of all the parameters may contribute to the LE. These errors may accumulate or counteract.

As seen in Fig. 7, most of the sensible heat fluxes are small because June, July, August and September constitute the growing season, when agricultural evapotranspiration greatly cools the air. The differential temperature between the land surface and air is small, leading to a low sensible heat flux. The points with large sensible heat fluxes are influenced by uncultivated land. In our study area, bare soil, the Gobi Desert, and desert are included in uncultivated land. The land cover of the points in the scatter plot with large sensible heat fluxes is desert, where the value reached approximately 250 W m$^{-2}$.

Some points in the sensible heat flux scatter plot are less than 0; this situation is caused by an inversion from the "oasis effect" or irrigation. For example, HiWATER's soil moisture data show irrigation on 22 August 2012. Irrigation is the main source of water within the oasis; it cools the land surface, causing to the surface temperature be lower than the air temperature. Irrigation also leads to errors in the LST retrieval caused by the increasing atmospheric water vapor, as discussed in Sect. 4.1. The model error is further analyzed in Sect. 4.4.

## 4.3 Comparison with IPUS turbulent heat flux

To verify whether the TSFA method has the ability to simulate the heterogeneities of the land surface, the IPUS method was compared with the ET estimation. The two methods were evaluated by qualitative analysis: (1) the spatial distribution and scatter

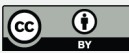

plots of the four energy balance components and (2) quantitative validation by in situ measurements.

### 4.3.1 Qualitative analysis between TSFA and IPUS

Using 13 September as an example, the spatial distribution of the four components of the energy balance calculated by IPUS are shown in Fig. 8, and TSFA minus IPUS, which shows the spatial distribution of the scale effect, is shown in Fig. 9. The quadrangular with a relatively large bias in Fig. 9a and b is caused by DLR, i.e. it is influenced by the MOD05 water vapor. Comparing Fig. 6 with Fig. 8, the spatial distribution of the flux greatly changes, except for $R_n$. Figure 9 has large range interval, except for in Fig. 9a, which is the difference of $R_n$. The TSFA results are synoptically smoother than the IPUS results because the land types in mixed pixels that cannot be considered in IPUS appear in TSFA. For example, the boundary between the oasis and uncultivated land becomes the intermediate belt of $G$, $H$ and LE because mixed pixels include uncultivated land and vegetation, but mixed pixels are classified as the dominate land use type in the parameterization process of IPUS. This result overlooks the heat flux contributions from complex land use types and overestimates or underestimates the heat flux. However, TSFA can integrate the effects of these land areas and reveal the relative actual surface conditions; the results have less dramatic variations compared with IPUS, as shown in the figures. The results are similar in the oasis.

Figure 10 shows the scatter plots between IPUS and TSFA for the entire image's four energy balance components. Figure 10a shows that $R_n$ does not vary much between the two methods, as the scatter is centralized around the 1 : 1 line. However, regarding the spatial scale effect, the differences in $G$, $H$ and LE calculated by IPUS and TSFA are obvious: the scatter plots disperse at the mixed pixels. However, LE is calculated as a residual; thus, the difference between the IPUS and TSFA LE is caused by $G$ and $H$. When the 300 m mixed pixel contains various land types, it can be categorized into one of the land types because of the coarse resolution. The pixels with highly different

Discussion Paper | Discussion Paper | Discussion Paper | Discussion Paper | Discussion Paper

**HESSD**

doi:10.5194/hess-2015-491

**Remote-sensing algorithm for surface evapotranspiration considering landscape**

Z. Q. Peng et al.

$G$ and $H$ values are mainly distributed near the mixed pixels, as shown in Fig. 9. An explanation for these deviations is provided below.

The parameterization of $G$ and $H$ is based on the land cover type; for example, for buildings, $G = 0$ and $H = R_n$, while for water, $G = 0.226 R_n$ and LE $= R_n - G$. From the land cover map in Fig. 4, there are four major classes in the study area: buildings with a high $H$, uncultivated land with a relatively high $H$, cropland with a relatively low $H$, and water with $H = 0$.

1. If the pixel contains cropland and buildings and it is categorized as cropland, then the building area within the pixel is ignored in IPUS; thus, $G$ is overestimated, while $H$ is underestimated, as shown by the green points in Fig. 10. However, if the pixel is categorized as built-up, then the building area within the pixel is exaggerated, causing $G$ to be underestimated and $H$ overestimated, as shown by the red points in Fig. 10.

2. At the margin of the oasis and uncultivated land, the mixed pixel is divided to cropland, and $G$ and $H$ are underestimated and vice versa. The underestimation of $G$ and $H$ also occurs in the pixels containing water and other land cover types, generally bare soil in our study area, when the pixel is categorized as water, as shown by the blue points in Fig. 10.

3. In mixed pixels that contain various crops, such as maize and vegetables, $H$ is overestimated if the maize area within the pixel is overestimated because the canopy height of the maize would be taller than that of the vegetables. And $G$ depends on the FVC of the crops.

Considering the land cover type, the TSFA method ensures that none of the end members (30 m pixel) is ignored or exaggerated. Thus, the distribution of the sensible heat flux calculated by the TSFA method is smoother. At the regional scale, the TSFA method can describe the heterogeneity of the land surface. However, how much the estimation accuracy can be improved is discussed in the following sections.

Discussion Paper | Discussion Paper | Discussion Paper | Discussion Paper |

**HESSD**

doi:10.5194/hess-2015-491

**Remote-sensing algorithm for surface evapotranspiration considering landscape**

Z. Q. Peng et al.

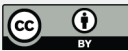

### 4.3.2 Comparison of TSFA and IPUS using in situ data

Table 8 provides the validation results of the turbulent heat flux calculated by IPUS and TSFA. TSFA has a better retrieval accuracy than IPUS for all days. MBE and RMSE decrease and $R^2$ increases on most days.

⁵ To estimate the effect of TSFA, stations with a typical severe heterogeneous surface, such as EC4, and a uniform surface, such as EC15, were selected to quantitatively analyze the temperature sharpening results below.

The EC4 is taken as an example because its land cover was complicated. Table 9 compares the turbulent heat flux calculated by IPUS and TSFA. There is a significant ¹⁰ difference between the TSFA and IPUS methods due to the heterogeneity of the surface. The sensible heat flux calculated by the TSFA method was consistent with the in situ measurements, and MBE and RMSE decreased greatly. $R^2$ increased compared with the IPUS value; the accuracy was improved by approximately 40 W m$^{-2}$.

Figure 11 shows that the classes and temperatures of 10 × 10 sub-pixels at 30 m cor¹⁵ respond to the 300 m resolution pixel at the EC tower. In the IPUS upscaling scheme, the 300 m pixel included buildings, maize and vegetables at the 30 m resolution, which was identified as maize. The canopy height gap between maize and vegetables was large during our study period, leading to the overestimation of the canopy height; for more details, see the sensitivity and error analysis in Sect. 4.4. However, because ²⁰ building $H = R_n$ in this paper, ignoring the building's contribution would lead to an underestimation of $H$. Figure 11a shows the temperature-sharpening result of EC4's pixel on 29 August. The temperature achieved at a 300 m resolution was 303.49 K. Compared with the in situ measurement of 313.24 K, the temperature at the 300 m scale was underestimated. Even when substituting the in situ temperature into the ET model, ²⁵ the sensible heat flux reached 399.60 W m$^{-2}$, which was also greatly overestimated with large error. After processing by temperature sharpening, the distribution of the temperature at the 30-m scale agreed with the classification. Temperature sharpening improves the description of heterogeneity from the view of the thermodynamic-driven

Discussion Paper | Discussion Paper | Discussion Paper | Discussion Paper |

**HESSD**

doi:10.5194/hess-2015-491

**Remote-sensing algorithm for surface evapotranspiration considering landscape**

Z. Q. Peng et al.



force of the turbulent heat flux. The results apply to the ET model with the classification map and high-resolution parameters; the accuracy of the ET estimation increased.

The land surface of EC15 was uniform and comprised pure pixels covered by maize. The temperature distribution at the 30 m scale was as homogeneous as the land cover,
and the variation range of the surface temperature was small. Table 10 shows that for the homogeneous surface, the gap between IPUS and TSFA was not large, within 10 W m$^{-2}$, and the accuracy did not improve (MBE and RMSE did not have obvious variations). Statistically sharpening the temperature may increase the uncertainty of the model results for a homogeneous surface, but this influence could be omitted.
For the study area scale, we compared TSFA and IPUS to quantify TSFA's ability to simulate the heterogeneities of the land surface on 13 September (see Table 11). For pure pixels, the sensible heat flux biases between IPUS and TSFA were small. More class types in the mixed pixel corresponded to larger biases. As seen in Table 12, which shows the scale error of the mixed pixels that contain or do not contain buildings, for
mixed pixels with buildings, IPUS usually underestimated the sensible heat flux with a large bias compared with TSFA. For mixed pixels without buildings, the bias between IPUS and TSFA was relatively small, meaning that the building effect is taken into account in TSFA. The aforementioned analyses demonstrate that TSFA can consider the effect of mixed pixels.

## 4.4   Error analysis

Land surface parameters (including LST, LAI, canopy height, and FVC) and meteorological parameters (including wind speed, air temperature, air pressure and relative humidity) are needed to estimate the sensible heat flux in this paper. To locate the error source of the sensible heat flux retrieval, a sensitivity analysis was performed by adding errors at each 10 % step. Figure 12 presents the sensitivity analysis results: LST = 306.84 K, LAI = 1.4, canopy height equals 1, FVC = 0.5, wind speed $u$ = 2.48 m s$^{-1}$, air temperature $T_a$ = 297.60 K, air pressure = 97.2 kPa, RH = 40.29 %, and the land use type is maize.

Discussion Paper | Discussion Paper | Discussion Paper | Discussion Paper |

**HESSD**

doi:10.5194/hess-2015-491

**Remote-sensing algorithm for surface evapotranspiration considering landscape**

Z. Q. Peng et al.

The air pressure is stable over a short period of time, and it has little effect on the evapotranspiration results. Although "excess resistance" was calculated by the friction velocity, the meteorological data were provided by ground observations; thus the meteorological data are relatively accurate. As shown in Fig. 12, LAI, canopy height and LST are sensitive parameters.

The parameterization of the momentum roughness length indicates that the LAI is sensitive to the sensible heat flux when it is less than 1, and the sensitivity decreases when it is greater than 1. When the LAI is less than 1, the momentum roughness length increases with the LAI, and the sensible heat flux is enhanced, along with the turbulent exchange. However, when the LAI is greater than 1, the plant canopy could be regarded as a continuum that is not sensitive to the sensible heat flux. Because our study area is dominated by agriculture and the study period is from July to September, the crops in the HRB middle stream grew quickly. The LAI was generally greater than 1. Thus, the LST and canopy height are the main sources of error.

### 4.4.1 The influence of LST

As shown in Fig. 12 with monitoring data, a 1 K LST bias would cause a $30\,\mathrm{W\,m^{-2}}$ error in the sensible heat flux. However, the sensitivity of the LST is unstable and is dependent on the strength of the turbulence. The strength of the turbulence determined the ability of the mass and energy transport and the resistance of heat transfer, thus influencing the sensitivity of the LST. A weaker turbulence corresponds to a weaker LST sensitivity and vice versa.

According to Eq. (10), the gradient of the surface–atmosphere temperature was used to estimate the sensible heat flux. The surface-atmosphere differential error caused by the LST retrieval ("noise") would affect the difference between the LST and air temperature ("signal"), and the influence of the sensible heat flux depends on the ratio between the "signal" and "noise", also known as the LST signal-noise ratio error (TRE):

**HESSD**

doi:10.5194/hess-2015-491

**Remote-sensing algorithm for surface evapotranspiration considering landscape**

Z. Q. Peng et al.

$$\text{TRE} = \frac{(T_{\text{rad}} - T_a) - (T_s - T_a)}{T_s - T_a} \times 100\,\% = \frac{T_{\text{rad}} - T_s}{T_s - T_a} \times 100\,\%. \qquad (17)$$

When validating the turbulent heat flux, TRE may be extremely large when the differential temperature is relatively small; thus, the LST bias has also been taken into consideration.

The analysis of the sensible heat flux validation result was based on the sensitivity analysis and sensible heat flux results. We chose homogeneous stations to analyze the LST error so that other errors could be ignored; the results are shown in Table 13. The results of $H$ from the observed LST are consistent with the in situ observations, with less bias. The sensible heat flux was overestimated when the LST was overestimated

and vice versa. Note that the sensible heat flux from retrieved LST of EC7 was less than 0 on 27 July and was caused by the temperature retrieval inversion in the oasis, which also leading to the downward transfer of the sensible heat in calculation.

## 4.4.2   The influence of canopy height

In this paper, canopy height was known a priori based on a phenophase and classifi-

cation map. Thus, the accuracy of the canopy height mainly depended on the classification accuracy and plant growth state. Even within the same region, a crop's canopy height differs because of the discrepancy in seeding times and soil attributes, such as moisture and fertilization.

      The land use of EC17 was orchard, but in our land classification map, the land use

was other crops, including vegetables and orchards. Thus, it was difficult to set the value of the canopy height. In our study area, the majority of the other crops were vegetables (canopy height of 0.2 m), while the height of the orchard was approximately 4 m; a value of 0.2 m would underestimate the sensible heat flux. The points of large LST TRE were removed. The sensible heat flux with incorrect canopy heights and

correct orchard canopy heights at EC17 is shown in Table 14. The bias between the

**HESSD**

doi:10.5194/hess-2015-491

**Remote-sensing algorithm for surface evapotranspiration considering landscape**

Z. Q. Peng et al.

Discussion Paper | Discussion Paper | Discussion Paper | Discussion Paper

model and ground observations decreased. The excess errors were caused by errors in the LST and other land use types, such as buildings and maize in the mixed pixels.

Except for the error source discussed before, other error sources were unavoidable:

1. remotely sensed turbulent heat flux is instantaneous, while the EC data are temporally average. Thus, the time scales do not match in the validation;

2. the calibration coefficient of HJ-1B satellite's CCD and IRS drifts because of the aging instruments;

3. geometric correction causes half-pixel bias equal to or less than the deviation of the artificially subjective interpretation.

A one-source model and simplified parameterization schemes of the surface roughness length and heat transfer coefficient were used in this paper. The one-source model combines the soil evaporation and plant transpiration, assuming SPAC is a one-source continuum for calculating ET. This assumption is reasonable when the surface is densely covered by vegetation, but it greatly relies on the accuracy of the difference between the LST and air temperature as mentioned. When a one-source model is applied to the area covered by sparse vegetation, such as semi-arid or arid area, the assumption is irrational.

## 5 Discussion

As mentioned in the results and analysis, TSFA produces the surface heterogeneity more clearly than IPUS. Compared with IPUS, the superiority of TSFA is as follows.

1. IPUS scheme aggregated the land surface parameters achieved by CCDs from 30 to 300 m, and the process lost surface information and led to the scale effect. The TSFA scheme could avoid this.

**HESSD**

doi:10.5194/hess-2015-491

**Remote-sensing algorithm for surface evapotranspiration considering landscape**

Z. Q. Peng et al.

Discussion Paper | Discussion Paper | Discussion Paper | Discussion Paper

**HESSD**

doi:10.5194/hess-2015-491

**Remote-sensing algorithm for surface evapotranspiration considering landscape**

Z. Q. Peng et al.

2. TSFA used the NDVI at 30 m to monitor the LST at 30 m and greatly decreased the heterogeneity of the LST. This result reflects the surface thermodynamics, which is the driving force of turbulent transportation.

3. In our one-source energy balance model, different land cover types used different parameterization schemes. IPUS assigns a single land cover type to a mixed pixel, which causes error. However, TSFA calculated the surface flux at 30 m and then aggregated to 300 m by the area-weighting method, which considers all of the sub-pixel contributions and improves the retrieval accuracy.

4. The canopy height varied by land cover use within the coarse pixels; using a single type to represent the whole pixel causes definite error, which overestimates or underestimates the turbulent heat flux. However, the TSFA ensures that the sub-pixel canopy height is more accurate by using a high-resolution classification map.

When estimating the ET by satellite, a spatial scale effect is caused by the heterogeneity of the land surface and the non-linearity of the operational model. TSFA attempts to decrease the heterogeneous influence of mixed pixels and generate high-resolution LST through temperature sharpening. However, finer resolution pixels are also mixed pixels; therefore, the mixed-pixel problem still exists. The temperature-downscaling method used in this paper causes boxy discontinuities in parts of the sharpened-temperature field because of the residual term in Eq. (3) (Agam et al., 2007). Bindhu et al. (2013) proposed a nonlinear method to generate the residual term by incorporating the corresponding variation in NDVI within the neighborhood; this could resolve the boxy discontinuities. In addition, the downscaled-temperature results are difficult to evaluate. The physical meaning of the finer-resolution LST obtained by the statistical regression method continues to be a complicated issue.

A footprint method was used in this paper in which the validation results were influenced by neighboring pixels that overlapped the corresponding source area. The relative weight calculated by the footprint was multiplied by the overlapped coarse pixel.

The value of the coarse pixel included the contribution of other sub-pixels that did not overlap the source area within the pixel.

A one-source model has difficulty describing the turbulent exchange for sparse vegetation and heterogeneous surfaces. A single-source assumption is the theoretical drawback of the one-source energy balance model. The two-source energy balance model that separates soil evaporation ($E$) and plant transpiration ($T$) may have a higher accuracy in our study area. In addition, in this paper, the parameterization scheme of the surface roughness length and heat transfer aerodynamic resistance performed well for a uniformly flat plant surface. Thus, models and parameterization schemes should be compared to select the optimum ones.

Because of the sensitive parameters of the one-source energy balance model used in this paper, the accuracy of the LST and canopy height greatly influenced the sensible heat flux. Because HJ-1B IRS is a single-thermal channel, the single-channel LST-retrieving algorithm may be unstable in wet atmospheric conditions (water vapor content higher than $3\,\mathrm{g\,cm^{-2}}$) (H. Li et al., 2010a), which may be the bottleneck of the ET estimation by HJ-B. The canopy height is a priori knowledge based on phenophase classifications and may influence the accuracy of the surface roughness length of a heterogeneous surface or seasonal transition. Multi-source remote sensing data could be used to improve the accuracy of calibrations and land surface parameter estimations. Fox example, active microwave and LiDAR data (Colin and Faivre, 2010) could be used to obtain the canopy height, which would decrease the dependence on the accuracy of the classification.

# 6 Conclusions

We employed a remote-sensing algorithm to estimate surface evapotranspiration over a heterogeneous surface and applied it to HJ-1B satellite data based on instrument characteristics.

**HESSD**

doi:10.5194/hess-2015-491

**Remote-sensing algorithm for surface evapotranspiration considering landscape**

Z. Q. Peng et al.

Compared with IPUS, the TSFA method is more consistent with in situ measurements. This method reduces the spatial scale uncertainty produced by surface heterogeneity. Because of the sensitive parameters of the ET model, the canopy height is mainly determined by the classification, and the application of the classification at a 30 m resolution can improve the accuracy of the canopy height. As another sensitive parameter, the sharpened surface temperature at a 30 m resolution decreases the thermodynamic uncertainty produced by land surface heterogeneities. TSFA can capture the land surface heterogeneities and integrate the effects of land types in a mixed pixel that are neglected at coarse spatial resolutions.

HJ-1B satellite data are advantageous because of their high spatiotemporal resolution and free access. Because the satellites are still in operation, the long-term data have promising applications in energy budget monitoring.

*Acknowledgements.* The authors would like to thank Franz Meixner of the Max Planck Institute for Biogeochemistry in Mainz and Albrecht Neftel of the Agroscope Research Station in Zurich for providing the footprint calculation tool. We are grateful to the research team of Shao Min Liu of Beijing Normal University for providing eddy covariance and automatic meteorological station data. This study was supported by the Special Fund from the Chinese Academy of Sciences (KZZD-EW-TZ-18) and the Chinese Natural Science Foundation Project (grant no. 41371360).

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

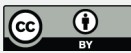

**Table 1.** Specifications of the HJ-1B main payloads.

| Sensor | Band | Spectral range (µm) | Spatial resolution (m) | Swath width (km) | Revisit time (days) |
|--------|------|---------------------|------------------------|------------------|---------------------|
| CCD | 1 | 0.43–0.52 | 30 | 360 (single) 700 (two) | |
| | 2 | 0.52–0.60 | | | |
| | 3 | 0.63–0.69 | | | |
| | 4 | 0.76–0.90 | | | |
| IRS | 5 | 0.75–1.10 | 150 | 720 | 4 |
| | 6 | 1.55–1.75 | | | |
| | 7 | 3.50–3.90 | | | |
| | 8 | 10.5–12.5 | 300 | | |

**Table 2.** The HiWATER-MUSOEXE in situ station information.

| Station | Longitude | Latitude | Tower height (m) | Altitude (m) | Land cover |
|---------|-----------|----------|------------------|--------------|------------|
| EC1 | 100.36° E | 38.89° N | 3.8 | 1552.75 | vegetation |
| EC2 | 100.35° E | 38.89° N | 3.7 | 1559.09 | maize |
| EC3 | 100.38° E | 38.89° N | 3.8 | 1543.05 | maize |
| EC4 | 100.36° E | 38.88° N | 4.2 | 1561.87 | building |
| EC5 | 100.35° E | 38.88° N | 3 | 1567.65 | maize |
| EC6 | 100.36° E | 38.87° N | 4.6 | 1562.97 | maize |
| EC7 | 100.37° E | 38.88° N | 3.8 | 1556.39 | maize |
| EC8 | 100.38° E | 38.87° N | 3.2 | 1550.06 | maize |
| EC9 | 100.39° E | 38.87° N | 3.9 | 1543.34 | maize |
| EC10 | 100.40° E | 38.88° N | 4.8 | 1534.73 | maize |
| EC11 | 100.34° E | 38.87° N | 3.5 | 1575.65 | maize |
| EC12 | 100.37° E | 38.87° N | 3.5 | 1559.25 | maize |
| EC13 | 100.38° E | 38.86° N | 5 | 1550.73 | maize |
| EC14 | 100.35° E | 38.86° N | 4.6 | 1570.23 | maize |
| EC15 | 100.37° E | 38.86° N | 4.5 | 1556.06 | maize |
| EC17 | 100.37° E | 38.85° N | 7 | 1559.63 | orchard |
| GB | 100.30° E | 38.91° N | 4.6 | 1562 | uncultivated land-Gobi |
| SSW | 100.49° E | 38.79° N | 4.6 | 1594 | uncultivated land-desert |
| SD | 100.45° E | 38.98° N | 5.2 | 1460 | swamp land |

Discussion Paper | Discussion Paper | Discussion Paper | Discussion Paper | Discussion Paper |

# HESSD

doi:10.5194/hess-2015-491

**Remote-sensing algorithm for surface evapotranspiration considering landscape**

Z. Q. Peng et al.

**Table 3.** Station validation results of the land surface temperature.

| Station | $R^2$ | MBE (K) | RMSE (K) | Station | $R^2$ | MBE (K) | RMSE (K) |
|---|---|---|---|---|---|---|---|
| EC1 | 0.82 | 0.18 | 1.74 | EC11 | 0.42 | 1.59 | 2.98 |
| EC2 | 0.82 | 0.59 | 1.54 | EC12 | 0.87 | 0.62 | 1.51 |
| EC3 | 0.69 | 0.38 | 1.90 | EC13 | 0.83 | 0.44 | 1.48 |
| EC4 | 0.83 | −9.87 | 10.04 | EC14 | 0.73 | 1.43 | 2.44 |
| EC5 | 0.83 | 1.71 | 2.34 | EC15 | 0.74 | 1.53 | 2.41 |
| EC6 | 0.61 | 0.30 | 2.44 | EC17 | 0.78 | 1.20 | 2.32 |
| EC7 | 0.82 | 0.39 | 1.40 | GB | 0.69 | 0.12 | 2.33 |
| EC8 | 0.83 | 0.45 | 1.55 | SSW | 0.59 | 1.38 | 3.82 |
| EC9 | 0.63 | 2.31 | 3.15 | SD | 0.76 | −3.83 | 4.84 |
| EC10 | 0.68 | 1.32 | 2.45 | | | | |

**HESSD**

doi:10.5194/hess-2015-491

**Remote-sensing algorithm for surface evapotranspiration considering landscape**

Z. Q. Peng et al.

**Table 4.** Station validation results of downward shortwave radiation.

| Station | $R^2$ | MBE (W m$^{-2}$) | RMSE (W m$^{-2}$) | Station | $R^2$ | MBE (W m$^{-2}$) | RMSE (W m$^{-2}$) |
|---------|-------|------------------|-------------------|---------|-------|------------------|-------------------|
| EC1 | 0.97 | 25.23 | 27.73 | EC11 | 0.90 | 30.11 | 33.76 |
| EC2 | 0.84 | 28.29 | 33.57 | EC12 | 0.96 | 24.35 | 26.43 |
| EC3 | 0.97 | 17.56 | 19.25 | EC13 | 0.93 | 12.41 | 17.92 |
| EC4 | 0.98 | 6.07 | 9.34 | EC14 | 0.98 | 32.40 | 33.49 |
| EC5 | 0.98 | 10.60 | 12.29 | EC15 | 0.94 | 26.71 | 29.71 |
| EC6 | 0.93 | 27.68 | 30.71 | EC17 | 0.94 | −20.25 | 24.54 |
| EC7 | 0.89 | −17.69 | 27.59 | GB | 0.89 | 25.34 | 30.63 |
| EC8 | 0.83 | 15.63 | 25.50 | SSW | 0.63 | 18.51 | 34.93 |
| EC9 | 0.96 | −2.27 | 9.96 | SD | 0.98 | 5.70 | 13.82 |
| EC10 | 0.94 | −3.50 | 11.97 | | | | |

# HESSD

doi:10.5194/hess-2015-491

**Remote-sensing algorithm for surface evapotranspiration considering landscape**

Z. Q. Peng et al.

Discussion Paper | Discussion Paper | Discussion Paper | Discussion Paper | Discussion Paper |

**HESSD**

doi:10.5194/hess-2015-491

**Remote-sensing algorithm for surface evapotranspiration considering landscape**

Z. Q. Peng et al.

**Table 5.** Station validation results of downward longwave radiation.

| Station | $R^2$ | MBE (W m$^{-2}$) | RMSE (W m$^{-2}$) | Station | $R^2$ | MBE (W m$^{-2}$) | RMSE (W m$^{-2}$) |
|---|---|---|---|---|---|---|---|
| EC1 | 0.85 | 4.16 | 17.21 | EC11 | 0.93 | −2.72 | 10.55 |
| EC2 | 0.88 | 0.11 | 14.23 | EC12 | 0.87 | −0.84 | 14.80 |
| EC3 | 0.91 | −35.65 | 37.88 | EC13 | 0.86 | −7.28 | 15.98 |
| EC4 | 0.88 | 3.36 | 16.38 | EC14 | 0.82 | 4.07 | 16.42 |
| EC5 | 0.88 | −0.79 | 15.02 | EC15 | 0.85 | 17.67 | 23.06 |
| EC6 | 0.84 | 2.55 | 15.43 | EC17 | 0.90 | −1.11 | 12.87 |
| EC7 | 0.75 | −5.90 | 19.72 | GB | 0.88 | 9.50 | 27.82 |
| EC8 | 0.80 | −1.35 | 17.49 | SSW | 0.85 | 25.33 | 34.50 |
| EC9 | 0.86 | 10.44 | 17.99 | SD | 0.85 | −26.54 | 34.08 |
| EC10 | 0.87 | 7.98 | 16.05 | | | | |



**HESSD**

doi:10.5194/hess-2015-491

**Remote-sensing algorithm for surface evapotranspiration considering landscape**

Z. Q. Peng et al.

**Table 6.** Station validation results of net radiation.

| Station | $R^2$ | MBE (W m$^{-2}$) | RMSE (W m$^{-2}$) | Station | $R^2$ | MBE (W m$^{-2}$) | RMSE (W m$^{-2}$) |
|---|---|---|---|---|---|---|---|
| EC1 | 0.76 | −2.55 | 30.61 | EC11 | 0.86 | −15.13 | 28.05 |
| EC2 | 0.79 | 2.52 | 25.24 | EC12 | 0.90 | −8.46 | 19.38 |
| EC3 | 0.86 | −35.84 | 42.97 | EC13 | 0.88 | −25.73 | 32.34 |
| EC4 | 0.84 | 76.64 | 80.25 | EC14 | 0.90 | 4.23 | 18.18 |
| EC5 | 0.85 | −24.41 | 32.34 | EC15 | 0.84 | 8.33 | 23.01 |
| EC6 | 0.82 | 4.35 | 23.44 | EC17 | 0.89 | −62.62 | 68.11 |
| EC7 | 0.61 | −58.66 | 67.83 | GB | 0.77 | −10.40 | 38.86 |
| EC8 | 0.83 | −20.62 | 32.45 | SSW | 0.44 | 23.05 | 62.93 |
| EC9 | 0.87 | −29.60 | 36.27 | SD | 0.75 | 19.98 | 35.24 |
| EC10 | 0.83 | −24.35 | 33.51 | | | | |

**Table 7.** Station validation results of soil heat flux.

| Station | $R^2$ | MBE (W m$^{-2}$) | RMSE (W m$^{-2}$) | Station | $R^2$ | MBE (W m$^{-2}$) | RMSE (W m$^{-2}$) |
|---|---|---|---|---|---|---|---|
| EC1 | 0.50 | 19.33 | 31.13 | EC11 | 0.71 | 4.23 | 19.23 |
| EC2 | 0.24 | 20.78 | 28.72 | EC12 | 0.53 | 20.29 | 24.79 |
| EC3 | 0.03 | −1.15 | 36.28 | EC13 | 0.91 | −0.89 | 17.27 |
| EC4 | 0.42 | 11.99 | 17.36 | EC14 | 0.82 | −1.89 | 18.72 |
| EC5 | 0.40 | 34.08 | 55.16 | EC15 | 0.78 | 6.68 | 15.80 |
| EC6 | 0.83 | −5.91 | 14.57 | EC17 | 0.51 | 3.46 | 32.39 |
| EC7 | 0.28 | 7.50 | 24.65 | GB | 0.29 | −17.86 | 26.81 |
| EC8 | 0.68 | −5.73 | 20.15 | SSW | 0.01 | 30.41 | 51.87 |
| EC9 | 0.62 | 3.84 | 26.56 | SD | 0.71 | −4.79 | 13.71 |
| EC10 | 0.41 | 7.68 | 28.67 | | | | |

**HESSD**

doi:10.5194/hess-2015-491

**Remote-sensing algorithm for surface evapotranspiration considering landscape**

Z. Q. Peng et al.

## HESSD

doi:10.5194/hess-2015-491

**Remote-sensing algorithm for surface evapotranspiration considering landscape**

Z. Q. Peng et al.

**Table 8.** The validation results of turbulent heat flux calculated by IPUS and TSFA.

| | IPUS – $H$ (W m$^{-2}$) | | | TSFA – $H$ (W m$^{-2}$) | | | IPUS – LE (W m$^{-2}$) | | | TSFA – LE (W m$^{-2}$) | | |
|---|---|---|---|---|---|---|---|---|---|---|---|---|
| Date | $R^2$ | MBE | RMSE | $R^2$ | MBE | RMSE | $R^2$ | MBE | RMSE | $R^2$ | MBE | RMSE |
| 0619 | 0.32 | 65.43 | 81.41 | 0.52 | 59.92 | 71.60 | 0.70 | −54.99 | 88.41 | 0.80 | −48.43 | 76.96 |
| 0630 | 0.60 | 34.94 | 59.15 | 0.76 | 20.35 | 34.17 | 0.85 | −57.37 | 78.10 | 0.89 | −41.29 | 60.30 |
| 0708 | 0.42 | 43.85 | 70.31 | 0.77 | 35.45 | 46.90 | 0.67 | −62.24 | 110.05 | 0.89 | −39.73 | 62.72 |
| 0727 | 0.73 | −30.96 | 54.99 | 0.90 | −26.41 | 37.42 | 0.82 | 27.33 | 75.16 | 0.91 | 24.97 | 58.24 |
| 0803 | 0.78 | −4.48 | 40.39 | 0.67 | −1.31 | 28.11 | 0.73 | −2.06 | 62.79 | 0.76 | −2.17 | 50.11 |
| 0815 | 0.55 | −22.90 | 52.30 | 0.70 | −10.19 | 37.12 | 0.84 | 12.61 | 57.48 | 0.92 | 2.04 | 38.79 |
| 0822 | 0.35 | 33.53 | 69.98 | 0.48 | 27.55 | 61.10 | 0.64 | −44.99 | 96.12 | 0.68 | −37.70 | 88.77 |
| 0829 | 0.72 | 25.28 | 43.09 | 0.81 | 19.71 | 34.53 | 0.81 | −54.29 | 78.84 | 0.82 | −50.24 | 74.50 |
| 0902 | 0.29 | −56.31 | 82.63 | 0.08 | −35.06 | 58.16 | 0.59 | 42.95 | 75.39 | 0.66 | 17.80 | 56.95 |
| 0913 | 0.01 | −20.65 | 65.90 | 0.43 | −5.76 | 31.05 | 0.52 | −44.41 | 78.98 | 0.70 | −60.64 | 77.49 |
| 0914 | 0.06 | −37.22 | 70.41 | 0.33 | −27.89 | 44.51 | 0.24 | 21.39 | 93.09 | 0.52 | 24.53 | 64.03 |

**Table 9.** Turbulent heat flux comparison results of IPUS and TSFA at EC4.

| Date | $H$ (W m$^{-2}$) | | | LE (W m$^{-2}$) | | |
|------|------|------|------|------|------|------|
| | EC | IPUS | TSFA | EC | IPUS | TSFA |
| 0619 | 150.65 | 100.88 | 187.69 | 278.55 | 407.58 | 360.62 |
| 0630 | 138.32 | 91.91 | 163.67 | 341.98 | 427.83 | 391.63 |
| 0708 | 117.04 | 55.59 | 153.65 | 361.16 | 510.49 | 449.77 |
| 0727 | 136.41 | 4.14 | 125.02 | 306.53 | 544.21 | 465.45 |
| 0803 | 68.97 | 36.51 | 116.30 | 389.63 | 498.21 | 454.23 |
| 0815 | 104.60 | 10.96 | 131.79 | 357.34 | 524.04 | 442.24 |
| 0822 | 125.34 | 77.23 | 138.77 | 318.08 | 423.85 | 402.21 |
| 0829 | 82.93 | 114.87 | 73.06 | 317.68 | 355.16 | 362.04 |
| 0902 | 162.05 | 86.79 | 174.27 | 280.41 | 382.37 | 330.72 |
| 0913 | 119.42 | 156.53 | 134.34 | 263.18 | 257.36 | 252.02 |
| 0914 | 110.02 | 78.89 | 138.16 | 262.33 | 343.17 | 317.85 |

| Station | IPUS – $H$ | | | TSFA – $H$ | | | IPUS – LE | | | TSFA – LE | | |
|---------|------|------|------|------|------|------|------|------|------|------|------|------|
| | $R^2$ | MBE | RMSE | $R^2$ | MBE | RMSE | $R^2$ | MBE | RMSE | $R^2$ | MBE | RMSE |
| EC4 | 0.02 | −45.59 | 65.42 | 0.65 | 20.09 | 26.81 | 0.53 | 108.86 | 125.03 | 0.65 | 68.36 | 78.90 |

unit: W m$^{-2}$

**HESSD**

doi:10.5194/hess-2015-491

**Remote-sensing algorithm for surface evapotranspiration considering landscape**

Z. Q. Peng et al.

Discussion Paper | Discussion Paper | Discussion Paper | Discussion Paper

Discussion Paper | Discussion Paper | Discussion Paper | Discussion Paper | Discussion Paper |

**HESSD**

doi:10.5194/hess-2015-491

**Remote-sensing algorithm for surface evapotranspiration considering landscape**

Z. Q. Peng et al.

**Table 10.** Turbulent heat flux validation results of IPUS and TSFA at EC15.

| Date | $H$ (W m$^{-2}$) | | | LE (W m$^{-2}$) | | |
|---|---|---|---|---|---|---|
| | EC | IPUS | TSFA | EC | IPUS | TSFA |
| 0619 | 92.55 | 106.60 | 99.81 | 419.47 | 427.19 | 429.98 |
| 0630 | 42.37 | 45.74 | 43.00 | 551.73 | 527.05 | 528.11 |
| 0708 | 18.34 | 67.53 | 59.90 | 620.95 | 575.71 | 574.86 |
| 0727 | 27.68 | 21.22 | 25.99 | 597.76 | 589.58 | 586.47 |
| 0803 | 2.33 | −2.53 | 2.91 | 592.37 | 604.04 | 599.74 |
| 0815 | 48.81 | 39.90 | 47.20 | 553.74 | 554.33 | 549.11 |
| 0822 | 54.59 | 154.34 | 158.60 | 473.68 | 408.37 | 405.07 |
| 0829 | 9.80 | 77.25 | 78.10 | 473.54 | 416.97 | 419.07 |
| 0913 | 176.96 | 209.96 | 204.90 | 307.72 | 221.05 | 227.61 |
| 0914 | 188.34 | 198.15 | 197.52 | 274.98 | 275.07 | 276.56 |

| Station | IPUS − $H$ | | | TSFA − $H$ | | | IPUS − LE | | | TSFA − LE | | |
|---|---|---|---|---|---|---|---|---|---|---|---|---|
| | $R^2$ | MBE | RMSE | $R^2$ | MBE | RMSE | $R^2$ | MBE | RMSE | $R^2$ | MBE | RMSE |
| EC15 | 0.76 | 25.64 | 42.96 | 0.75 | 25.61 | 42.58 | 0.94 | −26.66 | 42.30 | 0.94 | −26.94 | 41.34 |

unit: W m$^{-2}$

**HESSD**

doi:10.5194/hess-2015-491

**Remote-sensing algorithm for surface evapotranspiration considering landscape**

Z. Q. Peng et al.

**Table 11.** Sensible heat flux scale error of pixels containing different numbers of class types.

| Number of class types in mixed pixels | $R^2$ | MBD (W m$^{-2}$) | RMSD (W m$^{-2}$) | Pixel number |
|---|---|---|---|---|
| 1 | 1.00 | 0.21 | 2.04 | 11,398 |
| 2 | 0.75 | −6.54 | 33.44 | 8212 |
| 3 | 0.61 | −0.59 | 55.61 | 4762 |
| 4 | 0.42 | 13.63 | 77.44 | 2824 |
| 5 | 0.98 | −45.20 | 91.35 | 4 |

Notes: number of class types in mixed pixels means the number of classification types that were contained in the mixed pixels; for example, 1 represents pure pixels, while 2 represents two land use types that are contained in the mixed pixels and so forth. MBD and RMSD are the mean bias deviation and root mean square deviation between the TSFA and IPUS results, respectively.

Discussion Paper | Discussion Paper | Discussion Paper | Discussion Paper | Discussion Paper |

**HESSD**

doi:10.5194/hess-2015-491

**Remote-sensing algorithm for surface evapotranspiration considering landscape**

Z. Q. Peng et al.

**Table 12.** Sensible heat flux scale error of mixed pixels containing or not containing buildings between TSFA and IPUS.

| Mixed type in the pixel | $R^2$ | MBD ($\mathrm{W\,m^{-2}}$) | RMSD ($\mathrm{W\,m^{-2}}$) | Pixel number |
|---|---|---|---|---|
| Mixed pixels contain buildings | 0.53 | 9.71 | 72.38 | 4918 |
| Mixed pixels do not contain buildings | 0.69 | −6.06 | 37.26 | 10 884 |

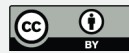

**HESSD**

doi:10.5194/hess-2015-491

**Remote-sensing algorithm for surface evapotranspiration considering landscape**

Z. Q. Peng et al.

**Table 13.** LST error analysis results at the homogeneous stations.

| Station | Date | Retrieved LST (K) | Observed LST (K) | LST bias (K) | TRE (%) | EC – $H$ (W m$^{-2}$) | $H$ from retrieved LST (W m$^{-2}$) | $H$ from observed LST (W m$^{-2}$) | $H$ relative error (%) |
|---|---|---|---|---|---|---|---|---|---|
| EC8 | 20120619 | 304.92 | 301.74 | 3.18 | 68.99 | 94.71 | 191.69 | 113.71 | 68.58 |
| EC7 | 20120630 | 302.5 | 299.35 | 3.15 | 517.22 | 40.33 | 116.16 | 11.78 | 886.08 |
| EC10 | 20120708 | 303.58 | 300.5 | 3.08 | 602.63 | 28.61 | 182.27 | 37.18 | 390.24 |
| EC15 | 20120708 | 303.55 | 300.13 | 3.42 | 445.52 | 18.34 | 217.53 | 39.51 | 450.57 |
| EC7 | 20120727 | 298.87 | 300.55 | −1.68 | −134.54 | 54.31 | −18.86 | 58.08 | −132.47 |
| EC19 | 20120727 | 307.86 | 316.82 | −8.96 | −55.22 | 286.56 | 104.86 | 264.5 | −60.36 |
| EC9 | 20120822 | 301.35 | 297.42 | 3.93 | 418.11 | 82.92 | 188.29 | 67.79 | 177.75 |
| EC2 | 20120822 | 299.79 | 298.05 | 1.74 | 66.95 | 72.32 | 186.23 | 111.38 | 67.20 |
| EC8 | 20120822 | 299.58 | 297.77 | 1.81 | 88.55 | 41.68 | 108.97 | 57.78 | 88.59 |
| EC10 | 20120822 | 301.61 | 298.04 | 3.57 | 219.6 | 90.38 | 197.71 | 82.86 | 138.61 |
| EC15 | 20120822 | 300.59 | 297.69 | 2.9 | 126.46 | 54.6 | 154.34 | 67.22 | 129.60 |
| EC8 | 20120829 | 301.54 | 300.44 | 1.1 | 76.01 | 19.7 | 66.3 | 40.45 | 63.91 |
| EC15 | 20120829 | 301.41 | 299.84 | 1.57 | 137.16 | 9.8 | 93.54 | 33.13 | 182.34 |
| EC19 | 20120902 | 304.9 | 303.42 | 1.48 | 9.61 | 131.01 | 214.33 | 192.46 | 11.36 |

Notes: "LST bias" is calculated as the retrieved LST minus the observed LST; "EC – $H$" is in situ sensible heat flux; $H$ relative error is the relative error between the retrieved and observed LST, expressed as (($H$ from retrieved LST) – ($H$ from retrieved LST))/($H$ from retrieved LST) × 100 %.

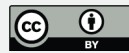

**Table 14.** The results of sensible heat flux at EC17.

| Date | EC – $H$ (W m$^{-2}$) | $H$ from incorrect canopy height (W m$^{-2}$) | $H$ from correct canopy height (W m$^{-2}$) |
|---|---|---|---|
| 20120815 | 159.77 | 6.89 | 47.12 |
| 20120822 | 165.78 | 20.74 | 143.21 |
| 20120902 | 235.96 | 26.97 | 162.13 |
| 20120914 | 117.64 | 76.23 | 170.94 |

Discussion Paper | Discussion Paper | Discussion Paper | Discussion Paper |

**HESSD**

doi:10.5194/hess-2015-491

**Remote-sensing algorithm for surface evapotranspiration considering landscape**

Z. Q. Peng et al.

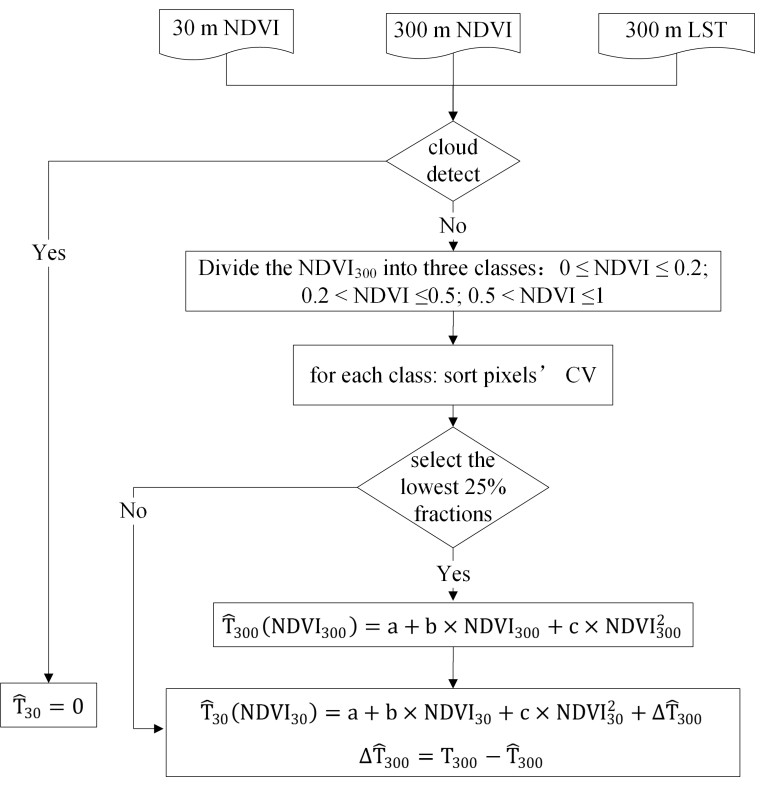

**Figure 1.** Flowchart of temperature sharpening.

Discussion Paper | Discussion Paper | Discussion Paper | Discussion Paper | Discussion Paper |

# HESSD

doi:10.5194/hess-2015-491

**Remote-sensing algorithm for surface evapotranspiration considering landscape**

Z. Q. Peng et al.

Discussion Paper | Discussion Paper | Discussion Paper | Discussion Paper | Discussion Paper |

# HESSD

doi:10.5194/hess-2015-491

**Remote-sensing algorithm for surface evapotranspiration considering landscape**

Z. Q. Peng et al.

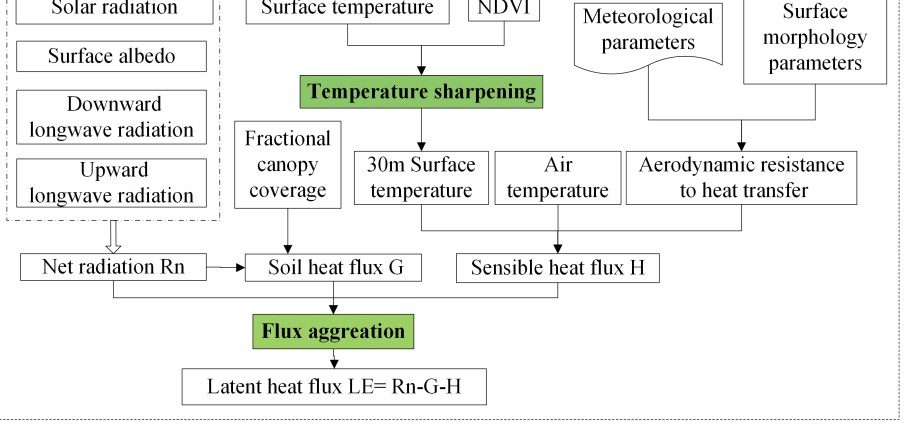

**Figure 2.** Flowchart of the ET retrieval using the "Temperature Sharpening and Flux Aggregation" method.

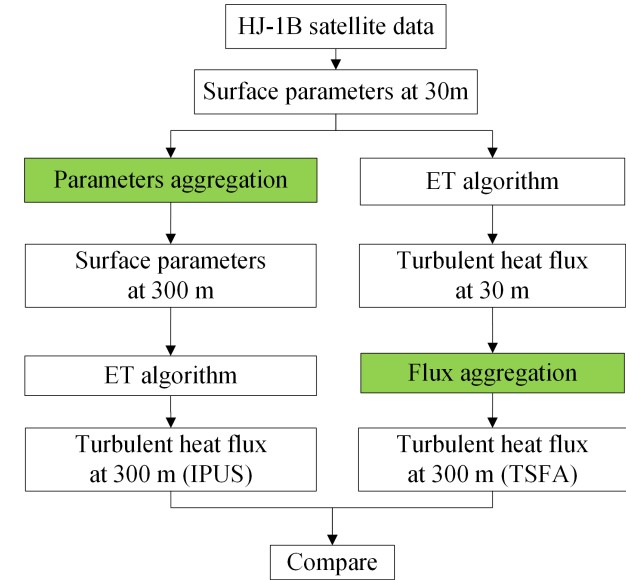

**Figure 3.** Flowchart of the two upscaling schemes for evapotranspiration retrieval.

# HESSD

doi:10.5194/hess-2015-491

**Remote-sensing algorithm for surface evapotranspiration considering landscape**

Z. Q. Peng et al.

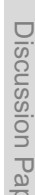

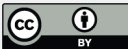

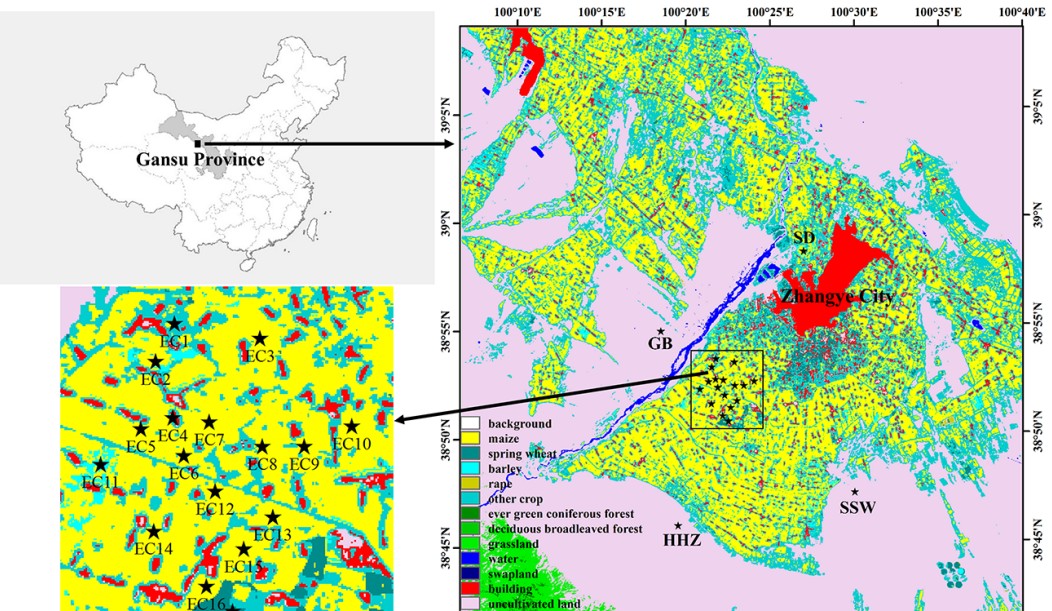

**Figure 4.** Study area and distribution of EC towers in HiWATER-MUSOEXE.

**HESSD**

doi:10.5194/hess-2015-491

**Remote-sensing algorithm for surface evapotranspiration considering landscape**

Z. Q. Peng et al.

## HESSD

doi:10.5194/hess-2015-491

**Remote-sensing algorithm for surface evapotranspiration considering landscape**

Z. Q. Peng et al.

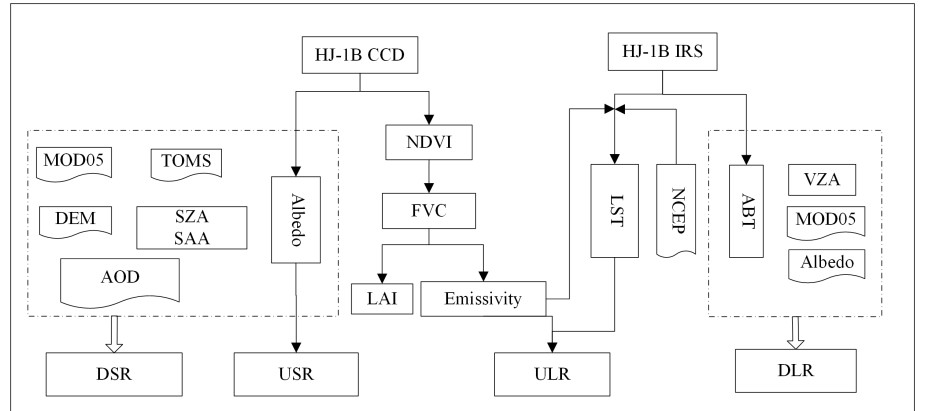

**Figure 5.** The flowchart of the land surface parameters retrievals. The abbreviations are as follows: SZA: solar zenith angle; SAA: solar azimuth angle; VZA: view zenith angle; AOD: aerosol optical depth; ABT: at-nadir brightness temperature; DSR: downward shortwave radiation; USR: upward shortwave radiation, ULR: upward longwave radiation; DLR: downward longwave radiation.

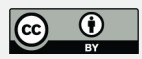



**Figure 6.** The spatial distribution of the four energy components, **(a)** $R_n$, **(b)** $G$, **(c)** $H$ and **(d)** LE, calculated by TSFA on 13 September 2012.

**HESSD**

doi:10.5194/hess-2015-491

**Remote-sensing algorithm for surface evapotranspiration considering landscape**

Z. Q. Peng et al.

## HESSD

doi:10.5194/hess-2015-491

**Remote-sensing algorithm for surface evapotranspiration considering landscape**

Z. Q. Peng et al.

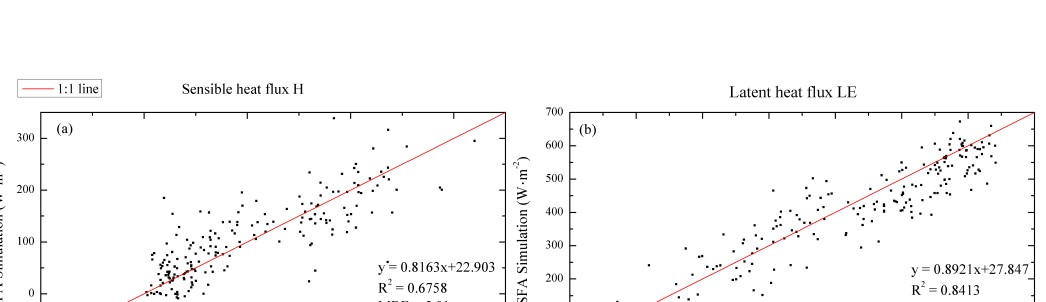

**Figure 7.** The scatter plot of the TSFA turbulent heat flux results.

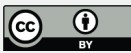

**HESSD**

doi:10.5194/hess-2015-491

**Remote-sensing algorithm for surface evapotranspiration considering landscape**

Z. Q. Peng et al.

**Figure 8.** The spatial distribution of the four energy components, **(a)** $R_n$, **(b)** $G$, **(c)** $H$ and **(d)** LE, as calculated by IPUS on 13 September 2012.

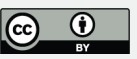

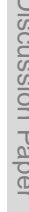

**HESSD**

doi:10.5194/hess-2015-491

**Remote-sensing algorithm for surface evapotranspiration considering landscape**

Z. Q. Peng et al.

**Figure 9.** The spatial distribution of the energy balance components bias as calculated by TSFA minus IPUS: **(a)** $R_n$, **(b)** $G$, **(c)** $H$ and **(d)** LE.

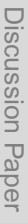
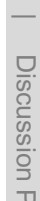
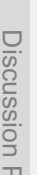

# HESSD

doi:10.5194/hess-2015-491

**Remote-sensing algorithm for surface evapotranspiration considering landscape**

Z. Q. Peng et al.

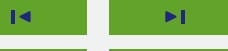
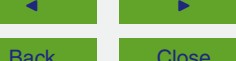
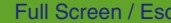

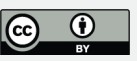

**Figure 10.** The scatter plots between TSFA and IPUS: (a) $R_n$, (b) $G$, (c) $H$ and (d) LE. MBD and RMSD are the mean bias deviation and root mean square deviation between the TSFA and IPUS results, respectively.

**HESSD**

doi:10.5194/hess-2015-491

**Remote-sensing algorithm for surface evapotranspiration considering landscape**

Z. Q. Peng et al.

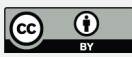

**Figure 11.** Distribution of classes and temperatures over the heterogeneous surface **(a)** EC4 and the homogeneous surface and **(b)** EC15 on 29 August 2012.

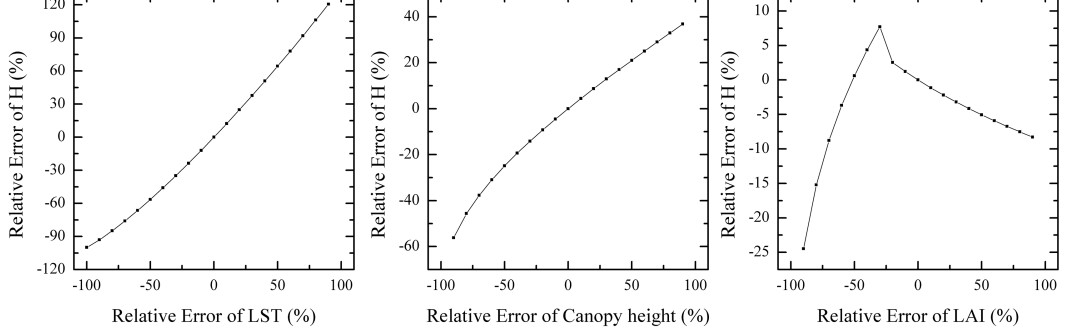

**Figure 12.** Sensitivity analysis of the surface parameters.

Discussion Paper | Discussion Paper | Discussion Paper | Discussion Paper |

## HESSD

doi:10.5194/hess-2015-491

**Remote-sensing algorithm for surface evapotranspiration considering landscape**

Z. Q. Peng et al.