# Peer review of "Remote-sensing algorithm for surface evapotranspiration"

_Hydrology and Earth System Sciences, 2015_

## Referee Comment (RC1) · Anonymous Referee #1 · 9 Feb 2016

General Comments:

This manuscript compares two different approaches to retrieve energy budget components (including sensible and latent heat flux) at the land surface using satellite data from the Chinese HJ-1B. One approach (IPUS) uses information aggregated to the 300m resolution as given by the thermal channel; the second approach (TSFA) uses a temperature sharpening approach, making use of a NDVI – TS relationship and downscaling 300m Ts information to the 30m scale. Authors illustrate the differences between both approaches and demonstrate within a validation exercise the advantages and improved prediction capacities of the latter approach. I think this comparison and the results obtained are in principle worth publishing and will be of use for the read-

ership of HESS. However, before a possible publication, author need to address and solve some significant concerns and questions that came up when working through the manuscript.

1. One of the major deficits of the manuscript is the following: The satellite data available are the 30m resolution data in the VIS/NIR spectral region and the 300m thermal information. As a "standard/normal" remote sensing user, I would try to make use of this available information. That means in a "reference application" (as it appears to me the IPUS scheme is meant to be) I would try to make use of the 30m data to derive NDVI and land use information (and all the other relevant parameters such as vegetation height, vegetation cover, roughness length etc., but also the simplification of individual fluxes for given LU type). Why are these parameters aggregated in the IPUS approach? Why don't use the high resolution information with an aggregated 300m Ts-signal. This should be compared to the TSFA approach in order to be able to evaluate the effect of purely temperature sharpening. Here actually the baseline situation is first worsened by aggregating information that is available in much higher resolution. In case the intention of the authors was to show what can happen when also in the VIS/NIR range only 300m resolution data were available, then all the (300m) average land surface parameters should have been derived from the aggregated reflectance information. So, I personally feel here are different aspects mixed and not properly separated.

2. The title of the manuscript suggests that the focus of the paper is on evapotranspiration – when looking through the manuscript and figures and tables, it seems to me that sensible heat flux is dominating the content and discussion. As a result, I would suggest to either change the title or put some more emphasis on ET in the presentation and discussion of results As a result of my evaluation I would suggest major revisions of the manuscript before a possible publication in HESS.

Specific Comments/Questions
[Figure]

- In general, there are a very large number of abbreviations used in the manuscript – not all of them are intuitive and it is painful to always try and find the first position where they are explained. So I would suggest generating a list of abbreviations.

- Figure and table legends are not self-explaining throughout the manuscript and need extension!

P2L14-22: While this paragraph is ok in principle, we as hydrologist all know how important ET – so in order to come quicker to the point it should be omitted.

P6L17: Why choosing the 25% fractions having the lowest CV? Please explain in the text!

P9L11: How is Ld calculated in the scheme?

P12L21ff: It remains unclear how albedo is calculated

P14L11: briefly describe how this is expressed/described (Ref)

P16L2: What reliable methods? This needs to be more specific and with references

P17ff: In the section 4.1 surface parameter and fluxes derived are evaluated against measurements. In order to put those results into a general context I think a discussion and comparison in relation to other international Remote sensing/Flux measurement campaigns should be given.

P20L9-10: This statement about errors is not very specific!

P22L5: How do you justify a ground heat lux of 0 for buildings?

P23L22ff: This statement is actually a result of what is summarized under point 1 in the general conclusion.

P24L25ff: Why do you use these specific day for calculating the sensitivities! In fig. 12 the x-axis shows variations in %. This makes it difficult to follow the interpretations of the curves in the section.

P29L5: Why do authors suddenly come up with the two source model – why didn't they use it initially?

P30: While the difference between Ts and Taero has been mentioned in the introduction, why isn't that problem discussed here!

P50: Table 13 – there is an error in the definition of the relative error (twice the same expression in the difference)

Minor Comments:

P3L4: Surface resistance is also needed for schemes classified under (1) because closure schemes need to calculate H where ra is required as well.

P3L20: Which models? All those listed in (1) - (5) or only those in (5)

P3L24-25: I do not understand "... inhomogeneity is a relative concept of homogeneity...!???

P3L26: Density of what?

P4L4ff: I do not understand that sentence/statement!

P13L18: what is <dáʈŃ> in equation (15)?

P14L1: Sentence (... H Li et al ...) does not make sense

P14L7: What is 6SLUT? Reference!

P28L1: Sentence ( ... greatly decreased the heterogeneity) does not make sense

---

## Referee Comment (RC2) · K. Mallick (Referee) · 15 Feb 2016

General Comments: In this manuscript, the authors compared two different spatial aggregation approaches to retrieve and evaluate the land surface energy balance fluxes using remote sensing data from the Chinese HJ-1B. One approach (IPUS) uses information aggregated to the 300m resolution as given by the thermal channel while the second approach (TSFA) uses a thermal sharpening approach by utilizing NDVI – TS relationship and downscaling 300m Ts into the 30m. Authors showed the differences between both approaches. Validation exercise is also performed to demonstrate the advantages and improved prediction capacities of the TSFA approach. This study is very useful to the community and worth publishing. However,

the authors need to address the following concerns before a possible publication. The sentence constructions also need to be better in some part of the manuscript. (1) A clear hypothesis and research question is missing in the manuscript. (2) Is it really necessary to aggregate the NDVI from 30 m to 300 m as described in the IPUS method? Why not using the 30 m NDVI with 300 m LST? (3) More emphasis is given on discussing the sensible heat flux (For example Table 11, 12, 13 and 14). A balanced discussion involving both LE and H would read better and rational. (4) Suggest including a table on different input data, their source and for what purpose they were used. (5) The table and figure captions need to be explicit. (6) Abstract: Some statistics need to be added in the abstract. At this moment it reads too general. (7) Page 2, line 16: Evapotranspiration is a variable, not a 'parameter' as stated by the authors. Authors should know the difference between a parameter and a variable. (8) Page 2, line 16: Reference is too old. Many recent references are available. (9) Page 2, line 22-22: This sentence does not carry anything meaningful. Please make your statement clear. (10) Page 3, line 37: it should be 'landscapes' instead of 'landscape'. (11) Page 3 (line 23 onwards to page 4): The last paragraph is quite confusing to understand. (12) Page 4, L26: 'Land based parameters'.….LAI, LST, DLR are not parameters, these are variables. This is becoming confusing now. (13) Page 5, L10: The resolution.…….. Need to be explicit on what is intended here by 'resolution'. (14) Throughout the entire manuscript, the authors are confused about 'parameter'. (15) Section 4.3.2, paragraph 3: The authors have not mentioned anything about the LE statistics of the two methods. (16) Spatial comparison of surface fluxes (as mentioned in section 4.3.2) should be done at least for 2 different vegetation cover conditions. (17) I made some edits and comments in the manuscript pdf (attached here), which the authors should consider.

Please also note the supplement to this comment:
http://www.hydrol-earth-syst-sci-discuss.net/hess-2015-491/hess-2015-491-RC2-supplement.pdf
[Figure]

**Supplement:**

Hydrol. Earth Syst. Sci. Discuss., 20, 1–63, 2016
www.hydrol-earth-syst-sci-discuss.net/20/1/2016/
doi:10.5194/hessd-20-1-2016

[revised manuscript text omitted]

**HESSD**

20, 1–63, 2016

[revised manuscript text omitted]

Z. Q. Peng et al.

[Figure]

**Figure 12.** Sensitivity analysis of the surface parameters.

[Figure]

---

## Author Comment (AC2) · 1 Apr 2016

**RESPONSES TO THE REFEREES**

We thank the reviewers for the comments. Below are our responses (in **blue** font) to the reviewers' comments and questions (in **black** font).

**K. Mallick (Referee)**

kaniska.mallick@gmail.com

General Comments:

In this manuscript, the authors compared two different spatial aggregation approaches to retrieve and evaluate the land surface energy balance fluxes using remote sensing data from the Chinese HJ-1B. One approach (IPUS) uses information aggregated to the 300m resolution as given by the thermal channel while the second approach (TSFA) uses a thermal sharpening approach by utilizing NDVI – TS relationship and downscaling 300m Ts into the 30m. Authors showed the differences between both approaches. Validation exercise is also performed to demonstrate the advantages and improved prediction capacities of the TSFA approach. This study is very useful to the community and worth publishing. However, the authors need to address the following concerns before a possible publication.

The sentence constructions also need to be better in some part of the manuscript.

(1) A clear hypothesis and research question is missing in the manuscript.

Response: Our basic hypothesis is that the inhomogeneity of surface landscapes and variables in the mixed pixels would result in large ET estimation error. In this study, we aimed to reduce the uncertainty of ET estimations caused by landscape and surface variables. We revised the introduction to clarify our hypothesis and goals.

(2) Is it really necessary to aggregate the NDVI from 30 m to 300 m as described in the IPUS method? Why not using the 30 m NDVI with 300 m LST?

Response: Thank you for your valuable comments. Although it is important to compare the TSFA method with the IPUS method, this comparison is not sufficient. We assume that ET estimation errors mainly result from the inhomogeneity of surface landscapes and variables. We aim to reduce the uncertainties of ET estimations due to surface heterogeneities and use the TSFA method as our final method. To evaluate the ability of the TSFA method to capture surface heterogeneity and reveal the scale effect, we used the IPUS method because it does not consider the effects of mixed pixels at all. According to your comments, we added the TRFA (temperature resampling and flux aggregation) method, which uses 30 m visible/near infrared and 300 m thermal infrared band data to estimate ET and simple spatial LST resampling (300 m to 30 m) instead of spatial sharpening based on NDVI information. Comparisons of the TFSA and TRFA methods can be used to evaluate the effects of temperature sharpening on estimating ET, as well as the significance of separating inhomogeneity of landscape from that of surface variables (such as LST), and that would make our logic clearer.

(3) More emphasis is given on discussing the sensible heat flux (For example Table 11, 12, 13 and 14). A balanced discussion involving both LE and H would read better and rational.

Response: We revised the manuscript by placing more emphasis on discussing the LE with a balance

analysis and discussion of H.

(4) Suggest including a table on different input data, their source and for what purpose they were used.
Response: Thank you for your suggestion. A table of abbreviations and the usage of input data were added in the appendix.

(5) The table and figure captions need to be explicit.
Response: We revised the table and figure captions.

(6) Abstract: Some statistics need to be added in the abstract. At this moment it reads too general.
Response: We revised the abstract by adding and updating statistical results as follows:
"Evapotranspiration (ET) plays an important role in surface-atmosphere interactions and can be monitored using remote sensing. However, surface heterogeneity, including the inhomogeneity of landscapes and variables, greatly affects the accuracy of ET retrieved by satellite. The objective of this study is to reduce the uncertainty that results from surface heterogeneity by using Chinese HJ-1B data. Three upscaling methods with area-weighted aggregation for different steps and variables were applied: input parameter upscaling (IPUS), which refers to parameter aggregation; temperature resampling and flux aggregation (TRFA), and temperature sharpening and flux aggregation (TSFA). Under a heterogeneous surface, the latent heat flux (LE) bias between the TSFA and IPUS methods varies statically from 35.36 to 65.66 W·m$^{-2}$, and the bias between the TSFA and TRFA methods varies statically from 4.41 to 22.53 W·m$^{-2}$. The footprint validation results show that the TSFA method could improve the accuracy of LE by approximately 20 W·m$^{-2}$ and 10 W·m$^{-2}$ relative to the IPUS and TRFA methods, respectively. Furthermore, additional analysis shows that the TSFA method can capture the sub-pixel variations of land surface temperature and integrate the effects of overlooked landscapes in mixed pixels. "

(7) Page 2, line 16: Evapotranspiration is a variable, not a 'parameter' as stated by the authors. Authors should know the difference between a parameter and a variable.
Response: We agree with your opinion that evapotranspiration is a variable rather than a 'parameter'. We have revised this phrasing throughout the manuscript.

(8) Page 2, line 16: Reference is too old. Many recent references are available.
Response: Thank you for this reminder. We agree that the presented references are old. This paragraph was mainly introduced to highlight the importance of ET. We deleted this paragraph because hydrologists should already understand the importance of ET.

(9) Page 2, line 22-22: This sentence does not carry anything meaningful. Please make your statement clear.
Response: We agree with your opinion and have deleted this meaningless sentence to introduce the models directly.

(10) Page 3, line 37: it should be 'landscapes' instead of 'landscape'.
Response: We have made this suggested correction.

(11) Page 3 (line 23 onwards to page 4): The last paragraph is quite confusing to understand.

Response: We have revised this paragraph as follows:

"However, heterogeneity is a natural attribute of the Earth's surface. Studies have shown that different landscapes (Blyth and Harding, 1995; Moran et al., 1997; Bonan et al., 2002; McCabe and Wood, 2006) and the sub-pixel variations of surface variables, such as stomatal conductance (Bin and Roni, 1994), leaf area index (Bonan et al., 1993; Maayar and Chen, 2006), and land surface temperature (Ershadi et al., 2013), can cause large errors in turbulent heat flux estimations. Surface landscape inhomogeneity can be classified using two scenarios: nonlinear vegetation density variations between sub-pixels (e.g., different types of vegetation mixed with each other or with bare soil) and coarse pixels containing different landscapes (e.g., vegetation or bare soil mixed with buildings or water). In mixed pixels, surface variables such as land surface temperature are set as singular to represent the entire pixel area in ET estimation models.

When these remotely sensed models are applied to calculate the regional ET via satellite data, large spatial scale errors occur. The non-linear operational model.is another important issue of remotely sensed spatial scale. However, it is difficult to develop linear operational models due to the complexity of mass and heat transfer processes between the atmosphere and land surface. In previous studies, researchers have coupled high- and low-resolution satellite data and statistically quantified the inhomogeneity of mixed pixels to correct the scale error in ET estimations (Zhou et al., 2016) by using temperature downscaling (Kustas et al., 2003; Norman et al., 2003; Cammalleri et al., 2013), the correction-factor method (Chen, 1999; Maayar and Chen, 2006) and the area-weighting method (Xin et al., 2012)."

References:

Bin, L., and Roni, A.: The Impact of Spatial Variability of Land-Surface Characteristics on Land-Surface Heat Fluxes, Journal of Climate, 7, 527-537, 10.1175/1520-0442(1994)007<0527:TIOSVO>2.0.CO;2, 1994.

Blyth, E. M., and Harding, R. J.: Application of aggregation models to surface heat flux from the Sahelian tiger bush, Agricultural and Forest Meteorology, 72, 213-235, http://dx.doi.org/10.1016/0168-1923(94)02164-F, 1995.

Bonan, G. B., Pollard, D., and Thompson, S. L.: Influence of Subgrid-Scale Heterogeneity in Leaf Area Index, Stomatal Resistance, and Soil Moisture on Grid-Scale Land–Atmosphere Interactions, Journal of Climate, 6, 1882-1897, 10.1175/1520-0442(1993)006<1882:IOSSHI>2.0.CO;2, 1993.

Bonan, G. B., Levis, S., Kergoat, L., and Oleson, K. W.: Landscapes as patches of plant functional types: An integrating concept for climate and ecosystem models, Global Biogeochemical Cycles, 16, 5-1-5-23, 10.1029/2000GB001360, 2002.

Cammalleri, C., Anderson, M. C., Gao, F., Hain, C. R., and Kustas, W. P.: A data fusion approach for mapping daily evapotranspiration at field scale, Water Resour. Res., 49, 4672-4686, 10.1002/wrcr.20349, 2013.

Chen, J. M.: Spatial Scaling of a Remotely Sensed Surface Parameter by Contexture, Remote Sens. Environ., 69, 30-42, http://dx.doi.org/10.1016/S0034-4257(99)00006-1, 1999.

Ershadi, A., McCabe, M. F., Evans, J. P., Mariethoz, G., and Kavetski, D.: A Bayesian analysis of sensible heat flux estimation: Quantifying uncertainty in meteorological forcing to improve model prediction, Water Resources Research, 49, 2343-2358, 10.1002/wrcr.20231, 2013.

Kustas, W. P., Norman, J. M., Anderson, M. C., and French, A. N.: Estimating subpixel surface

temperatures and energy fluxes from the vegetation index–radiometric temperature relationship, Remote Sens. Environ., 85, 429-440, http://dx.doi.org/10.1016/S0034-4257(03)00036-1, 2003.

Maayar, E. M., and Chen, J. M.: Spatial scaling of evapotranspiration as affected by heterogen eities in vegetation, topography, and soil texture, Remote Sensing of Environment, 102, 33-51, http://dx.doi.org/10.1016/j.rse.2006.01.017, 2006.

McCabe, M. F., and Wood, E. F.: Scale influences on the remote estimation of evapotranspiration using multiple satellite sensors, Remote Sensing of Environment, 105, 271-285, 10.1016/j.rse.2006.07.006, 2006.

Moran, M. S., Humes, K. S., and Pinter Jr, P. J.: The scaling characteristics of remotely-sensed variables for sparsely-vegetated heterogeneous landscapes, Journal of Hydrology, 190, 337-362, http://dx.doi.org/10.1016/S0022-1694(96)03133-2, 1997.

Norman, J. M., Anderson, M. C., Kustas, W. P., French, A. N., Mecikalski, J., Torn, R., Diak, G. R., Schmugge, T. J., and Tanner, B. C. W.: Remote sensing of surface energy fluxes at 101-m pixel resolutions, Water Resour. Res., 39, 1221, 10.1029/2002WR001775, 2003.

Xin, X., Liu, Y. N., Liu, Q., and Tang, Y.: Spatial-scale error correction methods for regional fluxes retrieval using MODIS data, J. Remote Sens., 16(2), 207-231, 2012.

Zhou, T., Peng, Z., Xin, X., and Li, F.: Remote sensing research of evapotranspiration over heterogeneous surface: A review, Journal of Remote Sensing, 20(2), 257-277, 10.11834/jrs.20155030, 2016.

(12) Page 4, L26: 'Land based parameters': : :..LAI, LST, DLR are not parameters, these are variables. This is becoming confusing now.

Response: Thank you for your suggestion. We have revised our use of 'parameters' throughout the manuscript.

(13) Page 5, L10: The resolution: : :: : :. Need to be explicit on what is intended here by 'resolution'.

Response: We revised this sentence as follows:

"The spatial resolution of TIR images is usually not as high as the spatial resolution of visible near-infrared bands (VNIR) because the energy of VNIR photons is higher than the energy of thermal photons. Thus, the inhomogeneity of TIR images would be greater than the inhomogeneity of VNIR images."

(14) Throughout the entire manuscript, the authors are confused about 'parameter'.

Response: Thank you for your suggestion. We have revised our use of 'parameter' throughout the manuscript.

(15) Section 4.3.2, paragraph 3: The authors have not mentioned anything about the LE statistics of the two methods.

Response: We revised this paragraph and emphasized the LE.

(16) Spatial comparison of surface fluxes (as mentioned in section 4.3.2) should be done at least for 2 different vegetation cover conditions.

Response: Thank you for your suggestion. We added comparisons of the turbulent heat fluxes for the two following weak heterogeneity conditions: (1) different vegetation cover conditions and (2)

vegetation mixed with bare soil.

(17) I made some edits and comments in the manuscript pdf (attached here), which the authors should consider.
Response: The provided edits and comments were addressed in the manuscript.

Specific comments in attached pdf:
How did you assign the crop height and ancillary parameter information in the stability corrections.
Response: A widely used parameterization scheme was used for stability correction. The equations used are listed below.
From the Monin-Obukhov similarity theory (MOST), the aerodynamic resistance $r_a$ can be calculated as follows:

$$r_a = \frac{1}{ku_*}\left[\ln\left(\frac{z-d}{z_{0m}}\right) - \psi_H\left(\frac{z-d}{L}\right)\right] \tag{1}$$

$$u_* = ku\left[\ln\left(\frac{z-d}{z_{0m}}\right) - \psi_M\left(\frac{z-d}{L}\right)\right]^{-1} \tag{2}$$

$$L = \rho c_p \frac{u_*^3 \theta_v}{kgH} \tag{3}$$

where $k = 0.4$ and is the von Karman's constant, $u_*$ is the friction velocity, $u$ is the wind speed at a reference height of $z$ above the surface, $d$ and $z_{0m}$ are the zero plane displacement height and the roughness length for momentum transfer, respectively, $L$ is the Monin-Obukhov length, $g$ is the acceleration due to gravity and $\theta_v$ is the potential virtual temperature near the surface. In addition, $\psi_M$ and $\psi_H$ are stability functions, where $\psi_M = \psi_H = 0$ under neutral conditions and $\psi_M$ and $\psi_H$ can be parameterized as follows under unstable conditions (Paulson, 1970; Ambast et al., 2002):

$$\psi_M = 2\ln\left[\frac{1+x}{2}\right] + \ln\left[\frac{1+x^2}{2}\right] - 2\tan^{-1}x + \frac{\pi}{2} \tag{4}$$

$$\psi_H = 2\ln[(1 + x^2)/2] \tag{5}$$

where $x = (1 - 16z/L)^{1/4}$. Under stable conditions, $\psi_M$ is equal to $\psi_H$ as follows (Webb, 1970):

$$\psi_M = \psi_H = -5 \cdot \frac{z-d}{L} \tag{6}$$

The parameterization of the zero plane displacement height d and the roughness length $z_{0m}$ are determined as follows (Choudhury and Monteith, 1988):

$$d = 1.1h\ln(1 + (c_d LAI)^{1/4}) \tag{7}$$

$$z_{0m} = \begin{cases} z_{0s} + 0.3h(c_d LAI)^{1/2} & 0 \leq c_d LAI \leq 0.2 \\ 0.3h\left(1 - \frac{d}{h}\right) & 0.2 < c_d LAI \leq 1.5 \end{cases} \tag{8}$$

where h is the canopy height and was set according to the area phenophase, classification and *a priori* knowledge. $c_d$ is the mean drag coefficient and is assumed uniform within the canopy, LAI is the leaf area index, and $z_{0s}$ is the substrate roughness length (for the bare soil surface, $z_{0s} = 0.01$).

References:
Ambast, S. K., Keshari, A. K., and Gosain, A. K.: An operational model for estimating Regional

Evapotranspiration through Surface Energy Partitioning (RESEP), International Journal of Remote Sensing, 23, 4917-4930, 10.1080/01431160110114501, 2002.

Choudhury, B. J., and Monteith, J. L.: A four-layer model for the heat budget of homogeneous land surfaces, Quarterly Journal of the Royal Meteorological Society, 114, 373-398, 10.1002/qj.49711448006, 1988.

Paulson, C. A.: The mathematical representation of wind speed and temperature profiles in the unstable atmospheric surface layer, Journal of Applied Meteorology, 9, 857-861, 1970.

Webb, E. K.: Profile relationships: The log-linear range, and extension to strong stability, Quarterly Journal of the Royal Meteorological Society, 96, 67-90, 10.1002/qj.49709640708, 1970.

How can you infer "The quadrangular with a relatively large bias in Fig. 9a and b is caused by DLR, i.e. it is influenced by the MOD05 water vapor."

Response: Bad lines appeared in the images scanned by MODIS Terra due to an instrumental malfunction that occurred beginning in 2002. After preprocessing the original data by interpolation, a weak quadrangular remained in the image. In addition, the MOD05 water vapor product was used to calculate downward longwave radiation in this paper, which is an important and sensitive variable of net radiation. We compared the results with the processed MOD05 product and observed that the quadrangular overlapped well. In addition, the order of magnitude at the quadrangular was within $\pm 5$ W·m$^{-2}$, which matches the bias caused by the downward longwave radiation between IPUS and TSFA.

We revised the expression as follows (Fig. 9 becomes Fig. 10 in the latest revised manuscript, in next page):

"The quadrangular with a relatively unstable bias shown in Fig. 10(a) is caused by the $L_d$ that was calculated from the MOD05 water vapor product which exists quadrangular even after preprocessing the instrument malfunction gap."

[Figure]

**Figure 10.** Maps of the bias of the energy balance components calculated using the TSFA method minus the IPUS method: (a) $R_n$, (b) G, (c) H, (d) LE, TSFA minus TRFA: (e) H and (f) LE.

---

## Author Response (AR1)

**RESPONSES TO THE REFEREES**

We thank the reviewers for the comments. Below are our responses (in **blue** font) to the reviewers' comments and questions (in **black** font).

**Anonymous Referee #1**

General Comments:

This manuscript compares two different approaches to retrieve energy budget components (including sensible and latent heat flux) at the land surface using satellite data from the Chinese HJ-1B. One approach (IPUS) uses information aggregated to the 300m resolution as given by the thermal channel; the second approach (TSFA) uses a temperature sharpening approach, making use of a NDVI – TS relationship and downscaling 300m Ts information to the 30m scale. Authors illustrate the differences between both approaches and demonstrate within a validation exercise the advantages and improved prediction capacities of the latter approach. I think this comparison and the results obtained are in principle worth publishing and will be of use for the readership of HESS. However, before a possible publication, author need to address and solve some significant concerns and questions that came up when working through the manuscript.

1. One of the major deficits of the manuscript is the following: The satellite data available are the 30m resolution data in the VIS/NIR spectral region and the 300m thermal information. As a "standard/normal" remote sensing user, I would try to make use of this available information. That means in a "reference application" (as it appears to me the IPUS scheme is meant to be) I would try to make use of the 30m data to derive NDVI and land use information (and all the other relevant parameters such as vegetation height, vegetation cover, roughness length etc., but also the simplification of individual fluxes for given LU type). Why are these parameters aggregated in the IPUS approach? Why don't use the high resolution information with an aggregated 300m Ts-signal. This should be compared to the TSFA approach in order to be able to evaluate the effect of purely temperature sharpening. Here actually the baseline situation is first worsened by aggregating information that is available in much higher resolution. In case the intention of the authors was to show what can happen when also in the VIS/NIR range only 300m resolution data were available, then all the (300m) average land surface parameters should have been derived from the aggregated reflectance information. So, I personally feel here are different aspects mixed and not properly separated.

Response: Thank you for your valuable comments. We assume that the ET estimation errors mainly come from inhomogeneity of surface landscape and variables. We aimed to reduce the uncertainties in estimated ET that caused by surface heterogeneities, and the TSFA is our final scheme. To evaluate the ability of the TSFA method to capture surface heterogeneity and reveal the scale effect, we used the IPUS method because it does not consider the effects of mixed pixels at all. According to your comments, we added the TRFA (temperature resampling and flux aggregation) method, which uses 30 m visible/near infrared band data with 300 m thermal infrared band data to estimate ET. In this method, simple spatial resampling (300 m to 30 m) of LST was used instead of spatial sharpening according to NDVI information. Comparisons between the TFSA and TRFA

methods can be used to evaluate the effects of temperature sharpening on estimating ET, as well as the significance of separating inhomogeneity of landscape from that of surface variables (such as LST), and that would make our logic clearer.

2. The title of the manuscript suggests that the focus of the paper is on evapotranspiration – when looking through the manuscript and figures and tables, it seems to me that sensible heat flux is dominating the content and discussion. As a result, I would suggest to either change the title or put some more emphasis on ET in the presentation and discussion of results. As a result of my evaluation I would suggest major revisions of the manuscript before a possible publication in HESS.
Response: Thank you for your suggestion. We revised the manuscript by placing more emphasis on LE with a balance analysis and discussion of H.

Specific Comments/Questions

- In general, there are a very large number of abbreviations used in the manuscript – not all of them are intuitive and it is painful to always try and find the first position where they are explained. So I would suggest generating a list of abbreviations.
Response: Thank you for your suggestion. A table of abbreviations and the usage of input data were added in the appendix.

- Figure and table legends are not self-explaining throughout the manuscript and need extension!
Response: The figure and table legends were revised.

P2L14-22: While this paragraph is ok in principle, we as hydrologist all know how important ET – so in order to come quicker to the point it should be omitted.
Response: Thank you for your suggestion. We agree with your suggestion and have deleted this paragraph.

P6L17: Why choosing the 25% fractions having the lowest CV? Please explain in the text!
Response: We have added our justification for this decision in the manuscript.
According to the temperature sharpening method "DisTrad" proposed by Kustas et al. (2003), 25% of the pure pixels with the lowest CV are selected from each class. Regarding heterogeneity, lower CVs correspond with more homogeneous land surfaces. In addition, a fraction should guarantee that a sufficient number of pixels was obtained to fit a least-squares expression between $NDVI_{300}$ and $T_{300}$; thus, we choose 25% of the fractions with the lowest CVs.

P9L11: How is Ld calculated in the scheme?
Response: The Ld calculation method was introduced in Section 3.2.1.1. (P14L10).
Reference: Yu, S., Xin, X., and Liu, Q.: Estimation of clear-sky longwave downward radiation from HJ-1B thermal data, Sci. China Earth Sci., 56, 829-842, 10.1007/s11430-012-4507-z, 2013.

P12L21ff: It remains unclear how albedo is calculated

Response: The expression of the surface albedo computing method was modified as follows:

"According to the algorithm proposed by Liang et al. (2005) and Q. Liu et al. (2011), surface albedo was obtained from the top of the atmosphere (TOA) reflectance by the HJ-1 satellite with a lookup table based on an angular bin regression relationship. The surface albedo and bidirectional reflectance distribution function (BRDF) of the HJ-1 satellite in the regression procedure were monitored by using POLDER-3/PARASOL BRDF datasets, and BRDF was used to obtain the TOA reflectance using the 6S (Second Simulation of a Satellite Signal in the Solar Spectrum) radiation transfer mode."

Reference: Liang, S., Stroeve, J., and Box, J. E.: Mapping daily snow/ice shortwave broadband albedo from Moderate Resolution Imaging Spectroradiometer (MODIS): The improved direct retrieval algorithm and validation with Greenland in situ measurement, Journal of Geophysical Research: Atmospheres, 110, D10109, 10.1029/2004JD005493, 2005.

Liu, Q., Qu, Y., Wang, L. Z., Liu, N. F., and Liang, S. L.: Glass-Global Land Surface Broadband Albedo Product: Algorithm Theoretical Basis Document. Version, 1, 1-50, College of Global Change and Earth System Science, Beijing Norman University, 2011.

P14L11: briefly describe how this is expressed/described (Ref)

Response: The top-of-atmosphere (TOA) brightness temperature of the HJ-1B thermal channel was used as the atmospheric effective temperature.

As shown in Yu et al. (2013), "To investigate the relation between TOA brightness temperature of the HJ-1B thermal channel and near-surface air temperature, TOA brightness temperature of HJ-1B is simulated using the Thermodynamic Initial Guess Retrieval (TIGR) atmospheric profile database TIGR2002 (http://ara.abct.lmd.Polytechnique.fr/index.php?page=tigr) and MODTRAN radiative transfer model; it has high correlation with near-surface air temperature."

Reference: Yu, S., Xin, X., and Liu, Q.: Estimation of clear-sky longwave downward radiation from HJ-1B thermal data, Sci. China Earth Sci., 56, 829-842, 10.1007/s11430-012-4507-z, 2013.

P16L2: What reliable methods? This needs to be more specific and with references.

Response: The methods were added as follows:

"Reliable methods were used to ensure the quality of the turbulent heat flux data. Before the main campaign, an intercomparison of all instruments was conducted in the Gobi Desert (Xu et al., 2013). After basic processing, including spike removal and corrections for density fluctuations (WPL-correction), a four-step procedure (data were rejected when (1) the sensor was malfunctioning, (2) precipitation occurred within 1 h before or after collection, (3) the missing ratio was greater than 3% in the 30-min raw record and (4) the friction velocity was below 0.1 ms-1 at night) was performed to control the quality of the EC data, and EC outputs were available every 30 min (for more details see Liu et al., 2011; Xu et al., 2013)."

P17ff: In the section 4.1 surface parameter and fluxes derived are evaluated against measurements. In order to put those results into a general context I think a discussion and comparison in relation to other international Remote sensing/Flux measurement campaigns should be given.

Response: Thank you for your comments. Introduction and discussion of the other ground campaigns was given.

Introduction of the study area:

The Heihe River Basin has long served as a test bed for integrated watershed studies as well as land surface or hydrological experiments. Comprehensive experiments, such as Watershed Allied Telemetry Experimental Research (WATER) (Li et al., 2009), and an international experiment - the Heihe Basin Field Experiment (HEIFE) in World Climate Research Programme (WCRP) have taken place in the Heihe River Basin. One major objective of HiWATER is to capture the strong land surface heterogeneities and associated uncertainties within a watershed (Li et al., 2013).

Discussion:

Our surface variable retrieval methods were validated against other areas considered in remote sensing measurement campaigns. For example, the albedo algorithm was previously applied to re-trieve Global Land Surface Satellite (GLASS) Products (Liang et al., 2014), the LST retrieval algo-rithm was validated in the Haihe River Basin in northern China (Li et al., 2011), and the soil heat flux correction algorithm was validated in the GAME-Tibet campaign (Yang and Wang, 2008). Since the surface of the Heihe River Basin is extreme heterogeneous, additional comparisons of our algorithm in other areas of research would be better.

References:

Hu Y Q, Gao Y X, Wang J M, Ji G L, Shen Z B, Chen L C, Chen J Y and Li S Q: Some achievements in scientific research during HEIFE, Plateau Meteorology, (03), 2-13, 1994.

Li, H., Liu, Q., Jiang, J., Wang, H., and Sun, L.: Validation of the land surface temperature derived from HJ-1B/IRS data with ground measurements, Geoscience and Remote Sensing Sympo-sium (IGARSS), 2011 IEEE International, Vancouver, Canada, 293-296, 2011.

Li, X., Cheng, G. D., Liu, S. M., Xiao, Q., Ma, M. G., Jin, R., Che, T., Liu, Q. H., Wang, W. Z., Qi, Y., Wen, J. G., Li, H. Y., Zhu, G. F., Guo, J. W., Ran, Y. H., Wang, S. G., Zhu, Z. L., Zhou, J., Hu, X. L., and Xu, Z. W.: Heihe Watershed Allied Telemetry Experimental Research (HiWATER): Scientific Objectives and Experimental Design, Bulletin of the American Meteorological Society, 94, 1145-1160, 10.1175/BAMS-D-12-00154.1, 2013.

Liang, S. L., Zhang, X. T., Xiao, Z. Q., Cheng, J., Liu , Q., and Zhao, X.: Global LAnd Surface Satellite (GLASS) Products: Algorithms, Validation and Analysis, 1 ed., SpringerBriefs in Earth Sciences, Springer International Publishing, 2014.

Yang, K., and Wang, J.: A temperature prediction-correction method for estimating surface soil heat flux from soil temperature and moisture data, Sci. China Ser. D-Earth Sci., 51, 721-729, 10.1007/s11430-008-0036-1, 2008.

P20L9-10: This statement about errors is not very specific!

Response: We deleted this statement because it was repetitive with the sentence preceding it.

P22L5: How do you justify a ground heat flux of 0 for buildings?

Response: In our study area, 'buildings' contain residents and roads. Influenced by local cli-mate situation, special materials with low heat conductance are used for residential buildings to maintain cool conditions during the summer and warm conditions during the winter. Thus, we jus-tified using a ground heat flux of 0 for buildings. According to your comments, the buildings of these residents were not prevalent. Thus, we recalculated all the data, and $G = 0.4R_n$ for buildings during the summer (Kato, 2005).

Reference: Kato, S., and Yamaguchi, Y.: Analysis of urban heat-island effect using ASTER and ETM+ Data: Separation of anthropogenic heat discharge and natural heat radiation from sensible heat flux, Remote Sensing of Environment, 99, 44-54, http://dx.doi.org/10.1016/j.rse.2005.04.026, 2005.

P23L22ff: This statement is actually a result of what is summarized under point 1 in the general conclusion.

Response: We did not observe any relations between the statement at P23L22 and the general conclusion.

P24L25ff: Why do you use these specific day for calculating the sensitivities! In fig. 12 the x-axis shows variations in %. This makes it difficult to follow the interpretations of the curves in the section.

Response: Sensitivity analysis is a general mathematic analysis procedure, and the presented input data for a specific day can indicate the influences of surface variables. We calculated the sensitivity results using large amounts of data from different phenophases, and our input data illustrated the sensitivities of our ET algorithm. To make the sensitivity analysis results universal, we drew a figure with % on the x-axis. We revised the paragraph and the x-axis in Figure 12 (Figure 13 in latest revised version) to make it easier to understand, especially for LST, because the discussed manuscript did not describe the x axis of LST variation clear, as follows:

"Since LE is calculated as a residual item in energy balance equations, the sensitivity of H is analyzed at first. Land surface variables (including LST, LAI, canopy height, and FVC) and meteorological variables (including wind speed, air temperature, air pressure and relative humidity) are needed to estimate H in this paper. To locate the error source when retrieving H, a sensitivity analysis was performed by adding errors at each 10% step (except LST). Fig. 13 presents the sensitivity analysis results: LST = 303.9 K (ranging from 298.4~309.4 K with a step size of 0.5 K), LAI=1.4 (ranging 0.14~2.66 with a step size of 0.14), canopy height equals 1 m (ranging 0.1~1.9 m with a step size of 0.1 m), FVC=0.5, wind speed u=2.48 m·s$^{-1}$, air temperature Ta=297.9 K, air pressure = 97.2 kPa, and RH=40.29%. In addition, the land use type is maize, and the reference H is 230.2 W·m$^{-2}$."

[Figure]

Figure 13. Sensitivity analysis of the surface variables for sensible heat flux

P29L5: Why do authors suddenly come up with the two source model – why didn't they use it initially?

Response: This sentence was deleted because it was not related to the objective of our study.

P30: While the difference between Ts and Taero has been mentioned in the introduction, why isn't that problem discussed here!

Response: Yes, we agree that the difference between Ts and Tareo should be discussed in this paper. "Excess" resistance $r_{ex}$ was added to $r_a$ to correct the discrepancy between Ts and Tareo in most remotely sensed evapotranspiration models. Thus, the error caused by the difference between Ts and Tareo was shifted to the parameterization scheme error of "excess" resistance, which we discussed in the discussion. We revised this section as follows to clarify the discussion.

"In addition, to correct the discrepancy between remotely sensed radiative surface temperature and aerodynamic temperature at the source of heat transport, a brief and well-performed parameterization scheme (under uniformly flat plant surface) of "excess" resistance was used to calculate the aerodynamic resistance of heat transfer (Jiao et al., 2014). Because the objects of our study are mixed pixels, more parameterization methods should be compared to select the optimum method."

Reference: Jiao, J. J, Xin, X. Z., Yu S. S., Zhou, T. and Peng, Z. Q.: Estimation of surface energy balance from HJ-1 satellite data. Journal of Remote Sensing, 18(5), 1048-1058, doi:10.11834/jrs.20143322, 2014

P50: Table 13 – there is an error in the definition of the relative error (twice the same expression in the difference)

Response: The mistake regarding the definition of relative error was corrected.

Minor Comments:

P3L4: Surface resistance is also needed for schemes classified under (1) because closure schemes need to calculate H where ra is required as well.

Response: We classified these remotely sensed models to discuss their drawbacks when used for heterogeneous surfaces. In addition, surface resistance is also needed for Penman-Monteith equations. Thus, we do not think surface resistance needs to be classified in this paper because it would disturb the flow of the manuscript and is not a focal point of our study.

P3L20: Which models? All those listed in (1) - (5) or only those in (5)

Response: All those models listed in (1) – (5).

P3L24-25: I do not understand ": : : inhomogeneity is a relative concept of homogeneity: : :!???

Response: We tried to express comparison concept of heterogeneous surface and weak heterogeneous surface and homogeneous surface. And we removed this sentence.

P3L26: Density of what?

Response: The density of the vegetation variations. We revised this paragraph for clarity as follows: "Surface landscape inhomogeneity can be classified using two scenarios: nonlinear vegetation density variations between sub-pixels (e.g., different types of vegetation mixed with each other or with bare soil) and coarse pixels containing total different landscapes (e.g., vegetation or bare soil mixed with buildings or water)."

P4L4ff: I do not understand that sentence/statement!

Response: We have revised this sentence as follows:

"However, it is difficult to develop linear operational models due to the complexity of mass and heat transfer processes between the atmosphere and land surface."

P13L18: what is <dε> in equation (15)?

Response: <dε> is an effective value of the cavity effect of emissivity and is the mean dε of all vegetation species. In this paper, <dε>=0.015. The definition of <dε> was updated in the manuscript.

P14L1: Sentence (: : : H Li et al : : : ) does not make sense.

Response: The names of the authors were located incorrectly due to typesetting. We corrected this problem as follows:

"A single-channel parametric model for retrieving LST based on HJ-1B/IRS TIR data developed by H. Li et al. (2010) was applied."

P14L7: What is 6SLUT? Reference!

Response: 6SLUT is a look up table that was generated by the 6S (Second Simulation of a Satellite Signal in the Solar Spectrum) radiation transfer mode (Vermote et al., 2006). The following reference was added in the paper.

Reference: Vermote E F, Tanre D, Deuze J L, et al. Second Simulation of a Satellite Signal in the Solar Spectrum-Vector. 6S User Guide Version 3, 2006.

P28L1: Sentence ( : : : greatly decreased the heterogeneity) does not make sense

Response: We have corrected this expression as follows:

"The temperature sharpening algorithm in TSFA uses the NDVI at 30 m to monitor the LST at 30 m and is capable of decreasing the influences of the heterogeneity of the LST."

**K. Mallick (Referee)**

kaniska.mallick@gmail.com

General Comments:

In this manuscript, the authors compared two different spatial aggregation approaches to retrieve and evaluate the land surface energy balance fluxes using remote sensing data from the Chinese HJ-1B. One approach (IPUS) uses information aggregated to the 300m resolution as given by the thermal channel while the second approach (TSFA) uses a thermal sharpening approach by utilizing NDVI – TS relationship and downscaling 300m Ts into the 30m. Authors showed the differences between both approaches. Validation exercise is also performed to demonstrate the advantages and improved prediction capacities of the TSFA approach. This study is very useful to the community and worth publishing. However, the authors need to address the following concerns before a possible publication.

The sentence constructions also need to be better in some part of the manuscript.

(1) A clear hypothesis and research question is missing in the manuscript.

Response: Our basic hypothesis is that the inhomogeneity of surface landscapes and variables in the mixed pixels would result in large ET estimation error. In this study, we aimed to reduce the uncertainty of ET estimations caused by landscape and surface variables. We revised the introduction to clarify our hypothesis and goals.

(2) Is it really necessary to aggregate the NDVI from 30 m to 300 m as described in the IPUS method? Why not using the 30 m NDVI with 300 m LST?

Response: Thank you for your valuable comments. Although it is important to compare the TSFA method with the IPUS method, this comparison is not sufficient. We assume that ET estimation errors mainly result from the inhomogeneity of surface landscapes and variables. We aim to reduce the uncertainties of ET estimations due to surface heterogeneities and use the TSFA method as our final method. To evaluate the ability of the TSFA method to capture surface heterogeneity and reveal the scale effect, we used the IPUS method because it does not consider the effects of mixed pixels at all. According to your comments, we added the TRFA (temperature resampling and flux aggregation) method, which uses 30 m visible/near infrared and 300 m thermal infrared band data to estimate ET and simple spatial LST resampling (300 m to 30 m) instead of spatial sharpening based on NDVI information. Comparisons of the TFSA and TRFA methods can be used to evaluate the effects of temperature sharpening on estimating ET, as well as the significance of separating inhomogeneity of landscape from that of surface variables (such as LST), and that would make our logic clearer.

(3) More emphasis is given on discussing the sensible heat flux (For example Table 11, 12, 13 and 14). A balanced discussion involving both LE and H would read better and rational.

Response: We revised the manuscript by placing more emphasis on discussing the LE with a balance analysis and discussion of H.

(4) Suggest including a table on different input data, their source and for what purpose they were used.

Response: Thank you for your suggestion. A table of abbreviations and the usage of input data were added in the appendix.

(5) The table and figure captions need to be explicit.

Response: We revised the table and figure captions.

(6) Abstract: Some statistics need to be added in the abstract. At this moment it reads too general.

Response: We revised the abstract by adding and updating statistical results as follows:

"Evapotranspiration (ET) plays an important role in surface-atmosphere interactions and can be monitored using remote sensing data. However, surface heterogeneity including inhomogeneity of landscapes and surface variables affects the accuracy of ET estimated from satellite data significantly. The objective of this study is to assess and reduce the uncertainties resulted from surface heterogeneity in remotely sensed ET using Chinese HJ-1B satellite data, which is of 30m spatial resolution in VIS/NIR bands and 300m spatial resolution in TIR band. A temperature sharpening and flux aggregation scheme (TSFA) was developed to obtain accurate heat fluxes from the HJ-1B satellite data. Two methods employing different upscaling policies of surface variables and fluxes were used to compare with TSFA, i.e., IPUS (input parameter upscaling) and TRFA (temperature resampling and flux aggregation). Moreover, the three methods can also be regarded as representing three typical schemes handling mixed pixels from the simplest to the most complex, i.e., all surface variables are at coarse resolution (300 m in this study) in IPUS and fine resolution (30 m in this study) in TSFA, while TRFA is in the middle (both 30m and 300m variables are used). Analysis and comparison between them can help us to get better understandings about spatial scale errors in remote sensing of surface heat fluxes. In situ data collected during HiWATER-MUSOEXE (Multi-Scale Observation Experiment on Evapotranspiration over heterogeneous land surfaces of The Heihe Watershed Allied Telemetry Experimental Research) were used for the validation and analysis of the methods. ET estimated by TSFA is of best agreement with in-situ observations, the footprint validation results show that the R2, MBE, and RMSE of the sensible heat flux (H) were 0.61, 0.90 W·m$^{-2}$ and 50.99 W·m$^{-2}$, respectively, and the corresponding terms for the latent heat flux (LE) were 0.82, -20.54 W·m$^{-2}$ and 71.24 W·m$^{-2}$, respectively, and IPUS showed the largest errors in ET estimation. The RMSE of LE between the TSFA and IPUS methods was 51.30 W·m$^{-2}$, and the RMSE of LE between the TSFA and TRFA methods was 16.48 W·m$^{-2}$. Furthermore, additional analysis shows that the TSFA method can capture the sub-pixel variations of land surface temperature and integrate the effects of overlooked landscapes in mixed pixels."

(7) Page 2, line 16: Evapotranspiration is a variable, not a 'parameter' as stated by the authors. Authors should know the difference between a parameter and a variable.

Response: We agree with your opinion that evapotranspiration is a variable rather than a 'parameter'. We have revised this phrasing throughout the manuscript.

(8) Page 2, line 16: Reference is too old. Many recent references are available.

Response: Thank you for this reminder. We agree that the presented references are old. This paragraph was mainly introduced to highlight the importance of ET. We deleted this paragraph because hydrologists should already understand the importance of ET.

(9) Page 2, line 22-22: This sentence does not carry anything meaningful. Please make your statement clear.

Response: We agree with your opinion and have deleted this meaningless sentence to introduce the models directly.

(10) Page 3, line 37: it should be 'landscapes' instead of 'landscape'.

Response: We have made this suggested correction.

(11) Page 3 (line 23 onwards to page 4): The last paragraph is quite confusing to understand.

Response: We have revised this paragraph.

(12) Page 4, L26: 'Land based parameters': : :..LAI, LST, DLR are not parameters, these are variables. This is becoming confusing now.

Response: Thank you for your suggestion. We have revised our use of 'parameters' throughout the manuscript.

(13) Page 5, L10: The resolution: : :: : :. Need to be explicit on what is intended here by 'resolution'.

Response: We revised this sentence as follows:

"The spatial resolution of TIR images is usually not as high as the spatial resolution of visible near-infrared bands (VNIR) because the energy of VNIR photons is higher than the energy of thermal photons. Thus, the inhomogeneity of TIR images would be greater than the inhomogeneity of VNIR images."

(14) Throughout the entire manuscript, the authors are confused about 'parameter'.

Response: Thank you for your suggestion. We have revised our use of 'parameter' throughout the manuscript.

(15) Section 4.3.2, paragraph 3: The authors have not mentioned anything about the LE statistics of the two methods.

Response: We revised this paragraph and emphasized the LE.

(16) Spatial comparison of surface fluxes (as mentioned in section 4.3.2) should be done at least for 2 different vegetation cover conditions.

Response: Thank you for your suggestion. We added comparisons of the turbulent heat fluxes for the two following weak heterogeneity conditions: (1) different vegetation cover conditions and (2) vegetation mixed with bare soil.

(17) I made some edits and comments in the manuscript pdf (attached here), which the authors should consider.

Response: The provided edits and comments were addressed in the manuscript.

Specific comments in attached pdf:

How did you assign the crop height and ancillary parameter information in the stability corrections.

Response: A widely used parameterization scheme was used for stability correction. The equations used are listed below.

From the Monin-Obukhov similarity theory (MOST), the aerodynamic resistance $r_a$ can be calculated as follows:

$$r_a = \frac{1}{ku_*}[\ln(\frac{z-d}{z_{0m}}) - \psi_H(\frac{z-d}{L})] \tag{1}$$

$$u_* = ku[\ln(\frac{z-d}{z_{0m}}) - \psi_M(\frac{z-d}{L})]^{-1} \tag{2}$$

$$L = -\rho c_p \frac{u_*^3 \theta_v}{kgH} \tag{3}$$

where $k = 0.4$ and is the von Karman's constant, $u_*$ is the friction velocity, $u$ is the wind speed at a reference height of $z$ above the surface, $d$ and $z_{0m}$ are the zero plane displacement height and the roughness length for momentum transfer, respectively, $L$ is the Monin-Obukhov length, $g$ is the acceleration due to gravity and $\theta_v$ is the potential virtual temperature near the surface. In addition, $\psi_M$ and $\psi_H$ are stability functions, where $\psi_M = \psi_H = 0$ under neutral conditions and $\psi_M$ and $\psi_H$ can be parameterized as follows under unstable conditions (Paulson, 1970; Ambast et al., 2002):

$$\psi_M = 2\ln\left[\frac{1+x}{2}\right] + \ln\left[\frac{1+x^2}{2}\right] - 2\tan^{-1}x + \frac{\pi}{2} \tag{4}$$

$$\psi_H = 2\ln[(1+x^2)/2] \tag{5}$$

where $x = (1 - 16z/L)^{1/4}$. Under stable conditions, $\psi_M$ is equal to $\psi_H$ as follows (Webb, 1970):

$$\psi_M = \psi_H = -5 \cdot \frac{z-d}{L} \tag{6}$$

The parameterization of the zero plane displacement height d and the roughness length $z_{0m}$ are determined as follows (Choudhury and Monteith, 1988):

$$d = 1.1h\ln(1 + (c_d LAI)^{1/4}) \tag{7}$$

$$z_{0m} = \begin{cases} z_{0s} + 0.3h(c_d LAI)^{1/2} & 0 \leq c_d LAI \leq 0.2 \\ 0.3h\left(1 - \frac{d}{h}\right) & 0.2 < c_d LAI \leq 1.5 \end{cases} \tag{8}$$

where h is the canopy height and was set according to the area phenophase, classification and a priori knowledge. $c_d$ is the mean drag coefficient and is assumed uniform within the canopy, LAI is the leaf area index, and $z_{0s}$ is the substrate roughness length (for the bare soil surface, $z_{0s} = 0.01$).

References:

Ambast, S. K., Keshari, A. K., and Gosain, A. K.: An operational model for estimating Regional Evapotranspiration through Surface Energy Partitioning (RESEP), International Journal of Remote Sensing, 23, 4917-4930, 10.1080/01431160110114501, 2002.

Choudhury, B. J., and Monteith, J. L.: A four-layer model for the heat budget of homogeneous land surfaces, Quarterly Journal of the Royal Meteorological Society, 114, 373-398, 10.1002/qj.49711448006, 1988.

Paulson, C. A.: The mathematical representation of wind speed and temperature profiles in the unstable atmospheric surface layer, Journal of Applied Meteorology, 9, 857-861, 1970.

Webb, E. K.: Profile relationships: The log-linear range, and extension to strong stability, Quarterly Journal of the Royal Meteorological Society, 96, 67-90, 10.1002/qj.49709640708, 1970.

How can you infer "The quadrangular with a relatively large bias in Fig. 9a and b is caused by DLR, i.e. it is influenced by the MOD05 water vapor."

Response: Bad lines appeared in the images scanned by MODIS Terra due to an instrumental malfunction that occurred beginning in 2002. After preprocessing the original data by interpolation, a weak quadrangular remained in the image. In addition, the MOD05 water vapor product was used to calculate downward longwave radiation in this paper, which is an important and sensitive variable of net radiation. We compared the results with the processed MOD05 product and observed that the quadrangular overlapped well. In addition, the order of magnitude at the quadrangular was within $\pm 5$ W·m$^{-2}$, which matches the bias caused by the downward longwave radiation between IPUS and TSFA.

We revised the expression as follows (Fig. 9 becomes Fig. 11 in the latest revised manuscript, in next page):

"The quadrangular with a relatively unstable bias shown in Fig. 11(a) is caused by the $L_d$ that was calculated from the MOD05 water vapor product which exists quadrangular even after prepro- cessing the instrument malfunction gap."

[Figure]

**Figure 11.** Maps of the bias of the energy balance components calculated using the TSFA method minus the IPUS method: (a) $R_n$, (b) G, (c) H, (d) LE, TSFA minus TRFA: (e) H and (f) LE.

**Relevant changes**

All the changes were marked as **red** color in the manuscript.

1. We have revised the abstract and introduction with hypothesis and research question stated.

2. We revised the manuscript by placing more emphasis on discussing the LE with a balance analysis and discussion of H.

3. Figure and table legends and names were revised.

4. A table of abbreviations and the usage of input data were added in the appendix.

5. New references were added.

**Remote-sensing algorithm for surface evapotranspiration considering landscape and statistical effects on mixed-pixels**

ZhiQing Peng [a, b], Xiaozhou Xin [a, *], JinJun Jiao [a, b], Ti Zhou [a, b], Qinhuo Liu [a, c]

a. State Key Laboratory of Remote Sensing Science, Institute of Remote Sensing and Digital Earth

Chinese Academy of Sciences, Beijing 100101, China b. University of Chinese Academy of Sciences, Beijing 100049, China c. Joint Center for Global Change Studies (JCGCS), Beijing 100875, China

**Abstract**

Evapotranspiration (ET) plays an important role in surface-atmosphere interactions and can be monitored using remote sensing data. However, surface heterogeneity including inhomogeneity of landscapes and surface variables affects the accuracy of ET estimated from satellite data significantly. The objective of this study is to assess and reduce the uncertainties resulted from surface heterogeneity in remotely sensed ET using Chinese HJ-1B satellite data, which is of 30m spatial resolution in VIS/NIR bands and 300m spatial resolution in TIR band. A temperature sharpening and flux aggregation scheme (TSFA) was developed to obtain accurate heat fluxes from the HJ-1B satellite data. Two methods employing different upscaling policies of surface variables and fluxes were used to compare with TSFA, i.e., IPUS (input parameter upscaling) and TRFA (temperature resampling and flux aggregation). Moreover, the three methods can also be regarded as representing three typical schemes handling mixed pixels from the simplest to the most complex, i.e., all surface variables are at coarse resolution (300 m in this study) in IPUS and fine resolution (30 m in this study) in TSFA, while TRFA is in the middle (both 30m and 300m variables are used). Analysis and comparison between them can help us to get better understandings about spatial scale errors in remote sensing of surface heat fluxes. In situ data collected during HiWATER-MUSOEXE (Multi-Scale Observation Experiment on Evapotranspiration over heterogeneous land surfaces of The Heihe Watershed Allied Telemetry Experimental Research) were used for the validation and analysis of the methods. ET estimated by TSFA is of best agreement with in-situ observations, the footprint validation results show that the $R^2$, MBE, and RMSE of the sensible heat flux (H) were 0.61,

0.90 W·m$^{-2}$ and 50.99 W·m$^{-2}$, respectively, and the corresponding terms for the latent heat flux (LE) were 0.82, -20.54 W·m$^{-2}$ and 71.24 W·m$^{-2}$, respectively, and IPUS showed the largest errors in ET estimation. The RMSE of LE between the TSFA and IPUS methods was 51.30 W·m$^{-2}$, and the RMSE of LE between the TSFA and TRFA methods was 16.48 W·m$^{-2}$. Furthermore, additional analysis shows that the TSFA method can capture the sub-pixel variations of land surface temperature and integrate the effects of overlooked landscapes in mixed pixels.

**Index Terms:** heterogeneous surface, temperature sharpening, area weighting, energy balance, evapotranspiration, spatial scale, HJ-1B satellite

**1. Introduction**

Five types of methods have been developed to estimate evapotranspiration (ET) or latent heat flux (LE) via remote sensing. (1) Surface energy balance models calculate LE as a residual term. According to the partitioning of the sources and sinks of the Soil-Plant-Atmosphere Continuum (SPAC), surface energy balance models can be classified as one-source (Bastiaanssen et al., 1998; Su, 2002; Allen et al., 2007; Long and Singh, 2012a) or two-source models (Shuttleworth and Wallace, 1985; Norman et al., 1995; Xin and Liu, 2010; Zhu et al., 2013). (2) Penman-Monteith models are used to calculate LE by using the Penman-Monteith equation and numerous surface resistance parameterization schemes that control the diffusion of evaporation from land surfaces and transpiration from plant canopies. These two-source Penman-Monteith models separate soil evaporation from plant transpiration (Cleugh et al., 2007; Mu et al., 2011; Leuning et al., 2008; Chen et al., 2013; Sun et al., 2013; Mallick et al., 2015). (3) Land surface temperature-vegetation index (LST-VI) space methods assign the dry and wet edges of the LST-VI feature space as minimum and maximum ET, respectively. These methods interpolate the media using the Penman-Monteith or Priestley-Taylor equation to calculate the LE (Jiang and Islam, 1999, 2001; Sun et al., 2011; Long and Singh, 2012b; Yang and Shang, 2013; Fan et al., 2015; Zhang et al., 2005). (4) Priestley-Taylor models expand the range of the Priestley-Taylor coefficient in the Priestley-Taylor equation (Jiang and Islam, 2003; Jin et al., 2011) or combine the physiological force factors with the energy component of ET (Fisher et al., 2008; Yao et al., 2013). (5) Additional methods include empirical/statistical methods (Wang and Liang, 2008; Yebra et al., 2013) and the use of complementary based models (Venturini et al., 2008) and land-process models with data assimilation schemes (Bateni and Liang, 2012; Xu et al., 2015).

All these ET estimation models are usually developed for simple and homogeneous surface conditions. When these remotely sensed models are applied to calculate the regional ET via satellite data, large spatial scale errors occur. Because heterogeneity is a natural attribute of the Earth's surface, non-linear operational model is another important issue of remotely sensed spatial scale error. However, it is difficult to develop linear operational models due to the complexity of mass and heat transfer processes between the atmosphere and land surface.

In previous studies, researchers have coupled high- and low-resolution satellite data and statistically quantified the inhomogeneity of mixed pixels to correct the scale error in ET estimations by using temperature downscaling that converts images from a lower (coarser) to higher (finer) spatial resolution using statistical-based models with regression or stochastic relationships among parameters (Kustas et al., 2003; Norman et al., 2003; Cammalleri et al., 2013; Ha et al., 2013), the correction-factor method that uses sub-pixel landscapes information to regress the correction factor of scale bias (Maayar and Chen, 2006) and the area-weighting method that calculates roughness length and sensible heat flux based on sub-pixel landscapes (Xin et al., 2012).These correction methods mainly focus on two problems: inhomogeneity of landscapes and inhomogeneity of surface variables.

Studies have shown that different landscapes (Blyth and Harding, 1995; Moran et al., 1997; Bonan et al., 2002; McCabe and Wood, 2006) and the sub-pixel variations of surface variables, such as stomatal conductance (Bin and Roni, 1994), leaf area index (Bonan et al., 1993; Maayar and Chen, 2006) can cause errors in turbulent heat flux estimations. Surface variables inhomogeneity is rather difficult to evaluate as the sub-pixel variation of surface variables could be large even in the pure pixel. For example, generally, temperatures over the land surfaces vary strongly in space and time, and it is not unusual for the LST to vary by more than 10 K over just a few centimeters of distance or by more than 1 K in less than a minute over certain cover types (Z. Li et al., 2013). But in mixed pixels, surface variables such as land surface temperature are set as singular to represent the entire pixel area in ET estimation models.

The focus of this study is on the effects of surface heterogeneity when estimating ET. According to the current satellites operation situation, three methods were used to analyze the uncertainty produced by surface heterogeneity. Input parameter upscaling (IPUS) does not consider the surface heterogeneities at all. It was designed to simulate the satellites that have identical spatial resolution both in visible near-infrared (VNIR) and thermal infrared bands (TIR), such as the land surface products of Moderate-Resolution Imaging Spectroradiometer (MODIS) satellites. Temperature resampling and flux aggregation (TRFA) only does not consider the heterogeneity of LST, and temperature sharpening and flux aggregation (TSFA) consider all the surface heterogeneities. They were designed for the majority of satellites data or products that have inconsistent spatial resolution between VNIR and TIR, such as Landsat and HJ-1B satellites.

Surface variables in this paper mainly derived from HJ-1B satellite data were used for this purpose. The Chinese HJ-1A/B satellites were launched on September 6, 2008, and were designed for disaster and environmental monitoring, as well as other applications. The HJ-1B satellites are equipped with two charge-coupled device (CCD) cameras and one infrared scanner (IRS) with spatial resolutions of 30 m and 300 m, respectively. Compared with high-temporal-resolution satellites, such as the MODIS satellite, or high-spatial-resolution satellites, such as the Landsat 7 or 8 satellites, HJ-1B has the advantage of a high spatial-temporal resolution. Since the satellites were launched, the HJ-1/CCD time series data have been widely used in China to accurately classify land cover (Zhong et al., 2014a) and monitor various environmental disasters (Wang et al., 2010). Land-based variables, such as leaf area index (LAI), land surface temperature (LST), and downward longwave radiation ($L_d$), have been retrieved by the HJ-1 satellites using algorithms developed by Chen et al. (2010), H. Li et al. (2010, 2011) and Yu et al. (2013), respectively. These variables lay the foundation for ET research.

Although the HJ-1B satellites provide CCD data with a high spatial resolution of 30 m, the spatial resolution of the thermal infrared (TIR) band is only 300 m. Thus, surface heterogeneity effects must be considered when estimating the heat flux.

**2. Methodology**

**2.1. Temperature-sharpening method based on statistical relationships**

Surface thermal dynamics are a driving force of ET. The spatial resolution of TIR images is usually not as high as the spatial resolution of visible near-infrared bands (VNIR) because the energy of VNIR photons is higher than the energy of thermal photons. Thus, the inhomogeneity of TIR

images would be greater than the inhomogeneity of VNIR images. Once the inhomogeneity of TIR images is enhanced, the uncertainty of the variables is calculated in the TIR band, and variables such as the land surface temperature become unpredictable. Therefore, we would like to derive land surface temperature data with a high spatial resolution.

The different spatial resolutions of TIR and VNIR images make it possible to obtain the land surface temperature at the spatial resolution of the VNIR images, which is referred to as temperature-sharpening. Kustas et al. (2003) proposed a statistical temperature-sharpening method that could be applied to remotely sensed evapotranspiration models. This method assumes that the negative correlation between the Normalized Difference Vegetation Index (NDVI) and LST is invariant. The NDVI reflects vegetation growth and cover, and the LST reflects surface thermal dynamics. The LST decreases with increasing vegetation cover. The resulting scatter plots form a feature space that is applicable at different scales when enough pixels exist.

HJ-1B satellite images can provide vegetation and thermal information at spatial resolutions of 30 m and 300 m, respectively. However, the 300 m resolution thermal data cannot be use to discriminate the surface temperatures of small targets within pixels. This deficiency can be addressed by using the functional relationship between NDVI and LST. A flowchart of temperature sharpening is shown in Fig. 1, and the LST at the NDVI pixel resolution can be derived based on the following steps (Kustas et al., 2003):

(1) The selection of a subset of pixels from the scene where the NDVI is the most uniform at a pixel resolution of 300 m. Calculate the coefficient of variation (CV) by using the original NDVI data ($NDVI_{30}$) with a resolution of 30 m and sort the values from smallest to largest. The CV is calculated as follows:

$$CV = \frac{STD}{mean} \tag{1}$$

where STD and mean are the standard deviation and the average values, respectively, among the $10 \times 10$ pixels that make up each 300-m NDVI ($NDVI_{300}$) aggregated from $NDVI_{30}$.

(2) Next, the $NDVI_{300}$ is divided into several classes ($0 \leq NDVI_{300} < 0.2$, $0.2 \leq NDVI_{300} < 0.5$ and $0.5 \leq NDVI_{300}$). Lower CV values correspond with more homogeneous land surface values, and a specific fraction should guarantee that a sufficient number of pixels is available for fitting a least-squares expression between $NDVI_{300}$ and $T_{300}$. Then, the fractions (25%) of the pixels having the lowest CV are selected from each class.

(3) A least-squares expression is fit between $NDVI_{300}$ and $T_{300}$ using the selected pixels.

$$\widehat{T}_{300}(NDVI_{300}) = a + b \times NDVI_{300} + c \times NDVI_{300}^2 \tag{2}$$

[Figure]

**Figure 1.** Flowchart of temperature sharpening.

(4) For each 30-m pixel within the 300-m pixel, $\widehat{T}_{30}$ can be computed according to Eq. (2) as follows:

$$\widehat{T}_{30}(NDVI_{30}) = a + b \times NDVI_{30} + c \times NDVI_{30}^2 + \Delta\widehat{T}_{300} \qquad (3)$$

where $\Delta\widehat{T}_{300} = T_{300} - \widehat{T}_{300}$ is the deviation between the regressed temperature and the temperature that was observed by the satellite at 300 m.

**2.2. Area-weighting method based on landscape information**

Coarse pixels are inhomogeneous because various types of land use may be included. Using a dominant type to represent such a large landscape is irrational. When a sharpened temperature is obtained, the spatial details could be provided by surface variables at a high resolution, and the inhomogeneous problem could be greatly diminished as the landscape is divided into finer pixels.

Combined with a high-resolution classification map, sub-pixel scale parameters can be applied to the ET algorithm, which is more rational than using a dominate-class type because different landscapes might require different ET algorithms. The surface energy flux can be averaged linearly due to the conservation of energy (Kustas et al., 2003), and a simple average that calculates the arithmetic mean over sub-pixels is the best choice for flux upscaling approaches (Ershadi et al., 2013b). Thus, the aggregated flux at a low resolution $F(x, y)$ is the arithmetic mean of all of the $n \times n$ sub-pixel fluxes that constitute the contributing flux $F(x_i, y_j)$ at coordinate $(x_i, y_j)$ as follows:

$$F(x, y) = \frac{1}{n \times n} \sum_{i=1}^{n} \sum_{j=1}^{n} F(x_i, y_j) \qquad (4)$$

Because the average of the sub-pixels fluxes is equal to the area-weighted sum of each land-type result, the final coarse result can be derived by the area-weighted sum of each land-type result within the landscape. The main steps of the area-weighting process are shown below (Xin et al., 2012):

(1) Geometric correction and registration of the VNIR and TIR input datasets.

(2) Count area ratio of different land-cover types within each pixel of a low-spatial-resolution classification image.

(3) According to the fine-classification data, different parameterization schemes can be used in the ET algorithm to calculate the sub-pixel flux, such as net radiation ($R_n$), soil heat flux ($G$) and sensible heat flux  (H).

(4) To calculate the regional flux, the flux of the large pixel is calculated by the area-weighting method as follows:

$$F = \sum_{i=1}^{n} w_i \cdot F_i \tag{5}$$

where $w_i$ is the fractional area contributing flux $F_i$ of class type  i, and  F  is the aggregated flux at the coarse resolution. The LE is computed as a residual of the surface energy balance in the TSFA

(Temperature Sharpening and Flux Aggregation, see Sect. 2.3) process, in which a high-spatial- resolution image is used to reduce the mixed pixels.

## 2.3. Pixel ET algorithm

The surface energy balance describes the energy between the land surface and atmosphere. The energy budget is commonly expressed as follows:

$$R_n = LE + H + G \tag{6}$$

where  $R_n$  is the net radiation,  G  is the soil heat flux,  H  is the sensible heat flux, and  LE  is the latent heat flux absorbed by water vapor when it evaporates from the soil surface and transpires from plants through stomata. The widely used one-source energy balance model considers the ho- mogeneous SPAC medium and ignores the inhomogeneity and structure. The  LE  can be expressed as follows:

$$LE = \frac{\rho c_p}{\gamma} \cdot \frac{e_s - e_a}{r_a + r_s} \tag{7}$$

where  $\gamma$  is the psychometric constant;  $e_s$  and  $e_a$  are the aerodynamic saturation vapor pressure and atmospheric water vapor pressure, respectively; and  $r_a$  and  $r_s$  are the water vapor transfer aerodynamic resistance and surface resistance, respectively. Surface resistance includes soil re- sistance and canopy resistance. The surface resistance is influenced by the physiological character- istics of the vegetation and the water supply of roots. Thus, it is difficult to obtain surface resistance by using remote sensing, and surface resistance is highly uncertain, particularly over heterogeneous surfaces. To avoid error introduced by the uncertainty of the surface resistance, the  LE  is computed as a residual of the surface energy balance equation.

$R_n$  is the difference between incoming and outgoing radiation and is calculated as follows:

$$R_n = S_d(1 - \alpha) + \varepsilon_s L_d - \varepsilon_s \sigma T_{rad}^4 \tag{8}$$

where  $S_d$  is the downward shortwave radiation,  $\alpha$  is the surface broadband albedo,  $\varepsilon_s$  is the emissivity of the land surface,  $L_d$  is the downward atmospheric longwave radiation,  $\sigma = 5.67 \times$

$10^{-8} W \cdot m^{-2} \cdot K^{-4}$  is the Stefan-Boltzmann constant, and  $T_{rad}$  is the surface radiation tempera- ture.

G  is commonly estimated by deriving empirical equations that consider surface variables, such as  $R_n$. Because the canopy exerts a significant influence on  G, the fractional canopy coverage FVC

is used to determine the ratio of  G  to  $R_n$  as follows:

$$G = R_n \times [\Gamma_c + (1 - FVC) \times (\Gamma_s - \Gamma_c)] \tag{9}$$

where  $\Gamma_s$  is 0.315 for bare soil and  $\Gamma_c$  is 0.05 for a full vegetation canopy (Su, 2002). H is the transfer of turbulent heat between the surface and atmosphere that is driven by a temperature differ- ence and is controlled by resistances that depend on local atmospheric conditions and land cover properties (Kalma et al., 2008). According to gradient diffusion theory,

$$H = \rho c_p \frac{T_{aero} - T_a}{r_a} \tag{10}$$

where $\rho$ is the density of the air; $c_p$ is the specific heat of the air at a constant pressure; $T_{aero}$ is the aerodynamic surface temperature obtained by extrapolating the logarithmic air-temperature profile to the roughness length for heat transport; $T_a$ is the air temperature at a reference height; and $r_a$ is the aerodynamic resistance, which influences the heat transfer between the source of turbulent heat flux and the reference height. Aerodynamic resistance was calculated based on the Monin-Obukhov similarity theory (MOST) using a stability correction function (Paulson, 1970; Ambast et al., 2002). The zero-plane displacement height, d, and roughness length, $z_{0m}$, were parameterized by the schemes proposed by Choudhury (Choudhury and Monteith, 1988).

In this approach, H must be accurately estimated. However, calculating H by using Eq. (10) is difficult. Because remote sensing cannot obtain $T_{aero}$, the value of $T_{aero}$ is usually replaced by the radiative surface temperature $T_{rad}$, which is not always equal to $T_{aero}$. The difference between these terms for homogeneous and fully covered vegetation is approximately 1-2℃ (Choudhury et al., 1986), or up to 10℃ in sparsely vegetative areas (Kustas, 1990). The method that corrects for this discrepancy adds "excess" resistance $r_{ex}$ to $r_a$. We used the brief method $r_{ex} = 4/u_*$, which was proposed by Chen (1988), to calculate $r_{ex}$.

Fig. 2 shows a flowchart for merging ET retrieval and temperature sharpening based on HJ-1B satellites.

[Figure]

**Figure 2.** Flowchart of ET retrieval using the "Temperature Sharpening and Flux Aggregation" method.

The spatial scale effect is usually revealed by a discrepancy between different upscaling methods: the upscaling of aggregate parameters to the large scale to calculate the heat flux and the calculation of the heat flux at the small scale before upscaling it to the large scale. In this paper, the resolution of the final output result is 300 m. To evaluate the reduced heterogeneity effect of TSFA, two other upscaling methods called IPUS and TRFA were used (see Fig. 3). When using IPUS, the surface-parameter retrieving algorithms (see Sect. 3.2.1.1) are applied to HJ-1 CCD data. Then, the variable results are aggregated at a spatial resolution of 300 m. These 300 m outputs are used as input parameters in the one-source energy balance model to obtain the four energy-balance components at 300 m. In TRFA, the LST at 300 m is resized to 30 m using nearest neighbor sampling. Then, the resampled LST and surface VNIR variables at 30 m are applied to ET algorithm. The outputs of the four energy-balance components of the TRFA are obtained using the area-weighting method shown in Sect. 2.2.

[Figure]

**Figure 3.** Flowchart of the three upscaling methods for retrieving evapotranspiration.

**3. Study area and Dataset**

**3.1. Study area**

Our study was conducted in the middle stream of the Heihe River Basin (HRB), which is located near the city of Zhangye in the arid region of Gansu Province in northwestern China (100.11°E-100.16°E, 39.10°N-39.15°N). The middle reach of the HRB is a typical desert-oasis agriculture ecosystem dominated by maize and wheat. A large portion of the Gobi Desert and the alpine vegetation of Qilian Mountain are located near the study area (see Fig. 4). The artificial oasis is highly heterogeneous, which impacts the thermal-dynamic and hydraulic features. Consequently, the water use efficiency and ET are variable. The Heihe River Basin has long served as a test bed for integrated watershed studies as well as land surface or hydrological experiments. Comprehensive experiments, such as Watershed Allied Telemetry Experimental Research (WATER) (Li et al., 2009), and an international experiment - the Heihe Basin Field Experiment (HEIFE) in World Climate Research Programme (WCRP) have taken place in the Heihe River Basin. One major objective of HiWATER is to capture the strong land surface heterogeneities and associated uncertainties within a watershed (Li et al., 2013).

[Figure]

**Figure 4.** Study area and distribution of EC towers in HiWATER-MUSOEXE

## 3.2. Dataset

In this paper, the data are mainly derived from the HJ-1B satellite. We combined these data
with ancillary data and the in situ "Multi-Scale Observation Experiment on Evapotranspiration over
heterogeneous land surfaces of The Heihe Watershed Allied Telemetry Experimental Research"
(HiWATER-MUSOEXE) data to estimate and validate the HJ-B land surface variables and heat
fluxes.

### 3.2.1. Remote sensing data

### 3.2.1.1. HJ-1B satellite data

The specifications of HJ-1B are shown in Table 1. These satellites have quasi-sun-synchronous
orbits at an altitude of 650 km, a swath width of 700 km and a revisit period of 4 days. Together, the
revisit period of the satellites is 48 h. Because HJ-1 CCDs lack an onboard calibration system,
scholars have proposed cross-calibration methods for calibrating the CCD instruments (Zhong et al.,
2014b; Zhang et al., 2013). The image quality of HJ-1A/B CCDs is stable, the performances of each
band are balanced (Zhang et al., 2013), and the radiometric performance of the HJ-1A/B CCD sen-
sors is similar to the performances of the Landsat-5 TM, Advanced Land Imager, and ASTER sen-
sors. The image quality of HJ-1 CCDs is very similar to the image quality of Landsat-5 TM (Jiang
et al., 2013). In addition, the accuracy of the TIR band's onboard calibration meets land surface
temperature retrieval requirements but not sea surface temperature retrieval requirements (J. Li et
al., 2011). China Center for Resources Satellite Data and Application (CRESDA) releases calibra-
tion coefficients once each year on its website (http://www.cresda.com). These data are freely avail-
able from the CRESDA website (http://218.247.138.121/DSSPlatform/index.html).

   **Table 1.** Specifications of the HJ-1B main payloads

| Sensor | Band | Spectral range (μm) | Spatial resolution (m) | Swath width (km) | Revisit time (days) |
|--------|------|---------------------|------------------------|------------------|---------------------|
|        | 1    | 0.43-0.52           |                        |                  |                     |

| | 2 | 0.52-0.60 | | | |
| --- | --- | --- | --- | --- | --- |
| CCD | 3 | 0.63-0.69 | 30 | 360 (single) | 4 |
| | 4 | 0.76-0.90 | | 700 (two) | |
| IRS | 5 | 0.75-1.10 | 150 | 720 | 4 |
| | 6 | 1.55-1.75 | | | |
| | 7 | 3.50-3.90 | | | |
| | 8 | 10.5-12.5 | 300 | | |

[Figure]

**Figure 5.** Flowchart of the land surface variable retrieval. The abbreviations are defined as follows: SZA: solar zenith angle; SAA: solar azimuth angle; VZA: view zenith angle; AOD: aerosol optical depth; ABT: at-nadir brightness temperature; $S_d$: downward shortwave radiation; USR: upward shortwave radiation, ULR: upward longwave radiation; and $L_d$: downward longwave radiation.

We used the HJ-1B satellite data for the HRB region in 2012. Because many variable-retrieving algorithms required clear-sky conditions for calculating ET, we combined data-quality information with visual interpretation to select satellite images without clouds. Considering the time period of the ground observations discussed in Sect. 3.2.2, we obtained data for 11 days: June 19, June 30, July 8, July 27, August 2, August 15, August 22, August 29, September 2, September 13 and September 14.

The HJ-1B satellite data from the HRB were pre-processed and included geometric correction, radiometric calibration, and atmosphere correction. For Eq. (1) to (10), the following surface variables are needed: downward shortwave radiation, downward longwave radiation, emissivity, albedo, fractional vegetation coverage (FVC), cloud mask data, meteorological data, LAI and LST. Fig. 5 contains a flowchart showing the retrieval of these variables.

(1) Surface albedo. According to the algorithm proposed by Liang et al. (2005) and Q. Liu et al. (2011), surface albedo was obtained from the top of the atmosphere (TOA) reflectance by the HJ-1 satellite with a lookup table based on an angular bin regression relationship. The surface albedo and bidirectional reflectance distribution function (BRDF) of the HJ-1 satellite in the regression procedure were monitored by using POLDER-3/PARASOL BRDF datasets, and BRDF was used to obtain the TOA reflectance using the 6S (Second Simulation of a Satellite Signal in the Solar Spectrum) radiation transfer mode.

(2) NDVI, FVC and LAI. The NDVI is the central regression of temperature sharpening and was used to calculate the FVC. Atmospherically corrected surface reflectance values were used to calculate the NDVI as follows:

$$\text{NDVI} = \frac{\rho_{\text{nir}} - \rho_{\text{red}}}{\rho_{\text{nir}} + \rho_{\text{red}}} \tag{11}$$

and

$$\text{FVC} = \frac{\text{NDVI} - \text{NDVI}_s}{\text{NDVI}_v + \text{NDVI}_s} \qquad (12)$$

where $\rho_{nir}$ and $\rho_{red}$ are the reflectances in the near-infrared and red band, respectively, and NDVI$_v$ and NDVI$_s$ are the fully vegetated and bare soil NDVI values, respectively. As an important input for the parameterization of surface roughness length and aerodynamic resistance, the LAI was determined using the following equation (Nilson, 1971):

$$P(\theta) = e^{-G(\theta) \cdot \Omega \cdot \text{LAI}/\cos(\theta)} \qquad (13)$$

$$P(\theta) = 1 - \text{FVC} \qquad (14)$$

where $\theta$ is the zenith angle, $P(\theta)$ is the angular distribution of the canopy gap fraction, $G(\theta)$ is the projection coefficient (0.5), and $\Omega$ is the total foliage clumping index, which can be obtained from the GLC global clumping index database according to the type of land use (He et al., 2012).

(3) Land surface emissivity (LSE). LSE is needed to calculate the $R_n$ and is extremely important for retrieving LST. In this paper, LSE was calculated using the FVC as follows (Valor and Caselles, 1996):

$$\varepsilon = \varepsilon_v \cdot \text{FVC} + \varepsilon_g(1 - \text{FVC}) + 4 < d\varepsilon > \cdot \text{FVC} \cdot (1 - \text{FVC}) \qquad (15)$$

where $\varepsilon$ is the LSE, $< d\varepsilon >$ is an effective value of the cavity effect of emissivity, the mean $d\varepsilon$ of all vegetation species in this study is <dε>=0.015, and $\varepsilon_v$ and $\varepsilon_g$ are the vegetation and ground emissivity, respectively.

(4) Land surface temperature. A single-channel parametric model for retrieving LST based on HJ-1B/IRS TIR data developed by H. Li et al. (2010) was applied. This model was developed from a parametric model based on MODTRAN4 using NCEP atmospheric profile data.

(5) Downward shortwave radiation. The algorithm proposed by L. Li et al. (2010) was applied. MOD05, TOMS, aerosol, and solar angle data were used to estimate the direct light flux and diffuse light flux by using a lookup table that was generated using the 6S radiation transfer mode (Vermote et al., 2006). This method considered the influences of complex terrain, and a topographic correction was performed by using products of the ASTER DEM.

(6) Downward longwave radiation ($L_d$). The TOA brightness temperature of the HJ-1B thermal channel was used to substitute the atmospheric effective temperature. Effective atmospheric emissivity was parameterized as an empirical function of the water vapor content. These values were substituted for atmospheric temperature and atmospheric emissivity to estimate the value of $L_d$. Because this $L_d$ retrieval method proposed by Yu et al. (2013) was only valid for clear-sky conditions, cloud masking information was used to determine clear skies. When cloud contamination existed in the image, the brightness temperature was relatively low, causing the $L_d$ to be lower than that in the cloudless images.

**3.2.1.2. Ancillary data**

Ancillary data were used because the bands of the satellite could not invert all of the variables needed for retrieving ET.

(1) Atmospheric water vapor data. MODIS provides water vapor data (MOD05), including a 1-km near-infrared product and a 5-km thermal-infrared product, every day. The 1-km near-infrared water vapor product was used to retrieve $L_d$ in this study.

(2) Surface elevation data. We used the 30-m resolution Global Digital Elevation Model (GDEM) based on ASTER, which covers 83 °N–83 °S, to derive $S_d$.

(3) Atmosphere ozone data. A Total Ozone Mapping Spectrometer (TOMS), which was carried on an Earth Probe (EP) satellite, was used to derive $S_d$. The TOMS-EP provided daily global atmosphere ozone data at a resolution of $1\,°×1.25\,°$ (Li et al., 2010b).

(4) Atmosphere profile data. Global reanalysis data from the National Centers for Environmental Prediction (NCEP) were used to derive LST. These data were generated globally every 6 hours (0:00, 06:00, 12:00, 18:00 UTC) for every $1\,°$ of latitude and longitude (Li et al., 2010a).

**3.2.2. HiWATER experiment dataset**

The in situ HRB observation data were provided by HiWATER. From June to September 2012, HiWATER designed two nested observation matrices over 30 km×30 km and 5.5 km×5.5 km within the middle stream oasis in Zhangye to focus on the heterogeneity of the scale effect in the so-called HiWATER-MUSOEXE.

In a larger observation matrix, four eddy covariance (EC) systems and one superstation were installed in the oasis–desert ecosystem. Each station was supplemented with an automatic meteorological station (AMS) to record meteorological and soil variables and monitor the spatial–temporal variations of ET and its impact factors (Li et al., 2013). The station information is shown in Table 2, and the distribution of the stations is shown in Fig. 4. Within the artificial oasis, an observation matrix composed of 17 EC towers and ordinary AMSs exists where the superstation was located. The land surface was heterogeneous and dominated by maize, maize inter-cropped with spring wheat, vegetables, orchards, and residential areas (Li et al., 2013). Because the EC16 and HHZ stations lacked $R_n$ and G observation data, they were excluded from this study.

**Table 2.** The in situ HiWATER-MUSOEXE station information

| Station | Longitude (°) | Latitude (°) | Tower height (m) | Altitude (m) | Land cover |
|---------|--------------|-------------|------------------|--------------|------------|
| EC1 | 100.36E | 38.89N | 3.8 | 1552.75 | vegetation |
| EC2 | 100.35E | 38.89N | 3.7 | 1559.09 | maize |
| EC3 | 100.38E | 38.89N | 3.8 | 1543.05 | maize |
| EC4 | 100.36E | 38.88N | 4.2 | 1561.87 | building |
| EC5 | 100.35E | 38.88N | 3 | 1567.65 | maize |
| EC6 | 100.36E | 38.87N | 4.6 | 1562.97 | maize |
| EC7 | 100.37E | 38.88N | 3.8 | 1556.39 | maize |
| EC8 | 100.38E | 38.87N | 3.2 | 1550.06 | maize |
| EC9 | 100.39E | 38.87N | 3.9 | 1543.34 | maize |
| EC10 | 100.40E | 38.88N | 4.8 | 1534.73 | maize |
| EC11 | 100.34E | 38.87N | 3.5 | 1575.65 | maize |
| EC12 | 100.37E | 38.87N | 3.5 | 1559.25 | maize |
| EC13 | 100.38E | 38.86N | 5 | 1550.73 | maize |
| EC14 | 100.35E | 38.86N | 4.6 | 1570.23 | maize |
| EC15 | 100.37E | 38.86N | 4.5 | 1556.06 | maize |
| EC17 | 100.37E | 38.85N | 7 | 1559.63 | orchard |
| GB | 100.30E | 38.91N | 4.6 | 1562 | uncultivated land-Gobi |
| SSW | 100.49E | 38.79N | 4.6 | 1594 | uncultivated land-desert |
| SD | 100.45E | 38.98N | 5.2 | 1460 | swamp land |

The ground observation data include the H and LE. Reliable methods were used to ensure the quality of the turbulent heat flux data. Before the main campaign, an intercomparison of all instruments was conducted in the Gobi Desert (Xu et al., 2013). After basic processing, including spike removal and corrections for density fluctuations (WPL-correction), a four-step procedure (data were rejected when (1) the sensor was malfunctioning, (2) precipitation occurred within 1 h before or after collection, (3) the missing ratio was greater than 3% in the 30-min raw record and (4) the friction velocity was below 0.1 ms$^{-1}$ at night) was performed to control the quality of the EC data, and EC outputs were available every 30 min (for more details see Liu et al., 2011b; Xu et al., 2013). G was measured by using three soil heat plates at a depth of 6 cm at each site, and the surface G was calculated using the method proposed by (Yang and Wang, 2008) based on the soil temperature and moisture above the plates. Surface meteorological variables, such as wind speed, wind direction, relative humidity and air pressure, were used to interpolate images using the inverse-distance weighted method. Researchers can obtain these data from the websites of the Cold and Arid Regions Science Data Center at LanZhou http://card.westgis.ac.cn/ or the Heihe Plan Data Management Center http://www.heihedata.org/.

An energy imbalance is common in ground flux observations. The conserving Bowen ratio (H/LE) and residual closure technique are often used to force energy balance. Computing the LE as a residual variable may be a better method for energy balance closure under conditions with large LEs (small or negative Bowen ratios due to strong advection) (Kustas et al., 2012). Thus, the residual closure method was applied because the "oasis effect" was distinctly observed in the desert-oasis system on clear days during the summer (Liu et al., 2011).

**4. Results and analysis**

**4.1. Evaluation of surface variables**

To control the model input variables and analyze sources of error, the coarse-resolution land surface temperature, downward shortwave radiation, downward longwave radiation, $R_n$ and G were evaluated using in situ data.

The ground-based land surface temperature, $T_s$, was calculated using the Stefan-Boltzman Law from the AMS measurements of the longwave radiation fluxes (Li et al., 2014) as follows:

$$T_s = \left[ \frac{L^{\uparrow} - (1 - \varepsilon_s) \cdot L^{\downarrow}}{\varepsilon_s \cdot \sigma} \right]^{\frac{1}{4}} \tag{16}$$

in which $L^{\uparrow}$ and $L^{\downarrow}$ are in situ surface upwelling and atmospheric downwelling longwave radiation, respectively, and $\varepsilon_s$ is the surface broadband emissivity, which is regarded as the pixel value of the HJ-1B at the AMS. The coefficient of determination $R^2$, mean bias error (MBE) and root mean square error (RMSE) of the LST are 0.71, -0.14 K and 3.37 K, respectively. As seen in Table 3, the accuracy of EC4 is low. The main causes of the large errors are as follows: (1) because buildings and soil/vegetation are distinct materials, the LSE algorithm may not be suitable for buildings and (2) the EC4 foundation is non-uniform and is not suitable for validation. After removing the EC4 data, the $R^2$, MBE, and RMSE of the LSTs were 0.83, 0.69 K and 2.51 K, respectively. The LST errors of SSW and SD were large due to large errors on particular days. For example, although it was briefly cloudy above station SSW on July 27, this area was not identified as cloudy in the cloud detection process.

**Table 3.** Station validation results of land surface temperature

| station | $R^2$ | MBE (K) | RMSE (K) | station | $R^2$ | MBE (K) | RMSE (K) |
|---|---|---|---|---|---|---|---|
| EC1 | 0.82 | 0.18 | 1.74 | EC11 | 0.42 | 1.59 | 2.98 |
| EC2 | 0.82 | 0.59 | 1.54 | EC12 | 0.87 | 0.62 | 1.51 |
| EC3 | 0.69 | 0.38 | 1.90 | EC13 | 0.83 | 0.44 | 1.48 |
| EC4 | 0.83 | -9.87 | 10.04 | EC14 | 0.73 | 1.43 | 2.44 |
| EC5 | 0.83 | 1.71 | 2.34 | EC15 | 0.74 | 1.53 | 2.41 |
| EC6 | 0.61 | 0.30 | 2.44 | EC17 | 0.78 | 1.20 | 2.32 |
| EC7 | 0.82 | 0.39 | 1.40 | GB | 0.69 | 0.12 | 2.33 |
| EC8 | 0.83 | 0.45 | 1.55 | SSW | 0.59 | 1.38 | 3.82 |
| EC9 | 0.63 | 2.31 | 3.15 | SD | 0.76 | -3.83 | 4.84 |
| EC10 | 0.68 | 1.32 | 2.45 | | | | |

The $R^2$, MBE, and RMSE values of $S_d$ were 0.81, 13.80 W·m$^{-2}$, and 25.35 W·m$^{-2}$, respectively.

The station validation results are shown in Table 4. The accuracy of SSW is low. Because cloudy conditions occurred briefly on July 27, few ground observations were obtained, and $S_d$ was signif- icantly overestimated. After removing these data, the $R^2$, MBE, and RMSE values of $S_d$ at SSW

were 0.87, 10.90 W·m$^{-2}$ and 21.13 W·m$^{-2}$, respectively.

**Table 4.** Station validation results of downward shortwave radiation

| station | $R^2$ | MBE (W·m$^{-2}$) | RMSE (W·m$^{-2}$) | station | $R^2$ | MBE (W·m$^{-2}$) | RMSE (W·m$^{-2}$) |
|---|---|---|---|---|---|---|---|
| EC1 | 0.97 | 25.23 | 27.73 | EC11 | 0.90 | 30.11 | 33.76 |
| EC2 | 0.84 | 28.29 | 33.57 | EC12 | 0.96 | 24.35 | 26.43 |
| EC3 | 0.97 | 17.56 | 19.25 | EC13 | 0.93 | 12.41 | 17.92 |
| EC4 | 0.98 | 6.07 | 9.34 | EC14 | 0.98 | 32.40 | 33.49 |
| EC5 | 0.98 | 10.60 | 12.29 | EC15 | 0.94 | 26.71 | 29.71 |
| EC6 | 0.93 | 27.68 | 30.71 | EC17 | 0.94 | -20.25 | 24.54 |
| EC7 | 0.89 | -17.69 | 27.59 | GB | 0.89 | 25.34 | 30.63 |
| EC8 | 0.83 | 15.63 | 25.50 | SSW | 0.63 | 18.51 | 34.93 |
| EC9 | 0.96 | -2.27 | 9.96 | SD | 0.98 | 5.70 | 13.82 |
| EC10 | 0.94 | -3.50 | 11.97 | | | | |

The $R^2$, MBE, and RMSE of the HRB $L_d$ were 0.73, 0.28 W·m$^{-2}$, and 21.24 W·m$^{-2}$, respectively.

As seen in Table 5, the accuracies at EC3, SD and SSW were low. The low accuracies at EC3 and

SD potentially resulted from (1) high humidity, which resulted in low at-nadir brightness tempera- tures and low retrieved $L_d$, or (2) instrument error, which occurred because the EC3 ground obser- vations were always greater than those of the other stations during the same period. Although SSW

was located in a desert, the ground-air temperature difference was large. The $L_d$ retrieval may have a large error because the models use surface temperature when estimating $L_d$ to approximate or substitute the near-surface temperature (Yu et al., 2013). The corrected error of our $L_d$ retrieving algorithm resulted from the ground-air temperature difference in non-vegetated areas. The inaccu- racy of the SSW LST may influence the $L_d$ results.

**Table 5.** Station validation results of downward longwave radiation

| station | $R^2$ | MBE (W·m$^{-2}$) | RMSE (W·m$^{-2}$) | station | $R^2$ | MBE (W·m$^{-2}$) | RMSE (W·m$^{-2}$) |
|---|---|---|---|---|---|---|---|

| | | | | | | | |
|---|---|---|---|---|---|---|---|
| EC1 | 0.85 | 4.16 | 17.21 | EC11 | 0.93 | -2.72 | 10.55 |
| EC2 | 0.88 | 0.11 | 14.23 | EC12 | 0.87 | -0.84 | 14.80 |
| EC3 | 0.91 | -35.65 | 37.88 | EC13 | 0.86 | -7.28 | 15.98 |
| EC4 | 0.88 | 3.36 | 16.38 | EC14 | 0.82 | 4.07 | 16.42 |
| EC5 | 0.88 | -0.79 | 15.02 | EC15 | 0.85 | 17.67 | 23.06 |
| EC6 | 0.84 | 2.55 | 15.43 | EC17 | 0.90 | -1.11 | 12.87 |
| EC7 | 0.75 | -5.90 | 19.72 | GB | 0.88 | 9.50 | 27.82 |
| EC8 | 0.80 | -1.35 | 17.49 | SSW | 0.85 | 25.33 | 34.50 |
| EC9 | 0.86 | 10.44 | 17.99 | SD | 0.85 | -26.54 | 34.08 |
| EC10 | 0.87 | 7.98 | 16.05 | | | | |

The $R^2$, MBE, and RMSE of the HRB $R_n$ were 0.70, -9.64 W·m$^{-2}$, and 42.77 W·m$^{-2}$, respec-
tively. The station $R_n$ validation results are shown in Table 6, which indicate that the accuracies of
EC4, EC7, EC17 and SSW were relatively low. According to the sensitivity analysis of Eq. (8), $L_d$
and $S_d$ are highly sensitive variables when calculating $R_n$, while the albedo, LSE and LST are not
as sensitive. Although LST was not a sensitive variable, the EC4's LST, MBE and RMSE reached -
9.87 K and 10.04 K because the land cover of EC4 was maize at the 300 m resolution. However,
the observation tower was in a built-up area, which potentially caused errors when estimating $R_n$.
The accuracies of the EC7 $S_d$ and $L_d$ were low on several days, and after removing these data,
MBE=-43.40 W·m$^{-2}$ and the RMSE=50.50 W·m$^{-2}$. EC17 was within an orchard, and the signal that
was received by the sensors at EC17 were affected by the complex vertical structure of the orchard
ecosystem. The information on substrate plants may be ignored, leading to albedo retrieval errors.
Although the albedo was not a sensitive variable, a 0.03 bias can lead to an $R_n$ error of approxi-
mately 20 W·m$^{-2}$ when the solar incoming radiation is large. As previously mentioned, it was briefly
cloudy on July 27, and after removing that data, the $R^2$, MBE, and RMSE values of the $R_n$ obtained
at SSW were 0.72, 8.20 W·m$^{-2}$, and 37.60 W·m$^{-2}$, respectively.

**Table 6.** Station net radiation validation results

| station | $R^2$ | MBE (W·m$^{-2}$) | RMSE (W·m$^{-2}$) | station | $R^2$ | MBE (W·m$^{-2}$) | RMSE (W·m$^{-2}$) |
|---|---|---|---|---|---|---|---|
| EC1 | 0.76 | -2.55 | 30.61 | EC11 | 0.86 | -15.13 | 28.05 |
| EC2 | 0.79 | 2.52 | 25.24 | EC12 | 0.90 | -8.46 | 19.38 |
| EC3 | 0.86 | -35.84 | 42.97 | EC13 | 0.88 | -25.73 | 32.34 |
| EC4 | 0.84 | 76.64 | 80.25 | EC14 | 0.90 | 4.23 | 18.18 |
| EC5 | 0.85 | -24.41 | 32.34 | EC15 | 0.84 | 8.33 | 23.01 |
| EC6 | 0.82 | 4.35 | 23.44 | EC17 | 0.89 | -62.62 | 68.11 |
| EC7 | 0.61 | -58.66 | 67.83 | GB | 0.77 | -10.40 | 38.86 |
| EC8 | 0.83 | -20.62 | 32.45 | SSW | 0.44 | 23.05 | 62.93 |
| EC9 | 0.87 | -29.60 | 36.27 | SD | 0.75 | 19.98 | 35.24 |
| EC10 | 0.83 | -24.35 | 33.51 | | | | |

The $R^2$, MBE, and RMSE of the G in the HRB were 0.57, 8.51 W·m$^{-2}$, and 29.73 W·m$^{-2}$, re-
spectively. The station $R_n$ validation results are shown in Table 7. For EC5, the soil temperature
and moisture were the same at different depths after July 19, which resulted in a surface G that was
equal to the G at a depth of 6 cm. The G below the surface was usually less than the G at the soil
surface; thus, the validation results of the G at EC5 indicate that G was overestimated. For SSW, the brief cloudy period decreased the observed soil surface temperature, which decreased the calcu- lated surface G. However, the remotely sensed G did not reflect this situation. In this case, the G

was overestimated because the $R_n$ was overestimated. After removing the data on July 27, the $R^2$,

MBE, and RMSE of the G at SSW were 0.17, 19.34 W·m$^{-2}$, and 33.30 W·m$^{-2}$, respectively.

**Table 7.** Station validation results of the soil heat flux

| station | $R^2$ | MBE (W·m$^{-2}$) | RMSE (W·m$^{-2}$) | station | $R^2$ | MBE (W·m$^{-2}$) | RMSE (W·m$^{-2}$) |
|---|---|---|---|---|---|---|---|
| EC1 | 0.50 | 19.73 | 31.53 | EC11 | 0.71 | 4.23 | 19.23 |
| EC2 | 0.24 | 20.78 | 28.72 | EC12 | 0.53 | 20.29 | 24.79 |
| EC3 | 0.03 | -1.15 | 36.28 | EC13 | 0.91 | -0.89 | 17.27 |
| EC4 | 0.45 | 18.50 | 22.29 | EC14 | 0.82 | -1.89 | 18.72 |
| EC5 | 0.38 | 41.87 | 60.19 | EC15 | 0.78 | 6.68 | 15.80 |
| EC6 | 0.83 | -5.91 | 14.57 | EC17 | 0.49 | 8.26 | 33.59 |
| EC7 | 0.28 | 7.50 | 24.65 | GB | 0.29 | -17.86 | 26.81 |
| EC8 | 0.68 | -5.73 | 20.15 | SSW | 0.01 | 30.41 | 51.87 |
| EC9 | 0.61 | 6.83 | 26.96 | SD | 0.71 | -4.79 | 13.71 |
| EC10 | 0.41 | 7.68 | 28.67 | | | | |

**4.2. Validation of heat fluxes by TSFA**

Fig. 6 provides the turbulent heat flux results calculated by TSFA on September 13, 2012. The spatial distribution of the turbulent heat flux is obvious. The H of buildings and uncultivated land, including the Gobi Desert, barren areas and other deserts, was high, in addition to the LEs of the water and agricultural areas in the oasis. The southern areas of the images show uncultivated barren land bordering the Qilian Mountains that resulted from snowmelt and the downward movement of water. In these areas, the groundwater levels are high and the soil moisture content is approximately

30% based on in situ measurements at a depth of 2 cm. Therefore, the LE is higher in the south than in the southeast desert, although both areas were classified as uncultivated land.

Studies have shown that validation methods that consider the source area are more appropriate for evaluating ET models than traditional validation methods based on a single pixel (Jia et al., 2012;

Song et al., 2012). In this study, a user-friendly tool presented by Neftel et al. (2008) and based on the Eulerian analytic flux footprint model proposed by Kormann and Meixner (2001) was used to calculate the footprints of the function parameters. The continuous footprint function was dispersed based on the relative weights of the pixels on which the source area fell.

[Figure]

**Figure 6.** Maps of the four energy components, (a) Rn, (b) G, (c) H and (d) LE, calculated by TSFA on September

13, 2012.

[Figure]

**Figure 7.** Scatter plot of the TSFA turbulent heat flux results

The footprint validation results of the TSFA turbulent heat fluxes are shown in Fig. 7 and Table

8. The $R^2$, MBE, and RMSE of the H were 0.61, 0.90 W·m$^{-2}$ and 50.99 W·m$^{-2}$, respectively, and the corresponding terms for the LE were 0.82, -20.54 W·m$^{-2}$ and 71.24 W·m$^{-2}$, respectively. Because the LE was calculated as a residual term, it was impacted by the $R_n$, surface G and H. The errors of all of these variables may contribute to the LE, which complicates the error source of the LE and is discussed in Sects. 4.3.2 and 4.4.

**Table 8.** In situ validation results of heat flux of TSFA

| | TSFA-H(W·m$^{-2}$) | | | TSFA-LE(W·m$^{-2}$) | | |
|------|-------|--------|-------|-------|--------|-------|
| date | $R^2$ | MBE | RMSE | $R^2$ | MBE | RMSE |
| 0619 | 0.39 | 44.73 | 66.38 | 0.69 | -44.15 | 80.60 |
| 0630 | 0.73 | 23.71 | 38.96 | 0.88 | -63.81 | 77.83 |
| 0708 | 0.55 | 32.70 | 58.72 | 0.85 | -43.02 | 72.32 |
| 0727 | 0.90 | -34.34 | 43.59 | 0.92 | 26.74 | 57.60 |
| 0803 | 0.80 | -4.77 | 18.92 | 0.78 | -4.58 | 47.86 |
| 0815 | 0.74 | -18.37 | 38.82 | 0.93 | 4.75 | 35.41 |
| 0822 | 0.40 | 31.64 | 66.21 | 0.65 | -44.44 | 93.81 |
| 0829 | 0.79 | 23.01 | 38.36 | 0.79 | -50.45 | 77.99 |
| 0902 | 0.21 | -45.10 | 74.81 | 0.54 | 24.39 | 69.31 |
| 0913 | 0.25 | -9.64 | 41.01 | 0.59 | -59.36 | 82.77 |
| 0914 | 0.31 | -34.11 | 50.88 | 0.47 | 27.99 | 67.50 |

As seen in Fig. 7, most of the H values are small because June, July, August and September constitute the growing season when ET greatly cools the air. The differential temperature between the land surface and air is small, leading to a low H. The points with large H values are influenced by uncultivated land. In our study area, bare soil, the Gobi Desert, and desert areas compose the uncultivated land. The points in the scatter plot with large H values represent desert, where the H values reach approximately 300 W·m$^{-2}$. Some points in the H scatter plot are less than 0 due to inversion from the "oasis effect" or irrigation. For example, HiWATER's soil moisture data show that irrigation occurred on August 22, 2012. Irrigation is the main source of water within the oasis and cools the land surface to temperatures below the air temperature. In addition, irrigation leads to errors in LST retrieval because it increases the atmospheric water vapor content, as discussed in Sect. 4.1. The model error is further analyzed in Sect. 4.4.

**4.3. Comparison between TSFA, TRFA and IPUS**

To verify whether the TSFA method can simulate the heterogeneities of the land surface, the TRFA and IPUS methods were compared for estimating the ET. These three methods were evaluated using (1) validation of TRFA and IPUS based on in situ measurements and (2) qualitative analysis based on the spatial distribution and scatter plots of the four energy balance components.

**4.3.1. Validation of TRFA and IPUS heat fluxes**

Table 9 provides the footprint in situ validation results of the H and LE calculated using the IPUS and TRFA methods. The $R^2$, MBE, and RMSE of the LE between TSFA and TRFA were 0.99, -7.81 W·m$^{-2}$ and 16.48 W·m$^{-2}$, respectively. And the $R^2$, MBE, and RMSE of the LE between TSFA and IPUS were 0.91, -4.10 W·m$^{-2}$ and 51.30W·m$^{-2}$, respectively. Comparing with validation results of TSFA in Table 8, the TSFA method had a better retrieval accuracy than the TRFA method, and TRFA method was better than the IPUS method on all days, because the MBE and RMSE of TSFA decreased and the $R^2$ of TSFA increased on most days. Table 9 shows that the improvement in the accuracy that resulted from temperature resampling (TRFA) when comparing with the IPUS method was relatively higher than the improvement observed from temperature sharpening (TSFA) when comparing with the TRFA method. Compared with the IPUS method, the TRFA results were similar to the TSFA results since the sub-pixel landscapes and sub-pixel variations of most variables were considered. Thus, TRFA could effectively decrease the scale error that resulted from heterogeneity because the VNIR data of satellite were fully used. However, the performance of the TRFA method is unstable. For example, on August 3 and August 29, the TRFA results were slightly worse than the IPUS results, and the TSFA results were obviously better. This difference occurred because the different sub-pixel landscape temperatures were treated as equal to the values estimated at the 300-m resolution. Thus, when the 300-m-resolution LST has large retrieving errors, the turbulent heat flux retrieving error may be amplified by the sub-pixel landscapes.

**Table 9.** In situ validation results of turbulent heat fluxes of IPUS and TRFA

| date | IPUS-H(W·m$^{-2}$) | | | IPUS-LE (W·m$^{-2}$) | | | TRFA-H (W·m$^{-2}$) | | | TRFA-LE (W·m$^{-2}$) | | |
|------|------|-------|-------|------|-------|--------|------|-------|-------|------|--------|-------|
| | $R^2$ | MBE | RMSE | $R^2$ | MBE | RMSE | $R^2$ | MBE | RMSE | $R^2$ | MBE | RMSE |
| 0619 | 0.32 | 48.53 | 71.70 | 0.66 | -47.68 | 86.02 | 0.39 | 52.28 | 70.98 | 0.65 | -46.71 | 85.93 |
| 0630 | 0.50 | 41.45 | 67.30 | 0.80 | -81.75 | 102.33 | 0.69 | 42.64 | 60.85 | 0.86 | -78.50 | 93.98 |
| 0708 | 0.34 | 44.17 | 77.45 | 0.63 | -66.75 | 118.63 | 0.44 | 54.20 | 76.00 | 0.82 | -63.82 | 89.11 |
| 0727 | 0.81 | -33.14 | 50.01 | 0.83 | 25.61 | 74.26 | 0.84 | -23.53 | 41.76 | 0.86 | 14.82 | 65.21 |
| 0803 | 0.84 | -5.23 | 33.50 | 0.74 | -3.98 | 60.49 | 0.80 | 7.76 | 37.51 | 0.76 | -18.23 | 62.71 |
| 0815 | 0.64 | -23.28 | 47.89 | 0.85 | 10.32 | 54.98 | 0.70 | -14.77 | 39.99 | 0.89 | 0.59 | 45.22 |
| 0822 | 0.31 | 41.50 | 74.81 | 0.61 | -53.60 | 102.12 | 0.40 | 40.63 | 69.94 | 0.65 | -54.17 | 98.97 |
| 0829 | 0.72 | 27.15 | 44.16 | 0.76 | -54.76 | 83.20 | 0.75 | 30.79 | 44.97 | 0.77 | -59.43 | 86.22 |
| 0902 | 0.28 | -52.44 | 83.25 | 0.51 | 32.89 | 76.48 | 0.21 | -45.77 | 75.84 | 0.52 | 24.37 | 71.69 |
| 0913 | 0.08 | -11.45 | 57.50 | 0.61 | -57.38 | 81.83 | 0.06 | -11.89 | 49.63 | 0.54 | -57.78 | 84.58 |
| 0914 | 0.12 | -36.52 | 67.38 | 0.28 | 19.46 | 89.30 | 0.03 | -34.34 | 64.85 | 0.38 | 25.41 | 75.96 |

Surface landscape inhomogeneity can be classified using two conditions: nonlinear vegetation density variations between sub-pixels (e.g., different types of vegetation mixed with each other or with bare soil) and coarse pixels containing different landscapes (e.g., vegetation or bare soil mixed with buildings or water). And landscapes variation always corresponding to inhomogeneity of surface variable. To evaluate the effects of TSFA, stations with a typical severe heterogeneous surface, such as EC4, a weak heterogeneous surface, such as EC11 and a typical pixel (called "TP" hereafter) at the boundary of the oasis and bare soil (sample 62, line 102 in the image of study area), and a uniform surface, such as EC15, were selected to analyze the temperature sharpening results.

EC4 is used as an example because its land cover and sub-pixel variation of temperature were complicated. Table 11 compares the turbulent heat fluxes calculated using the IPUS, TRFA and TSFA methods. Significant differences were observed between the TSFA and IPUS results and between the TRFA and IPUS results due to the heterogeneity of the surface. The LE calculated using the TSFA method was more consistent with in situ measurements than the LE calculated using the IPUS method because the MBE and RMSE decreased greatly, the $R^2$ increased, and the accuracy was improved by approximately 40 W·m$^{-2}$. However, the LE calculated by using the TRFA was more accurate than the LE calculated by using the TSFA, as discussed below.

The H calculated by using the TSFA method was more accurate than the H calculated by using the TRFA and IPUS methods. The accuracy of the results from the TRFA method was relatively close to the accuracy of the results from the TSFA method because the TRFA method also considers the effects of the heterogeneity of landscapes. In addition, the H values obtained from the TRFA method were always greater than those obtained from the TSFA method. Because the TSFA turbulent heat flux results are the same as the TRFA turbulent heat flux results for buildings and water bodies in our pixel ET algorithm, so the difference between TSFA and TRFA depends on the vegetation and bare soil. And the 300-m-resolution LST is larger than the LST of the sub-pixels, such as pixels containing vegetation or bare soil, for two reasons: (1) the coarse pixels contain buildings and result in a larger 300-m-resolution LST and (2) the LSTs were underestimated at EC4 (as shown in Table 3), which would underestimate the value of $\Delta \widehat{T}_{300}$ in Eq.(3) and, consequently, the sharpening temperature at 30 m and H. Because the LE was calculated as a residual item in the energy balance equation, the errors of the other three energy balance components would accumulate in the LE. At EC4, the $R_n$ was overestimated by approximately 80 W·m$^{-2}$, but the scale effect of $R_n$ was not obvious, as discussed in Sect. 4.3.1, and the G was overestimated by approximately 20 W·m$^{-2}$. These results would lead to low accuracy of the available energy and overestimate the error by 60 W·m$^{-2}$. As TRFA overestimates H, the underestimation of H in TSFA would result in larger overestimation of LE than TRFA. Consequently, the LE calculated by using the TSFA method is less accurate than the LE calculated by using the TRFA method.

**Table 10.** Comparison of the turbulent heat flux results at EC4

| EC4 | H(W·m$^{-2}$) | | | | LE(W·m$^{-2}$) | | | |
|---|---|---|---|---|---|---|---|---|
| Date | EC | IPUS | TRFA | TSFA | EC | IPUS | TRFA | TSFA |
| 0619 | 150.65 | 105.86 | 154.71 | 142.13 | 278.55 | 402.60 | 344.05 | 357.79 |
| 0630 | 138.32 | 99.91 | 153.53 | 126.88 | 341.98 | 419.83 | 358.12 | 386.07 |
| 0708 | 117.04 | 63.47 | 131.79 | 112.16 | 361.16 | 502.60 | 424.85 | 444.01 |
| 0727 | 136.41 | 4.87 | 85.99 | 72.33 | 306.53 | 543.48 | 452.01 | 467.96 |
| 0803 | 68.97 | 36.51 | 111.73 | 74.76 | 389.63 | 498.21 | 414.67 | 454.23 |
| 0815 | 104.60 | 12.69 | 88.26 | 82.56 | 357.34 | 522.31 | 436.43 | 441.95 |
| 0822 | 125.34 | 85.93 | 120.68 | 93.18 | 318.08 | 415.15 | 370.76 | 400.99 |
| 0829 | 82.93 | 73.06 | 103.84 | 74.76 | 317.68 | 362.04 | 322.77 | 355.16 |
| 0902 | 162.05 | 93.74 | 144.49 | 132.60 | 280.41 | 375.42 | 315.16 | 326.29 |
| 0913 | 119.42 | 151.44 | 157.07 | 130.85 | 263.18 | 234.93 | 222.62 | 249.59 |
| 0914 | 110.02 | 88.24 | 128.37 | 99.33 | 262.33 | 333.82 | 285.04 | 314.91 |

units: W·m$^{-2}$

| | IPUS | | | TRFA | | | TSFA | | |
|---|---|---|---|---|---|---|---|---|---|
| Variable | $R^2$ | MBE | RMSE | $R^2$ | MBE | RMSE | $R^2$ | MBE | RMSE |
| EC4-H | 0.11 | -44.65 | 61.73 | 0.25 | 5.88 | 26.33 | 0.51 | -16.93 | 26.54 |
| EC4-LE | 0.49 | 99.21 | 119.55 | 0.56 | 42.69 | 62.40 | 0.60 | 63.92 | 76.78 |

Fig. 8 shows that the classes and temperatures of $10\times10$ sub-pixels at 30 m correspond to the pixels with a resolution of 300 m at the EC tower. In the IPUS upscaling scheme, the 300-m pixels included buildings and maize and vegetable crops at the 30-m resolution and were identified as maize. The canopy height gap between maize and vegetables was large during our study period, resulting in the overestimation of the canopy height. For more details see the sensitivity and error analysis in Sect. 4.4. However, because buildings corresponded with $H = 0.6R_n$ in this paper, ignoring the contributions of buildings would result in the underestimation of H. Fig. 8(a) shows the temperature-sharpening results for the EC4 pixel on August 29. The temperature achieved at a resolution of 300 m was 303.49 K. Compared with the in situ measurement of 313.24 K, the temperature at a resolution of 300 m was underestimated. Even when substituting the in situ temperature into the ET model, the value of H reached 399.60 W·m$^{-2}$ and the LE became 0 W·m$^{-2}$. When substituting the in situ temperature in the TRFA method, H was 396.49 W·m$^{-2}$ and LE was 18.7 W·m$^{-2}$, indicating that the LE was underestimated and the H was overestimated with large errors. After processing by temperature sharpening, the distribution of the temperature at the 30-m resolution agreed with the classification. Temperature sharpening improved the description of heterogeneity based on the thermodynamic-driven force of the turbulent heat flux. These results apply to the ET

model with the classification map and high-resolution variables and correspond with more accurate sensible heat flux estimations.

[Figure]

**Figure 8.** Distribution of classes and temperatures over the extreme heterogeneous surface (a) EC4, homogeneous surface (b) EC15, weak heterogeneous surface (c) EC11 and (d) a typical pixel on August 29, 2012.

The land surface of EC15 was uniform and comprised of pure pixels covered by maize. The temperature distribution at the 30-m resolution was as homogeneous as the land cover, and the var- iation range of the surface temperature was small (approximately 1.6 K). Table 11 shows the in situ validation results of EC15, for which the overall accuracy is not high due to the low LST retrieval accuracy on July 8, which is discussed in Sect. 4.4.1. For the homogeneous surface, the gaps be- tween IPUS, TRFA and TSFA were not large (within 10 W·m$^{-2}$), and the accuracy did not improve (MBE and RMSE did not have obvious variations). Statistically sharpening the temperature may increase the uncertainty of the model results for a homogeneous surface; however, this influence could be omitted.

**Table 11.** Comparison of the turbulent heat fluxes results at EC15

| EC15 | H (W·m$^{-2}$) | | | | LE (W·m$^{-2}$) | | | |
|------|------|------|------|------|------|------|------|------|
| Date | EC | IPUS | TRFA | TSFA | EC | IPUS | TRFA | TSFA |
| 0619 | 92.55 | 106.60 | 109.25 | 99.81 | 419.47 | 427.19 | 419.99 | 429.98 |
| 0630 | 42.37 | 43.99 | 45.51 | 44.67 | 551.73 | 527.12 | 525.17 | 526.09 |
| 0708 | 18.34 | 217.53 | 235.48 | 209.90 | 620.95 | 425.71 | 397.49 | 424.86 |
| 0727 | 27.68 | 21.22 | 31.11 | 24.30 | 597.76 | 589.58 | 579.43 | 586.47 |
| 0803 | 2.33 | 33.32 | -0.07 | 0.01 | 592.37 | 565.20 | 601.33 | 601.33 |
| 0815 | 48.81 | 32.31 | 46.28 | 44.62 | 553.74 | 561.92 | 547.48 | 549.11 |
| 0822 | 54.59 | 154.34 | 151.77 | 158.60 | 473.68 | 408.37 | 410.80 | 405.07 |
| 0829 | 9.80 | 94.97 | 95.01 | 90.91 | 473.54 | 399.25 | 398.52 | 402.93 |
| 0913 | 176.96 | 265.62 | 209.65 | 257.81 | 307.72 | 165.40 | 221.68 | 173.58 |
| 0914 | 188.34 | 198.15 | 197.04 | 196.60 | 274.98 | 275.07 | 276.05 | 276.56 |

units: W·m$^{-2}$

| | IPUS | | | TRFA | | | TSFA | | |
|------|------|------|------|------|------|------|------|------|------|
| Variable | $R^2$ | MBE | RMSE | $R^2$ | MBE | RMSE | $R^2$ | MBE | RMSE |
| EC15-H | 0.40 | 40.64 | 74.64 | 0.33 | 45.93 | 80.81 | 0.40 | 40.36 | 72.88 |
| EC15-LE | 0.74 | -52.11 | 83.48 | 0.71 | -48.80 | 82.51 | 0.74 | -49.00 | 81.94 |

The weak heterogeneous land surface EC11 contained barley, maize and vegetables in a coarse pixel with a fractional area of 48:41:1 and was classified as barley at the 300-m resolution. The distributions of the classes and temperatures are shown in Fig. 8(c), and the pixel belongs to the first conditions of heterogeneity (nonlinear vegetation density variation between sub-pixels) that are classified in the introduction. Table 12 shows the in situ validation results of EC11, for which the improvements in the accuracies of H and LE by temperature resampling or sharpening were not as obvious as the improvements at EC4, which contained total different landscapes (the other inhomo- geneous scenario in introduction).

Theoretically, the LE pixel values from the TSFA and TRFA methods at EC11 should be smaller than the IPUS values in the energy balance system. The height of maize (range 0.3 ~ 2 m)

was usually higher than the height of barley (range 0.9 ~ 1.1 m) in the study area from June to

August. Taller vegetation resulted in greater surface roughness and smaller aerodynamic resistance, which led to larger H values, smaller LE values, and vice versa (e.g., vegetables with a canopy height of 0.2 m). When using the TSFA and TRFA methods, patch landscapes consisting of different crops, such as maize and vegetables, were considered. Thus, the LE was smaller than the IPUS LE.

On June 19, the canopy height of maize was 0.74 m, which was lower than the canopy height of barley (1 m) and indicated that the H values resulting from the TRFA and TSFA methods were less than the H resulting from the IPUS method. Because our validation method considered the influence of source area, the in situ turbulent heat flux validation results included the effects of neighboring pixels (i.e., on August 3, the turbulent heat flux values of the pixel corresponding with the location of EC11 was only weighted 37% in the source area).

The differences between the TSFA and TRFA methods was small and resulted from the LST differences between the 30-m resolution temperature sharpening results and the LST retrieved at the 300-m resolution and were not evident at EC11. For example, on August 29, the temperature range was 1.4 K, as shown in Fig. 8(c). This temperature was even less than the temperature range at EC15 because the observation system at EC15 was a superstation with a 40-m tall tower that may cause a large shadow and a large temperature range. Hence the temperature sharpening effect is not obvious after aggregating flux at the 300-m resolution under dense vegetation canopies. However, temperature sharpening can still decrease the heterogeneity that results from thermal dynamics.

The excess errors resulted from the relatively low LST accuracy, with $R^2$, MBE, and RMSE values of 0.42, 1.59 K and 2.98 K, respectively. On August 29, the temperature at a resolution of 300 m was 301.6 K, and the observed temperature of the ground was 300.20 K. The LST at the 300-m resolution was slightly overestimated. When the in situ temperature was substituted into the IPUS algorithm, the value of H decreased to 16.06 W·m$^{-2}$ and the LE became 467.43 W·m$^{-2}$. When substituting the in situ temperature in the TRFA scheme, the value of H was 22.43 W·m$^{-2}$ and the LE was 461.58 W·m$^{-2}$, which were more similar to the ground observations.

**Table 12.** Comparison of the turbulent heat flux results at EC11

| EC11 | H(W·m$^{-2}$) | | | | LE(W·m$^{-2}$) | | | |
|---|---|---|---|---|---|---|---|---|
| Date | EC | IPUS | TRFA | TSFA | EC | IPUS | TRFA | TSFA |
| 0619 | 33.94 | 173.69 | 158.12 | 158.18 | 531.46 | 391.60 | 407.42 | 407.40 |
| 0630 | 25.03 | 3.29 | 23.12 | 21.37 | 635.22 | 586.37 | 566.48 | 568.28 |
| 0708 | 32.29 | 68.17 | 97.16 | 96.13 | 601.98 | 567.73 | 538.77 | 539.81 |
| 0727 | 21.42 | -1.17 | -1.58 | -3.77 | 587.70 | 618.80 | 619.19 | 621.46 |
| 0803 | 7.01 | 24.85 | 20.34 | 19.52 | 614.28 | 575.03 | 585.29 | 586.16 |
| 0815 | 38.94 | 12.51 | 15.52 | 16.02 | 567.07 | 584.31 | 581.31 | 580.82 |
| 0822 | 69.25 | 73.45 | 83.11 | 84.38 | 516.07 | 483.23 | 473.60 | 472.40 |
| 0829 | 29.77 | 48.21 | 60.9 | 60.81 | 473.22 | 427.92 | 415.32 | 415.45 |
| 0902 | 193.97 | 154.58 | 197.01 | 197.49 | 306.62 | 361.96 | 319.54 | 319.03 |
| 0913 | 288.37 | 168.42 | 176.4 | 177.71 | 160.29 | 216.53 | 208.49 | 207.19 |
| 0914 | 240.33 | 268.91 | 256.29 | 256.40 | 199.52 | 156.00 | 168.63 | 168.55 |

units: W·m$^{-2}$

| | IPUS | | | TRFA | | | TSFA | | |
|---|---|---|---|---|---|---|---|---|---|
| Variable | $R^2$ | MBE | RMSE | $R^2$ | MBE | RMSE | $R^2$ | MBE | RMSE |
| EC11-H | 0.61 | -1.07 | 61.31 | 0.57 | -0.36 | 63.24 | 0.67 | -0.21 | 55.50 |
| EC11-LE | 0.88 | -19.83 | 63.16 | 0.89 | -18.12 | 60.02 | 0.90 | -21.29 | 58.11 |

Another typical pixel located at the boundary of the bare soil and the oasis with no flux measurements was used to evaluate the correction effects of landscapes and temperature sharpening. The land surface of TP contained maize, vegetables and bare soil at a fraction of 35:31:34. Table 13 shows that when neither the heterogeneity of the landscape nor the LST are considered, the relative error of LE could reach 180 W·m$^{-2}$. In addition, if only the LST heterogeneity is not considered, the LE relative error could reach 48 W·m$^{-2}$. This result also reveals that the influences of landscape inhomogeneity are greater than the influences of inhomogeneity on the LST in mixed pixels.

| | **H (W·m⁻²)** | | | **LE (W·m⁻²)** | | |
|---|---|---|---|---|---|---|

**Table 13.** Comparison of the turbulent heat flux results at TP

| Date | IPUS | TRFA | TSFA | IPUS | TRFA | TSFA |
|---|---|---|---|---|---|---|
| 0619 | 186.31 | 149.73 | 143.98 | 321.04 | 358.22 | 364.79 |
| 0630 | 383.65 | 191.59 | 158.79 | 67.03 | 259.36 | 292.89 |
| 0708 | 498.36 | 240.20 | 204.18 | 0.29 | 259.25 | 293.41 |
| 0727 | 276.79 | 136.06 | 84.01 | 206.52 | 347.64 | 402.23 |
| 0803 | 214.14 | 75.45 | 53.72 | 252.37 | 392.08 | 416.41 |
| 0815 | 214.14 | 98.24 | 72.05 | 252.37 | 368.64 | 393.68 |
| 0822 | 436.48 | 369.28 | 276.70 | 0.00 | 67.79 | 162.80 |
| 0829 | 235.29 | 117.16 | 67.21 | 183.62 | 302.41 | 356.75 |
| 0902 | 423.61 | 212.15 | 180.92 | 0.00 | 211.77 | 241.36 |
| 0913 | 338.00 | 285.04 | 216.26 | 0.00 | 53.62 | 122.58 |
| 0914 | 270.44 | 148.20 | 100.19 | 115.19 | 238.43 | 286.51 |

units: W·m⁻²

| | **IPUS** | | | **TRFA** | | |
|---|---|---|---|---|---|---|
| Variable | $R^2$ | MBE | RMSE | $R^2$ | MBE | RMSE |
| TP-H | 0.62 | 174.47 | 185.49 | 0.95 | 42.28 | 48.01 |
| TP-LE | 0.71 | -175.91 | 186.63 | 0.97 | -43.11 | 49.04 |

**4.3.2. Comparison of TRFA and IPUS methods**

Using September 13 as an example, the spatial distributions of the four components of the energy balance calculated by IPUS and TRFA are shown in Fig. 9 and Fig. 10, respectively. TSFA minus IPUS and TSFA minus TRFA, which show the spatial distributions of the heterogeneity effect, are shown in Fig. 11. Scatterplots of TSFA versus IPUS and TRFA are shown in Fig. 12.

Comparing Fig. 6 with Fig. 9, the spatial distribution of the fluxes greatly changes, except for $R_n$. The TSFA results are synoptically smoother than the IPUS results because the land types and temperature distributions in mixed pixels that cannot be considered in IPUS appear in TSFA. For example, the boundary between the oasis and uncultivated land becomes a belt of intermediate G, H and LE because mixed pixels include uncultivated land and vegetation. However, mixed pixels are classified as the dominant land use type in the parameterization process of IPUS. This result overlooks the contributions of heat flux from complex land use types and overestimates or underestimates the heat flux by approximately 50 W·m⁻². However, TSFA can integrate the effects of these land areas and reveals the relative actual surface conditions. The results of this analysis vary less dramatically than the results obtained using IPUS, as shown in the figures. The results are similar in the oasis.

Based on the overviews presented in Fig. 6 and Fig. 10, the TRFA and TSFA methods are similar. Because the TRFA method considers the sub-pixel landscapes that could be a significant source of error in ET models, the difference between the TSFA and TRFA methods result from the differences between the sharpened and retrieved LST for the sub-pixels at the 300 m resolution. In addition, the bias between the TSFA and TRFA is not as obvious as the bias between the TSFA and IPUS methods, as shown in Fig. 11(c)(d)(e)(f). Furthermore, Fig. 11(f) shows that the LEs calculated by using the TSFA method for most oasis areas were slightly greater than the LEs calculated by using the TRFA method, which were approximately 20 W·m⁻².

The quadrangular with a relatively unstable bias shown in Fig. 11(a) is caused by the $L_d$ that was calculated from the MOD05 water vapor product which exists quadrangular even after preprocessing the instrument malfunction gap. From Fig. 11, the differences of the four energy components of the pure pixels between these three methods are within 5 W·m$^{-2}$, and the mixed pixels have different ranges.

[Figure]

**Figure 9.** Maps of the four energy components, (a) $R_n$, (b) G, (c) H and (d) LE, calculated using the IPUS method on September 13, 2012.

[Figure]

**Figure 10.** Maps of the four energy components, (a) $R_n$, (b) G, (c) H and (d) LE, calculated using the TRFA method
on September 13, 2012.

[Figure]

**Figure 11.** Maps of the bias of the energy balance components calculated using the TSFA method minus the IPUS

method: (a) $R_n$, (b) G, (c) H, (d) LE, TSFA minus TRFA: (e) H and (f) LE.

[Figure]

**Figure 12.** Scatter plots between the TSFA and IPUS results: (a) $R_n$, (b) G, (c) H and (d) LE; TSFA and TRFA (e)

$R_n$, (f) G, (g) H and (h) LE. MBD and RMSD are the mean bias deviation and root mean square deviation between the TSFA and IPUS results, respectively.

Fig. 12 shows the scatter plots between the results from the TSFA method and the other two methods for all four energy balance components in the image. Fig. 11(a)(e) shows that $R_n$ does not vary much between the three methods because the scatter is centralized around the 1:1 line. However, regarding the spatial scale effect, the differences in $G$, $H$ and $LE$ calculated by using the IPUS and TSFA methods are obvious: the scatter plots are dispersed at the mixed pixels, and the differences between the TRFA and TSFA results are relatively smaller. When using the TSFA method, the temperature sharpening results can be divided into results that are higher and lower than the LST retrieved at 300 m. Compared with the LST retrieved at 300 m when using the TRFA method, a higher LST would be counterbalanced by a lower LST when calculating H. Thus, the heterogeneous effect of temperature is neutralized in this case. This observation potentially resulted from the temperature sharpening algorithms because they tend to overestimate the sub-pixel LST for cooler landscapes and underestimate the sub-pixel LST for warmer areas in the image (Kustas et al., 2003).

However, $LE$ is calculated as a residual; thus, the difference of $LE$ resulted from the $G$ and $H$. When the 300 m mixed pixels contain various types of land, they may be categorized as one type of land because of the coarse resolution of the IPUS results and because a single temperature value is used to evaluate the thermal dynamic effects when using the TRFA method. Pixels with highly different $G$, $H$ and $LE$ values are mainly distributed near the mixed pixels, as shown in Fig. 10. An explanation for these deviations is provided below.

The parameterization of $G$ and $H$ is based on the land cover type. For example, for buildings, $G = 0.4R_n$(Kato and Yamaguchi, 2005) (which is usually greater than the $G$ of vegetation and bare soil deduced from Eq.(9)) and $H = 0.6R_n$, and for water, $G = 0.226R_n$ and $LE = R_n - G$. From the land cover map shown in Fig. 4, four major classes exist in the study area, buildings with a high $H$, uncultivated land with a relatively high $H$, cropland with a relatively low $H$, and water with $H = 0$.

(1) If a pixel contains cropland and buildings and is categorized as cropland the building area within the pixel is ignored when using the IPUS method. In this case, $G$ and $H$ are underestimated and $LE$ is overestimated. In addition, after considering the landscapes by using the TRFA method, the $LE$ is underestimated and $H$ is overestimated because the pixels contain buildings that are still reflected indistinctly by LST at 300 m because the detailed temperature heterogeneity cannot be represented by the TRFA method. These points are shown in green in Fig. 11. However, if the pixel is categorized as built-up, the building area within a pixel is exaggerated, which causes $G$ and $H$ to be overestimated and $LE$ to be underestimated when using the IPUS method. This situation is similar to the points shown in green for the TRFA results and is shown by red points in Fig. 11.

(2) At the margin of the oasis and uncultivated land, the mixed pixels are divided into cropland, the $LE$ is overestimated, $G$ and $H$ are underestimated in the IPUS method, and vice versa. The $LE$ is also overestimated in the pixels containing water and other types of land cover (generally bare soil in our study area). These pixels are categorized as water and are shown as blue points in Fig. 11. Some of the blue $LE$ points calculated by using the TSFA method are slightly smaller than those calculated by using the TRFA method for pixels containing vegetation, and the temperature of vegetation is lower than the temperature of water bodies at noon in our study area.

(3) In mixed pixels that contain various crops, such as maize and vegetables, the $LE$ is underestimated if the area of maize within the pixel is overestimated because the canopy height of the maize would be taller than that of vegetables, which would result in the overestimation of $H$ when using the IPUS and TRFA methods. In addition, $G$ depends on the FVC of the crops when using the IPUS method, and is nearly the same as the values of $G$ obtained when using the TRFA and

TSFA methods because it depends on $R_n$.

At the study area scale, we compared TRFA and IPUS to quantify the ability of the TSFA
method to simulate the heterogeneities of the land surface on September 13 (see Table 14). For pure
pixels, the LE biases among the IPUS, TRFA and TSFA methods were small. In mixed pixels, the
LE bias between the TSFA and IPUS methods varied from 35.36 to 65.66 W·m$^{-2}$, and the bias be-
tween the TSFA and TRFA methods varied from 4.41 to 22.53 W·m$^{-2}$. More class types in mixed
pixels correspond to larger biases. Table 15 shows the bias of the mixed pixels that contain buildings
and bare soil between the three methods. For mixed pixels with buildings, the IPUS and TRFA
methods usually underestimated the LE, with a large bias compared with the TSFA method. For
mixed pixels without buildings and bare soil, the bias between TRFA (or IPUS) and TSFA was
relatively small, which indicates that the landscape and temperature inhomogeneity are accounted
for by the TSFA method. The aforementioned analyses demonstrate that the TSFA method can con-
sider the heterogeneous effects of mixed pixels.

**Table 14.** Comparison of the latent heat flux in pixels containing different numbers of class types

| Number of class types in pixels | IPUS (W·m$^{-2}$) | | | TRFA (W·m$^{-2}$) | | | Pixel number |
|---|---|---|---|---|---|---|---|
| | $R^2$ | MBD | RMSD | $R^2$ | MBD | RMSD | |
| 1 | 1.00 | 0.21 | 0.21 | 1.00 | 0.05 | 0.61 | 11,398 |
| 2 | 0.85 | -7.18 | 35.36 | 1.00 | -0.35 | 4.41 | 8212 |
| 3 | 0.66 | -2.32 | 52.55 | 0.98 | -7.33 | 12.56 | 4762 |
| 4 | 0.49 | 1.88 | 65.66 | 0.96 | -11.56 | 16.55 | 2824 |
| 5 | 0.98 | -30.92 | 62.69 | 0.96 | -16.90 | 22.53 | 4 |

Notes: Number of class types in mixed pixels means the number of classification types that were contained in
the pixels. For example, 1 represents the pure pixels, 2 represents mixed pixels containing two land use types, etc.
MBD and RMSD are the mean bias deviation and root mean square deviation, respectively, between the TSFA
results and the TRFA and IPUS results.

**Table 15.** Comparison of the latent heat fluxes of typical mixed pixels

| Types of mixed pixels | IPUS (W·m$^{-2}$) | | | TRFA (W·m$^{-2}$) | | | Pixel number |
|---|---|---|---|---|---|---|---|
| | $R^2$ | MBD | RMSD | $R^2$ | MBD | RMSD | |
| mixed pixels contain buildings | 0.58 | -1.02 | 61.94 | 0.97 | -9.64 | 14.66 | 4918 |
| mixed pixels do not contain buildings | 0.81 | -5.49 | 39.21 | 0.99 | -2.12 | 7.60 | 10,884 |
| mixed pixels contain bare soil | 0.73 | -1.52 | 49.04 | 0.98 | -5.96 | 11.86 | 9049 |
| mixed pixels do not contain bare soil | 0.65 | -7.55 | 45.28 | 0.98 | -2.46 | 7.83 | 6753 |

Considering the landscapes and inhomogeneous distribution of LST, the TSFA method ensures
that none of the end members (30 m pixel) are ignored or exaggerated. Thus, the distribution of LE
calculated using the TSFA method is smoother and more rational than the distributions of LE cal-
culated using the other methods. At the regional scale, the TSFA method describes the heterogeneity
of the land surface more precisely. And how much the estimation accuracy can be improved is dis-
cussed in the following sections.

**4.4. Error analysis**

Since LE is calculated as a residual term in the energy balance equations, the sensitivity of H
was analyzed at first. Land surface variables (including LST, LAI, canopy height, and FVC) and
meteorological variables (including wind speed, air temperature, air pressure and relative humidity)
are needed to estimate H in this paper. To locate the error source when retrieving H, a sensitivity analysis was performed by adding errors at each 10% step (except LST). Fig. 13 presents the sensi- tivity analysis results: LST = 303.9 K (ranging from 298.4~309.4 K with a step size of 0.5 K),

LAI=1.4 (ranging 0.14~2.66 with a step size of 0.14), canopy height equals 1 m (ranging 0.1~1.9 m with a step size of 0.1 m), FVC=0.5, wind speed u=2.48 m·s$^{-1}$, air temperature Ta=297.9 K, air pressure = 97.2 kPa, and RH=40.29%. In addition, the land use type is maize, and the reference H

is 230.2 W·m$^{-2}$.

[Figure]

**Figure 13.** Sensitivity analysis of the surface variables for sensible heat flux

The air pressure is stable over a short period and has little effect on the ET results. Although

"excess resistance" was calculated from the friction velocity, the meteorological data were provided by ground observations; thus, the meteorological data are relatively accurate. As shown in Fig. 13,

LAI, canopy height and LST are sensitive variables.

The parameterization of the momentum roughness length indicates that LAI is sensitive to H, with decreasing sensitivity when the LAI is greater than 1. When the LAI is less than 1, the momen- tum roughness length increases as the LAI increases and the H and turbulent exchange are enhanced.

However, when the LAI is greater than 1, the plant canopy could be regarded as a continuum that is not sensitive to H. Because our study area is dominated by agriculture and the study period was from July to September, the crops in the HRB middle stream grew quickly, so the LAI was generally greater than 1. Thus, LST and canopy height are the main sources of error.

**4.4.1. The error of LST**

As shown in Fig. 13 using monitoring data, a 1 K LST bias would result in 21% error of H, about 48.3 W·m$^{-2}$. However, the sensitivity of the LST is unstable and depends on the strength of the turbulence. The strength of the turbulence determines the mass and energy transport and the resistance of heat transfer, which influences the sensitivity of the LST. A weaker turbulence corre- sponds to a weaker LST sensitivity and vice versa.

The influence of LST was analyzed based on the sensitivity analysis and LE results. We chose homogeneous stations to analyze the LST error so that other errors could be ignored. These results are shown in Table 16. The LE results obtained from the observed LST are consistent with the in situ observations but have less bias. The LE was overestimated when the LST was underestimated and vice versa. Because the magnitude of LE was greater than H, the relative error of LE was less than the relative error of H. However, 1 K of LST bias would result in an average LE error of 30

W·m$^{-2}$, which is consistent with the sensitivity analysis of H shown in Fig. 13. Specifically, 1 K of

LST bias would result in LE biases of 8.7 W·m$^{-2}$ (in desert, SSW) to 84.4 W·m$^{-2}$ (in oasis, EC8), which may prove that the sensitivity of LST is unstable.

**Table 16.** Results of the LST error analyses at the homogeneous stations

| Station | Date | retrieved LST (K) | observed LST (K) | LST bias (K) | EC-LE (W·m⁻²) | LE from retrieved LST (W·m⁻²) | LE from observed LST (W·m⁻²) | LE relative error (%) | H relative error (%) |
|---------|------|------|------|------|------|------|------|------|------|
| EC8 | 0619 | 304.92 | 301.74 | 3.18 | 415.89 | 321.80 | 399.78 | -22.62 | 68.58 |
| EC7 | 0630 | 302.5 | 299.35 | 3.15 | 611.22 | 453.59 | 557.97 | -25.79 | 886.08 |
| EC10 | 0708 | 303.58 | 300.5 | 3.08 | 617.83 | 504.44 | 549.53 | -18.35 | 390.24 |
| EC15 | 0708 | 303.55 | 300.13 | 3.42 | 620.95 | 425.71 | 603.73 | -31.44 | 450.57 |
| EC7 | 0727 | 298.87 | 300.55 | -1.68 | 577.59 | 643.56 | 566.62 | 11.42 | -132.47 |
| SSW | 0727 | 307.86 | 316.82 | -8.96 | 119.35 | 238.07 | 78.43 | 99.48 | -60.36 |
| EC2 | 0822 | 299.79 | 298.05 | 1.74 | 501.12 | 411.43 | 486.28 | -17.90 | 67.20 |
| EC8 | 0822 | 299.58 | 297.77 | 1.81 | 543.56 | 416.23 | 467.42 | -23.42 | 88.59 |
| EC10 | 0822 | 301.61 | 298.04 | 3.57 | 503.82 | 398.82 | 513.67 | -20.84 | 138.61 |
| EC15 | 0822 | 300.59 | 297.69 | 2.9 | 473.68 | 408.37 | 495.49 | -13.79 | 129.60 |
| EC8 | 0829 | 301.54 | 300.44 | 1.1 | 514.31 | 402.93 | 428.78 | -21.66 | 63.91 |
| EC15 | 0829 | 301.41 | 299.84 | 1.57 | 473.54 | 399.25 | 459.66 | -15.69 | 182.34 |
| SSW | 0902 | 304.9 | 303.42 | 1.48 | 226.88 | 127.96 | 149.83 | -43.60 | 11.36 |

Notes: "LST bias" is calculated as the retrieved LST minus the observed LST; "EC-LE" is the in situ latent heat flux; "LE relative error" is the relative error between the retrieved and observed LST and is expressed as ((LE from retrieved LST)-(LE from observed LST))/(LE from observed LST)×100%, "H relative error" is calculated in the same way.

**4.4.2. The error of canopy height**

In this paper, canopy height was obtained from a phenophase and classification map. Thus, the accuracy of the canopy height was mainly dependent on the classification accuracy and plant growth state. Even within the same region, the canopy height of a crop can differ due to differences in seeding times and soil attributes, such as soil moisture and fertilization.

The land use at EC17 was orchard. However, in our land classification map, the land use at EC17 was other crops, which includes vegetables and orchards. Thus, it was difficult to set the canopy height. In our study area, most of the other crops were vegetables (canopy height of 0.2 m), and the height of the orchard was approximately 4 m; thus, a value of 0.2 m would overestimate the LE. The LE results with incorrect canopy heights and correct orchard canopy heights at EC17 are shown in Table 17. The days of large LST bias were removed, and the bias between the model and ground observations decreased. The excess errors were caused by errors in the LST and other land use types, such as buildings and maize in the mixed pixels.

**Table 17.** Results of the canopy height error analyses at EC17

| Date | EC-LE (W·m⁻²) | LE from incorrect canopy height (W·m⁻²) | LE from correct canopy height (W·m⁻²) | LE relative error (%) |
|------|------|------|------|------|
| 20120815 | 499.62 | 562.06 | 521.83 | 7.71 |
| 20120822 | 366.27 | 519.01 | 396.54 | 30.88 |
| 20120902 | 377.96 | 471.68 | 336.52 | 40.16 |
| 20120914 | 465.38 | 352.78 | 258.07 | 36.70 |

Except for the error source discussed before, the following sources of error were unavoidable:
(1) Although the remotely sensed turbulent heat flux is instantaneous, the EC data are averaged over time. Thus, the time scales do not match in the validation.

(2) The calibration coefficient of HJ-1B satellite's CCD and IRS drifts because of the aging instruments.

(3) Geometric correction causes half-pixel bias equal to or less than the deviation of the artificially subjective interpretation.

A one-source model and simplified parameterization schemes for determining surface roughness lengths and heat transfer coefficients were used in this paper. The one-source model combines soil evaporation and plant transpiration and assumes that SPAC is a one-source continuum for calculating ET. This assumption is reasonable when the surface is densely covered by vegetation but relies on the accuracy of the difference between the LST and air temperature, as previously mentioned. When a one-source model is applied to an area covered by sparse vegetation, such a semiarid or arid areas, this assumption is irrational.

**5. Discussion**

As mentioned in the results and analysis, the TSFA method describes the surface heterogeneity more clearly than the IPUS and TRFA methods. The IPUS method aggregates the land surface parameters achieved by CCDs from 30 m to 300 m, which results in the loss of surface information and leads to the scale effect. Although the TRFA method uses VNIR information and partially decreases the heterogeneity caused by landscape and VNIR variables, it treats the pivotal variable LST as homogeneous within mixed pixels, which results in considerable error. In summary, the superiority of the TSFA method is described as follows:

(1) The temperature sharpening algorithm in TSFA uses the NDVI at 30 m to monitor the LST at 30 m and is capable of decreasing the influences of the heterogeneity of the LST, which agrees with previous research results (Kustas et al., 2003; Bayala and Rivas, 2014; Mukherjee et al., 2014). As analyzed in Sect. 4.3, the ignorance of the heterogeneity of LST in mixed pixels is irrational and causes errors when estimating ET.

(2) In the one-source energy balance model, different landscapes used different parameterization schemes. In the IPUS method, a single land cover type is assigned to a mixed pixel, which results in a large error. However, the TSFA method is used to calculate the surface flux at 30 m and is aggregated to 300 m using the area-weighting method, which considers all of the sub-pixel landscapes and improves the retrieval accuracy.

Some problems exist in the temperature sharpening algorithms. The temperature-downscaling method used in this paper caused boxy anomalies in parts of the sharpened-temperature field because of the constant residual term, $\Delta \widehat{T}_{300}$, in Eq. (3) within large pixels. This situation also occurred in the temperature sharpening algorithm proposed by Agam et al. (2007). In addition, our temperature sharpening algorithm tends to overestimate the sub-pixel LST for cooler landscapes and underestimate the sub-pixel LST for warmer areas (Kustas et al., 2003). This inaccurate estimation causes errors that are difficult to evaluate when estimating turbulent heat flux. For example, the small turbulent heat flux bias between TSFA and TRFA was caused by the counterbalanced effect as analyzed in Sect. 4.3.1. The evaluation of more temperature sharpening algorithms under heterogeneous surfaces with real datasets when applied in ET models would be helpful (Ha et al., 2011).

Our surface variable retrieval methods were validated against other areas considered in remote sensing measurement campaigns. For example, the albedo algorithm was previously applied to retrieve Global Land Surface Satellite (GLASS) Products (Liang et al., 2014), the LST retrieval algorithm was validated in the Haihe River Basin in northern China (Li et al., 2011), and the soil heat flux correction algorithm was validated in the GAME-Tibet campaign (Yang and Wang, 2008). Since the surface of the Heihe River Basin is extreme heterogeneous, additional comparisons of our algorithm in other areas of research would be better.

In addition, to correct the discrepancy between remotely sensed radiative surface temperature and aerodynamic temperature at the source of heat transport, a brief and well-performed parameterization scheme (under uniformly flat plant surface) of "excess" resistance was used to calculate the aerodynamic resistance of heat transfer (Jiao et al., 2014). Since the objects of our study are mixed pixels, more parameterization methods should be compared to select the optimum method.

Because of the sensitive variables of the one-source energy balance model used in this paper, the accuracy of the LST and canopy height greatly influenced the turbulent heat flux. HJ-1B IRS is a single-thermal channel, the single-channel LST-retrieving algorithm may be unstable under wet atmospheric conditions (water vapor contents higher than 3 $g/cm^2$) (H. Li et al., 2010), which may create a bottleneck for ET estimations by HJ-1B. The canopy height is a priori knowledge based on phenophase classifications and would influence the accuracy of the surface roughness, the length of a heterogeneous surface or the seasonal transition. Multi-source remote sensing data could be used to improve the accuracy of calibrations and land surface variable estimations. Active microwave and LiDAR data (Colin and Faivre, 2010) could be used to obtain the canopy height, which would decrease the dependence on the accuracy of the classification.

**6. Conclusion**

We studied the effects of surface heterogeneity in ET estimation by the IPUS, TRFA and TSFA methods over heterogeneous surface based on spatial resolution characteristic of different satellites, and applied them to HJ-1B satellite data based on operational satellites' instrumental characteristics.

Compared with the IPUS and TRFA methods, the TSFA method is more consistent with in situ measurements. If ET estimating algorithm does not consider surface heterogeneity at all (i.e. IPUS), it would cause significant error (i.e. 186 W·m$^{-2}$) of heat fluxes. If ET estimating algorithm does not consider heterogeneity of LST only (i.e. TRFA), it would cause non-negligible error (i.e. 49 W·m$^{-2}$) in heat fluxes calculating. The TSFA method reduces the uncertainties produced by surface landscapes and LST inhomogeneity. As a sensitive variable of the ET model, canopy height is mainly determined by classification, and the application of classification at a 30-m resolution can improve the accuracy of the canopy height. As another sensitive variable, the sharpened surface temperature at a resolution of 30 m decreases the thermodynamic uncertainty caused by land surface heterogeneities. The TSFA method can capture the heterogeneities of the land surface and integrate the effects of landscapes in mixed pixels that are neglected at coarse spatial resolutions.

HJ-1B satellite data are advantageous because of their high spatiotemporal resolution and free access. Because the satellites are still in operation, long-term data have promising applications for monitoring energy budgets.

**Acknowledgements**

The authors would like to thank Dr. Franz Meixner of the Max Planck Institute for Biogeochemistry in Mainz and Dr. Albrecht Neftel of the Agroscope Research Station in Zurich for providing the footprint calculation tool. We are grateful to the research team of Professor ShaoMin Liu at

Beijing Normal University for providing eddy covariance and automatic meteorological station data.
This study was supported by the Special Fund from the Chinese Academy of Sciences (KZZD-EW-
TZ-18) and the Chinese Natural Science Foundation Project (grant no. 41371360). Generous help
for revising the paper was provided by the editors and reviewers.

**Appendix**

[revised manuscript text omitted]

---

## Editor Decision (ED1)

**Remote-sensing algorithm for surface evapotranspiration considering landscape and statistical effects on mixed-pixels**

ZhiQing Peng [a, b], Xiaozhou Xin [a, *], JinJun Jiao [a, b], Ti Zhou [a, b], Qinhuo Liu [a, c]

*a. State Key Laboratory of Remote Sensing Science, Institute of Remote Sensing and Digital Earth*

*Chinese Academy of Sciences, Beijing 100101, China*

*b. University of Chinese Academy of Sciences, Beijing 100049, China*

*c. Joint Center for Global Change Studies (JCGCS), Beijing 100875, China*

**Abstract**

Evapotranspiration (ET) plays an important role in surface-atmosphere interactions and can be monitored using remote sensing data. However, surface heterogeneity including inhomogeneity of landscapes and surface variable affects the accuracy of ET estimated from satellite data significantly. The objective of this study is to assess and reduce the uncertainties resulted from surface heterogeneity in remotely sensed ET using Chinese HJ-1B satellite data, which is of 30m spatial resolution in VIS/NIR bands and 300m spatial resolution in TIR band. A temperature sharpening and flux aggregation scheme (TSFA) was developed to obtain accurate heat fluxes from the HJ-1B satellite data. Two methods employing different upscaling policies of surface variables and fluxes were used to compare with TSFA, i.e., IPUS (input parameter upscaling) and TRFA (temperature resampling and flux aggregation). Moreover, the three methods can also be regarded as representing three typical schemes handling mixed pixels from the simplest to the most complex, i.e., all surface variables are at coarse resolution (300 m in this study) in IPUS and fine resolution (30 m in this study) in TSFA, while TRFA is in the middle (both 30m and 300m variables are used). Analysis and comparison between them can help us to get better understandings about spatial scale errors in remote sensing of surface heat fluxes. In situ data collected during HiWATER-MUSOEXE (Multi-Scale Observation Experiment on Evapotranspiration over heterogeneous land surfaces of The Heihe Watershed Allied Telemetry Experimental Research) were used for the validation and analysis of the methods. ET estimated by TSFA is of best agreement with in-situ observations, the footprint validation results show that the $R^2$, MBE, and RMSE of the sensible heat flux (H) were 0.61, 0.90 W·m$^{-2}$ and 50.99 W·m$^{-2}$, respectively, and the corresponding terms for the latent heat flux (LE) were 0.82, -20.54 W·m$^{-2}$ and 71.24 W·m$^{-2}$, respectively, and IPUS showed the largest errors in ET estimation. The RMSE of LE between the TSFA and IPUS methods was 51.30 W·m$^{-2}$, and the RMSE of LE between the TSFA and TRFA methods was 16.48 W·m$^{-2}$. Furthermore, additional analysis shows that the TSFA method can capture the sub-pixel variations of land surface temperature and integrate the effects of overlooked landscapes in mixed pixels.

**Index Terms:** heterogeneous surface, temperature sharpening, area weighting, energy balance, evapotranspiration, spatial scale, HJ-1B satellite

**1. Introduction**

Five types of methods have been developed to estimate evapotranspiration (ET) or latent heat flux (LE) via remote sensing. (1) Surface energy balance models calculate LE as a residual term. According to the partitioning of the sources and sinks of the Soil-Plant-Atmosphere Continuum

(SPAC), surface energy balance models can be classified as one-source (Bastiaanssen et al., 1998; Su, 2002; Allen et al., 2007; Long and Singh, 2012a) or two-source models (Shuttleworth and Wallace, 1985; Norman et al., 1995; Xin and Liu, 2010; Zhu et al., 2013). (2) Penman-Monteith models are used to calculate LE by using the Penman-Monteith equation and numerous surface resistance parameterization schemes that control the diffusion of evaporation from land surfaces and transpiration from plant canopies. These two-source Penman-Monteith models separate soil evaporation from plant transpiration (Cleugh et al., 2007; Mu et al., 2011; Leuning et al., 2008; Chen et al., 2013; Sun et al., 2013; Mallick et al., 2015). (3) Land surface temperature-vegetation index (LST-VI) space methods assign the dry and wet edges of the LST-VI feature space as minimum and maximum ET, respectively. These methods interpolate the media using the Penman-Monteith or Priestley-Taylor equation to calculate the LE (Jiang and Islam, 1999, 2001; Sun et al., 2011; Long and Singh, 2012b; Yang and Shang, 2013; Fan et al., 2015; Zhang et al., 2005). (4) Priestley-Taylor models expand the range of the Priestley-Taylor coefficient in the Priestley-Taylor equation (Jiang and Islam, 2003; Jin et al., 2011) or combine the physiological force factors with the energy component of ET (Fisher et al., 2008; Yao et al., 2013). (5) Additional methods include empirical/statistical methods (Wang and Liang, 2008; Yebra et al., 2013) and the use of complementary based models (Venturini et al., 2008) and land-process models with data assimilation schemes (Bateni and Liang, 2012; Xu et al., 2015).

All these ET estimation models are usually developed for simple and homogeneous surface conditions. When these remotely sensed models are applied to calculate the regional ET via satellite data, large spatial scale errors occur. Because heterogeneity is a natural attribute of the Earth's surface, non-linear operational model is another important issue of remotely sensed spatial scale error. However, it is difficult to develop linear operational models due to the complexity of mass and heat transfer processes between the atmosphere and land surface.

In previous studies, researchers have coupled high- and low-resolution satellite data and statistically quantified the inhomogeneity of mixed pixels to correct the scale error in ET estimations by using temperature downscaling that converts images from a lower (coarser) to higher (finer) spatial resolution using statistical-based models with regression or stochastic relationships among parameters (Kustas et al., 2003; Norman et al., 2003; Cammalleri et al., 2013; Ha et al., 2013), the correction-factor method that uses sub-pixel landscapes information to regress the correction factor of scale bias (Maayar and Chen, 2006) and the area-weighting method that calculates roughness length and sensible heat flux based on sub-pixel landscapes (Xin et al., 2012).These correction methods mainly focus on two problems: inhomogeneity of landscapes and inhomogeneity of surface variables.

Studies have shown that different landscapes (Blyth and Harding, 1995; Moran et al., 1997; Bonan et al., 2002; McCabe and Wood, 2006) and the sub-pixel variations of surface variables, such as stomatal conductance (Bin and Roni, 1994), leaf area index (Bonan et al., 1993; Maayar and Chen, 2006) can cause errors in turbulent heat flux estimations. Surface variables inhomogeneity is rather difficult to evaluate as the sub-pixel variation of surface variables could be large even in the pure pixel. For example, generally, temperatures over the land surfaces vary strongly in space and time, and it is not unusual for the LST to vary by more than 10 K over just a few centimeters of distance or by more than 1 K in less than a minute over certain cover types (Z. Li et al., 2013). But in mixed pixels, surface variables such as land surface temperature are set as singular to represent the entire pixel area in ET estimation models.

The focus of this study is on the effects of surface heterogeneity when estimating ET. According to the current satellites operation situation, three methods were used to analyze the uncertainty produced by surface heterogeneity. Input parameter upscaling (IPUS) does not consider the surface heterogeneities at all. It was designed to simulate the satellites that have identical spatial resolution both in visible near-infrared (VNIR) and thermal infrared bands (TIR), such as the land surface products of Moderate-Resolution Imaging Spectroradiometer (MODIS) satellites. Temperature resampling and flux aggregation (TRFA) only does not consider the heterogeneity of LST, and temperature sharpening and flux aggregation (TSFA) consider all the surface heterogeneities. They were designed for the majority of satellites data or products that have inconsistent spatial resolution between VNIR and TIR, such as Landsat and HJ-1B satellites.

Surface variables in this paper mainly derived from HJ-1B satellite data were used for this purpose. The Chinese HJ-1A/B satellites were launched on September 6, 2008, and were designed for disaster and environmental monitoring, as well as other applications. The HJ-1B satellites are equipped with two charge-coupled device (CCD) cameras and one infrared scanner (IRS) with spatial resolutions of 30 m and 300 m, respectively. Compared with high-temporal-resolution satellites, such as the MODIS satellite, or high-spatial-resolution satellites, such as the Landsat 7 or 8 satellites, HJ-1B has the advantage of a high spatial-temporal resolution. Since the satellites were launched, the HJ-1/CCD time series data have been widely used in China to accurately classify land cover (Zhong et al., 2014a) and monitor various environmental disasters (Wang et al., 2010). Land-based variables, such as leaf area index (LAI), land surface temperature (LST), and downward longwave radiation ($L_d$), have been retrieved by the HJ-1 satellites using algorithms developed by Chen et al. (2010), H. Li et al. (2010, 2011) and Yu et al. (2013), respectively. These variables lay the foundation for ET research.

Although the HJ-1B satellites provide CCD data with a high spatial resolution of 30 m, the spatial resolution of the thermal infrared (TIR) band is only 300 m. Thus, surface heterogeneity effects must be considered when estimating the heat flux.

**2. Methodology**

**2.1. Temperature-sharpening method based on statistical relationships**

Surface thermal dynamics are a driving force of ET. The spatial resolution of TIR images is usually not as high as the spatial resolution of visible near-infrared bands (VNIR) because the energy of VNIR photons is higher than the energy of thermal photons. Thus, the inhomogeneity of TIR images would be greater than the inhomogeneity of VNIR images. Once the inhomogeneity of TIR images is enhanced, the uncertainty of the variables is calculated in the TIR band, and variables such as the land surface temperature become unpredictable. Therefore, we would like to derive land surface temperature data with a high spatial resolution.

The different spatial resolutions of TIR and VNIR images make it possible to obtain the land surface temperature at the spatial resolution of the VNIR images, which is referred to as temperature-sharpening. Kustas et al. (2003) proposed a statistical temperature-sharpening method that could be applied to remotely sensed evapotranspiration models. This method assumes that the negative correlation between the Normalized Difference Vegetation Index (NDVI) and LST is invariant. The NDVI reflects vegetation growth and cover, and the LST reflects surface thermal dynamics. The LST decreases with increasing vegetation cover. The resulting scatter plots form a feature space that is applicable at different scales when enough pixels exist.

HJ-1B satellite images can provide vegetation and thermal information at spatial resolutions of 30 m and 300 m, respectively. However, the 300 m resolution thermal data cannot be use to discriminate the surface temperatures of small targets within pixels. This deficiency can be addressed by using the functional relationship between NDVI and LST. A flowchart of temperature sharpening is shown in Fig. 1, and the LST at the NDVI pixel resolution can be derived based on the following steps (Kustas et al., 2003):

(1) The selection of a subset of pixels from the scene where the NDVI is the most uniform at a pixel resolution of 300 m. Calculate the coefficient of variation (CV) by using the original NDVI data ($NDVI_{30}$) with a resolution of 30 m and sort the values from smallest to largest. The CV is calculated as follows:

$$CV = \frac{STD}{mean} \qquad (1)$$

where STD and mean are the standard deviation and the average values, respectively, among the $10 \times 10$ pixels that make up each 300-m NDVI ($NDVI_{300}$) aggregated from $NDVI_{30}$.

(2) Next, the $NDVI_{300}$ is divided into several classes ($0 \leq NDVI_{300} < 0.2$, $0.2 \leq NDVI_{300} < 0.5$ and $0.5 \leq NDVI_{300}$). Lower CV values correspond with more homogeneous land surface values, and a specific fraction should guarantee that a sufficient number of pixels is available for fitting a least-squares expression between $NDVI_{300}$ and $T_{300}$. Then, the fractions (25%) of the pixels having the lowest CV are selected from each class.

(3) A least-squares expression is fit between $NDVI_{300}$ and $T_{300}$ using the selected pixels.

$$\widehat{T}_{300}(NDVI_{300}) = a + b \times NDVI_{300} + c \times NDVI_{300}^2 \qquad (2)$$

[Figure]

**Figure 1.** Flowchart of temperature sharpening.

(4) For each 30-m pixel within the 300-m pixel, $\widehat{T}_{30}$ can be computed according to Eq. (2) as follows:

$$\widehat{T}_{30}(NDVI_{30}) = a + b \times NDVI_{30} + c \times NDVI_{30}^2 + \Delta\widehat{T}_{300} \qquad (3)$$

where $\Delta\widehat{T}_{300} = T_{300} - \widehat{T}_{300}$ is the deviation between the regressed temperature and the temperature that was observed by the satellite at 300 m.

**2.2. Area-weighting method based on landscape information**

Coarse pixels are inhomogeneous because various types of land use may be included. Using a

1 dominant type to represent such a large landscape is irrational. When a sharpened temperature is
2 obtained, the spatial details could be provided by surface variables at a high resolution, and the
3 inhomogeneous problem could be greatly diminished as the landscape is divided into finer pixels.

4 Combined with a high-resolution classification map, sub-pixel scale parameters can be applied
5 to the ET algorithm, which is more rational than using a dominate-class type because different land-
6 scapes might require different ET algorithms. The surface energy flux can be averaged linearly due
7 to the conservation of energy (Kustas et al., 2003), and a simple average that calculates the arithme-
8 tic mean over sub-pixels is the best choice for flux upscaling approaches (Ershadi et al., 2013b).
9 Thus, the aggregated flux at a low resolution $F(x, y)$ is the arithmetic mean of all of the $n \times n$
10 sub-pixel fluxes that constitute the contributing flux $F(x_i, y_j)$ at coordinate $(x_i, y_j)$ as follows:

$$F(x, y) = \frac{1}{n \times n} \sum_{i=1}^{n} \sum_{j=1}^{n} F(x_i, y_j) \tag{4}$$

12 Because the average of the sub-pixels fluxes is equal to the area-weighted sum of each land-
13 type result, the final coarse result can be derived by the area-weighted sum of each land-type result
14 within the landscape. The main steps of the area-weighting process are shown below (Xin et al.,
15 2012):
16 (1) Geometric correction and registration of the VNIR and TIR input datasets.
17 (2) Count area ratio of different land-cover types within each pixel of a low-spatial-resolution
18 classification image.
19 (3) According to the fine-classification data, different parameterization schemes can be used in
20 the ET algorithm to calculate the sub-pixel flux, such as net radiation ($R_n$), soil heat flux ($G$) and
21 sensible heat flux ($H$).
22 (4) To calculate the regional flux, the flux of the large pixel is calculated by the area-weighting
23 method as follows:

$$F = \sum_{i=1}^{n} w_i \cdot F_i \tag{5}$$

25 where $w_i$ is the fractional area contributing flux $F_i$ of class type $i$, and $F$ is the aggregated flux
26 at the coarse resolution. The LE is computed as a residual of the surface energy balance in the TSFA
27 (Temperature Sharpening and Flux Aggregation, see Sect. 2.3) process, in which a high-spatial-
28 resolution image is used to reduce the mixed pixels.

29 **2.3. Pixel ET algorithm**

30 The surface energy balance describes the energy between the land surface and atmosphere. The
31 energy budget is commonly expressed as follows:

$$R_n = LE + H + G \tag{6}$$

33 where $R_n$ is the net radiation, $G$ is the soil heat flux, $H$ is the sensible heat flux, and $LE$ is the
34 latent heat flux absorbed by water vapor when it evaporates from the soil surface and transpires
35 from plants through stomata. The widely used one-source energy balance model considers the ho-
36 mogeneous SPAC medium and ignores the inhomogeneity and structure. The $LE$ can be expressed
37 as follows:

$$LE = \frac{\rho c_p}{\gamma} \cdot \frac{e_s - e_a}{r_a + r_s} \tag{7}$$

39 where $\gamma$ is the psychometric constant; $e_s$ and $e_a$ are the aerodynamic saturation vapor pressure
40 and atmospheric water vapor pressure, respectively; and $r_a$ and $r_s$ are the water vapor transfer

aerodynamic resistance and surface resistance, respectively. Surface resistance includes soil resistance and canopy resistance. The surface resistance is influenced by the physiological characteristics of the vegetation and the water supply of roots. Thus, it is difficult to obtain surface resistance by using remote sensing, and surface resistance is highly uncertain, particularly over heterogeneous surfaces. To avoid error introduced by the uncertainty of the surface resistance, the LE is computed as a residual of the surface energy balance equation.

$R_n$ is the difference between incoming and outgoing radiation and is calculated as follows:

$$R_n = S_d(1 - \alpha) + \varepsilon_s L_d - \varepsilon_s \sigma T_{rad}^4 \tag{8}$$

where $S_d$ is the downward shortwave radiation, $\alpha$ is the surface broadband albedo, $\varepsilon_s$ is the emissivity of the land surface, $L_d$ is the downward atmospheric longwave radiation, $\sigma = 5.67 \times 10^{-8} W \cdot m^{-2} \cdot K^{-4}$ is the Stefan-Boltzmann constant, and $T_{rad}$ is the surface radiation temperature.

$G$ is commonly estimated by deriving empirical equations that consider surface variables, such as $R_n$. Because the canopy exerts a significant influence on $G$, the fractional canopy coverage FVC is used to determine the ratio of $G$ to $R_n$ as follows:

$$G = R_n \times [\Gamma_c + (1 - FVC) \times (\Gamma_s - \Gamma_c)] \tag{9}$$

where $\Gamma_s$ is 0.315 for bare soil and $\Gamma_c$ is 0.05 for a full vegetation canopy (Su, 2002). H is the transfer of turbulent heat between the surface and atmosphere that is driven by a temperature difference and is controlled by resistances that depend on local atmospheric conditions and land cover properties (Kalma et al., 2008). According to gradient diffusion theory,

$$H = \rho c_p \frac{T_{aero} - T_a}{r_a} \tag{10}$$

where $\rho$ is the density of the air; $c_p$ is the specific heat of the air at a constant pressure; $T_{aero}$ is the aerodynamic surface temperature obtained by extrapolating the logarithmic air-temperature profile to the roughness length for heat transport; $T_a$ is the air temperature at a reference height; and $r_a$ is the aerodynamic resistance, which influences the heat transfer between the source of turbulent heat flux and the reference height. Aerodynamic resistance was calculated based on the Monin-Obukhov similarity theory (MOST) using a stability correction function (Paulson, 1970; Ambast et al., 2002). The zero-plane displacement height, d, and roughness length, $z_{0m}$, were parameterized by the schemes proposed by Choudhury (Choudhury and Monteith, 1988).

In this approach, H must be accurately estimated. However, calculating H by using Eq. (10) is difficult. Because remote sensing cannot obtain $T_{aero}$, the value of $T_{aero}$ is usually replaced by the radiative surface temperature $T_{rad}$, which is not always equal to $T_{aero}$. The difference between these terms for homogeneous and fully covered vegetation is approximately 1-2℃ (Choudhury et al., 1986), or up to 10℃ in sparsely vegetative areas (Kustas, 1990). The method that corrects for this discrepancy adds "excess" resistance $r_{ex}$ to $r_a$. We used the brief method $r_{ex} = 4/u_*$, which was proposed by Chen (1988), to calculate $r_{ex}$.

Fig. 2 shows a flowchart for merging ET retrieval and temperature sharpening based on HJ-1B satellites.

[Figure]

2      **Figure 2.** Flowchart of ET retrieval using the "Temperature Sharpening and Flux Aggregation" method.

3        The spatial scale effect is usually revealed by a discrepancy between different upscaling meth-
4      ods: the upscaling of aggregate parameters to the large scale to calculate the heat flux and the cal-
5      culation of the heat flux at the small scale before upscaling it to the large scale. In this paper, the
6      resolution of the final output result is 300 m. To evaluate the reduced heterogeneity effect of TSFA,
7      two other upscaling methods called IPUS and TRFA were used (see Fig. 3). When using IPUS, the
8      surface-parameter retrieving algorithms (see Sect. 3.2.1.1) are applied to HJ-1 CCD data. Then, the
9      variable results are aggregated at a spatial resolution of 300 m. These 300 m outputs are used as
10     input parameters in the one-source energy balance model to obtain the four energy-balance compo-
11     nents at 300 m. In TRFA, the LST at 300 m is resized to 30 m using nearest neighbor sampling.
12     Then, the resampled LST and surface VNIR variables at 30 m are applied to ET algorithm. The
13     outputs of the four energy-balance components of the TRFA are obtained using the area-weighting
14     method shown in Sect. 2.2.

[Figure]

16      **Figure 3.** Flowchart of the three upscaling methods for retrieving evapotranspiration.

**3. Study area and Dataset**

**3.1. Study area**

Our study was conducted in the middle stream of the Heihe River Basin (HRB), which is located near the city of Zhangye in the arid region of Gansu Province in northwestern China (100.11˚E-100.16˚E, 39.10˚N-39.15˚N). The middle reach of the HRB is a typical desert-oasis agriculture ecosystem dominated by maize and wheat. A large portion of the Gobi Desert and the alpine vegetation of Qilian Mountain are located near the study area (see Fig. 4). The artificial oasis is highly heterogeneous, which impacts the thermal-dynamic and hydraulic features. Consequently, the water use efficiency and ET are variable. The Heihe River Basin has long served as a test bed for integrated watershed studies as well as land surface or hydrological experiments. Comprehensive experiments, such as Watershed Allied Telemetry Experimental Research (WATER) (Li et al., 2009), and an international experiment - the Heihe Basin Field Experiment (HEIFE) in World Climate Research Programme (WCRP) have taken place in the Heihe River Basin. One major objective of HiWATER is to capture the strong land surface heterogeneities and associated uncertainties within a watershed (Li et al., 2013).

[Figure]

**Figure 4.** Study area and distribution of EC towers in HiWATER-MUSOEXE

**3.2. Dataset**

In this paper, the data are mainly derived from the HJ-1B satellite. We combined these data with ancillary data and the in situ "Multi-Scale Observation Experiment on Evapotranspiration over heterogeneous land surfaces of The Heihe Watershed Allied Telemetry Experimental Research" (HiWATER-MUSOEXE) data to estimate and validate the HJ-B land surface variables and heat fluxes.

**3.2.1. Remote sensing data**

**3.2.1.1. HJ-1B satellite data**

The specifications of HJ-1B are shown in Table 1. These satellites have quasi-sun-synchronous orbits at an altitude of 650 km, a swath width of 700 km and a revisit period of 4 days. Together, the revisit period of the satellites is 48 h. Because HJ-1 CCDs lack an onboard calibration system, scholars have proposed cross-calibration methods for calibrating the CCD instruments (Zhong et al., 2014b; Zhang et al., 2013). The image quality of HJ-1A/B CCDs is stable, the performances of each band are balanced (Zhang et al., 2013), and the radiometric performance of the HJ-1A/B CCD sensors is similar to the performances of the Landsat-5 TM, Advanced Land Imager, and ASTER sensors. The image quality of HJ-1 CCDs is very similar to the image quality of Landsat-5 TM (Jiang et al., 2013). In addition, the accuracy of the TIR band's onboard calibration meets land surface temperature retrieval requirements but not sea surface temperature retrieval requirements (J. Li et al., 2011). China Center for Resources Satellite Data and Application (CRESDA) releases calibration coefficients once each year on its website (http://www.cresda.com). These data are freely available from the CRESDA website (http://218.247.138.121/DSSPlatform/index.html).

**Table 1.** Specifications of the HJ-1B main payloads

| Sensor | Band | Spectral range (μm) | Spatial resolution (m) | Swath width (km) | Revisit time (days) |
|--------|------|---------------------|------------------------|------------------|---------------------|
| CCD | 1 | 0.43-0.52 | | | |
| | 2 | 0.52-0.60 | | 360 (single) | |
| | 3 | 0.63-0.69 | 30 | 700 (two) | 4 |
| | 4 | 0.76-0.90 | | | |
| IRS | 5 | 0.75-1.10 | | | |
| | 6 | 1.55-1.75 | 150 | 720 | 4 |
| | 7 | 3.50-3.90 | | | |
| | 8 | 10.5-12.5 | 300 | | |

[Figure]

**Figure 5.** Flowchart of the land surface variable retrieval. The abbreviations are defined as follows: SZA: solar zenith angle; SAA: solar azimuth angle; VZA: view zenith angle; AOD: aerosol optical depth; ABT: at-nadir brightness temperature; $S_d$: downward shortwave radiation; USR: upward shortwave radiation, ULR: upward longwave radiation; and $L_d$: downward longwave radiation.

We used the HJ-1B satellite data for the HRB region in 2012. Because many variable-retrieving algorithms required clear-sky conditions for calculating ET, we combined data-quality information with visual interpretation to select satellite images without clouds. Considering the time period of the ground observations discussed in Sect. 3.2.2, we obtained data for 11 days: June 19, June 30, July 8, July 27, August 2, August 15, August 22, August 29, September 2, September 13 and September 14.

The HJ-1B satellite data from the HRB were pre-processed and included geometric correction, radiometric calibration, and atmosphere correction. For Eq. (1) to (10), the following surface variables are needed: downward shortwave radiation, downward longwave radiation, emissivity, albedo, fractional vegetation coverage (FVC), cloud mask data, meteorological data, LAI and LST. Fig. 5 contains a flowchart showing the retrieval of these variables.

(1) Surface albedo. According to the algorithm proposed by Liang et al. (2005) and Liu et al. (2011), surface albedo was obtained from the top of the atmosphere (TOA) reflectance by the HJ-1 satellite with a lookup table based on an angular bin regression relationship. The surface albedo and bidirectional reflectance distribution function (BRDF) of the HJ-1 satellite in the regression procedure were monitored by using POLDER-3/PARASOL BRDF datasets, and BRDF was used to obtain the TOA reflectance using the 6S (Second Simulation of a Satellite Signal in the Solar Spectrum) radiation transfer mode.

(2) NDVI, FVC and LAI. The NDVI is the central regression of temperature sharpening and was used to calculate the FVC. Atmospherically corrected surface reflectance values were used to calculate the NDVI as follows:

$$NDVI = \frac{\rho_{nir} - \rho_{red}}{\rho_{nir} + \rho_{red}} \quad (11)$$

and

$$FVC = \frac{NDVI - NDVI_s}{NDVI_v + NDVI_s} \quad (12)$$

where $\rho_{nir}$ and $\rho_{red}$ are the reflectances in the near-infrared and red band, respectively, and $NDVI_v$ and $NDVI_s$ are the fully vegetated and bare soil NDVI values, respectively. As an important input for the parameterization of surface roughness length and aerodynamic resistance, the LAI was determined using the following equation (Nilson, 1971):

$$P(\theta) = e^{-G(\theta) \cdot \Omega \cdot LAI / \cos(\theta)} \quad (13)$$
$$P(\theta) = 1 - FVC \quad (14)$$

where $\theta$ is the zenith angle, $P(\theta)$ is the angular distribution of the canopy gap fraction, $G(\theta)$ is the projection coefficient (0.5), and $\Omega$ is the total foliage clumping index, which can be obtained from the GLC global clumping index database according to the type of land use (He et al., 2012).

(3) Land surface emissivity (LSE). LSE is needed to calculate the $R_n$ and is extremely important for retrieving LST. In this paper, LSE was calculated using the FVC as follows (Valor and Caselles, 1996):

$$\varepsilon = \varepsilon_v \cdot FVC + \varepsilon_g(1 - FVC) + 4 < d\varepsilon > \cdot FVC \cdot (1 - FVC) \quad (15)$$

where $\varepsilon$ is the LSE, $< d\varepsilon >$ is an effective value of the cavity effect of emissivity, the mean $d\varepsilon$ of all vegetation species in this study is <dε>=0.015, and $\varepsilon_v$ and $\varepsilon_g$ are the vegetation and ground emissivity, respectively.

(4) Land surface temperature. A single-channel parametric model for retrieving LST based on HJ-1B/IRS TIR data developed by Li et al. (2010) was applied. This model was developed from a parametric model based on MODTRAN4 using NCEP atmospheric profile data.

(5) Downward shortwave radiation. The algorithm proposed by L. Li et al. (2010) was applied. MOD05, TOMS, aerosol, and solar angle data were used to estimate the direct light flux and diffuse light flux by using a lookup table that was generated using the 6S radiation transfer mode (Vermote et al., 2006). This method considered the influences of complex terrain, and a topographic correction

was performed by using products of the ASTER DEM.

(6) Downward longwave radiation ($L_d$). The TOA brightness temperature of the HJ-1B thermal channel was used to substitute the atmospheric effective temperature. Effective atmospheric emissivity was parameterized as an empirical function of the water vapor content. These values were substituted for atmospheric temperature and atmospheric emissivity to estimate the value of $L_d$. Because this $L_d$ retrieval method proposed by Yu et al. (2013) was only valid for clear-sky conditions, cloud masking information was used to determine clear skies. When cloud contamination existed in the image, the brightness temperature was relatively low, causing the $L_d$ to be lower than that in the cloudless images.

**3.2.1.2. Ancillary data**

Ancillary data were used because the bands of the satellite could not invert all of the variables needed for retrieving ET.

(1) Atmospheric water vapor data. MODIS provides water vapor data (MOD05), including a 1-km near-infrared product and a 5-km thermal-infrared product, every day. The 1-km near-infrared water vapor product was used to retrieve $L_d$ in this study.

(2) Surface elevation data. We used the 30-m resolution Global Digital Elevation Model (GDEM) based on ASTER, which covers 83 °N–83 °S, to derive $S_d$.

(3) Atmosphere ozone data. A Total Ozone Mapping Spectrometer (TOMS), which was carried on an Earth Probe (EP) satellite, was used to derive $S_d$. The TOMS-EP provided daily global atmosphere ozone data at a resolution of 1 °×1.25 °(Li et al., 2010b).

(4) Atmosphere profile data. Global reanalysis data from the National Centers for Environmental Prediction (NCEP) were used to derive LST. These data were generated globally every 6 hours (0:00, 06:00, 12:00, 18:00 UTC) for every 1 °of latitude and longitude (Li et al., 2010a).

**3.2.2. HiWATER experiment dataset**

The in situ HRB observation data were provided by HiWATER. From June to September 2012, HiWATER designed two nested observation matrices over 30 km×30 km and 5.5 km×5.5 km within the middle stream oasis in Zhangye to focus on the heterogeneity of the scale effect in the so-called HiWATER-MUSOEXE.

In a larger observation matrix, four eddy covariance (EC) systems and one superstation were installed in the oasis–desert ecosystem. Each station was supplemented with an automatic meteorological station (AMS) to record meteorological and soil variables and monitor the spatial–temporal variations of ET and its impact factors (Li et al., 2013). The station information is shown in Table 2, and the distribution of the stations is shown in Fig. 4. Within the artificial oasis, an observation matrix composed of 17 EC towers and ordinary AMSs exists where the superstation was located. The land surface was heterogeneous and dominated by maize, maize inter-cropped with spring wheat, vegetables, orchards, and residential areas (Li et al., 2013). Because the EC16 and HHZ stations lacked $R_n$ and G observation data, they were excluded from this study.

**Table 2.** The in situ HiWATER-MUSOEXE station information

| Station | Longitude (°) | Latitude (°) | Tower height (m) | Altitude (m) | Land cover |
| --- | --- | --- | --- | --- | --- |
| EC1 | 100.36E | 38.89N | 3.8 | 1552.75 | vegetation |
| EC2 | 100.35E | 38.89N | 3.7 | 1559.09 | maize |
| EC3 | 100.38E | 38.89N | 3.8 | 1543.05 | maize |

| | | | | | |
|---|---|---|---|---|---|
| EC4 | 100.36E | 38.88N | 4.2 | 1561.87 | building |
| EC5 | 100.35E | 38.88N | 3 | 1567.65 | maize |
| EC6 | 100.36E | 38.87N | 4.6 | 1562.97 | maize |
| EC7 | 100.37E | 38.88N | 3.8 | 1556.39 | maize |
| EC8 | 100.38E | 38.87N | 3.2 | 1550.06 | maize |
| EC9 | 100.39E | 38.87N | 3.9 | 1543.34 | maize |
| EC10 | 100.40E | 38.88N | 4.8 | 1534.73 | maize |
| EC11 | 100.34E | 38.87N | 3.5 | 1575.65 | maize |
| EC12 | 100.37E | 38.87N | 3.5 | 1559.25 | maize |
| EC13 | 100.38E | 38.86N | 5 | 1550.73 | maize |
| EC14 | 100.35E | 38.86N | 4.6 | 1570.23 | maize |
| EC15 | 100.37E | 38.86N | 4.5 | 1556.06 | maize |
| EC17 | 100.37E | 38.85N | 7 | 1559.63 | orchard |
| GB | 100.30E | 38.91N | 4.6 | 1562 | uncultivated land-Gobi |
| SSW | 100.49E | 38.79N | 4.6 | 1594 | uncultivated land-desert |
| SD | 100.45E | 38.98N | 5.2 | 1460 | swamp land |

The ground observation data include the H and LE. Reliable methods were used to ensure the quality of the turbulent heat flux data. Before the main campaign, an intercomparison of all instruments was conducted in the Gobi Desert (Xu et al., 2013). After basic processing, including spike removal and corrections for density fluctuations (WPL-correction), a four-step procedure (data were rejected when (1) the sensor was malfunctioning, (2) precipitation occurred within 1 h before or after collection, (3) the missing ratio was greater than 3% in the 30-min raw record and (4) the friction velocity was below 0.1 ms$^{-1}$ at night) was performed to control the quality of the EC data, and EC outputs were available every 30 min (for more details see Liu et al., 2011b; Xu et al., 2013; Liu et al., 2016). G was measured by using three soil heat plates at a depth of 6 cm at each site, and the surface G was calculated using the method proposed by Yang and Wang(2008) based on the soil temperature and moisture above the plates. Surface meteorological variables, such as wind speed, wind direction, relative humidity and air pressure, were used to interpolate images using the inverse-distance weighted method. Researchers can obtain these data from the websites of the Cold and Arid Regions Science Data Center at LanZhou http://card.westgis.ac.cn/ or the Heihe Plan Data Management Center http://www.heihedata.org/.

An energy imbalance is common in ground flux observations. The conserving Bowen ratio (H/LE) and residual closure technique are often used to force energy balance. Computing the LE as a residual variable may be a better method for energy balance closure under conditions with large LEs (small or negative Bowen ratios due to strong advection) (Kustas et al., 2012). Thus, the residual closure method was applied because the "oasis effect" was distinctly observed in the desert-oasis system on clear days during the summer (Liu et al., 2011b).

**4. Results and analysis**

**4.1. Evaluation of surface variables**

To control the model input variables and analyze sources of error, the coarse-resolution land surface temperature, downward shortwave radiation, downward longwave radiation, $R_n$ and G

1  were evaluated using in situ data.

2    The ground-based land surface temperature, $T_s$, was calculated using the Stefan-Boltzman

3  Law from the AMS measurements of the longwave radiation fluxes (Li et al., 2014) as follows:

$$T_s = \left[\frac{L^\uparrow - (1-\varepsilon_s) \cdot L^\downarrow}{\varepsilon_s \cdot \sigma}\right]^{\frac{1}{4}} \tag{16}$$

5  in which $L^\uparrow$ and $L^\downarrow$ are in situ surface upwelling and atmospheric downwelling longwave radiation,

6  respectively, and $\varepsilon_s$ is the surface broadband emissivity, which is regarded as the pixel value of the

7  HJ-1B at the AMS. The coefficient of determination $R^2$, mean bias error (MBE) and root mean

8  square error (RMSE) of the LST are 0.71, -0.14 K and 3.37 K, respectively. As seen in Table 3, the

9  accuracy of EC4 is low. The main causes of the large errors are as follows: (1) because buildings

10  and soil/vegetation are distinct materials, the LSE algorithm may not be suitable for buildings and

11  (2) the EC4 foundation is non-uniform and is not suitable for validation. After removing the EC4

12  data, the $R^2$, MBE, and RMSE of the LSTs were 0.83, 0.69 K and 2.51 K, respectively. The LST

13  errors of SSW and SD were large due to large errors on particular days. For example, although it

14  was briefly cloudy above station SSW on July 27, this area was not identified as cloudy in the cloud

15  detection process.

16  **Table 3.** Station validation results of land surface temperature

| station | $R^2$ | MBE (K) | RMSE (K) | station | $R^2$ | MBE (K) | RMSE (K) |
|---------|-------|---------|----------|---------|-------|---------|----------|
| EC1 | 0.82 | 0.18 | 1.74 | EC11 | 0.42 | 1.59 | 2.98 |
| EC2 | 0.82 | 0.59 | 1.54 | EC12 | 0.87 | 0.62 | 1.51 |
| EC3 | 0.69 | 0.38 | 1.90 | EC13 | 0.83 | 0.44 | 1.48 |
| EC4 | 0.83 | -9.87 | 10.04 | EC14 | 0.73 | 1.43 | 2.44 |
| EC5 | 0.83 | 1.71 | 2.34 | EC15 | 0.74 | 1.53 | 2.41 |
| EC6 | 0.61 | 0.30 | 2.44 | EC17 | 0.78 | 1.20 | 2.32 |
| EC7 | 0.82 | 0.39 | 1.40 | GB | 0.69 | 0.12 | 2.33 |
| EC8 | 0.83 | 0.45 | 1.55 | SSW | 0.59 | 1.38 | 3.82 |
| EC9 | 0.63 | 2.31 | 3.15 | SD | 0.76 | -3.83 | 4.84 |
| EC10 | 0.68 | 1.32 | 2.45 | | | | |

17    The $R^2$, MBE, and RMSE values of $S_d$ were 0.81, 13.80 W·m$^{-2}$, and 25.35 W·m$^{-2}$, respectively.

18  The station validation results are shown in Table 4. The accuracy of SSW is low. Because cloudy

19  conditions occurred briefly on July 27, few ground observations were obtained, and $S_d$ was signif-

20  icantly overestimated. After removing these data, the $R^2$, MBE, and RMSE values of $S_d$ at SSW

21  were 0.87, 10.90 W·m$^{-2}$ and 21.13 W·m$^{-2}$, respectively.

22  **Table 4.** Station validation results of downward shortwave radiation

| station | $R^2$ | MBE (W·m$^{-2}$) | RMSE (W·m$^{-2}$) | station | $R^2$ | MBE (W·m$^{-2}$) | RMSE (W·m$^{-2}$) |
|---------|-------|------------------|-------------------|---------|-------|------------------|-------------------|
| EC1 | 0.97 | 25.23 | 27.73 | EC11 | 0.90 | 30.11 | 33.76 |
| EC2 | 0.84 | 28.29 | 33.57 | EC12 | 0.96 | 24.35 | 26.43 |
| EC3 | 0.97 | 17.56 | 19.25 | EC13 | 0.93 | 12.41 | 17.92 |
| EC4 | 0.98 | 6.07 | 9.34 | EC14 | 0.98 | 32.40 | 33.49 |
| EC5 | 0.98 | 10.60 | 12.29 | EC15 | 0.94 | 26.71 | 29.71 |
| EC6 | 0.93 | 27.68 | 30.71 | EC17 | 0.94 | -20.25 | 24.54 |

| station | R² | MBE | RMSE | station | R² | MBE | RMSE |
|---|---|---|---|---|---|---|---|
| EC7 | 0.89 | -17.69 | 27.59 | GB | 0.89 | 25.34 | 30.63 |
| EC8 | 0.83 | 15.63 | 25.50 | SSW | 0.63 | 18.51 | 34.93 |
| EC9 | 0.96 | -2.27 | 9.96 | SD | 0.98 | 5.70 | 13.82 |
| EC10 | 0.94 | -3.50 | 11.97 | | | | |

The $R^2$, MBE, and RMSE of the HRB $L_d$ were 0.73, 0.28 W·m$^{-2}$, and 21.24 W·m$^{-2}$, respectively. As seen in Table 5, the accuracies at EC3, SD and SSW were low. The low accuracies at EC3 and SD potentially resulted from (1) high humidity, which resulted in low at-nadir brightness temperatures and low retrieved $L_d$, or (2) instrument error, which occurred because the EC3 ground observations were always greater than those of the other stations during the same period. Although SSW was located in a desert, the ground-air temperature difference was large. The $L_d$ retrieval may have a large error because the models use surface temperature when estimating $L_d$ to approximate or substitute the near-surface temperature (Yu et al., 2013). The corrected error of our $L_d$ retrieving algorithm resulted from the ground-air temperature difference in non-vegetated areas. The inaccuracy of the SSW LST may influence the $L_d$ results.

**Table 5.** Station validation results of downward longwave radiation

| station | R² | MBE (W·m$^{-2}$) | RMSE (W·m$^{-2}$) | station | R² | MBE (W·m$^{-2}$) | RMSE (W·m$^{-2}$) |
|---|---|---|---|---|---|---|---|
| EC1 | 0.85 | 4.16 | 17.21 | EC11 | 0.93 | -2.72 | 10.55 |
| EC2 | 0.88 | 0.11 | 14.23 | EC12 | 0.87 | -0.84 | 14.80 |
| EC3 | 0.91 | -35.65 | 37.88 | EC13 | 0.86 | -7.28 | 15.98 |
| EC4 | 0.88 | 3.36 | 16.38 | EC14 | 0.82 | 4.07 | 16.42 |
| EC5 | 0.88 | -0.79 | 15.02 | EC15 | 0.85 | 17.67 | 23.06 |
| EC6 | 0.84 | 2.55 | 15.43 | EC17 | 0.90 | -1.11 | 12.87 |
| EC7 | 0.75 | -5.90 | 19.72 | GB | 0.88 | 9.50 | 27.82 |
| EC8 | 0.80 | -1.35 | 17.49 | SSW | 0.85 | 25.33 | 34.50 |
| EC9 | 0.86 | 10.44 | 17.99 | SD | 0.85 | -26.54 | 34.08 |
| EC10 | 0.87 | 7.98 | 16.05 | | | | |

The $R^2$, MBE, and RMSE of the HRB $R_n$ were 0.70, -9.64 W·m$^{-2}$, and 42.77 W·m$^{-2}$, respectively. The station $R_n$ validation results are shown in Table 6, which indicate that the accuracies of EC4, EC7, EC17 and SSW were relatively low. According to the sensitivity analysis of Eq. (8), $L_d$ and $S_d$ are highly sensitive variables when calculating $R_n$, while the albedo, LSE and LST are not as sensitive. Although LST was not a sensitive variable, the EC4's LST, MBE and RMSE reached -9.87 K and 10.04 K because the land cover of EC4 was maize at the 300 m resolution. However, the observation tower was in a built-up area, which potentially caused errors when estimating $R_n$. The accuracies of the EC7 $S_d$ and $L_d$ were low on several days, and after removing these data, MBE=-43.40 W·m$^{-2}$ and the RMSE=50.50 W·m$^{-2}$. EC17 was within an orchard, and the signal that was received by the sensors at EC17 were affected by the complex vertical structure of the orchard ecosystem. The information on substrate plants may be ignored, leading to albedo retrieval errors. Although the albedo was not a sensitive variable, a 0.03 bias can lead to an $R_n$ error of approximately 20 W·m$^{-2}$ when the solar incoming radiation is large. As previously mentioned, it was briefly cloudy on July 27, and after removing that data, the $R^2$, MBE, and RMSE values of the $R_n$ obtained at SSW were 0.72, 8.20 W·m$^{-2}$, and 37.60 W·m$^{-2}$, respectively.

**Table 6.** Station net radiation validation results

| station | $R^2$ | MBE ($W·m^{-2}$) | RMSE ($W·m^{-2}$) | station | $R^2$ | MBE ($W·m^{-2}$) | RMSE ($W·m^{-2}$) |
|---|---|---|---|---|---|---|---|
| EC1 | 0.76 | -2.55 | 30.61 | EC11 | 0.86 | -15.13 | 28.05 |
| EC2 | 0.79 | 2.52 | 25.24 | EC12 | 0.90 | -8.46 | 19.38 |
| EC3 | 0.86 | -35.84 | 42.97 | EC13 | 0.88 | -25.73 | 32.34 |
| EC4 | 0.84 | 76.64 | 80.25 | EC14 | 0.90 | 4.23 | 18.18 |
| EC5 | 0.85 | -24.41 | 32.34 | EC15 | 0.84 | 8.33 | 23.01 |
| EC6 | 0.82 | 4.35 | 23.44 | EC17 | 0.89 | -62.62 | 68.11 |
| EC7 | 0.61 | -58.66 | 67.83 | GB | 0.77 | -10.40 | 38.86 |
| EC8 | 0.83 | -20.62 | 32.45 | SSW | 0.44 | 23.05 | 62.93 |
| EC9 | 0.87 | -29.60 | 36.27 | SD | 0.75 | 19.98 | 35.24 |
| EC10 | 0.83 | -24.35 | 33.51 | | | | |

The $R^2$, MBE, and RMSE of the G in the HRB were 0.57, 8.51 $W·m^{-2}$, and 29.73 $W·m^{-2}$, respectively. The station G validation results are shown in Table 7. For EC5, the soil temperature and moisture were the same at different depths after July 19, which resulted in a surface G that was equal to the G at a depth of 6 cm. The G below the surface was usually less than the G at the soil surface; thus, the validation results of the G at EC5 indicate that G was overestimated. For SSW, the brief cloudy period decreased the observed soil surface temperature, which decreased the calculated surface G. However, the remotely sensed G did not reflect this situation. In this case, the G was overestimated because the $R_n$ was overestimated. After removing the data on July 27, the $R^2$, MBE, and RMSE of the G at SSW were 0.17, 19.34 $W·m^{-2}$, and 33.30 $W·m^{-2}$, respectively.

**Table 7.** Station validation results of the soil heat flux

| station | $R^2$ | MBE ($W·m^{-2}$) | RMSE ($W·m^{-2}$) | station | $R^2$ | MBE ($W·m^{-2}$) | RMSE ($W·m^{-2}$) |
|---|---|---|---|---|---|---|---|
| EC1 | 0.50 | 19.73 | 31.53 | EC11 | 0.71 | 4.23 | 19.23 |
| EC2 | 0.24 | 20.78 | 28.72 | EC12 | 0.53 | 20.29 | 24.79 |
| EC3 | 0.03 | -1.15 | 36.28 | EC13 | 0.91 | -0.89 | 17.27 |
| EC4 | 0.45 | 18.50 | 22.29 | EC14 | 0.82 | -1.89 | 18.72 |
| EC5 | 0.38 | 41.87 | 60.19 | EC15 | 0.78 | 6.68 | 15.80 |
| EC6 | 0.83 | -5.91 | 14.57 | EC17 | 0.49 | 8.26 | 33.59 |
| EC7 | 0.28 | 7.50 | 24.65 | GB | 0.29 | -17.86 | 26.81 |
| EC8 | 0.68 | -5.73 | 20.15 | SSW | 0.01 | 30.41 | 51.87 |
| EC9 | 0.61 | 6.83 | 26.96 | SD | 0.71 | -4.79 | 13.71 |
| EC10 | 0.41 | 7.68 | 28.67 | | | | |

**4.2. Validation of heat fluxes by TSFA**

Fig. 6 provides the turbulent heat flux results calculated by TSFA on September 13, 2012. The spatial distribution of the turbulent heat flux is obvious. The H of buildings and uncultivated land, including the Gobi Desert, barren areas and other deserts, was high, in addition to the LEs of the water and agricultural areas in the oasis. The southern areas of the images show uncultivated barren land bordering the Qilian Mountains that resulted from snowmelt and the downward movement of water. In these areas, the groundwater levels are high and the soil moisture content is approximately 30% based on in situ measurements at a depth of 2 cm. Therefore, the LE is higher in the south than

1    in the southeast desert, although both areas were classified as uncultivated land.

2         Studies have shown that validation methods that consider the source area are more appropriate

3    for evaluating ET models than traditional validation methods based on a single pixel (Jia et al., 2012;

4    Song et al., 2012). In this study, a user-friendly tool presented by Neftel et al. (2008) and based on

5    the Eulerian analytic flux footprint model proposed by Kormann and Meixner (2001) was used to

6    calculate the footprints of the function parameters. The continuous footprint function was dispersed

7    based on the relative weights of the pixels on which the source area fell.

[Figure]

9    **Figure 6.** Maps of the four energy components, (a) Rn, (b) G, (c) H and (d) LE, calculated by TSFA on September

10    13, 2012.

[Figure]

12    **Figure 7.** Scatter plot of the TSFA turbulent heat flux results

The footprint validation results of the TSFA turbulent heat fluxes are shown in Fig. 7 and Table 8. The $R^2$, MBE, and RMSE of the H were 0.61, 0.90 W·m$^{-2}$ and 50.99 W·m$^{-2}$, respectively, and the corresponding terms for the LE were 0.82, -20.54 W·m$^{-2}$ and 71.24 W·m$^{-2}$, respectively. Because the LE was calculated as a residual term, it was impacted by the $R_n$, surface G and H. The errors of all of these variables may contribute to the LE, which complicates the error source of the LE and is discussed in Sects. 4.3 and 4.4.

**Table 8.** In situ validation results of heat flux of TSFA

| | TSFA-H(W·m$^{-2}$) | | | TSFA-LE(W·m$^{-2}$) | | |
|------|------|--------|-------|------|--------|-------|
| date | $R^2$ | MBE | RMSE | $R^2$ | MBE | RMSE |
| 0619 | 0.39 | 44.73 | 66.38 | 0.69 | -44.15 | 80.60 |
| 0630 | 0.73 | 23.71 | 38.96 | 0.88 | -63.81 | 77.83 |
| 0708 | 0.55 | 32.70 | 58.72 | 0.85 | -43.02 | 72.32 |
| 0727 | 0.90 | -34.34 | 43.59 | 0.92 | 26.74 | 57.60 |
| 0803 | 0.80 | -4.77 | 18.92 | 0.78 | -4.58 | 47.86 |
| 0815 | 0.74 | -18.37 | 38.82 | 0.93 | 4.75 | 35.41 |
| 0822 | 0.40 | 31.64 | 66.21 | 0.65 | -44.44 | 93.81 |
| 0829 | 0.79 | 23.01 | 38.36 | 0.79 | -50.45 | 77.99 |
| 0902 | 0.21 | -45.10 | 74.81 | 0.54 | 24.39 | 69.31 |
| 0913 | 0.25 | -9.64 | 41.01 | 0.59 | -59.36 | 82.77 |
| 0914 | 0.31 | -34.11 | 50.88 | 0.47 | 27.99 | 67.50 |

As seen in Fig. 7, most of the H values are small because June, July, August and September constitute the growing season when ET greatly cools the air. The differential temperature between the land surface and air is small, leading to a low H. The points with large H values are influenced by uncultivated land. In our study area, bare soil, the Gobi Desert, and desert areas compose the uncultivated land. The points in the scatter plot with large H values represent desert, where the H values reach approximately 300 W·m$^{-2}$. Some points in the H scatter plot are less than 0 due to inversion from the "oasis effect" or irrigation. For example, HiWATER's soil moisture data show that irrigation occurred on August 22, 2012. Irrigation is the main source of water within the oasis and cools the land surface to temperatures below the air temperature. In addition, irrigation leads to errors in LST retrieval because it increases the atmospheric water vapor content, as discussed in Sect. 4.1. The model error is further analyzed in Sect. 4.4.

**4.3. Comparison between TSFA, TRFA and IPUS**

To verify whether the TSFA method can simulate the heterogeneities of the land surface, the TRFA and IPUS methods were compared for estimating the ET. These three methods were evaluated using (1) validation of TRFA and IPUS based on in situ measurements and (2) qualitative analysis based on the spatial distribution and scatter plots of the four energy balance components.

**4.3.1. Validation of TRFA and IPUS heat fluxes**

Table 9 provides the footprint in situ validation results of the H and LE calculated using the IPUS and TRFA methods. The $R^2$, MBE, and RMSE of the LE between TSFA and TRFA were 0.99, -7.81 W·m$^{-2}$ and 16.48 W·m$^{-2}$, respectively. And the $R^2$, MBE, and RMSE of the LE between TSFA and IPUS were 0.91, -4.10 W·m$^{-2}$ and 51.30W·m$^{-2}$, respectively. Comparing with validation results of TSFA in Table 8, the TSFA method had a better retrieval accuracy than the TRFA method, and

TRFA method was better than the IPUS method on all days, because the MBE and RMSE of TSFA decreased and the $R^2$ of TSFA increased on most days. Table 9 shows that the improvement in the accuracy that resulted from temperature resampling (TRFA) when comparing with the IPUS method was relatively higher than the improvement observed from temperature sharpening (TSFA) when comparing with the TRFA method. Compared with the IPUS method, the TRFA results were similar to the TSFA results since the sub-pixel landscapes and sub-pixel variations of most variables were considered. Thus, TRFA could effectively decrease the scale error that resulted from heterogeneity because the VNIR data of satellite were fully used. However, the performance of the TRFA method is unstable. For example, on August 3 and August 29, the TRFA results were slightly worse than the IPUS results, and the TSFA results were obviously better. This difference occurred because the different sub-pixel landscape temperatures were treated as equal to the values estimated at the 300-m resolution. Thus, when the 300-m-resolution LST has large retrieving errors, the turbulent heat flux retrieving error may be amplified by the sub-pixel landscapes.

**Table 9.** In situ validation results of turbulent heat fluxes of IPUS and TRFA

| date | IPUS-H(W·m⁻²) | | | IPUS-LE (W·m⁻²) | | | TRFA-H (W·m⁻²) | | | TRFA-LE (W·m⁻²) | | |
|---|---|---|---|---|---|---|---|---|---|---|---|---|
| | $R^2$ | MBE | RMSE | $R^2$ | MBE | RMSE | $R^2$ | MBE | RMSE | $R^2$ | MBE | RMSE |
| 0619 | 0.32 | 48.53 | 71.70 | 0.66 | -47.68 | 86.02 | 0.39 | 52.28 | 70.98 | 0.65 | -46.71 | 85.93 |
| 0630 | 0.50 | 41.45 | 67.30 | 0.80 | -81.75 | 102.33 | 0.69 | 42.64 | 60.85 | 0.86 | -78.50 | 93.98 |
| 0708 | 0.34 | 44.17 | 77.45 | 0.63 | -66.75 | 118.63 | 0.44 | 54.20 | 76.00 | 0.82 | -63.82 | 89.11 |
| 0727 | 0.81 | -33.14 | 50.01 | 0.83 | 25.61 | 74.26 | 0.84 | -23.53 | 41.76 | 0.86 | 14.82 | 65.21 |
| 0803 | 0.84 | -5.23 | 33.50 | 0.74 | -3.98 | 60.49 | 0.80 | 7.76 | 37.51 | 0.76 | -18.23 | 62.71 |
| 0815 | 0.64 | -23.28 | 47.89 | 0.85 | 10.32 | 54.98 | 0.70 | -14.77 | 39.99 | 0.89 | 0.59 | 45.22 |
| 0822 | 0.31 | 41.50 | 74.81 | 0.61 | -53.60 | 102.12 | 0.40 | 40.63 | 69.94 | 0.65 | -54.17 | 98.97 |
| 0829 | 0.72 | 27.15 | 44.16 | 0.76 | -54.76 | 83.20 | 0.75 | 30.79 | 44.97 | 0.77 | -59.43 | 86.22 |
| 0902 | 0.28 | -52.44 | 83.25 | 0.51 | 32.89 | 76.48 | 0.21 | -45.77 | 75.84 | 0.52 | 24.37 | 71.69 |
| 0913 | 0.08 | -11.45 | 57.50 | 0.61 | -57.38 | 81.83 | 0.06 | -11.89 | 49.63 | 0.54 | -57.78 | 84.58 |
| 0914 | 0.12 | -36.52 | 67.38 | 0.28 | 19.46 | 89.30 | 0.03 | -34.34 | 64.85 | 0.38 | 25.41 | 75.96 |

Surface landscape inhomogeneity can be classified using two conditions: nonlinear vegetation density variations between sub-pixels (e.g., different types of vegetation mixed with each other or with bare soil) and coarse pixels containing different landscapes (e.g., vegetation or bare soil mixed with buildings or water). And landscapes variation always corresponding to inhomogeneity of surface variable. To evaluate the effects of TSFA, stations with a typical severe heterogeneous surface, such as EC4, a weak heterogeneous surface, such as EC11 and a typical pixel (called "TP" hereafter) at the boundary of the oasis and bare soil (sample 62, line 102 in the image of study area), and a uniform surface, such as EC15, were selected to analyze the temperature sharpening results.

EC4 is used as an example because its land cover and sub-pixel variation of temperature were complicated. Table 11 compares the turbulent heat fluxes calculated using the IPUS, TRFA and TSFA methods. Significant differences were observed between the TSFA and IPUS results and between the TRFA and IPUS results due to the heterogeneity of the surface. The LE calculated using the TSFA method was more consistent with in situ measurements than the LE calculated using the IPUS method because the MBE and RMSE decreased greatly, the $R^2$ increased, and the accuracy was improved by approximately 40 W·m⁻². However, the LE calculated by using the TRFA was more accurate than the LE calculated by using the TSFA, as discussed below.

The H calculated by using the TSFA method was more accurate than the H calculated by using

the TRFA and IPUS methods. The accuracy of the results from the TRFA method was relatively close to the accuracy of the results from the TSFA method because the TRFA method also considers the effects of the heterogeneity of landscapes. In addition, the H values obtained from the TRFA method were always greater than those obtained from the TSFA method. Because the TSFA turbulent heat flux results are the same as the TRFA turbulent heat flux results for buildings and water bodies in our pixel ET algorithm, so the difference between TSFA and TRFA depends on the vegetation and bare soil. And the 300-m-resolution LST is larger than the LST of the sub-pixels, such as pixels containing vegetation or bare soil, for two reasons: (1) the coarse pixels contain buildings and result in a larger 300-m-resolution LST and (2) the LSTs were underestimated at EC4 (as shown in Table 3), which would underestimate the value of $\Delta\widehat{T}_{300}$ in Eq.(3) and, consequently, the sharpening temperature at 30 m and H. Because the LE was calculated as a residual item in the energy balance equation, the errors of the other three energy balance components would accumulate in the LE. At EC4, the $R_n$ was overestimated by approximately 80 W·m$^{-2}$, but the scale effect of $R_n$ was not obvious, as discussed in Sect. 4.1, and the G was overestimated by approximately 20 W·m$^{-2}$. These results would lead to low accuracy of the available energy and overestimate the error by 60 W·m$^{-2}$. As TRFA overestimates H, the underestimation of H in TSFA would result in larger overestimation of LE than TRFA. Consequently, the LE calculated by using the TSFA method is less accurate than the LE calculated by using the TRFA method.

**Table 10.** Comparison of the turbulent heat flux results at EC4

| EC4 | H(W·m$^{-2}$) | | | | LE(W·m$^{-2}$) | | | |
|------|------|------|------|------|------|------|------|------|
| Date | EC | IPUS | TRFA | TSFA | EC | IPUS | TRFA | TSFA |
| 0619 | 150.65 | 105.86 | 154.71 | 142.13 | 278.55 | 402.60 | 344.05 | 357.79 |
| 0630 | 138.32 | 99.91 | 153.53 | 126.88 | 341.98 | 419.83 | 358.12 | 386.07 |
| 0708 | 117.04 | 63.47 | 131.79 | 112.16 | 361.16 | 502.60 | 424.85 | 444.01 |
| 0727 | 136.41 | 4.87 | 85.99 | 72.33 | 306.53 | 543.48 | 452.01 | 467.96 |
| 0803 | 68.97 | 36.51 | 111.73 | 74.76 | 389.63 | 498.21 | 414.67 | 454.23 |
| 0815 | 104.60 | 12.69 | 88.26 | 82.56 | 357.34 | 522.31 | 436.43 | 441.95 |
| 0822 | 125.34 | 85.93 | 120.68 | 93.18 | 318.08 | 415.15 | 370.76 | 400.99 |
| 0829 | 82.93 | 73.06 | 103.84 | 74.76 | 317.68 | 362.04 | 322.77 | 355.16 |
| 0902 | 162.05 | 93.74 | 144.49 | 132.60 | 280.41 | 375.42 | 315.16 | 326.29 |
| 0913 | 119.42 | 151.44 | 157.07 | 130.85 | 263.18 | 234.93 | 222.62 | 249.59 |
| 0914 | 110.02 | 88.24 | 128.37 | 99.33 | 262.33 | 333.82 | 285.04 | 314.91 |

units: W·m$^{-2}$

| | IPUS | | | TRFA | | | TSFA | | |
|----------|------|------|------|------|------|------|------|------|------|
| Variable | $R^2$ | MBE | RMSE | $R^2$ | MBE | RMSE | $R^2$ | MBE | RMSE |
| EC4-H | 0.11 | -44.65 | 61.73 | 0.25 | 5.88 | 26.33 | 0.51 | -16.93 | 26.54 |
| EC4-LE | 0.49 | 99.21 | 119.55 | 0.56 | 42.69 | 62.40 | 0.60 | 63.92 | 76.78 |

Fig. 8 shows that the classes and temperatures of 10×10 sub-pixels at 30 m correspond to the pixels with a resolution of 300 m at the EC tower. In the IPUS upscaling scheme, the 300-m pixels included buildings and maize and vegetable crops at the 30-m resolution and were identified as maize. The canopy height gap between maize and vegetables was large during our study period, resulting in the overestimation of the canopy height. For more details see the error analysis in Sect. 4.4. However, because buildings corresponded with $H = 0.6R_n$ in this paper, ignoring the contributions of buildings would result in the underestimation of H. Fig. 8(a) shows the temperaturesharpening results for the EC4 pixel on August 29. The temperature achieved at a resolution of 300 m was 303.49 K. Compared with the in situ measurement of 313.24 K, the temperature at a resolution of 300 m was underestimated. Even when substituting the in situ temperature into the ET model, the value of H reached 399.60 W·m$^{-2}$ and the LE became 0 W·m$^{-2}$. When substituting the in situ temperature in the TRFA method, H was 396.49 W·m$^{-2}$ and LE was 18.7 W·m$^{-2}$, indicating that the LE was underestimated and the H was overestimated with large errors. After processing by temperature sharpening, the distribution of the temperature at the 30-m resolution agreed with the classification. Temperature sharpening improved the description of heterogeneity based on the thermodynamic-driven force of the turbulent heat flux. These results apply to the ET model with the classification map and high-resolution variables and correspond with more accurate sensible heat flux estimations.

[Figure]

Figure 8. Distribution of classes and temperatures over the extreme heterogeneous surface (a) EC4, homogeneous surface (b) EC15, weak heterogeneous surface (c) EC11 and (d) a typical pixel on August 29, 2012.

The land surface of EC15 was uniform and comprised of pure pixels covered by maize. The temperature distribution at the 30-m resolution was as homogeneous as the land cover, and the variation range of the surface temperature was small (approximately 1.6 K). Table 11 shows the in situ validation results of EC15, for which the overall accuracy is not high due to the low LST retrieval

accuracy on July 8, which is discussed in Sec. 4.4.1. For the homogeneous surface, the gaps between IPUS, TRFA and TSFA were not large (within 10 W·m⁻²), and the accuracy did not improve (MBE and RMSE did not have obvious variations). Statistically sharpening the temperature may increase the uncertainty of the model results for a homogeneous surface; however, this influence could be omitted.

**Table 11.** Comparison of the turbulent heat fluxes results at EC15

| EC15 | H (W·m⁻²) | | | | LE (W·m⁻²) | | | |
|------|-----|------|------|------|-----|------|------|------|
| Date | EC | IPUS | TRFA | TSFA | EC | IPUS | TRFA | TSFA |
| 0619 | 92.55 | 106.60 | 109.25 | 99.81 | 419.47 | 427.19 | 419.99 | 429.98 |
| 0630 | 42.37 | 43.99 | 45.51 | 44.67 | 551.73 | 527.12 | 525.17 | 526.09 |
| 0708 | 18.34 | 217.53 | 235.48 | 209.90 | 620.95 | 425.71 | 397.49 | 424.86 |
| 0727 | 27.68 | 21.22 | 31.11 | 24.30 | 597.76 | 589.58 | 579.43 | 586.47 |
| 0803 | 2.33 | 33.32 | -0.07 | 0.01 | 592.37 | 565.20 | 601.33 | 601.33 |
| 0815 | 48.81 | 32.31 | 46.28 | 44.62 | 553.74 | 561.92 | 547.48 | 549.11 |
| 0822 | 54.59 | 154.34 | 151.77 | 158.60 | 473.68 | 408.37 | 410.80 | 405.07 |
| 0829 | 9.80 | 94.97 | 95.01 | 90.91 | 473.54 | 399.25 | 398.52 | 402.93 |
| 0913 | 176.96 | 265.62 | 209.65 | 257.81 | 307.72 | 165.40 | 221.68 | 173.58 |
| 0914 | 188.34 | 198.15 | 197.04 | 196.60 | 274.98 | 275.07 | 276.05 | 276.56 |

units: W·m⁻²

| | IPUS | | | TRFA | | | TSFA | | |
|----------|-------|-------|-------|-------|-------|-------|-------|-------|-------|
| Variable | $R^2$ | MBE | RMSE | $R^2$ | MBE | RMSE | $R^2$ | MBE | RMSE |
| EC15-H | 0.40 | 40.64 | 74.64 | 0.33 | 45.93 | 80.81 | 0.40 | 40.36 | 72.88 |
| EC15-LE | 0.74 | -52.11 | 83.48 | 0.71 | -48.80 | 82.51 | 0.74 | -49.00 | 81.94 |

The weak heterogeneous land surface EC11 contained barley, maize and vegetables in a coarse pixel with a fractional area of 48:41:1 and was classified as barley at the 300-m resolution. The distributions of the classes and temperatures are shown in Fig. 8(c), and the pixel belongs to the first conditions of heterogeneity (nonlinear vegetation density variation between sub-pixels) that are classified in the introduction. Table 12 shows the in situ validation results of EC11, for which the improvements in the accuracies of H and LE by temperature resampling or sharpening were not as obvious as the improvements at EC4, which contained total different landscapes (the other inhomogeneous scenario in introduction).

Theoretically, the LE pixel values from the TSFA and TRFA methods at EC11 should be smaller than the IPUS values in the energy balance system. The height of maize (range 0.3 ~ 2 m) was usually higher than the height of barley (range 0.9 ~ 1.1 m) in the study area from June to August. Taller vegetation resulted in greater surface roughness and smaller aerodynamic resistance, which led to larger H values, smaller LE values, and vice versa (e.g., vegetables with a canopy height of 0.2 m). When using the TSFA and TRFA methods, patch landscapes consisting of different crops, such as maize and vegetables, were considered. Thus, the LE was smaller than the IPUS LE. On June 19, the canopy height of maize was 0.74 m, which was lower than the canopy height of barley (1 m) and indicated that the H values resulting from the TRFA and TSFA methods were less than the H resulting from the IPUS method. Because our validation method considered the influence of source area, the in situ turbulent heat flux validation results included the effects of neighboring pixels (i.e., on August 3, the turbulent heat flux values of the pixel corresponding with the location of EC11 was only weighted 37% in the source area).

The differences between the TSFA and TRFA methods was small and resulted from the LST differences between the 30-m resolution temperature sharpening results and the LST retrieved at the 300-m resolution and were not evident at EC11. For example, on August 29, the temperature range was 1.4 K, as shown in Fig. 8(c). This temperature was even less than the temperature range at EC15 because the observation system at EC15 was a superstation with a 40-m tall tower that may cause a large shadow and a large temperature range. Hence the temperature sharpening effect is not obvious after aggregating flux at the 300-m resolution under dense vegetation canopies. However, temperature sharpening can still decrease the heterogeneity that results from thermal dynamics.

The excess errors resulted from the relatively low LST accuracy, with $R^2$, MBE, and RMSE values of 0.42, 1.59 K and 2.98 K, respectively. On August 29, the temperature at a resolution of 300 m was 301.6 K, and the observed temperature of the ground was 300.20 K. The LST at the 300-m resolution was slightly overestimated. When the in situ temperature was substituted into the IPUS algorithm, the value of H decreased to 16.06 $W·m^{-2}$ and the LE became 467.43 $W·m^{-2}$. When substituting the in situ temperature in the TRFA scheme, the value of H was 22.43 $W·m^{-2}$ and the LE was 461.58 $W·m^{-2}$, which were more similar to the ground observations.

**Table 12.** Comparison of the turbulent heat flux results at EC11

| EC11 | H($W·m^{-2}$) | | | | LE($W·m^{-2}$) | | | |
|------|------|------|------|------|------|------|------|------|
| Date | EC | IPUS | TRFA | TSFA | EC | IPUS | TRFA | TSFA |
| 0619 | 33.94 | 173.69 | 158.12 | 158.18 | 531.46 | 391.60 | 407.42 | 407.40 |
| 0630 | 25.03 | 3.29 | 23.12 | 21.37 | 635.22 | 586.37 | 566.48 | 568.28 |
| 0708 | 32.29 | 68.17 | 97.16 | 96.13 | 601.98 | 567.73 | 538.77 | 539.81 |
| 0727 | 21.42 | -1.17 | -1.58 | -3.77 | 587.70 | 618.80 | 619.19 | 621.46 |
| 0803 | 7.01 | 24.85 | 20.34 | 19.52 | 614.28 | 575.03 | 585.29 | 586.16 |
| 0815 | 38.94 | 12.51 | 15.52 | 16.02 | 567.07 | 584.31 | 581.31 | 580.82 |
| 0822 | 69.25 | 73.45 | 83.11 | 84.38 | 516.07 | 483.23 | 473.60 | 472.40 |
| 0829 | 29.77 | 48.21 | 60.9 | 60.81 | 473.22 | 427.92 | 415.32 | 415.45 |
| 0902 | 193.97 | 154.58 | 197.01 | 197.49 | 306.62 | 361.96 | 319.54 | 319.03 |
| 0913 | 288.37 | 168.42 | 176.4 | 177.71 | 160.29 | 216.53 | 208.49 | 207.19 |
| 0914 | 240.33 | 268.91 | 256.29 | 256.40 | 199.52 | 156.00 | 168.63 | 168.55 |

units: $W·m^{-2}$

| | IPUS | | | TRFA | | | TSFA | | |
|------|------|------|------|------|------|------|------|------|------|
| Variable | $R^2$ | MBE | RMSE | $R^2$ | MBE | RMSE | $R^2$ | MBE | RMSE |
| EC11-H | 0.61 | -1.07 | 61.31 | 0.57 | -0.36 | 63.24 | 0.67 | -0.21 | 55.50 |
| EC11-LE | 0.88 | -19.83 | 63.16 | 0.89 | -18.12 | 60.02 | 0.90 | -21.29 | 58.11 |

Another typical pixel located at the boundary of the bare soil and the oasis with no flux measurements was used to evaluate the correction effects of landscapes and temperature sharpening. The land surface of TP contained maize, vegetables and bare soil at a fraction of 35:31:34. Table 13 shows that when neither the heterogeneity of the landscape nor the LST are considered, the relative error of LE could reach 180 $W·m^{-2}$. In addition, if only the LST heterogeneity is not considered, the LE relative error could reach 48 $W·m^{-2}$. This result also reveals that the influences of landscape inhomogeneity are greater than the influences of inhomogeneity on the LST in mixed pixels.

**Table 13.** Comparison of the turbulent heat flux results at TP

| | H ($W·m^{-2}$) | | | LE ($W·m^{-2}$) | | |
|------|------|------|------|------|------|------|
| Date | IPUS | TRFA | TSFA | IPUS | TRFA | TSFA |

| 0619 | 186.31 | 149.73 | 143.98 | 321.04 | 358.22 | 364.79 |
| 0630 | 383.65 | 191.59 | 158.79 | 67.03 | 259.36 | 292.89 |
| 0708 | 498.36 | 240.20 | 204.18 | 0.29 | 259.25 | 293.41 |
| 0727 | 276.79 | 136.06 | 84.01 | 206.52 | 347.64 | 402.23 |
| 0803 | 214.14 | 75.45 | 53.72 | 252.37 | 392.08 | 416.41 |
| 0815 | 214.14 | 98.24 | 72.05 | 252.37 | 368.64 | 393.68 |
| 0822 | 436.48 | 369.28 | 276.70 | 0.00 | 67.79 | 162.80 |
| 0829 | 235.29 | 117.16 | 67.21 | 183.62 | 302.41 | 356.75 |
| 0902 | 423.61 | 212.15 | 180.92 | 0.00 | 211.77 | 241.36 |
| 0913 | 338.00 | 285.04 | 216.26 | 0.00 | 53.62 | 122.58 |
| 0914 | 270.44 | 148.20 | 100.19 | 115.19 | 238.43 | 286.51 |

units: $W \cdot m^{-2}$

| Variable | IPUS | | | TRFA | | |
| --- | --- | --- | --- | --- | --- | --- |
| | $R^2$ | MBE | RMSE | $R^2$ | MBE | RMSE |
| TP-H | 0.62 | 174.47 | 185.49 | 0.95 | 42.28 | 48.01 |
| TP-LE | 0.71 | -175.91 | 186.63 | 0.97 | -43.11 | 49.04 |

**4.3.2. Comparison of TRFA and IPUS methods**

Using September 13 as an example, the spatial distributions of the four components of the energy balance calculated by IPUS and TRFA are shown in Fig. 9 and Fig. 10, respectively. TSFA minus IPUS and TSFA minus TRFA, which show the spatial distributions of the heterogeneity effect, are shown in Fig. 11. Scatterplots of TSFA versus IPUS and TRFA are shown in Fig. 12.

Comparing Fig. 6 with Fig. 9, the spatial distribution of the fluxes greatly changes, except for $R_n$. The TSFA results are synoptically smoother than the IPUS results because the land types and temperature distributions in mixed pixels that cannot be considered in IPUS appear in TSFA. For example, the boundary between the oasis and uncultivated land becomes a belt of intermediate G, H and LE because mixed pixels include uncultivated land and vegetation. However, mixed pixels are classified as the dominant land use type in the parameterization process of IPUS. This result overlooks the contributions of heat flux from complex land use types and overestimates or underestimates the heat flux by approximately 50 $W \cdot m^{-2}$. However, TSFA can integrate the effects of these land areas and reveals the relative actual surface conditions. The results of this analysis vary less dramatically than the results obtained using IPUS, as shown in the figures. The results are similar in the oasis.

Based on the overviews presented in Fig. 6 and Fig. 10, the TRFA and TSFA methods are similar. Because the TRFA method considers the sub-pixel landscapes that could be a significant source of error in ET models, the difference between the TSFA and TRFA methods result from the differences between the sharpened and retrieved LST for the sub-pixels at the 300 m resolution. In addition, the bias between the TSFA and TRFA is not as obvious as the bias between the TSFA and IPUS methods, as shown in Fig. 11(c)(d)(e)(f). Furthermore, Fig. 11(f) shows that the LEs calculated by using the TSFA method for most oasis areas were slightly greater than the LEs calculated by using the TRFA method, which were approximately 20 $W \cdot m^{-2}$.

The quadrangular with a relatively unstable bias shown in Fig. 11(a) is caused by the $L_d$ that was calculated from the MOD05 water vapor product which exists quadrangular even after preprocessing the instrument malfunction gap. From Fig. 11, the differences of the four energy components

  of the pure pixels between these three methods are within 5 W·m⁻², and the mixed pixels have dif-

  ferent ranges.

[Figure]

**Figure 9.** Maps of the four energy components, (a) $R_n$, (b) G, (c) H and (d) LE, calculated using the IPUS method on September 13, 2012.

[Figure]

2 **Figure 10.** Maps of the four energy components, (a) $R_n$, (b) G, (c) H and (d) LE, calculated using the TRFA method

3 on September 13, 2012.

[Figure]

2  **Figure 11.** Maps of the bias of the energy balance components calculated using the TSFA method minus the IPUS

3  method: (a) $R_n$, (b) G, (c) H, (d) LE, TSFA minus TRFA: (e) H and (f) LE.

[Figure]

2 **Figure 12.** Scatter plots between the TSFA and IPUS results: (a) $R_n$, (b) G, (c) H and (d) LE; TSFA and TRFA (e)

3 $R_n$, (f) G, (g) H and (h) LE. MBD and RMSD are the mean bias deviation and root mean square deviation between

4 the TSFA and IPUS results, respectively.

5    Fig. 12 shows the scatter plots between the results from the TSFA method and the other two

methods for all four energy balance components in the image. Fig. 11(a)(e) shows that $R_n$ does not vary much between the three methods because the scatter is centralized around the 1:1 line. However, regarding the spatial scale effect, the differences in $G$, $H$ and $LE$ calculated by using the IPUS and TSFA methods are obvious: the scatter plots are dispersed at the mixed pixels, and the differences between the TRFA and TSFA results are relatively smaller. When using the TSFA method, the temperature sharpening results can be divided into results that are higher and lower than the LST retrieved at 300 m. Compared with the LST retrieved at 300 m when using the TRFA method, a higher LST would be counterbalanced by a lower LST when calculating H. Thus, the heterogeneous effect of temperature is neutralized in this case. This observation potentially resulted from the temperature sharpening algorithms because they tend to overestimate the sub-pixel LST for cooler landscapes and underestimate the sub-pixel LST for warmer areas in the image (Kustas et al., 2003).

However, $LE$ is calculated as a residual; thus, the difference of $LE$ resulted from the $G$ and $H$. When the 300 m mixed pixels contain various types of land, they may be categorized as one type of land because of the coarse resolution of the IPUS results and because a single temperature value is used to evaluate the thermal dynamic effects when using the TRFA method. Pixels with highly different $G$, $H$ and $LE$ values are mainly distributed near the mixed pixels, as shown in Fig. 10. An explanation for these deviations is provided below.

The parameterization of $G$ and $H$ is based on the land cover type. For example, for buildings, $G = 0.4R_n$(Kato and Yamaguchi, 2005) (which is usually greater than the $G$ of vegetation and bare soil deduced from Eq.(9)) and $H = 0.6R_n$, and for water, $G = 0.226R_n$ and $LE = R_n - G$. From the land cover map shown in Fig. 4, four major classes exist in the study area, buildings with a high $H$, uncultivated land with a relatively high $H$, cropland with a relatively low $H$, and water with $H = 0$.

(1) If a pixel contains cropland and buildings and is categorized as cropland the building area within the pixel is ignored when using the IPUS method. In this case, $G$ and $H$ are underestimated and $LE$ is overestimated. In addition, after considering the landscapes by using the TRFA method, the $LE$ is underestimated and $H$ is overestimated because the pixels contain buildings that are still reflected indistinctly by LST at 300 m because the detailed temperature heterogeneity cannot be represented by the TRFA method. These points are shown in green in Fig. 11. However, if the pixel is categorized as built-up, the building area within a pixel is exaggerated, which causes $G$ and $H$ to be overestimated and $LE$ to be underestimated when using the IPUS method. This situation is similar to the points shown in green for the TRFA results and is shown by red points in Fig. 11.

(2) At the margin of the oasis and uncultivated land, the mixed pixels are divided into cropland, the $LE$ is overestimated, $G$ and $H$ are underestimated in the IPUS method, and vice versa. The $LE$ is also overestimated in the pixels containing water and other types of land cover (generally bare soil in our study area). These pixels are categorized as water and are shown as blue points in Fig. 11. Some of the blue $LE$ points calculated by using the TSFA method are slightly smaller than those calculated by using the TRFA method for pixels containing vegetation, and the temperature of vegetation is lower than the temperature of water bodies at noon in our study area.

(3) In mixed pixels that contain various crops, such as maize and vegetables, the $LE$ is underestimated if the area of maize within the pixel is overestimated because the canopy height of the maize would be taller than that of vegetables, which would result in the overestimation of $H$ when using the IPUS and TRFA methods. In addition, $G$ depends on the FVC of the crops when using the IPUS method, and is nearly the same as the values of $G$ obtained when using the TRFA and

TSFA methods because it depends on $R_n$.

At the study area scale, we compared TRFA and IPUS to quantify the ability of the TSFA method to simulate the heterogeneities of the land surface on September 13 (see Table 14). For pure pixels, the LE biases among the IPUS, TRFA and TSFA methods were small. In mixed pixels, the LE bias between the TSFA and IPUS methods varied from 35.36 to 65.66 W·m$^{-2}$, and the bias between the TSFA and TRFA methods varied from 4.41 to 22.53 W·m$^{-2}$. More class types in mixed pixels correspond to larger biases. Table 15 shows the bias of the mixed pixels that contain buildings and bare soil between the three methods. For mixed pixels with buildings, the IPUS and TRFA methods usually underestimated the LE, with a large bias compared with the TSFA method. For mixed pixels without buildings and bare soil, the bias between TRFA (or IPUS) and TSFA was relatively small, which indicates that the landscape and temperature inhomogeneity are accounted for by the TSFA method. The aforementioned analyses demonstrate that the TSFA method can consider the heterogeneous effects of mixed pixels.

**Table 14.** Comparison of the latent heat flux in pixels containing different numbers of class types

| Number of class types in pixels | IPUS (W·m$^{-2}$) | | | TRFA (W·m$^{-2}$) | | | Pixel number |
|---|---|---|---|---|---|---|---|
| | $R^2$ | MBD | RMSD | $R^2$ | MBD | RMSD | |
| 1 | 1.00 | 0.21 | 0.21 | 1.00 | 0.05 | 0.61 | 11,398 |
| 2 | 0.85 | -7.18 | 35.36 | 1.00 | -0.35 | 4.41 | 8212 |
| 3 | 0.66 | -2.32 | 52.55 | 0.98 | -7.33 | 12.56 | 4762 |
| 4 | 0.49 | 1.88 | 65.66 | 0.96 | -11.56 | 16.55 | 2824 |
| 5 | 0.98 | -30.92 | 62.69 | 0.96 | -16.90 | 22.53 | 4 |

Notes: Number of class types in mixed pixels means the number of classification types that were contained in the pixels. For example, 1 represents the pure pixels, 2 represents mixed pixels containing two land use types, etc. MBD and RMSD are the mean bias deviation and root mean square deviation, respectively, between the TSFA results and the TRFA and IPUS results.

**Table 15.** Comparison of the latent heat fluxes of typical mixed pixels

| Types of mixed pixels | IPUS (W·m$^{-2}$) | | | TRFA (W·m$^{-2}$) | | | Pixel number |
|---|---|---|---|---|---|---|---|
| | $R^2$ | MBD | RMSD | $R^2$ | MBD | RMSD | |
| mixed pixels contain buildings | 0.58 | -1.02 | 61.94 | 0.97 | -9.64 | 14.66 | 4918 |
| mixed pixels do not contain buildings | 0.81 | -5.49 | 39.21 | 0.99 | -2.12 | 7.60 | 10,884 |
| mixed pixels contain bare soil | 0.73 | -1.52 | 49.04 | 0.98 | -5.96 | 11.86 | 9049 |
| mixed pixels do not contain bare soil | 0.65 | -7.55 | 45.28 | 0.98 | -2.46 | 7.83 | 6753 |

Considering the landscapes and inhomogeneous distribution of LST, the TSFA method ensures that none of the end members (30 m pixel) are ignored or exaggerated. Thus, the distribution of LE calculated using the TSFA method is smoother and more rational than the distributions of LE calculated using the other methods. At the regional scale, the TSFA method describes the heterogeneity of the land surface more precisely. And how much the estimation accuracy can be improved is discussed in the following sections.

**4.4. Error analysis**

Since LE is calculated as a residual term in the energy balance equations, the sensitivity of H was analyzed first. Land surface variables (including LST, LAI, canopy height, and FVC) and meteorological variables (including wind speed, air temperature, air pressure and relative humidity) are needed to estimate H in this paper. To locate the error source when retrieving H, a sensitivity

analysis was performed by adding errors at each 10% step (except LST). Fig. 13 presents the sensitivity analysis results: LST = 303.9 K (ranging from 298.4~309.4 K with a step size of 0.5 K), LAI=1.4 (ranging 0.14~2.66 with a step size of 0.14), canopy height equals 1 m (ranging 0.1~1.9 m with a step size of 0.1 m), FVC=0.5, wind speed u=2.48 m·s$^{-1}$, air temperature Ta=297.9 K, air pressure = 97.2 kPa, and RH=40.29%. In addition, the land use type is maize, and the reference H is 230.2 W·m$^{-2}$.

[Figure]

**Figure 13.** Sensitivity analysis of the surface variables for sensible heat flux

The air pressure is stable over a short period and has little effect on the ET results. Although "excess resistance" was calculated from the friction velocity, the meteorological data were provided by ground observations; thus, the meteorological data are relatively accurate. As shown in Fig. 13, LAI, canopy height and LST are sensitive variables.

The parameterization of the momentum roughness length indicates that LAI is sensitive to H, with decreasing sensitivity when the LAI is greater than 1. When the LAI is less than 1, the momentum roughness length increases as the LAI increases and the H and turbulent exchange are enhanced. However, when the LAI is greater than 1, the plant canopy could be regarded as a continuum that is not sensitive to H. Because our study area is dominated by agriculture and the study period was from July to September, the crops in the HRB middle stream grew quickly, so the LAI was generally greater than 1. Thus, LST and canopy height are the main sources of error.

**4.4.1. The error of LST**

As shown in Fig. 13 using monitoring data, a 1 K LST bias would result in 21% error of H, about 48.3 W·m$^{-2}$. However, the sensitivity of the LST is unstable and depends on the strength of the turbulence. The strength of the turbulence determines the mass and energy transport and the resistance of heat transfer, which influences the sensitivity of the LST. A weaker turbulence corresponds to a weaker LST sensitivity and vice versa.

The influence of LST was analyzed based on the sensitivity analysis and LE results. We chose homogeneous stations to analyze the LST error so that other errors could be ignored. These results are shown in Table 16. The LE results obtained from the observed LST are consistent with the in situ observations but have less bias. The LE was overestimated when the LST was underestimated and vice versa. Because the magnitude of LE was greater than H, the relative error of LE was less than the relative error of H. However, 1 K of LST bias would result in an average LE error of 30 W·m$^{-2}$, which is consistent with the sensitivity analysis of H shown in Fig. 13. Specifically, 1 K of LST bias would result in LE biases of 8.7 W·m$^{-2}$ (in desert, SSW) to 84.4 W·m$^{-2}$ (in oasis, EC8), which may prove that the sensitivity of LST is unstable.

**Table 16.** Results of the LST error analyses at the homogeneous stations

| Station | Date | retrieved LST (K) | observed LST (K) | LST bias (K) | EC-LE (W·m$^{-2}$) | LE from retrieved LST (W·m$^{-2}$) | LE from observed LST (W·m$^{-2}$) | LE relative error (%) | H relative error (%) |
|---------|------|-------------------|------------------|--------------|---------------------|-------------------------------------|-------------------------------------|-----------------------|----------------------|
| EC8 | 0619 | 304.92 | 301.74 | 3.18 | 415.89 | 321.80 | 399.78 | -22.62 | 68.58 |
| EC7 | 0630 | 302.5 | 299.35 | 3.15 | 611.22 | 453.59 | 557.97 | -25.79 | 886.08 |
| EC10 | 0708 | 303.58 | 300.5 | 3.08 | 617.83 | 504.44 | 549.53 | -18.35 | 390.24 |
| EC15 | 0708 | 303.55 | 300.13 | 3.42 | 620.95 | 425.71 | 603.73 | -31.44 | 450.57 |
| EC7 | 0727 | 298.87 | 300.55 | -1.68 | 577.59 | 643.56 | 566.62 | 11.42 | -132.47 |
| SSW | 0727 | 307.86 | 316.82 | -8.96 | 119.35 | 238.07 | 78.43 | 99.48 | -60.36 |
| EC2 | 0822 | 299.79 | 298.05 | 1.74 | 501.12 | 411.43 | 486.28 | -17.90 | 67.20 |
| EC8 | 0822 | 299.58 | 297.77 | 1.81 | 543.56 | 416.23 | 467.42 | -23.42 | 88.59 |
| EC10 | 0822 | 301.61 | 298.04 | 3.57 | 503.82 | 398.82 | 513.67 | -20.84 | 138.61 |
| EC15 | 0822 | 300.59 | 297.69 | 2.9 | 473.68 | 408.37 | 495.49 | -13.79 | 129.60 |
| EC8 | 0829 | 301.54 | 300.44 | 1.1 | 514.31 | 402.93 | 428.78 | -21.66 | 63.91 |
| EC15 | 0829 | 301.41 | 299.84 | 1.57 | 473.54 | 399.25 | 459.66 | -15.69 | 182.34 |
| SSW | 0902 | 304.9 | 303.42 | 1.48 | 226.88 | 127.96 | 149.83 | -43.60 | 11.36 |

Notes: "LST bias" is calculated as the retrieved LST minus the observed LST; "EC-LE" is the in situ latent heat flux; "LE relative error" is the relative error between the retrieved and observed LST and is expressed as ((LE from retrieved LST)-(LE from observed LST))/(LE from observed LST)×100%, "H relative error" is calculated in the same way.

**4.4.2. The error of canopy height**

In this paper, canopy height was obtained from a phenophase and classification map. Thus, the accuracy of the canopy height was mainly dependent on the classification accuracy and plant growth state. Even within the same region, the canopy height of a crop can differ due to differences in seeding times and soil attributes, such as soil moisture and fertilization.

The land use at EC17 was orchard. However, in our land classification map, the land use at EC17 was other crops, which includes vegetables and orchards. Thus, it was difficult to set the canopy height. In our study area, most of the other crops were vegetables (canopy height of 0.2 m), and the height of the orchard was approximately 4 m; thus, a value of 0.2 m would overestimate the LE. The LE results with incorrect canopy heights and correct orchard canopy heights at EC17 are shown in Table 17. The days of large LST bias were removed, and the bias between the model and ground observations decreased. The excess errors were caused by errors in the LST and other land use types, such as buildings and maize in the mixed pixels.

**Table 17.** Results of the canopy height error analyses at EC17

| Date | EC-LE (W·m$^{-2}$) | LE from incorrect canopy height (W·m$^{-2}$) | LE from correct canopy height (W·m$^{-2}$) | LE relative error (%) |
|------|---------------------|-----------------------------------------------|---------------------------------------------|------------------------|
| 20120815 | 499.62 | 562.06 | 521.83 | 7.71 |
| 20120822 | 366.27 | 519.01 | 396.54 | 30.88 |
| 20120902 | 377.96 | 471.68 | 336.52 | 40.16 |
| 20120914 | 465.38 | 352.78 | 258.07 | 36.70 |

Except for the error source discussed before, the following sources of error were unavoidable:
(1) Although the remotely sensed turbulent heat flux is instantaneous, the EC data are averaged

over time. Thus, the time scales do not match in the validation.

(2) The calibration coefficient of HJ-1B satellite's CCD and IRS drifts because of the aging instruments.

(3) Geometric correction causes half-pixel bias equal to or less than the deviation of the artificially subjective interpretation.

A one-source model and simplified parameterization schemes for determining surface roughness lengths and heat transfer coefficients were used in this paper. The one-source model combines soil evaporation and plant transpiration and assumes that SPAC is a one-source continuum for calculating ET. This assumption is reasonable when the surface is densely covered by vegetation but relies on the accuracy of the difference between the LST and air temperature, as previously mentioned. When a one-source model is applied to an area covered by sparse vegetation, such a semi-arid or arid areas, this assumption is irrational.

**5. Discussion**

As mentioned in the results and analysis, the TSFA method describes the surface heterogeneity more clearly than the IPUS and TRFA methods. The IPUS method aggregates the land surface parameters achieved by CCDs from 30 m to 300 m, which results in the loss of surface information and leads to the scale effect. Although the TRFA method uses VNIR information and partially decreases the heterogeneity caused by landscape and VNIR variables, it treats the pivotal variable LST as homogeneous within mixed pixels, which results in considerable error. In summary, the superiority of the TSFA method is described as follows:

(1) The temperature sharpening algorithm in TSFA uses the NDVI at 30 m to monitor the LST at 30 m and is capable of decreasing the influences of the heterogeneity of the LST, which agrees with previous research results (Kustas et al., 2003; Bayala and Rivas, 2014; Mukherjee et al., 2014). As analyzed in Sect 4.3, the ignorance of the heterogeneity of LST in mixed pixels is irrational and causes errors when estimating ET.

(2) In the one-source energy balance model, different landscapes used different parameterization schemes. In the IPUS method, a single land cover type is assigned to a mixed pixel, which results in a large error. However, the TSFA method is used to calculate the surface flux at 30 m and is aggregated to 300 m using the area-weighting method, which considers all of the sub-pixel landscapes and improves the retrieval accuracy.

Some problems exist in the temperature sharpening algorithms. The temperature-downscaling method used in this paper caused boxy anomalies in parts of the sharpened-temperature field because of the constant residual term, $\Delta \widehat{T}_{300}$, in Eq. (3) within large pixels. This situation also occurred in the temperature sharpening algorithm proposed by Agam et al. (2007). In addition, our temperature sharpening algorithm tends to overestimate the sub-pixel LST for cooler landscapes and underestimate the sub-pixel LST for warmer areas (Kustas et al., 2003). This inaccurate estimation causes errors that are difficult to evaluate when estimating turbulent heat flux. For example, the small turbulent heat flux bias between TSFA and TRFA was caused by the counterbalanced effect as analyzed in Sect 4.3.1. The evaluation of more temperature sharpening algorithms under heterogeneous surfaces with real datasets when applied in ET models would be helpful (Ha et al., 2011).

Our surface variable retrieval methods were validated against other areas considered in remote sensing measurement campaigns. For example, the albedo algorithm was previously applied to retrieve Global Land Surface Satellite (GLASS) Products (Liang et al., 2014), the LST retrieval algorithm was validated in the Haihe River Basin in northern China (Li et al., 2011), and the soil heat flux correction algorithm was validated in the GAME-Tibet campaign (Yang and Wang, 2008). Since the surface of the Heihe River Basin is extreme heterogeneous, additional comparisons of our algorithm in other areas of research would be better.

In addition, to correct the discrepancy between remotely sensed radiative surface temperature and aerodynamic temperature at the source of heat transport, a brief and well-performed parameterization scheme (under uniformly flat plant surface) of "excess" resistance was used to calculate the aerodynamic resistance of heat transfer (Jiao et al., 2014). Since the objects of our study are mixed pixels, more parameterization methods should be compared to select the optimum method.

Because of the sensitive variables of the one-source energy balance model used in this paper, the accuracy of the LST and canopy height greatly influenced the turbulent heat flux. HJ-1B IRS is a single-thermal channel, the single-channel LST-retrieving algorithm may be unstable under wet atmospheric conditions (water vapor contents higher than 3 $g/cm^2$) (H. Li et al., 2010), which may create a bottleneck for ET estimations by HJ-1B. The canopy height is a priori knowledge based on phenophase classifications and would influence the accuracy of the surface roughness, the length of a heterogeneous surface or the seasonal transition. Multi-source remote sensing data could be used to improve the accuracy of calibrations and land surface variable estimations. Active microwave and LiDAR data (Colin and Faivre, 2010) could be used to obtain the canopy height, which would decrease the dependence on the accuracy of the classification.

The energy balance closure has significant influence on evaluation of the model calculated heat flux results. In our study area, the EC energy balance closure ratio was greater than 0.75 (Liu et al., 2011b). Studies have shown that the not-captured low-frequency eddies(Von Randow et al., 2008), extension of averaging time (Charuchittipan et al., 2014), and lack of an accurate accounting of heat storage terms(Meyers and Hollinger, 2004) are potential reasons for the energy imbalance and so forth. The conserving Bowen ratio and residual closure technique are often used to force energy balance. We chose the residual closure at last because the conserving Bowen ratio method conducted irrational sensible heat flux due to small or negative Bowen ratios (large LEs due to "oasis effect") in the oasis-desert system. Energy balance closure was problematic at times for turbulent flux system and tended to be associated with significant discrepancies in LE (Prueger et al., 2005).

Since a footprint model was used in the validation, the footprints discrepancy between in situ measurements and remote sensing pixel may cause biases. For example, model validation results were calculated by the relative weights of the footprint model, and multiply heat flux results of the coarse pixels which were covered by source area from upwind direction. However, the heat fluxes of coarse pixels included the contribution of not-overlapped sub-pixels within the coarse pixel. Influenced by the heterogeneity of underlying surface, it would cause uncertainties in the validation.

**6. Conclusion**

We studied the effects of surface heterogeneity in ET estimation by the IPUS, TRFA and TSFA methods over heterogeneous surface based on spatial resolution characteristic of different satellites, and applied them to HJ-1B satellite data based on operational satellites' instrumental characteristics.

Compared with the IPUS and TRFA methods, the TSFA method is more consistent with in situ measurements (energy balance forced by residual closure method) according to the footprint validation results. If ET estimating algorithm does not consider surface heterogeneity at all (i.e. IPUS),

it would cause significant error (i.e. 186 W·m$^{-2}$) of heat fluxes. If ET estimating algorithm does not consider heterogeneity of LST only (i.e. TRFA), it would cause non-negligible error (i.e. 49 W·m$^{-2}$) in heat fluxes calculating. The TSFA method reduces the uncertainties produced by surface landscapes and LST inhomogeneity. As a sensitive variable of the ET model, canopy height is mainly determined by classification, and the application of classification at a 30-m resolution can improve the accuracy of the canopy height. As another sensitive variable, the sharpened surface temperature at a resolution of 30 m decreases the thermodynamic uncertainty caused by land surface heterogeneities. The TSFA method can capture the heterogeneities of the land surface and integrate the effects of landscapes in mixed pixels that are neglected at coarse spatial resolutions.

HJ-1B satellite data are advantageous because of their high spatiotemporal resolution and free access. Because the satellites are still in operation, long-term data have promising applications for monitoring energy budgets.

**Acknowledgements**

The authors would like to thank Dr. Franz Meixner of the Max Planck Institute for Biogeochemistry in Mainz and Dr. Albrecht Neftel of the Agroscope Research Station in Zurich for providing the footprint calculation tool. We are grateful to the research team of Professor ShaoMin Liu at Beijing Normal University for providing eddy covariance and automatic meteorological station data. This study was supported by the Special Fund from the Chinese Academy of Sciences (KZZD-EW-TZ-18) and the Chinese Natural Science Foundation Project (grant no. 41371360). Generous help for revising the paper was provided by the editors and reviewers.

**Appendix**

| Notation | | Application (for calculating) |
|---|---|---|
| 6S radiation transfer mode | Second Simulation of a Satellite Signal in the Solar Spectrum radiation transfer mode | Albedo, $S_d$ |
| α | Surface broadband albedo | $S_d$, $R_n$ |
| ABT | At-nadir brightness temperature (K) | $L_d$ |
| AMS | Automatic meteorological station | |
| AOD | Aerosol optical depth | $S_d$ |
| BRDF | Bidirectional reflectance distribution function | α |
| CCD | Charge-coupled device | |
| CV | Coefficient of variation | Sharpened LST |
| EC | Eddy covariance | |
| FVC | Fractional vegetation coverage | LSE, G, LAI |
| G | Soil heat flux (W·m$^{-2}$) | |
| $G(\theta)$ | G function, Foliage angle distribution | LAI |
| H | Sensible heat flux (W·m$^{-2}$) | |
| HRB | The Heihe River Basin | |
| IPUS | Input parameter upscaling scheme | |
| IRS | Infrared scanner | |
| $L_d$ | Downward atmospheric longwave radiation (W·m$^{-2}$) | $R_n$ |
| LSE/ε | Land surface emissivity | LST |

| | | |
|---|---|---|
| $\varepsilon_v/\varepsilon_g$ | The vegetation/ground emissivity | |
| LST/$T_{rad}$ | Land surface temperature/Surface radiation temperature (K) | H |
| MBE/MBD | Mean bias error (deviation) | |
| NCEP | National Centers for Environmental Prediction | LST |
| NDVI/$NDVI_{30}$ | Normalized difference vegetation index | FVC, Sharpened LST |
| $NDVI_{300}$ | 300-m NDVI aggregated from NDVI | Sharpened LST |
| $NDVI_s/NDVI_v$ | Normalized difference vegetation index of bare soil/fully covered vegetation | FVC |
| $P(\theta)$ | Angular distribution of the canopy gap fraction | LAI |
| $r_a$ | Aerodynamic resistance (s·m$^{-1}$) | H |
| $r_{ex}$ | "Excess" resistance (s·m$^{-1}$) | heat transfer resistance |
| $R_n$ | Net radiation (W·m$^{-2}$) | |
| RMSE/RMSD | Root mean square error (deviation) | |
| $S_d$ | Downward shortwave radiation (W·m$^{-2}$) | $R_n$ |
| SPAC | The soil-plant-atmosphere continuum | |
| SZA | Solar zenith angle | $S_d$ |
| $T_a$ | Air temperature (K) | H |
| $T_{aero}$ | Aerodynamic surface temperature obtained by extrapolating the logarithmic air-temperature profile to the roughness length for heat transport (K) | H |
| TOA | Top of the atmosphere | |
| TOMS | Total ozone mapping spectrometer | $S_d$ |
| TRFA | Temperature resampling and flux aggregation | |
| TSFA | Temperature sharpening and flux aggregation | |
| ULR | Upward longwave radiation (W·m$^{-2}$) | $R_n$ |
| USR | Upward shortwave radiation (W·m$^{-2}$) | $R_n$ |
| VNIR | Visible/near-infrared | |
| VZA/$\theta$ | View zenith angle | $L_d$, LAI |

**References**

Agam, N., Kustas, W. P., Anderson, M. C., Li, F., and Colaizzi, P. D.: Utility of thermal sharpening over Texas high plains irrigated agricultural fields, J. Geophys. Res.-Atmos., 112, D19110, doi:10.1029/2007JD008407, 2007.

Allen, R., Tasumi, M., and Trezza, R.: Satellite-Based Energy Balance for Mapping Evapotranspiration with Internalized Calibration (METRIC)—Model, J. Irrig. Drain. E.-ASCE, 133, 380-394, 10.1061/(ASCE)0733-9437(2007)133:4(380), 2007.

Ambast, S. K., Keshari, A. K., and Gosain, A. K.: An operational model for estimating Regional Evapotranspiration through Surface Energy Partitioning (RESEP), Int. J. Remote Sens., 23, 4917-4930, 10.1080/01431160110114501, 2002.

Bastiaanssen, W. G. M., Menenti, M., Feddes, R. A., and Holtslag, A. A. M.: A remote sensing surface energy balance algorithm for land (SEBAL). 1. Formulation, J. Hydrol., 212–213, 198-212, http://dx.doi.org/10.1016/S0022-1694(98)00253-4, 1998.

Bateni, S. M., and Liang, S.: Estimating surface energy fluxes using a dual-source data assimilation approach adjoined to the heat diffusion equation, Journal of Geophysical Research: Atmospheres, 117, D17118, 10.1029/2012JD017618, 2012.

Bayala, M. I., and Rivas, R. E.: Enhanced sharpening procedures on edge difference and water stress index basis over heterogeneous landscape of sub-humid region, The Egyptian Journal of Remote Sensing and Space Science, 17, 17-27, http://dx.doi.org/10.1016/j.ejrs.2014.05.002, 2014.

Bin, L., and Roni, A.: The Impact of Spatial Variability of Land-Surface Characteristics on Land-Surface Heat Fluxes, Journal of Climate, 7, 527-537, 10.1175/1520-0442(1994)007<0527:TIOSVO>2.0.CO;2, 1994.

Blyth, E. M., and Harding, R. J.: Application of aggregation models to surface heat flux from the Sahelian tiger bush, Agricultural and Forest Meteorology, 72, 213-235, http://dx.doi.org/10.1016/0168-1923(94)02164-F, 1995.

Bonan, G. B., Pollard, D., and Thompson, S. L.: Influence of Subgrid-Scale Heterogeneity in Leaf Area Index, Stomatal Resistance, and Soil Moisture on Grid-Scale Land–Atmosphere Interactions, Journal of Climate, 6, 1882-1897, 10.1175/1520-0442(1993)006<1882:IOSSHI>2.0.CO;2, 1993.

Bonan, G. B., Levis, S., Kergoat, L., and Oleson, K. W.: Landscapes as patches of plant functional types: An integrating concept for climate and ecosystem models, Global Biogeochemical Cycles, 16, 5-1-5-23, 10.1029/2000GB001360, 2002.

Cammalleri, C., Anderson, M. C., Gao, F., Hain, C. R., and Kustas, W. P.: A data fusion approach for mapping daily evapotranspiration at field scale, Water Resour. Res., 49, 4672-4686, 10.1002/wrcr.20349, 2013.

Charuchittipan, D., Babel, W., Mauder, M., Leps, J.-P., and Foken, T.: Extension of the Averaging Time in Eddy-Covariance Measurements and Its Effect on the Energy Balance Closure, Boundary-Layer Meteorology, 152, 303-327, 10.1007/s10546-014-9922-6, 2014.

Chen, J.: An important drawback and the improvement of the evapotranspiration model with remote sensing, Chinese Sci. Bull., 6, 454-457, 1988.

Chen, J., Chen, B. Z., Black, T. A., Innes, J. L., Wang, G. Y., Kiely, G., Hirano, T., and Wohlfahrt, G.: Comparison of terrestrial evapotranspiration estimates using the mass transfer and Penman-Monteith equations in land surface models, J. Geophys. Res.-Biogeo., 118, 1715-1731, 10.1002/2013jg002446, 2013.

Chen, W., Cao, C., He, Q., Guo, H., Zhang, H., Li, R., Zheng, S., Xu, M., Gao, M., Zhao, J., Li, S., Ni, X., Jia, H., Ji, W., Tian, R., Liu, c., Zhao, Y., and Li, J.: Quantitative estimation of the shrub canopy LAI from atmosphere-corrected HJ-1 CCD data in Mu Us Sandland, Sci. China Earth Sci., 53, 26-33, 2010.

Choudhury, B. J., Reginato, R. J., and Idso, S. B.: An analysis of infrared temperature observations over wheat and calculation of latent heat flux, Agr. Forest Meteorol., 37, 75-88, http://dx.doi.org/10.1016/0168-1923(86)90029-8, 1986.

Choudhury, B. J., and Monteith, J. L.: A four-layer model for the heat budget of homogeneous land surfaces, Q. J. Roy. Meteor. Soc., 114, 373-398, 10.1002/qj.49711448006, 1988.

Cleugh, H. A., Leuning, R., Mu, Q. Z., and Running, S. W.: Regional evaporation estimates from flux tower and MODIS satellite data, Remote Sens. Environ., 106, 285-304, http://dx.doi.org/10.1016/j.rse.2006.07.007, 2007.

Colin, J., and Faivre, R.: Aerodynamic roughness length estimation from very high-resolution imaging LIDAR observations over the Heihe basin in China, Hydrol. Earth Syst. Sci., 14, 2661-2669, 10.5194/hess-14-2661-2010, 2010.

Ershadi, A., McCabe, M. F., Evans, J. P., Mariethoz, G., and Kavetski, D.: A Bayesian analysis of sensible heat flux estimation: Quantifying uncertainty in meteorological forcing to improve model prediction, Water Resources Research, 49, 2343-2358, 10.1002/wrcr.20231, 2013a.

Ershadi, A., McCabe, M. F., Evans, J. P., and Walker, J. P.: Effects of spatial aggregation on the multi-scale estimation of evapotranspiration, Remote Sens. Environ., 131, 51-62, http://dx.doi.org/10.1016/j.rse.2012.12.007, 2013b.

Fan, L., Xiao, Q., Wen, J., Liu, Q., Tang, Y., You, D., Wang, H., Gong, Z., and Li, X.: Evaluation of the Airborne CASI/TASI Ts-VI Space Method for Estimating Near-Surface Soil Moisture, Remote Sensing, 7, 3114-3137, 10.3390/rs70303114, 2015.

Fisher, J. B., Tu, K. P., and Baldocchi, D. D.: Global estimates of the land–atmosphere water flux based on monthly AVHRR and ISLSCP-II data, validated at 16 FLUXNET sites, Remote Sens. Environ., 112, 901-919, http://dx.doi.org/10.1016/j.rse.2007.06.025, 2008.

Ha, W., Gowda, P. H., and Howell, T. A.: Downscaling of Land Surface Temperature Maps in the Texas High Plains with the TsHARP Method, GIScience & Remote Sensing, 48, 583-599, 10.2747/1548-1603.48.4.583, 2011.

Ha, W., Gowda, P. H., and Howell, T. A.: A review of downscaling methods for remote sensing-based irrigation management: part I, Irrigation Science, 31, 831-850, 10.1007/s00271-012-0331-7, 2013.

He, L., Chen, J. M., Pisek, J., Schaaf, C. B., and Strahler, A. H.: Global clumping index map derived from the MODIS BRDF product, Remote Sens. Environ., 119, 118-130, http://dx.doi.org/10.1016/j.rse.2011.12.008, 2012.

Hu, Y., Gao, Y., Wang, J., Ji, G., Shen, Z., Cheng, L., Chen, J., and Li, S.: Some achievements in scientific research during HEIFE, Plateau Meteorology, 13, 225-236, 1994.

Jia, Z. Z., Liu, S. M., Xu, Z. W., Chen, Y. J., and Zhu, M. J.: Validation of remotely sensed evapotranspiration over the Hai River Basin, China, Journal of Geophysical Research: Atmospheres, 117, D13113, 10.1029/2011JD017037, 2012.

Jiao, J. J, Xin, X. Z., Yu S. S., Zhou, T. and Peng, Z. Q.: Estimation of surface energy balance from HJ-1 satellite data. Journal of Remote Sensing, 18(5), 1048-1058, doi:10.11834/jrs.20143322, 2014.

Jiang, B., Liang, S., Townshend, J. R., and Dodson, Z. M.: Assessment of the Radiometric Performance of Chinese HJ-1 Satellite CCD Instruments, IEEE J. Sel. Top. Appl., 6, 840-85

0, 10.1109/JSTARS.2012.2212236, 2013.

Jiang, L., and Islam, S.: A methodology for estimation of surface evapotranspiration over l arge areas using remote sensing observations, Geophys. Res. Lett., 26, 2773-2776, 10.1029/199 9GL006049, 1999.

Jiang, L., and Islam, S.: Estimation of surface evaporation map over Southern Great Plains using remote sensing data, Water Resour. Res., 37, 329-340, 10.1029/2000WR900255, 2001.

Jiang, L., and Islam, S.: An intercomparison of regional latent heat flux estimation using r emote sensing data, Int. J. Remote Sens., 24, 2221-2236, 10.1080/01431160210154821, 2003.

Jin, Y. F., Randerson, J. T., and Goulden, M. L.: Continental-scale net radiation and evapo transpiration estimated using MODIS satellite observations, Remote Sens. Environ., 115, 2302-2 319, http://dx.doi.org/10.1016/j.rse.2011.04.031, 2011.

Kalma, J., McVicar, T., and McCabe, M.: Estimating Land Surface Evaporation: A Review of Methods Using Remotely Sensed Surface Temperature Data, Surveys in Geophysics, 29, 42 1-469, 10.1007/s10712-008-9037-z, 2008.

Kato, S., and Yamaguchi, Y.: Analysis of urban heat-island effect using ASTER and ETM + Data: Separation of anthropogenic heat discharge and natural heat radiation from sensible he at flux, Remote Sensing of Environment, 99, 44-54, http://dx.doi.org/10.1016/j.rse.2005.04.026, 2 005.

Kormann, R., and Meixner, F.: An Analytical Footprint Model For Non-Neutral Stratificati on, Bound.-Lay. Meteorol., 99, 207-224, 10.1023/A:1018991015119, 2001.

Kustas, W. P.: Estimates of Evapotranspiration with a One- and Two-Layer Model of Heat Transfer over Partial Canopy Cover, J. Appl. Meteorol., 29, 704-715, 10.1175/1520-0450(1990) 029<0704:EOEWAO>2.0.CO;2, 1990.

Kustas, W. P., Norman, J. M., Anderson, M. C., and French, A. N.: Estimating subpixel s urface temperatures and energy fluxes from the vegetation index–radiometric temperature relatio nship, Remote Sens. Environ., 85, 429-440, http://dx.doi.org/10.1016/S0034-4257(03)00036-1, 20 03.

Kustas, W. P., Moran, M. S., and Meyers, T. P.: The Bushland Evapotranspiration and Agr icultural Remote Sensing Experiment 2008 (BEAREX08) Special Issue, Adv. Water Resour., 50, 1-3, http://dx.doi.org/10.1016/j.advwatres.2012.11.006, 2012.

Leuning, R., Zhang, Y. Q., Rajaud, A., Cleugh, H., and Tu, K.: A simple surface conduct ance model to estimate regional evaporation using MODIS leaf area index and the Penman-Mo nteith equation, Water Resour. Res., 44, W10419, 10.1029/2007WR006562, 2008.

Li, H., Liu, Q. H., Zhong, B., Du, Y. M., Wang, H. S., and Wang, Q.: A single-channel algorithm for land surface temperature retrieval from HJ-1B/IRS data based on a parametric m odel, Geoscience and Remote Sensing Symposium (IGARSS), 2010 IEEE International, Honolul u, Hawaii, USA, 2448-2451, 2010.

Li, H., Liu, Q., Jiang, J., Wang, H., and Sun, L.: Validation of the land surface temperatu re derived from HJ-1B/IRS data with ground measurements, Geoscience and Remote Sensing S ymposium (IGARSS), 2011 IEEE International, Vancouver, Canada, 293-296, 2011.

Li, H., Sun, D., Yu, Y., Wang, H., Liu, Y., Liu, Q., Du, Y., Wang, H., and Cao, B.: Eval uation of the VIIRS and MODIS LST products in an arid area of Northwest China, Remote S ens. Environ., 142, 111-121, http://dx.doi.org/10.1016/j.rse.2013.11.014, 2014.

Li, J., Gu, X., Li, X., Yu, T., Chen, H., and Long, M.: Validation of HJ-1B Thermal Ban d On-board Calibration and Its Sensitivity Analysis, Remote Sensing Information, 1, 3-9, doi:10 .3969/j.issn.1000-3177.2011.01.001, 2011.

Li, L., Xin, X. Z., Su, G. L., and Liu, Q. H.: Photosynthetically active radiation retrieval based on HJ-1A/B satellite data, Sci. China Earth Sci., 53, 81-91, 10.1007/s11430-010-4142-5, 2010.

Li, X., Li, X. W., Li, Z. Y., Ma, M. G., Wang, J., Xiao, Q., Liu, Q., Che, T., Chen, E. X., Yan, G. J., Hu, Z. Y., Zhang, L. X., Chu, R. Z., Su, P. X., Liu, Q. H., Liu, S. M., Wang , J. D., Niu, Z., Chen, Y., Jin, R., Wang, W. Z., Ran, Y. H., Xin, X. Z., and Ren, H. Z.: Wa tershed Allied Telemetry Experimental Research, Journal of Geophysical Research: Atmospheres, 114, D22103, 10.1029/2008JD011590, 2009.

Li, X., Cheng, G. D., Liu, S. M., Xiao, Q., Ma, M. G., Jin, R., Che, T., Liu, Q. H., Wa ng, W. Z., Qi, Y., Wen, J. G., Li, H. Y., Zhu, G. F., Guo, J. W., Ran, Y. H., Wang, S. G., Z hu, Z. L., Zhou, J., Hu, X. L., and Xu, Z. W.: Heihe Watershed Allied Telemetry Experimenta l Research (HiWATER): Scientific Objectives and Experimental Design, B. Am. Meteorol. Soc., 94, 1145-1160, 10.1175/BAMS-D-12-00154.1, 2013.

Li, Z. L., Tang, B. H., Wu, H., Ren, H., Yan, G., Wan, Z., Trigo, I. F., and Sobrino, J. A.: Satellite-derived land surface temperature: Current status and perspectives, Remote Sensing of Environment, 131, 14-37, http://dx.doi.org/10.1016/j.rse.2012.12.008, 2013.

Liang, S., Stroeve, J., and Box, J. E.: Mapping daily snow/ice shortwave broadband albed o from Moderate Resolution Imaging Spectroradiometer (MODIS): The improved direct retrieval algorithm and validation with Greenland in situ measurement, Journal of Geophysical Research : Atmospheres, 110, D10109, 10.1029/2004JD005493, 2005.

Liang, S. L., Zhang, X. T., Xiao, Z. Q., Cheng, J., Liu , Q., and Zhao, X.: Global LAnd Surface Satellite (GLASS) Products: Algorithms, Validation and Analysis, 1 ed., SpringerBriefs in Earth Sciences, Springer International Publishing, 2014.

Liu, Q., Qu, Y., Wang, L. Z., Liu, N. F., and Liang, S. L.: Glass-Global Land Surface Br oadband Albedo Product: Algorithm Theoretical Basis Document. Version, 1, 1-50, College of Global Change and Earth System Science, Beijing Norman University, 2011.

Liu, S. M., Xu, Z. W., Wang, W. Z., Jia, Z. Z., Zhu, M. J., Bai, J., and Wang, J. M.: A comparison of eddy-covariance and large aperture scintillometer measurements with respect to t he energy balance closure problem, Hydrol. Earth Syst. Sci., 15, 1291-1306, 2011b.

Liu, S., Xu, Z., Song, L., Zhao, Q., Ge, Y., Xu, T., Ma, Y., Zhu, Z., Jia, Z., and Zhang, F.: Upscaling evapotranspiration measurements from multi-site to the satellite pixel scale over h eterogeneous land surfaces, Agricultural and Forest Meteorology, http://dx.doi.org/10.1016/j.agrfor met.2016.04.008, 2016.

Long, D., and Singh, V. P.: A modified surface energy balance algorithm for land (M-SEB AL) based on a trapezoidal framework, Water Resour. Res., 48, W02528, 10.1029/2011WR0106 07, 2012a.

Long, D., and Singh, V. P.: A Two-source Trapezoid Model for Evapotranspiration (TTME) from satellite imagery, Remote Sens. Environ., 121, 370-388, http://dx.doi.org/10.1016/j.rse.201 2.02.015, 2012b.

Maayar, E. M., and Chen, J. M.: Spatial scaling of evapotranspiration as affected by heter ogeneities in vegetation, topography, and soil texture, Remote Sens. Environ., 102, 33-51, http:/ /dx.doi.org/10.1016/j.rse.2006.01.017, 2006.

Mallick, K., Boegh, E., Trebs, I., Alfieri, J. G., Kustas, W. P., Prueger, J. H., Niyogi, D., Das, N., Drewry, D. T., Hoffmann, L., and Jarvis, A. J.: Reintroducing radiometric surface tem perature into the Penman-Monteith formulation, Water Resources Research, 51, 6214-6243, 10.1 002/2014WR016106, 2015.

McCabe, M. F., and Wood, E. F.: Scale influences on the remote estimation of evapotrans piration using multiple satellite sensors, Remote Sensing of Environment, 105, 271-285, 10.101 6/j.rse.2006.07.006, 2006.

Meyers, T. P., and Hollinger, S. E.: An assessment of storage terms in the surface energy balance of maize and soybean, Agricultural and Forest Meteorology, 125, 105-115, http://dx.doi. org/10.1016/j.agrformet.2004.03.001, 2004.

Moran, M. S., Humes, K. S., and Pinter Jr, P. J.: The scaling characteristics of remotely-s ensed variables for sparsely-vegetated heterogeneous landscapes, Journal of Hydrology, 190, 337 -362, http://dx.doi.org/10.1016/S0022-1694(96)03133-2, 1997.

Mu, Q. Z., Zhao, M. S., and Running, S. W.: Improvements to a MODIS global terrestria l evapotranspiration algorithm, Remote Sens. Environ., 115, 1781-1800, http://dx.doi.org/10.1016/ j.rse.2011.02.019, 2011.

Mukherjee, S., Joshi, P. K., and Garg, R. D.: A comparison of different regression models for downscaling Landsat and MODIS land surface temperature images over heterogeneous lan dscape, Advances in Space Research, 54, 655-669, http://dx.doi.org/10.1016/j.asr.2014.04.013, 20 14.

Neftel, A., Spirig, C., and Ammann, C.: Application and test of a simple tool for operatio nal footprint evaluations, Environ. Pollut., 152, 644-652, http://dx.doi.org/10.1016/j.envpol.2007.0 6.062, 2008.

Nilson, T.: A theoretical analysis of the frequency of gaps in plant stands, Agr. Meteorol., 8, 25-38, 1971.

Norman, J. M., Kustas, W. P., and Humes, K. S.: Source approach for estimating soil and vegetation energy fluxes in observations of directional radiometric surface temperature, Agr. F orest Meteorol., 77, 263-293, http://dx.doi.org/10.1016/0168-1923(95)02265-Y, 1995.

Norman, J. M., Anderson, M. C., Kustas, W. P., French, A. N., Mecikalski, J., Torn, R., Diak, G. R., Schmugge, T. J., and Tanner, B. C. W.: Remote sensing of surface energy fluxes at 101-m pixel resolutions, Water Resour. Res., 39, 1221, 10.1029/2002WR001775, 2003.

Paulson, C. A.: The mathematical representation of wind speed and temperature profiles in the unstable atmospheric surface layer, J. Appl. Meteorol., 9, 857-861, 1970.

Prueger, J. H., Hatfield, J. L., Parkin, T. B., Kustas, W. P., Hipps, L. E., Neale, C. M. U. , MacPherson, J. I., Eichinger, W. E., and Cooper, D. I.: Tower and Aircraft Eddy Covariance Measurements of Water Vapor, Energy, and Carbon Dioxide Fluxes during SMACEX, Journal o f Hydrometeorology, 6, 954-960, 10.1175/JHM457.1, 2005.

Shuttleworth, W. J., and Wallace, J.: Evaporation from sparse crops - an energy combinatio n theory, Q. J. Roy. Meteor. Soc., 111, 839-855, 1985.

Song, Y., Wang, J. M., Yang, K., Ma, M. G., Li, X., Zhang, Z. H., and Wang, X. F.: A revised surface resistance parameterisation for estimating latent heat flux from remotely sensed data, Int. J. Appl. Earth Obs., 17, 76-84, http://dx.doi.org/10.1016/j.jag.2011.10.011, 2012.

Su, Z.: The Surface Energy Balance System (SEBS) for estimation of turbulent heat fluxes , Hydrol. Earth Syst. Sci., 6, 85-100, 10.5194/hess-6-85-2002, 2002.

Sun, L., Sun, R., Li, X. W., Chen, H. L., and Zhang, X. F.: Estimating Evapotranspiration using Improved Fractional Vegetation Cover and Land Surface Temperature Space, Journal of Resources and Ecology, 2, 225-231, 10.3969/j.issn.1674-764x.2011.03.005, 2011.

Sun, L., Liang, S. L., Yuan, W. P., and Chen, Z. X.: Improving a Penman–Monteith evap otranspiration model by incorporating soil moisture control on soil evaporation in semiarid area s, International Journal of Digital Earth, 6, 134-156, 10.1080/17538947.2013.783635, 2013.

Valor, E., and Caselles, V.: Mapping land surface emissivity from NDVI: Application to European, African, and South American areas, Remote Sens. Environ., 57, 167-184, http://dx.doi.org/10.1016/0034-4257(96)00039-9, 1996.

Venturini, V., Islam, S., and Rodriguez, L.: Estimation of evaporative fraction and evapotranspiration from MODIS products using a complementary based model, Remote Sens. Environ., 112, 132-141, http://dx.doi.org/10.1016/j.rse.2007.04.014, 2008.

Vermote E F, Tanre D, Deuze J L, et al. Second Simulation of a Satellite Signal in the Solar Spectrum-Vector. 6S User Guide Version 3, 2006.

Von Randow, C., Kruijt, B., Holtslag, A. A. M., and de Oliveira, M. B. L.: Exploring eddy-covariance and large-aperture scintillometer measurements in an Amazonian rain forest, Agricultural and Forest Meteorology, 148, 680-690, http://dx.doi.org/10.1016/j.agrformet.2007.11.011, 2008.

Wang, K. C., and Liang, S. L.: An improved method for estimating global evapotranspiration based on satellite determination of surface net radiation, vegetation index, temperature, and soil moisture, Geoscience and Remote Sensing Symposium, 2008. IGARSS 2008. IEEE International, Boston, Massachusetts, USA, III - 875-III - 878, 2008.

Wang, Q., Wu, C., Li, Q., and Li, J.: Chinese HJ-1A/B satellites and data characteristics, Sci. China Earth Sci., 53, 51-57, 10.1007/s11430-010-4139-0, 2010.

Xin, X., and Liu, Q.: The Two-layer Surface Energy Balance Parameterization Scheme (TSEBPS) for estimation of land surface heat fluxes, Hydrol. Earth Syst. Sci., 14, 491-504, 10.5194/hess-14-491-2010, 2010.

Xin, X., Liu, Y. N., Liu, Q., and Tang, Y.: Spatial-scale error correction methods for regional fluxes retrieval using MODIS data, J. Remote Sens., 16(2), 207-231, 2012.

Xu, T. R., Bateni, S. M., and Liang, S. L.: Estimating Turbulent Heat Fluxes With a Weak-Constraint Data Assimilation Scheme: A Case Study (HiWATER-MUSOEXE), IEEE Geosci. Remote S., 12, 68-72, 10.1109/LGRS.2014.2326180, 2015.

Xu, Z. W., Liu, S. M., Li, X., Shi, S. J., Wang, J. M., Zhu, Z. L., Xu, T. R., Wang, W. Z., and Ma, M. G.: Intercomparison of surface energy flux measurement systems used during the HiWATER‐MUSOEXE, Journal of Geophysical Research: Atmospheres, 118, 13,140-113,157, 10.1002/2013JD020260, 2013.

Yang, K., and Wang, J.: A temperature prediction-correction method for estimating surface soil heat flux from soil temperature and moisture data, Sci. China Ser. D-Earth Sci., 51, 721-729, 10.1007/s11430-008-0036-1, 2008.

Yang, Y. T., and Shang, S. H.: A hybrid dual-source scheme and trapezoid framework–based evapotranspiration model (HTEM) using satellite images: Algorithm and model test, Journal of Geophysical Research: Atmospheres, 118, 2284-2300, 10.1002/jgrd.50259, 2013.

Yao, Y. J., Liang, S. L., Cheng, J., Liu, S. M., Fisher, J. B., Zhang, X. D., Jia, K., Zhao, X., Qin, Q. M., Zhao, B., Han, S., Jie, Zhou, G. S., Zhou, G. Y., Li, Y. L., and Zhao, S. H.: MODIS-driven estimation of terrestrial latent heat flux in China based on a modified Priestley–Taylor algorithm, Agr. Forest Meteorol., 171–172, 187-202, http://dx.doi.org/10.1016/j.agrformet.2012.11.016, 2013.

Yebra, M., Van Dijk, A., Leuning, R., Huete, A., and Guerschman, J. P.: Evaluation of optical remote sensing to estimate actual evapotranspiration and canopy conductance, Remote Sens. Environ., 129, 250-261, http://dx.doi.org/10.1016/j.rse.2012.11.004, 2013.

Yu, S., Xin, X., and Liu, Q.: Estimation of clear-sky longwave downward radiation from HJ-1B thermal data, Sci. China Earth Sci., 56, 829-842, 10.1007/s11430-012-4507-z, 2013.

Zhang, R., Sun, X., Wang, W., Xu, J., Zhu, Z., and Tian, J.: An operational two-layer remote sensing model to estimate surface flux in regional scale: physical background, Sci. China Ser. D, 34, 200-216, 2005.

Zhang, X., Zhao, X., Liu, G., Kang, Q., and Wu, D.: Radioactive Quality Evaluation and Cross Validation of Data from the HJ-1A/B Satellites' CCD Sensors, Sensors, 13, 8564, doi:10.3390/s130708564, 2013.

Zhong, B., Ma, P., Nie, A., Yang, A., Yao, Y., Lü, W., Zhang, H., and Liu, Q.: Land cover mapping using time series HJ-1/CCD data, Sci. China Earth Sci., 57, 1790-1799, 10.1007/s11430-014-4877-5, 2014a.

Zhong, B., Zhang, Y., Du, T., Yang, A., Lv, W., and Liu, Q.: Cross-Calibration of HJ-1/CCD Over a Desert Site Using Landsat ETM+ Imagery and ASTER GDEM Product, Geoscience and Remote Sensing, IEEE Transactions on, 52, 7247-7263, 10.1109/TGRS.2014.2310233, 2014b.

Zhu, G. F., Su, Y. H., Li, X., Zhang, K., and Li, C. B.: Estimating actual evapotranspiration from an alpine grassland on Qinghai-Tibetan plateau using a two-source model and parameter uncertainty analysis by Bayesian approach, J. Hydrol., 476, 42-51, 10.1016/j.jhydrol.2012.10.006, 2013.

---

## Author Response (AR4)

**Response to the Editor**

Many thanks for your time to review our manuscript. We highly appreciate your suggestion which will help to improve the submitted manuscript. Below are our responses (in blue font) and relevant changes (in red font) to the comments (in **black** font). We also attached a marked-up manuscript with track changes.

**Comments**

Page 2 - Line 20: … and TRFA handles them at 30m and 300m resolution, which depends …
Response: The sentence was revised as suggestion.

Page 2 - Line 22: … delete 's' of 'understandings'
Response: The word 'understandings' was revised as 'understanding'.

Page 2 - Line 32: Question – is it really the difference of varied landscapes within mixed pixels? Should it not rather read '… and the influence of various landscapes inside mixed pixels.'?
Response: The meaning of this sentence is 'the influence of various landscapes inside mixed pixels.' The sentence has been revised as follow:
Furthermore, additional analysis showed that the TSFA method can capture the sub-pixel variations of land surface temperature and the influences of various landscapes within mixed pixels.

Page 3 - Line 17-18: The sentence 'Non-linear operation model and surface heterogeneity are main issues of remotely sensed spatial scale error …' This sentence is not 100% clear – it should be rephrased.
Response: The sentence has been rephrased as follow:
If the operational algorithm can be described as a linear combination of inputs, or if the surface variables and landscapes are homogeneous at the pixel scale, scale error does not exist.

Page 3 - Line 40-42: SUGGESTION: 'However, in case of mixed pixels surface variables such as land surface temperature are set to singular [SENSE NOT 100% CLEAR] to represent the entire pixel area in ET estimation models.'
Response: The sentence has been revised as follow:
However, in case of mixed pixels, surface variables such as land surface temperature are commonly considered as single value to represent the entire pixel area in ET estimation models, which results in large errors.

Page 4 – Line 37-38: SUGGESTION 'The scatter plot between LST and NDVI values forms a …'
Response: The sentence was revised as suggestion.

Page 4 – Line 42: WHAT DO YOU MEAN BY 'inferiority' ?
Response: 'Inferiority' means the issue that the 300 m resolution thermal data cannot sufficiently distinguish the surface temperatures of small targets within pixels.

The sentence has been revised as follow:

However, this issue can be addressed by temperature sharpening based on the functional relationship between NDVI and LST.

Page 5 – Line 10: WHAT DOES 'uniformed' MEAN? COULD IT BE 'standardised'?

Response: 'Uniform' means 'homogeneous'.

We have revised the paragraph, and exchange the position with the next paragraph to make the sense clear.

(1) The $NDVI_{30}$ is aggregated to 300 m NDVI ($NDVI_{300}$). Then, the $NDVI_{300}$ is divided into three classes ($0 \leq NDVI_{300} < 0.2$, $0.2 \leq NDVI_{300} < 0.5$ and $0.5 \leq NDVI_{300}$).

(2) A subset of pixels is selected from the scene where the NDVI is as homogeneous as possible at a pixel resolution of 300 m based on the coefficient of variation (CV). The CVs are calculated using the original 30 m NDVI data ($NDVI_{30}$) as follows:

$$CV = \frac{STD}{mean} \tag{1}$$

where STD and mean are the standard deviation and the average values of $10 \times 10$ pixels of $NDVI_{30}$, respectively. The CVs are sorted from smallest to largest. Lower CVs corresponds to more homogeneous land surface values, and a threshold should be determined to guarantee that a sufficient number of pixels is available for least squares fitting between $NDVI_{300}$ and $T_{300}$. Therefore, the fractions of 25% of the lowest CVs are selected from each class.

Page 6 – Line 27: SUGGESTION 'The spatial resolution of LST is significantly increased by …' SPLIT THE SENTENCE IN TWO.

Response: The sentence has been revised as follow:

The spatial resolution of LST is significantly increased by temperature sharpening in section 2.1. Consequently, all inputs of ET algorithms can be obtained at high spatial resolutions. Then, inhomogeneity issues can be greatly diminished by dividing the landscape into finer pixels.

Page 8 – Lines 4 to 6: this sentence is too long and should somehow be structured in a different way (e.g. by using commas)

Response: The sentence has been revised as follow:

The spatial scale effect is usually generally revealed by a discrepancy between different up-scaling methods. In one method, parameters are upscaled to a large scale before calculating the heat flux. In the other method, heat flux is calculated at a small scale, and the results are then upscaled.

Page 8 – Line 8: SUGGESTION 'In the case of using IPUS, the input for the energy balance model are first retrieved at … and then aggregated to 300 m resolution.'

Response: The sentence has been revised as follow:

In the case of IPUS, the inputs of the energy balance model are first retrieved at 30 m resolution (see section 3.2.1.1) and then aggregated to 300 m resolution.

Page 8 – Line 11-12: SUGGESTION '… using the nearest neighbour method and the 30 m resolution inputs are used for estimating ET.'

Response: The sentence has been revised as follow:

In TRFA, the LST at 300 m is first resampled to 30 m using the nearest neighbour method and the 30 m resolution inputs are used for estimating ET.

Page 11 – Line 13: SUGGESTION '… is the regression kernel of the temperature sharpening procedure and …'

Response: The sentence has been revised as follow:

The NDVI is the regression kernel of the temperature-sharpening procedure and is used to calculate the FVC.

Page 13 – Line 24: SUGGESTION 'To control model inputs and analyse error sources, the …'

Response: The sentence has been revised as suggestion.

Page 16 – Line 14: SUGGESTION '… including land patches in the Gobi region, barren areas and …'

Response: The sentence has been revised as suggestion.

The H values of buildings and uncultivated land, including land patches in the Gobi region, barren areas and desert areas, were high, in addition to the LEs of water and agricultural areas in the oasis.

Page 16-17: SUGGESTION 'Therefore, the LE of barren areas in the south is higher than the LE of desert areas in the southeast, …'

Response: The sentence has been revised as suggestion.

Therefore, the LE values of barren areas in the south are higher than the LE values of desert areas in the southeast, although both areas were classified as uncultivated land.

Page 18 – Line 12: SUGGESTION replace 'Gobi areas' by 'Gobi region'

Response: The sentence has been revised as suggestion.

In our study area, Gobi region, barren area and desert area comprise the uncultivated land.

Page 18 – Line 22: SUGGESTION '… were also implemented for comparison purposes.'

Response: The sentence has been revised as suggestion.

Page 19 – Line 2: SUGGESTION '… the improvements in accuracy between TRFA and IPUS were relatively larger than those between TSFA and TRFA.'

Response: The sentence has been revised as suggestion.

Page 19 – Line 13: SUGGESTION change the sentence to 'Variations in landscape characteristics systematically trigger variations in surface variables.'

Response: The sentence has been revised as suggestion.

Page 23 – Line 3: SUGGESTION 'The land surface of EC15 was uniform and consisted of pure pixels covering maize fields. Consequently, the temperature distribution at 30 m resolution was very homogeneous and the surface temperature variations were comprised within a range of 1.6 K.'

Response: The sentence has been revised as suggestion.

Page 24 – Line 7: SUGGESTION '… and the observed ground temperature was …'

Response: The sentence has been revised as suggestion.

Page 25 – Line 8-9: this sentence is difficult to read and should be rephrased.

Response: The sentence has been revised as follow:

The TSFA results are synoptically smoother than the IPUS results because the land cover types and temperature distributions in mixed pixels are not considered in IPUS.

Page 30 – Line 39-40: SUGGESTION, split sentence 'At noon, the vegetation temperature in those pixels is lower than that of water bodies.'

Response: The sentence has been revised as follow:

Some of the blue LE points calculated by using the TSFA method are slightly smaller than those calculated using the TRFA method for pixels containing vegetation. At noon, the temperature of vegetation in those pixels is lower than that of water bodies.

Page 31 – Line 1: Moreover, G depends on Rn when using the TRFA method and is nearly identical to the values of G obtained with the TSFA method.'

Response: The sentence has been revised as follow:

Moreover, G depends on $R_n$ when using the TRFA method and is nearly identical to the values of G obtained using the TSFA method.

Page 32 – Line 20: SUGGESTION change to '4.4.1. Errors in LST'

Response: The title has been changed as suggestion.

Page 32 – Line 26: A sensitivity analysis of LE inferred from LST was also carried out. In order to exclude potential influences of other factors, only homogeneous stations were chosen'. COMMENT: what does homogeneous exactly mean in this context?

Response: 'Homogeneous' means 'homogeneous landscapes' in this context.

The sentence has been revised as:

A sensitivity analysis of LE induced by LST was also performed. In order to exclude the influence of other factors, stations were chosen with homogeneous landscapes within coarse pixels.

Page 34 – Line 18: SUGGESTION 'In summary, the advantages of the TSFA method are described as follows:'

Response: The sentence has been revised as suggestion.

Page 34 – Line 21: SUGGESTION '…, which is consistent with results from previous research'

Response: The sentence has been revised as suggestion.

Page 35 – Line 39-40: Sentence not clear – should be reformulated.

Response: The sentence has been revised as follow:

The IPUS approach does not consider surface heterogeneity at all, which causes significant error in the heat fluxes (i.e., 186 W·m$^{-2}$). The TRFA considers heterogeneity of landscapes besides LST heterogeneity, with a heat flux error (i.e., 49 W·m$^{-2}$) that is less than that of IPUS. However, this error is non-negligible.

Page 35 – Line 41: set 'variables' to singular

Response: 'Variables' has been set to singular.

**Relevant Changes**

1. We have revised all the sentences and paragraphs with ambiguous or redundant expression.

2. All the highlights in the manuscript have been revised.

3. The column of "LE relative error (%)" in Table 17 has been replaced by "LE-I relative error" and "LE-C relative error (%)", because the "LE relative error (%)" means the relative error between the LE from incorrect canopy height and the LE from correct canopy height in the last manuscript, and we did not make this sense clear. LE-I relative error" is the relative error between the LE from incorrect canopy height and observed LE and is expressed as ((LE from incorrect canopy height)-(EC-LE))/( EC-LE)$\times$100%, "LE-C relative error" is the relative error between the LE from correct canopy height and observed LE and is expressed as ((LE from correct canopy height)-(EC-LE))/( EC-LE)$\times$100%.

Here below attached a marked-up manuscript with track changes.

**Remote-sensing algorithm for surface evapotranspiration considering landscape and statistical effects on mixed-pixels**

ZhiQing Peng [a, b], Xiaozhou Xin [a, *], JinJun Jiao [a, b], Ti Zhou [a, b], Qinhuo Liu [a, c]

*a. State Key Laboratory of Remote Sensing Science, Institute of Remote Sensing and Digital Earth*

*Chinese Academy of Sciences, Beijing 100101, China*

*b. University of Chinese Academy of Sciences, Beijing 100049, China*

*c. Joint Center for Global Change Studies (JCGCS), Beijing 100875, China*

**Abstract**

Evapotranspiration (ET) plays an important role in surface-atmosphere interactions and can be monitored using remote sensing data. However, surface heterogeneity, including the inhomogeneity of landscapes and surface variables, significantly affects the accuracy of ET estimated from satellite data. The objective of this study is to assess and reduce the uncertainties resulting from surface heterogeneity in remotely sensed ET using Chinese HJ-1B satellite data, which is of 30 m spatial resolution in VIS/NIR bands and 300 m spatial resolution in the TIR band. A temperature sharpening and flux aggregation scheme (TSFA) was developed to obtain accurate heat fluxes from the HJ-1B satellite data. The IPUS (input parameter upscaling) and TRFA (temperature resampling and flux aggregation) methods were used to compare with the TSFA in this study. The three methods represent three typical schemes  used to handle mixed pixels from the simplest to the most complex. IPUS handles all surface variables  at coarse resolution  of 300 m in this study, TSFA handles them at  30 m resolution, and TRFA handles them at  30m and 300m resolution, which depends on the actual spatial resolution.  Analyzing and comparing the three methods can help us to get a better understanding of spatial scale errors in remote sensing of surface heat fluxes. *In situ* data collected during HiWATER-MUSOEXE (Multi-Scale Observation Experiment on Evapotranspiration over heterogeneous land surfaces of The Heihe Watershed Allied Telemetry Experimental Research) were used  to validate and analyze the methods. ET estimated by TSFA  exhibited the best agreement with *in situ* observations, and the footprint validation results showed that the $R^2$, MBE, and RMSE values of the sensible heat flux (H) were 0.61, 0.90 W·m$^{-2}$ and 50.99 W·m$^{-2}$, respectively, and those for the latent heat flux (LE) were 0.82, -20.54 W·m$^{-2}$ and 71.24 W·m$^{-2}$, respectively. IPUS  yielded the largest errors in ET estimation. The RMSE of LE between the TSFA and IPUS methods was 51.30 W·m$^{-2}$, and the RMSE of LE between the TSFA and TRFA methods was 16.48 W·m$^{-2}$. Furthermore, additional analysis showed that the TSFA method can capture the sub-pixel variations of land surface temperature and the  influences of various landscapes within mixed pixels.

**Index Terms:** surface heterogeneity, temperature sharpening, area weighting, energy balance, evapotranspiration, spatial scale, HJ-1B satellite

**1. Introduction**

Five types of methods have been developed to estimate evapotranspiration (ET) or latent heat flux (LE) via remote sensing. (1) Surface energy balance models calculate LE as a residual term. According to the partitioning of the sources and sinks of the Soil-Plant-Atmosphere Continuum (SPAC), surface energy balance models can be classified as one-source (Bastiaanssen et al., 1998; Su, 2002; Allen et al., 2007; Long and Singh, 2012a) or two-source models (Shuttleworth and Wallace, 1985; Norman et al., 1995; Xin and Liu, 2010; Zhu et al., 2013). (2) Penman-Monteith models are used to calculate LE by using the Penman-Monteith equation and numerous surface resistance parameterization schemes that control the diffusion of evaporation from  soil surfaces and transpiration from plant canopies. These two-source Penman-Monteith models separate soil evaporation from plant transpiration (Cleugh et al., 2007; Mu et al., 2011; Leuning et al., 2008; Chen et al., 2013; Sun et al., 2013; Mallick et al., 2015). (3) Land surface temperature-vegetation index (LST-VI) space methods assign the dry and wet edges of the LST-VI feature space as minimum and maximum ET, respectively. These methods interpolate the media, and use the Penman-Monteith or Priestley-Taylor equation to calculate the LE (Jiang and Islam, 1999, 2001; Sun et al., 2011; Long and Singh, 2012b; Yang and Shang, 2013; Fan et al., 2015; Zhang et al., 2005). (4) Priestley-Taylor models expand the range of the Priestley-Taylor coefficient in the Priestley-Taylor equation (Jiang and Islam, 2003; Jin et al., 2011) or combine the physiological force factors with the energy component of ET (Fisher et al., 2008; Yao et al., 2013). (5) Additional methods include empirical/statistical methods (Wang and Liang, 2008; Yebra et al., 2013) and the use of complementary based models (Venturini et al., 2008) and land-process models with data assimilation schemes (Bateni and Liang, 2012; Xu et al., 2015).

 If the operational algorithm can be described as a linear combination of inputs, or if the surface variables and landscapes are homogeneous at the pixel scale, scale error does not exist (Hu and Islam, 1997). However, it is difficult to develop linear operational models due to the complexity of mass and heat transfer processes between the atmosphere and land surface.  ET estimation models  have been generally developed for simple and homogeneous surface conditions. However, heterogeneity is a natural attribute of  the surface of the Earth. Therefore, larger spatial scale errors occur when  these remotely sensed models are applied to calculate the regional ET using satellite data.

In previous studies, researchers have coupled high- and low-resolution satellite data and statistically quantified the inhomogeneity of mixed pixels to correct the scale error in ET estimations  using (1) temperature downscaling , which converts images from a lower (coarser) to higher (finer) spatial resolution using statistical-based models with regression or stochastic relationships among parameters (Kustas et al., 2003; Norman et al., 2003; Cammalleri et al., 2013; Ha et al., 2013), (2) the correction-factor method , which uses sub-pixel landscapes information to  determine the correction factor of scale bias (Maayar and Chen, 2006) and (3) the area-weighting method , which calculates roughness length and sensible heat flux based on sub-pixel landscapes (Xin et al., 2012). These correction methods mainly focus on two problems: inhomogeneity of landscapes and inhomogeneity of surface variables.

Studies have shown that different landscapes (Blyth and Harding, 1995; Moran et al., 1997; Bonan et al., 2002; McCabe and Wood, 2006) and the sub-pixel variations of surface variables, such as stomatal conductance (Bin and Roni, 1994), or leaf area index (Bonan et al., 1993; Maayar and Chen, 2006), can cause errors in turbulent heat flux estimations. Surface variables inhomogeneity is rather difficult to evaluate, as the sub-pixel variation of surface variables  can be large, even in the pure pixels. For example, generally, temperatures over land surfaces vary strongly in space and time, and it is  common for the LST to vary by more than 10 K over just a few centimeters of distance or by more than 1 K in less than a minute over certain cover types (Li et al.,

2013b). However, in case of mixed pixels, surface variables such as land surface temperature are  commonly considered as single value to represent the entire pixel area in ET estimation models , which results in large errors.

The focus of this study is on the effects of surface heterogeneity when estimating ET.  Based on the satellite products that are currently available, three methods were used to analyze the uncertainties produced by surface heterogeneity: (1) input parameter upscaling (IPUS) does not consider the surface heterogeneities at all. It was designed to simulate the satellites that have identical spatial resolution both the visible near-infrared (VNIR) and thermal infrared  (TIR) bands; (2) temperature resampling and flux aggregation (TRFA) does not consider the heterogeneity of LST; and (3) temperature sharpening and flux aggregation (TSFA) consider all the surface heterogeneities.  These methods were designed for use with the majority of satellite data or products that have inconsistent spatial resolution between the VNIR and TIR bands, such as the Landsat and HJ-1B satellites.

The surface variables in this paper were mainly derived from HJ-1B satellite data . The Chinese HJ-1A/B satellites were launched on September 6, 2008, and were designed for disaster and environmental monitoring, as well as other applications. The HJ-1B satellites are equipped with two charge-coupled device (CCD) cameras and one infrared scanner (IRS) with spatial resolutions of 30 m and 300 m, respectively. Compared  to high-temporal-resolution satellite data, such as the MODIS satellite data, or high-spatial resolution satellite data, such as the Landsat 7 or 8 satellites data, HJ-1B data has the advantage of a high  spatiotemporal resolution. Since the satellites were launched, the HJ-1/CCD time series data have been widely used in China to accurately classify land cover (Zhong et al., 2014a) and monitor various environmental disasters (Wang et al., 2010). Land-based variables, such as leaf area index (LAI), land surface temperature (LST), and downward longwave radiation ($L_d$), have been retrieved by the HJ-1 satellites using algorithms developed by Chen et al. (2010), Li et al. (2010a, 2011a) and Yu et al. (2013), respectively. These variables lay the foundation for ET research.

Although the HJ-1B satellites provide CCD data with a high spatial resolution of 30 m, the spatial resolution of the thermal infrared (TIR) band is only 300 m. Thus, surface heterogeneity effects must be considered when estimating the heat flux.

**2. Methodology**

**2.1. Temperature-sharpening method based on statistical relationships**

Surface thermal dynamic affect  ET. The spatial resolution of TIR images is usually not as high as the spatial resolution of visible near-infrared  (VNIR) bands because the energy of VNIR photons is higher than the energy of thermal photons. Thus, the inhomogeneity of TIR images would be larger than the inhomogeneity of VNIR images. Since the land surface temperature is calculated from the TIR band, the uncertainty of the variables becomes unpredictable  when the inhomogeneity of TIR images is enhanced. Therefore,  land surface temperature data should be derived with a high spatial resolution.

The land surface temperature can be reconstructed at the spatial resolution of the VNIR images by using a statistical temperature-sharpening strategy proposed by Kustas et al. (2003). This method assumes that the negative correlation between the Normalized Difference Vegetation Index (NDVI) and LST is invariant. The NDVI reflects vegetation growth and coverage, and the LST reflects surface thermal dynamics. The LST decreases with increasing vegetation cover. The scatter plot be-tween the LST and NDVI values forms a feature space that is applicable at different scales when  sufficient number of pixels exist.

HJ-1B satellite images can provide vegetation and thermal information at spatial resolutions of 30 m and 300 m, respectively. The 300 m resolution thermal data  cannot sufficiently  distinguish the surface temperatures of small targets within pixels. However, this  issue can be addressed by temperature sharpening based on the functional relationship be-tween NDVI and LST. A flowchart of temperature sharpening is shown in Fig. 1, and  LST at the NDVI pixel resolution can be derived based on the following steps (Kustas et al., 2003):

$$\text{CV} = \frac{\text{STD}}{\text{mean}} \tag{1}$$

The NDVI$_{30}$ is aggregated to 300 m NDVI (NDVI$_{300}$). Then, the NDVI$_{300}$ is divided into three classes ($0 \leq \text{NDVI}_{300} < 0.2$, $0.2 \leq \text{NDVI}_{300} < 0.5$ and $0.5 \leq \text{NDVI}_{300}$).

A subset of pixels is selected from the scene where the NDVI is as homogeneous as possible at a pixel resolution of 300 m based on the coefficient of variation (CV). The CVs are calculated using the original 30 m NDVI data (NDVI$_{30}$) as follows:

$$\text{CV} = \frac{\text{STD}}{\text{mean}} \tag{1}$$

where STD and mean are the standard deviation and the average values of 10×10 pixels of NDVI$_{30}$, respectively. The CVs are sorted from smallest to largest. Lower CVs corresponds to more homo-geneous land surface values, and a threshold should be determined to guarantee that a sufficient number of pixels is available for  least squares fitting between NDVI$_{300}$ and T$_{300}$. Therefore, the fractions of 25%  of the lowest CVs  are selected from each class.

(3) A least squares expression is  established between NDVI$_{300}$ and T$_{300}$ using the selected pixels.

$$\widehat{T}_{300}(\text{NDVI}_{300}) = a + b \times \text{NDVI}_{300} + c \times \text{NDVI}_{300}^2 \tag{2}$$

[Figure]

**Figure 1.** Flowchart of temperature sharpening.

(4) For each 30 m pixel within a 300 m pixel, $\widehat{T}_{30}$ can be calculated according to Eq. (2) as follows:

$$\widehat{T}_{30}(NDVI_{30}) = a + b \times NDVI_{30} + c \times NDVI_{30}^2 + \Delta\widehat{T}_{300} \tag{3}$$

where $\Delta\widehat{T}_{300} = T_{300} - \widehat{T}_{300}$ is the deviation between the regressed temperature and the temperature that was observed by the satellite at 300 m.

**2.2. Area-weighting method based on landscape information**

Coarse pixels are inhomogeneous because various types of land use may be included. Using a dominant type to represent such a large landscape is irrational. The spatial resolution of LST is  significantly increased by temperature sharpening in section 2.1.  Consequently, all inputs of ET algorithms can be obtained at high spatial resolutions. Then, inhomogeneity issues can be greatly diminished  by dividing the landscape  into finer pixels.

Combined with a high-resolution classification map, sub-pixel scale parameters can be  used in the ET algorithm, which is more rational than using a  dominant class type because different landscapes  may require different ET algorithms. The surface energy flux can be averaged linearly due to the conservation of energy (Kustas et al., 2003), and a simple average that calculates the arithmetic mean over sub-pixels is the best choice for flux upscaling  (Ershadi et al., 2013b). Thus, the aggregated flux at a low resolution $F(x, y)$ is the arithmetic mean of all  the $n \times n$ sub-pixel fluxes that constitute the contributing flux $F(x_i, y_j)$ at coordinate $(x_i, y_j)$ :

$$F(x, y) = \frac{1}{n \times n} \sum_{i=1}^{n} \sum_{j=1}^{n} F(x_i, y_j) \tag{4}$$

Because the average of the sub-pixels fluxes is equal to the area-weighted sum of each land-type result, the final coarse result can be derived  from the area-weighted sum of each land type result within the landscape. The main steps  in the area-weighting process are shown below (Xin et al., 2012):

(1) Geometric correction and registration of the VNIR and TIR input datasets.

(2) Count the area ratios of different land cover types within each pixel of a low-spatial-resolution classification image.

(3) According to the fine-classification data, different parameterization schemes can be used in the ET algorithm to calculate the sub-pixel flux, such as the net radiation ($R_n$), soil heat flux (G) and sensible heat flux (H).

(4) To calculate the regional flux, the flux of the large pixel is calculated by the area-weighting method as follows:

$$F = \sum_{i=1}^{n} w_i \cdot F_i \tag{5}$$

where $w_i$ is the fractional area contributing flux $F_i$ of class type i, and F is the aggregated flux at the coarse resolution. The LE is computed as a residual of the surface energy balance in the TSFA (Temperature Sharpening and Flux Aggregation, see section 2.3) process, in which a high-spatial resolution image is used to reduce the number of mixed pixels.

**2.3. Pixel ET algorithm**

The surface energy balance describes the energy between the land surface and atmosphere. The energy budget is commonly expressed as follows:

$$R_n = LE + H + G \tag{6}$$

where $R_n$ is the net radiation, G is the soil heat flux, H is the sensible heat flux, and LE is the latent heat flux absorbed by water vapor when it evaporates from the soil surface and transpires from plants through stomata. The widely used one-source energy balance model considers the a homogeneous SPAC medium and ignores the inhomogeneity and structure. In this case, the LE can be expressed as follows:

$$LE = \frac{\rho c_p}{\gamma} \cdot \frac{e_s - e_a}{r_a + r_s} \tag{7}$$

where $\gamma$ is the psychometric constant; $e_s$ and $e_a$ are the aerodynamic saturation vapor pressure and atmospheric water vapor pressure, respectively; and $r_a$ and $r_s$ are the water vapor transfer aerodynamic resistance and surface resistance, respectively. Surface resistance includes soil resistance and canopy resistance. The surface resistance is influenced by the physiological characteristics of the vegetation and the water supply of roots. Thus, it is difficult to obtain surface resistance by using via remote sensing, and surface resistance is highly uncertain, particularly over heterogeneous surfaces. To avoid error introduced by the uncertainty of the surface resistance, the LE is computed as a residual of the surface energy balance equation.

$R_n$ is the difference between incoming and outgoing radiation and is calculated as follows:

$$R_n = S_d(1 - \alpha) + \varepsilon_s L_d - \varepsilon_s \sigma T_{rad}^4 \tag{8}$$

where $S_d$ is the downward shortwave radiation, $\alpha$ is the surface broadband albedo, $\varepsilon_s$ is the emissivity of the land surface, $L_d$ is the downward atmospheric longwave radiation, $\sigma = 5.67 \times 10^{-8} W \cdot m^{-2} \cdot K^{-4}$ is the Stefan-Boltzmann constant, and $T_{rad}$ is the surface radiation temperature.

G is usually commonly estimated using the empirical relationship with $R_n$. Because the canopy exerts a significant influence on G, the fractional canopy coverage FVC is used to determine the ratio of G to $R_n$ as follows:

$$G = R_n \times [\Gamma_c + (1 - FVC) \times (\Gamma_s - \Gamma_c)] \tag{9}$$

where $\Gamma_s$ is 0.315 for bare soil and $\Gamma_c$ is 0.05 for a full vegetation canopy (Su, 2002). H is the transfer of turbulent heat between the surface and atmosphere that, which is driven by a temperature difference and  controlled by resistances that depend on local atmospheric conditions and land cover properties (Kalma et al., 2008). According to gradient diffusion theory, the equation for H is as follows:

$$H = \rho c_p \frac{T_{aero} - T_a}{r_a} \tag{10}$$

where $\rho$ is the density of the air; $c_p$ is the specific heat of the air at a constant pressure; $T_{aero}$ is the aerodynamic surface temperature obtained by extrapolating the logarithmic air temperature profile to the roughness length for heat transport; $T_a$ is the air temperature at a reference height; and $r_a$ is the aerodynamic resistance, which influences the heat transfer between the source of the turbulent heat flux and the reference height. Aerodynamic resistance was calculated based on the Monin-Obukhov similarity theory (MOST) using a stability correction function (Paulson, 1970; Ambast et al., 2002). The zero-plane displacement height, d, and roughness length, $z_{0m}$, were parameterized by the schemes proposed by Choudhury and Monteith (1988).

In this approach, H must be accurately estimated. However, calculating H  using Eq. (10) is difficult. Because remote sensing cannot obtain $T_{aero}$, the value of $T_{aero}$ is  generally replaced  with the radiative surface temperature $T_{rad}$, which is not always equal to $T_{aero}$. The difference between these terms for homogeneous and  full-coverage vegetation is approximately 1-2℃ (Choudhury et al., 1986),  and it can reach 10℃ in sparsely vegetative areas (Kustas, 1990). The method that corrects for this discrepancy adds "excess" resistance $r_{ex}$ to $r_a$. We used the brief method proposed by Chen (1988) to calculate $r_{ex}$: $r_{ex} = 4/u_*$, .

Fig. 2 shows  the flowchart for merging ET retrieval and temperature sharpening based on HJ-1B satellites.

[Figure]

**Figure 2.** Flowchart of ET retrieval using the "Temperature Sharpening and Flux Aggregation" method.

The spatial scale effect is  generally revealed by a discrepancy between different upscaling methods. In one method, parameters are upscaled to a large scale before calculating the heat flux. In the other method, heat flux is calculated at a small scale, and the results are then upscaled. In this study, the resolution of the final output result is 300 m.  To evaluate the heterogeneity reducing effect of TSFA, two other upscaling methods called IPUS and TRFA were implemented (see Fig. 3).

In the case of  IPUS, the inputs of the energy balance model are first retrieved at 30 m resolution (see section 3.2.1.1) and  then aggregated to 300 m resolution. Subsequently, these 300 m inputs are used in the one-source energy balance model to obtain the four energy balance components at 300 m resolution. In TRFA, the LST at 300 m is first resampled to 30 m using the nearest neighbour method and  the 30 m resolution inputs are  used for estimating ET . The outputs of the four energy-balance components of the TRFA are obtained using the area-weighting method shown in section 2.2.

[Figure]

**Figure 3.** Flowchart of the three upscaling methods for retrieving evapotranspiration

**3. Study area and Dataset**

**3.1. Study area**

Our study was conducted in the middle stream of the Heihe River Basin (HRB), which is located near the city of Zhangye in the arid region of Gansu Province in northwestern China (100.11 °E-100.16 °E, 39.10 °N-39.15 °N). The middle reach of the HRB is a typical desert-oasis agriculture ecosystem dominated by maize and wheat.  Areas of the Gobi Desert and the alpine vegetation  in the Qilian Mountains are located near the study area (see Fig. 4). The artificial oasis is highly heterogeneous, which impacts the thermal-dynamics and hydraulic features. Consequently, the water use efficiency and ET are variable. The Heihe River Basin has long served as a test bed for integrated watershed studies, as well as land surface  and hydrological experiments. Comprehensive experiments, such as the Watershed Allied Telemetry Experimental Research (WATER) project (Li et al., 2009), and an international experiment - the Heihe Basin Field Experiment (HEIFE)  as part of the World Climate Research Programme (WCRP), have  been conducted in this basin. One major objective of HiWATER was to capture the strong land surface heterogeneities and associated uncertainties within a watershed (Li et al., 2013a).

[Figure]

**Figure 4.** Study area and distribution of EC towers in HiWATER-MUSOEXE

**3.2. Dataset**

In this study,  data are mainly derived from the HJ-1B satellite. We combined these data with ancillary data and  *in situ* "Multi-Scale Observation Experiment on Evapotranspiration over heterogeneous land surfaces of The Heihe Watershed Allied Telemetry Experimental Research"

(HiWATER-MUSOEXE) data to estimate and validate the HJ-B land surface variables and heat fluxes.

**3.2.1. Remote sensing data**

**3.2.1.1. HJ-1B satellite data**

The specifications of HJ-1B are shown in Table 1. The satellite has quasi-sun-synchronous orbits at an altitude of 650 km, a swath width of 700 km and a revisit period of 4 days.

Combined, the revisit period of the satellites is 48 h. Because HJ-1 CCDs lack an onboard calibra- tion system, cross-calibration methods  were proposed to calibrate the CCD instru- ments  (Zhang et al., 2013; Zhong et al., 2014b). The image quality of the HJ-1A/B

CCD is stable, the performances of each band are balanced (Zhang et al., 2013) and the radiometric performance of the HJ-1A/B CCD sensors is similar to the performances of the Landsat-5 TM,

Observer-1 (EO-1) Advanced Land Imager and Terra ASTER. The image quality of the HJ-1 CCD

is very similar to the image quality of Landsat-5 TM (Jiang et al., 2013). In addition, the accuracy of the TIR band's onboard calibration meets the land surface temperature retrieval requirements but not the sea surface temperature retrieval requirements (Li et al., 2011b).  The Center for Re- sources Satellite Data and Application (CRESDA) in China releases calibration coefficients annually on its website (http://www.cresda.com). These data are freely available from the

CRESDA website (http://218.247.138.121/DSSPlatform/index.html).

**Table 1.** Specifications of the HJ-1B main payloads

| Sensor | Band | Spectral range (μm) | Spatial resolution (m) | Swath width (km) | Revisit time (days) |
|---|---|---|---|---|---|

| | | | | | |
|---|---|---|---|---|---|
| CCD | 1 | 0.43-0.52 | | | |
| | 2 | 0.52-0.60 | 30 | 360 (single) | 4 |
| | 3 | 0.63-0.69 | | 700 (two) | |
| | 4 | 0.76-0.90 | | | |
| IRS | 5 | 0.75-1.10 | | | |
| | 6 | 1.55-1.75 | 150 | 720 | 4 |
| | 7 | 3.50-3.90 | | | |
| | 8 | 10.5-12.5 | 300 | | |

[Figure]

**Figure 5.** Flowchart of  land surface variable retrieval. The abbreviations are defined as follows: SZA: solar zenith angle; SAA: solar azimuth angle; VZA: view zenith angle; AOD: aerosol optical depth; ABT: at-nadir brightness temperature; $S_d$: downward shortwave radiation; USR: upward shortwave radiation, ULR: upward longwave radiation; and $L_d$: downward longwave radiation.

We used the HJ-1B satellite data  from the HRB region in 2012. Because many variable-retrieving algorithms require clear-sky conditions  when calculating ET, we combined data quality information with visual interpretation to select satellite images without clouds. Considering the  period of  ground observations discussed in section 3.2.2, we obtained data for 11 days: June 19, June 30, July 8, July 27, August 2, August 15, August 22, August 29, September 2, September 13 and September 14.

The HJ-1B satellite data  of the HRB were pre-processed , including geometric correction, radiometric calibration, and  atmospheric correction.   The following surface variables are needed in Eqs. (1) to (10): downward shortwave radiation, downward longwave radiation, emissivity, albedo, fractional vegetation coverage (FVC), cloud mask data, meteorological data, LAI and LST. Fig. 5  illustrates a flowchart  of the retrieval of these variables.

(1) Surface albedo. According to the algorithm proposed by Liang et al. (2005) and Liu et al. (2011a), surface albedo was obtained from the top of the atmosphere (TOA) reflectance by the HJ-1 satellite  using a lookup table based on an angular bin regression relationship. The surface albedo and bidirectional reflectance distribution function (BRDF) of the HJ-1 satellite in the regression procedure were monitored  using POLDER-3/PARASOL BRDF datasets, and BRDF was used to obtain the TOA reflectance  in the 6S (Second Simulation of a Satellite Signal in the Solar Spectrum) radiation transfer mode.

(2) NDVI, FVC and LAI. The NDVI is the regression kernel of the temperature-sharpening procedure and is used to calculate the FVC. Atmospherically corrected surface reflectance values were used to calculate the NDVI as follows:

$$\text{NDVI} = \frac{\rho_{\text{nir}} - \rho_{\text{red}}}{\rho_{\text{nir}} + \rho_{\text{red}}} \tag{11}$$

and

$$\text{FVC} = \frac{\text{NDVI} - \text{NDVI}_s}{\text{NDVI}_v + \text{NDVI}_s} \tag{12}$$

where $\rho_{nir}$ and $\rho_{red}$ are the reflectances in the near-infrared and red band, respectively, and $\text{NDVI}_v$ and $\text{NDVI}_s$ are the fully vegetated and bare soil NDVI values, respectively. As an important input for the parameterization of surface roughness length and aerodynamic resistance, the LAI was determined using the following equation (Nilson, 1971):

$$P(\theta) = e^{-G(\theta) \cdot \Omega \cdot \text{LAI} / \cos(\theta)} \tag{13}$$
$$P(\theta) = 1 - \text{FVC} \tag{14}$$

where $\theta$ is the zenith angle, $P(\theta)$ is the angular distribution of the canopy gap fraction, $G(\theta)$ is the projection coefficient (0.5), and $\Omega$ is the total foliage clumping index, which can be obtained from the GLC global clumping index database according to the  land use type (He et al., 2012).

(3) Land surface emissivity (LSE). LSE is needed to calculate the $R_n$ and is extremely important for retrieving LST. In this paper, LSE was calculated using the FVC as follows (Valor and Caselles, 1996):

$$\varepsilon = \varepsilon_v \cdot \text{FVC} + \varepsilon_g (1 - \text{FVC}) + 4 < d\varepsilon > \cdot \text{FVC} \cdot (1 - \text{FVC}) \tag{15}$$

where $\varepsilon$ is the LSE, $< d\varepsilon >$ is an effective value of the cavity effect of emissivity, the mean $d\varepsilon$ of all vegetation species in this study is <dε>=0.015,  $\varepsilon_v$ and $\varepsilon_g$ are the vegetation and ground emissivity, respectively.

(4) Land surface temperature. A single-channel parametric model for retrieving LST based on HJ-1B/IRS TIR data developed by Li et al. (2010a) was employed to  obtain the LST. This model was developed from a parametric model based on MODTRAN4 using NCEP atmospheric profile data.

(5) Downward shortwave radiation. In this study, the algorithm proposed by Li et al. (2010b) was applied. MOD05, TOMS, aerosol and solar angle data were used to estimate the direct light flux and diffuse light flux  using a lookup table that was generated  via the 6S radiation transfer mode (Vermote et al., 2006). This method considered the influences of complex terrain, and a topographic correction was performed by using products of the ASTER digital elevation model (DEM).

(6) Downward longwave radiation ($L_d$). The TOA brightness temperature of the HJ-1B thermal channel was used to substitute the atmospheric effective temperature. Effective atmospheric emissivity was parameterized as an empirical function of the water vapor content. These values were substituted for atmospheric temperature and atmospheric emissivity to estimate the value of $L_d$. Because this $L_d$ retrieval method proposed by Yu et al. (2013) was only valid for clear-sky conditions, cloud masking information was used to determine clear skies. When cloud contamination existed in the image, the brightness temperature was relatively low, causing the $L_d$ to be lower than that in the cloudless images.

**3.2.1.2. Ancillary data**

Ancillary data were used because the bands of the satellite could not invert all of the variables needed  to retrieve ET.

(1) Atmospheric water vapor data. MODIS provides water vapor data (MOD05), including a

1-km near-infrared product and a 5-km thermal-infrared product, every day. The 1-km near-infrared water vapor product was used to retrieve $L_d$ in this study.

(2) Surface elevation data. We used the 30 m resolution Global Digital Elevation Model (GDEM) based on ASTER, which covers 83°N–83°S, to derive $S_d$.

(3) Atmosphere ozone data. A Total Ozone Mapping Spectrometer (TOMS), which was carried on an Earth Probe (EP) satellite, was used to derive $S_d$. The TOMS-EP provided daily global  atmospheric ozone data at a resolution of 1°×1.25° (Li et al., 2010b).

(4) Atmosphere profile data. Global reanalysis data from the National Centers for Environmental Prediction (NCEP) were used to derive LST. These data were generated globally every 6 hours (0:00, 06:00, 12:00, 18:00 UTC) for every 1° of latitude and longitude (Li et al., 2010a).

**3.2.2. HiWATER experiment dataset**

The *in situ* HRB observation data were provided by HiWATER. From June to September 2012, HiWATER designed  nested observation matrices over 30 km×30 km and 5.5 km×5.5 km within the middle stream oasis in Zhangye to focus on the heterogeneity of the scale effect in  HiWATER-MUSOEXE.

In the larger observation matrix, four eddy covariance (EC) systems and one superstation were installed in the oasis–desert ecosystem. Each station was supplemented with an automatic meteorological station (AMS) to record meteorological and soil variables and monitor the spatial–temporal variations of ET and its  associated factors (Li et al., 2013a). The station information is shown in Table 2**,** and the distribution of the stations is shown in Fig. 4. Within the artificial oasis, an observation matrix composed of 17 EC towers and ordinary AMSs exists where the superstation was located. The land surface was heterogeneous and dominated by maize, maize inter-cropped with spring wheat, vegetables, orchards and residential areas (Li et al., 2013a). Because the EC16 and HHZ stations lacked $R_n$ and G observation data, they were excluded from this study.

**Table 2.** The *in situ* HiWATER-MUSOEXE station information

| Station | Longitude (°) | Latitude (°) | Tower height (m) | Altitude (m) | Land cover |
|---------|---------------|--------------|------------------|--------------|------------|
| EC1 | 100.36E | 38.89N | 3.8 | 1552.75 | vegetation |
| EC2 | 100.35E | 38.89N | 3.7 | 1559.09 | maize |
| EC3 | 100.38E | 38.89N | 3.8 | 1543.05 | maize |
| EC4 | 100.36E | 38.88N | 4.2 | 1561.87 | building |
| EC5 | 100.35E | 38.88N | 3 | 1567.65 | maize |
| EC6 | 100.36E | 38.87N | 4.6 | 1562.97 | maize |
| EC7 | 100.37E | 38.88N | 3.8 | 1556.39 | maize |
| EC8 | 100.38E | 38.87N | 3.2 | 1550.06 | maize |
| EC9 | 100.39E | 38.87N | 3.9 | 1543.34 | maize |
| EC10 | 100.40E | 38.88N | 4.8 | 1534.73 | maize |
| EC11 | 100.34E | 38.87N | 3.5 | 1575.65 | maize |
| EC12 | 100.37E | 38.87N | 3.5 | 1559.25 | maize |
| EC13 | 100.38E | 38.86N | 5 | 1550.73 | maize |
| EC14 | 100.35E | 38.86N | 4.6 | 1570.23 | maize |
| EC15 | 100.37E | 38.86N | 4.5 | 1556.06 | maize |

| | | | | | |
|---|---|---|---|---|---|
| EC17 | 100.37E | 38.85N | 7 | 1559.63 | orchard |
| GB | 100.30E | 38.91N | 4.6 | 1562 | uncultivated land-Gobi |
| SSW | 100.49E | 38.79N | 4.6 | 1594 | uncultivated land-desert |
| SD | 100.45E | 38.98N | 5.2 | 1460 | swamp land |

The ground observation data included the H and LE. Reliable methods were used to ensure the
quality of the turbulent heat flux data. Before the main campaign, an intercomparison of all instru-
ments was conducted in the Gobi Desert (Xu et al., 2013). After basic processing, including spike
removal and corrections for density fluctuations (WPL correction), a four-step procedure was per-
formed to control the quality of the EC data. In this procedure, data were rejected when (1) the
sensor was  malfunctioned, (2) precipitation occurred within 1 h before or after col-
lection, (3) the  ratio of missing data was greater than 3% in the 30 min raw record and (4)
the friction velocity was below 0.1 ms$^{-1}$ at night (for more details see Liu et al., 2011b; Xu et al.,
2013; Liu et al., 2016). EC outputs are available every 30 min. G was measured by using three soil
heat plates at a depth of 6 cm at each site, and the surface G was calculated using the method pro-
posed by Yang and Wang(2008) based on the soil temperature and moisture above the plates. Sur-
face meteorological variables, such as wind speed, wind direction, relative humidity and air pressure,
were used to interpolate images using the inverse distance  weighting. Researchers
can obtain these data from the websites of the Cold and Arid Regions Science Data Center at
Lanzhou (http://card.westgis.ac.cn/) or the Heihe Plan Data Management Center
(http://www.heihedata.org/).

Energy imbalances  are common in ground flux observations. The conserving Bowen
ratio (H/LE) and residual closure technique are often used to force the energy balance. Computing
the LE as a residual variable may be a better method for energy balance closure under conditions
with large LEs (small or negative Bowen ratios due to strong advection) (Kustas et al., 2012). Thus,
the residual closure method was applied because the "oasis effect" was distinctly observed in the
desert-oasis system on clear days during the summer (Liu et al., 2011b).

**4. Results and analysis**

**4.1. Evaluation of surface variables**

To control model inputs and analyse error sources, the
coarse-resolution land surface temperature, downward shortwave radiation, downward longwave
radiation, $R_n$ and G were evaluated using *in situ* data.

The ground-based land surface temperature, $T_s$, was calculated using the Stefan-Boltzman
Law from the AMS measurements of the longwave radiation fluxes (Li et al., 2014) as follows:

$$T_s = \left[\frac{L^\uparrow - (1-\varepsilon_s) \cdot L^\downarrow}{\varepsilon_s \cdot \sigma}\right]^{\frac{1}{4}} \tag{16}$$

in which $L^\uparrow$ and $L^\downarrow$ are *in situ* surface upwelling and atmospheric downwelling longwave radiation,
respectively, and $\varepsilon_s$ is the surface broadband emissivity, which is regarded as the pixel value of the
HJ-1B at the AMS. The coefficient of determination $R^2$, mean bias error (MBE) and root mean
square error (RMSE) of the LST are 0.71, -0.14 K and 3.37 K, respectively. As  shown in Table
3, the accuracy of EC4 is low. The main causes of the large errors are as follows: (1) buildings and
soil/vegetation are distinct materials, the LSE algorithm may not be suitable for buildings and (2)
the EC4 foundation is non-uniform and is not suitable for validation. After removing the EC4 data, the $R^2$, MBE, and RMSE values of the LSTs were 0.83, 0.69 K and 2.51 K, respectively. The LST
errors of SSW and SD were large due to large errors on particular days. For example, although it
was briefly cloudy above station SSW on July 27, this area was not identified as cloudy in the cloud
detection process.

**Table 3.** The station validation results of land surface temperature

| Station | $R^2$ | MBE (K) | RMSE (K) | Station | $R^2$ | MBE (K) | RMSE (K) |
|---|---|---|---|---|---|---|---|
| EC1 | 0.82 | 0.18 | 1.74 | EC11 | 0.42 | 1.59 | 2.98 |
| EC2 | 0.82 | 0.59 | 1.54 | EC12 | 0.87 | 0.62 | 1.51 |
| EC3 | 0.69 | 0.38 | 1.90 | EC13 | 0.83 | 0.44 | 1.48 |
| EC4 | 0.83 | -9.87 | 10.04 | EC14 | 0.73 | 1.43 | 2.44 |
| EC5 | 0.83 | 1.71 | 2.34 | EC15 | 0.74 | 1.53 | 2.41 |
| EC6 | 0.61 | 0.30 | 2.44 | EC17 | 0.78 | 1.20 | 2.32 |
| EC7 | 0.82 | 0.39 | 1.40 | GB | 0.69 | 0.12 | 2.33 |
| EC8 | 0.83 | 0.45 | 1.55 | SSW | 0.59 | 1.38 | 3.82 |
| EC9 | 0.63 | 2.31 | 3.15 | SD | 0.76 | -3.83 | 4.84 |
| EC10 | 0.68 | 1.32 | 2.45 | | | | |

The $R^2$, MBE, and RMSE values of $S_d$ were 0.81, 13.80 W·m$^{-2}$, and 25.35 W·m$^{-2}$, respectively.
The station validation results are shown in Table 4. The accuracy of SSW is low. Because cloudy
conditions occurred briefly on July 27, few ground observations were obtained, and $S_d$ was signif-
icantly overestimated. After removing these data, the $R^2$, MBE, and RMSE values of $S_d$ at SSW
were 0.87, 10.90 W·m$^{-2}$ and 21.13 W·m$^{-2}$, respectively.

**Table 4.** The station validation results of downward shortwave radiation

| Station | $R^2$ | MBE (W·m$^{-2}$) | RMSE (W·m$^{-2}$) | Station | $R^2$ | MBE (W·m$^{-2}$) | RMSE (W·m$^{-2}$) |
|---|---|---|---|---|---|---|---|
| EC1 | 0.97 | 25.23 | 27.73 | EC11 | 0.90 | 30.11 | 33.76 |
| EC2 | 0.84 | 28.29 | 33.57 | EC12 | 0.96 | 24.35 | 26.43 |
| EC3 | 0.97 | 17.56 | 19.25 | EC13 | 0.93 | 12.41 | 17.92 |
| EC4 | 0.98 | 6.07 | 9.34 | EC14 | 0.98 | 32.40 | 33.49 |
| EC5 | 0.98 | 10.60 | 12.29 | EC15 | 0.94 | 26.71 | 29.71 |
| EC6 | 0.93 | 27.68 | 30.71 | EC17 | 0.94 | -20.25 | 24.54 |
| EC7 | 0.89 | -17.69 | 27.59 | GB | 0.89 | 25.34 | 30.63 |
| EC8 | 0.83 | 15.63 | 25.50 | SSW | 0.63 | 18.51 | 34.93 |
| EC9 | 0.96 | -2.27 | 9.96 | SD | 0.98 | 5.70 | 13.82 |
| EC10 | 0.94 | -3.50 | 11.97 | | | | |

The $R^2$, MBE, and RMSE values of the HRB $L_d$ were 0.73, 0.28 W·m$^{-2}$, and 21.24 W·m$^{-2}$,
respectively. As seen in Table 5, the accuracies at EC3, SD and SSW were low. The low accuracies
at EC3 and SD potentially resulted from (1) high humidity, which resulted in low at-nadir brightness
temperatures and low retrieved $L_d$, or (2) instrument error, which occurred because the EC3 ground
observations were always greater than those of the other stations during the same period. Although
SSW was located in a desert, the ground-air temperature difference was large. The $L_d$ retrieval may
have a large error because the models use surface temperature when estimating $L_d$ to approximate
or substitute the near-surface temperature (Yu et al., 2013). The corrected error of our $L_d$ retrieving algorithm resulted from the ground-air temperature difference in non-vegetated areas. The inaccu-
racy of the SSW LST may influence the $L_d$ results.

**Table 5.** The station validation results of downward longwave radiation

| Station | $R^2$ | MBE (W·m$^{-2}$) | RMSE (W·m$^{-2}$) | Station | $R^2$ | MBE (W·m$^{-2}$) | RMSE (W·m$^{-2}$) |
|---|---|---|---|---|---|---|---|
| EC1 | 0.85 | 4.16 | 17.21 | EC11 | 0.93 | -2.72 | 10.55 |
| EC2 | 0.88 | 0.11 | 14.23 | EC12 | 0.87 | -0.84 | 14.80 |
| EC3 | 0.91 | -35.65 | 37.88 | EC13 | 0.86 | -7.28 | 15.98 |
| EC4 | 0.88 | 3.36 | 16.38 | EC14 | 0.82 | 4.07 | 16.42 |
| EC5 | 0.88 | -0.79 | 15.02 | EC15 | 0.85 | 17.67 | 23.06 |
| EC6 | 0.84 | 2.55 | 15.43 | EC17 | 0.90 | -1.11 | 12.87 |
| EC7 | 0.75 | -5.90 | 19.72 | GB | 0.88 | 9.50 | 27.82 |
| EC8 | 0.80 | -1.35 | 17.49 | SSW | 0.85 | 25.33 | 34.50 |
| EC9 | 0.86 | 10.44 | 17.99 | SD | 0.85 | -26.54 | 34.08 |
| EC10 | 0.87 | 7.98 | 16.05 | | | | |

The $R^2$, MBE and RMSE values of the HRB $R_n$ were 0.70, -9.64 W·m$^{-2}$, and 42.77 W·m$^{-2}$,
respectively. The station-validated  results of $R_n$ are shown in Table 6, which indi-
cates that the accuracies of EC4, EC7, EC17 and SSW were relatively low. According to the sensi-
tivity analysis of Eq. (8), $L_d$ and $S_d$ are highly sensitive variables when calculating $R_n$, while the
albedo, LSE and LST are not as sensitive. Although LST was not a sensitive variable, the
MBE and RMSE values of LST at EC4 reached -9.87 K and 10.04 K because the land cover
of EC4 was maize at  300 m resolution. However, the observation tower was located in a built-
up area, which potentially caused errors when estimating $R_n$. The accuracies of  $S_d$ and
$L_d$ at EC7 were low on several days, and  MBE=-43.40 W·m$^{-2}$ and
RMSE=50.50 W·m$^{-2}$ after removing these data. EC17 was  located in an orchard, and the
signal that was received by the sensors at EC17  was affected by the complex vertical structure
of the orchard ecosystem. The information  of substrate plants may be ignored, leading to albedo
retrieval errors.  An albedo bias of 0.03 can lead to an $R_n$ error of approximately 20 W·m$^{-2}$
when the solar incoming radiation is large. As previously  discussed, it was briefly cloudy
on July 27, and after removing  those data, the $R^2$, MBE and RMSE values of the $R_n$ obtained
at station SSW were 0.72, 8.20 W·m$^{-2}$, and 37.60 W·m$^{-2}$, respectively.

**Table 6.** The station net radiation validation results

| Station | $R^2$ | MBE (W·m$^{-2}$) | RMSE (W·m$^{-2}$) | Station | $R^2$ | MBE (W·m$^{-2}$) | RMSE (W·m$^{-2}$) |
|---|---|---|---|---|---|---|---|
| EC1 | 0.76 | -2.55 | 30.61 | EC11 | 0.86 | -15.13 | 28.05 |
| EC2 | 0.79 | 2.52 | 25.24 | EC12 | 0.90 | -8.46 | 19.38 |
| EC3 | 0.86 | -35.84 | 42.97 | EC13 | 0.88 | -25.73 | 32.34 |
| EC4 | 0.84 | 76.64 | 80.25 | EC14 | 0.90 | 4.23 | 18.18 |
| EC5 | 0.85 | -24.41 | 32.34 | EC15 | 0.84 | 8.33 | 23.01 |
| EC6 | 0.82 | 4.35 | 23.44 | EC17 | 0.89 | -62.62 | 68.11 |
| EC7 | 0.61 | -58.66 | 67.83 | GB | 0.77 | -10.40 | 38.86 |
| EC8 | 0.83 | -20.62 | 32.45 | SSW | 0.44 | 23.05 | 62.93 |
| EC9 | 0.87 | -29.60 | 36.27 | SD | 0.75 | 19.98 | 35.24 |

| | | | | | | | |
|---|---|---|---|---|---|---|---|
| EC10 | 0.83 | -24.35 | 33.51 | | | | |

The $R^2$, MBE, and RMSE values of the G in the HRB were 0.57, 8.51 W·m⁻² , and 29.73 W·m⁻², respectively. The station-validated G  results are shown in Table 7.  At EC5, the soil temperature and moisture were the same at different depths after July 19, which resulted in a surface G that was equal to the G at a depth of 6 cm.  G below the surface was usually less than the G at the soil surface; thus, the validation results of  G at EC5 indicate that G was overestimated.  At SSW, the brief cloudy period decreased the observed soil surface temperature, which decreased the calculated surface G. However, the remotely sensed G did not reflect this situation. In this case, the G was overestimated because the $R_n$ was overestimated. After removing the data on July 27, the $R^2$, MBE, and RMSE values of the G at SSW were 0.17, 19.34 W·m⁻², and 33.30 W·m⁻², respectively.

**Table 7.** The station validation results of the soil heat flux

| Station | $R^2$ | MBE (W·m⁻²) | RMSE (W·m⁻²) | Station | $R^2$ | MBE (W·m⁻²) | RMSE (W·m⁻²) |
|---|---|---|---|---|---|---|---|
| EC1 | 0.50 | 19.73 | 31.53 | EC11 | 0.71 | 4.23 | 19.23 |
| EC2 | 0.24 | 20.78 | 28.72 | EC12 | 0.53 | 20.29 | 24.79 |
| EC3 | 0.03 | -1.15 | 36.28 | EC13 | 0.91 | -0.89 | 17.27 |
| EC4 | 0.45 | 18.50 | 22.29 | EC14 | 0.82 | -1.89 | 18.72 |
| EC5 | 0.38 | 41.87 | 60.19 | EC15 | 0.78 | 6.68 | 15.80 |
| EC6 | 0.83 | -5.91 | 14.57 | EC17 | 0.49 | 8.26 | 33.59 |
| EC7 | 0.28 | 7.50 | 24.65 | GB | 0.29 | -17.86 | 26.81 |
| EC8 | 0.68 | -5.73 | 20.15 | SSW | 0.01 | 30.41 | 51.87 |
| EC9 | 0.61 | 6.83 | 26.96 | SD | 0.71 | -4.79 | 13.71 |
| EC10 | 0.41 | 7.68 | 28.67 | | | | |

**4.2. Validation of heat fluxes by TSFA**

Fig. 6 provides the turbulent heat flux results calculated by TSFA on September 13, 2012. The spatial distribution of the turbulent heat flux is obvious. The H values of buildings and uncultivated land, including  land patches in the Gobi region, barren areas and desert areas,  were high, in addition to the LEs of  water and agricultural areas in the oasis. The southern areas of the images show uncultivated barren land bordering the Qilian Mountains that resulted from snowmelt and the downward movement of water. In these areas, the groundwater levels are high and the soil moisture content is approximately 30% based on *in situ* measurements at a depth of 2 cm. Therefore, the LE values of barren areas in the south  are higher than the LE values of desert areas in the southeast, although both areas were classified as uncultivated land.

Studies have shown that validation methods that consider the source area are more appropriate for evaluating ET models than traditional validation methods based on a single pixel (Jia et al., 2012; Song et al., 2012). In this study, a user-friendly tool presented by Neftel et al. (2008) , which is based on the Eulerian analytic flux footprint model proposed by Kormann and Meixner (2001) was used to calculate the footprints of the function parameters. The continuous footprint function was dispersed based on the relative weights of the pixels  in which the source area  was located.

[Figure]

**Figure 6.** Maps of the four energy components, (a) Rn, (b) G, (c) H and (d) LE, calculated by TSFA on September

13, 2012.

[Figure]

**Figure 7.** Scatter plot of the TSFA turbulent heat flux results

The footprint validation results of the TSFA turbulent heat fluxes are shown in Fig. 7 and Table

8. The $R^2$, MBE, and RMSE of  H were 0.61, 0.90 W·m$^{-2}$ and 50.99 W·m$^{-2}$, respectively, and those  of LE were 0.82, -20.54 W·m$^{-2}$ and 71.24 W·m$^{-2}$, respectively. Because  LE was calculated as a residual term, it was impacted by  $R_n$, surface G and H. The errors of all inputs may contribute to the LE, which complicates the error sources of the LE.  These errors are discussed in detail in sections 4.3.2 and 4.4.

**Table 8.** *In situ* validation results of heat flux  using the TSFA

| | TSFA-H (W·m$^{-2}$) | | | TSFA-LE (W·m$^{-2}$) | | |
|---|---|---|---|---|---|---|
| Date | $R^2$ | MBE | RMSE | $R^2$ | MBE | RMSE |
| 0619 | 0.39 | 44.73 | 66.38 | 0.69 | -44.15 | 80.60 |
| 0630 | 0.73 | 23.71 | 38.96 | 0.88 | -63.81 | 77.83 |
| 0708 | 0.55 | 32.70 | 58.72 | 0.85 | -43.02 | 72.32 |
| 0727 | 0.90 | -34.34 | 43.59 | 0.92 | 26.74 | 57.60 |
| 0803 | 0.80 | -4.77 | 18.92 | 0.78 | -4.58 | 47.86 |
| 0815 | 0.74 | -18.37 | 38.82 | 0.93 | 4.75 | 35.41 |
| 0822 | 0.40 | 31.64 | 66.21 | 0.65 | -44.44 | 93.81 |
| 0829 | 0.79 | 23.01 | 38.36 | 0.79 | -50.45 | 77.99 |
| 0902 | 0.21 | -45.10 | 74.81 | 0.54 | 24.39 | 69.31 |
| 0913 | 0.25 | -9.64 | 41.01 | 0.59 | -59.36 | 82.77 |
| 0914 | 0.31 | -34.11 | 50.88 | 0.47 | 27.99 | 67.50 |

As seen shown in Fig. 7, most the majority of the H values are small because June, July, August
and September constitute the growing season when ET greatly cools the air. The differential tem-
perature difference between the land surface and air is small, leading to a low H. The points with
large H values are influenced by uncultivated land. In our study area, Gobi area region, barren area
and desert area compose comprise the uncultivated land. The points in the scatter plot with large H
values represent desert, where the H values reach approximately 300 W·m$^{-2}$. Some points in the H
scatter plot are less than 0 due to inversion from the "oasis effect" or irrigation. For example, the
HiWATER's soil moisture data show that irrigation occurred on August 22, 2012. Irrigation is the
main source of water within the oasis and cools the land surface to temperatures below the air tem-
perature. In addition, irrigation leads to errors in LST retrieval because it increases the atmospheric
water vapor content, as discussed in section 4.1. The model error is further analyzed in section 4.4.

**4.3. Comparison between TSFA, TRFA and IPUS**

To verify whether the TSFA method can simulate the heterogeneity of the land surface, the
TRFA and IPUS methods were also implemented for comparison purposes. These three methods
were evaluated using (1) validation of TRFA and IPUS based on *in situ* measurements and (2) quan-
titative analysis based on the spatial distribution and scatter plots of the four energy balance com-
ponents.

**4.3.1. Validation of the TRFA and IPUS heat fluxes**

Table 9 provides the footprint *in situ* validation results of the H and LE calculated using the
IPUS and TRFA methods. Comparing with Compared to validation results of TSFA in Table 8, the
TSFA had produced a better retrieval accuracy than the TRFA, and the TRFA was better than the
IPUS on all days, because and the MBE and RMSE values of TSFA decreased and the $R^2$ of TSFA
increased on most days. Table 9 shows that the improvements of in accuracy between TRFA and
IPUS was were relatively larger than the ones those between TSFA and TRFA. Compared with to
the IPUS results, the TRFA results were similar to the TSFA results since because the sub-pixel
landscapes and sub-pixel variations of most variables were considered. Thus, TRFA could effec-
tively decreased the scale error that resulted from heterogeneity because the 30 m VNIR data were
fully used. However, the performance of the TRFA method is unstable. For example, on August 3
and August 29, the TRFA results were slightly worse than the IPUS results. This situation occurred because the different sub-pixel landscape temperatures were  considered as equal to the val- ues estimated at the 300 m resolution. Thus, when  values of LST at 300 m scale  have large  retrieval errors, the turbulent heat flux  retrieval error may be amplified by the sub-pixel landscapes.

**Table 9.** *In situ* validation results of the turbulent heat fluxes of IPUS and TRFA

| | IPUS-H($W \cdot m^{-2}$) | | | IPUS-LE ($W \cdot m^{-2}$) | | | TRFA-H ($W \cdot m^{-2}$) | | | TRFA-LE ($W \cdot m^{-2}$) | | |
|---|---|---|---|---|---|---|---|---|---|---|---|---|
| date | $R^2$ | MBE | RMSE | $R^2$ | MBE | RMSE | $R^2$ | MBE | RMSE | $R^2$ | MBE | RMSE |
| 0619 | 0.32 | 48.53 | 71.70 | 0.66 | -47.68 | 86.02 | 0.39 | 52.28 | 70.98 | 0.65 | -46.71 | 85.93 |
| 0630 | 0.50 | 41.45 | 67.30 | 0.80 | -81.75 | 102.33 | 0.69 | 42.64 | 60.85 | 0.86 | -78.50 | 93.98 |
| 0708 | 0.34 | 44.17 | 77.45 | 0.63 | -66.75 | 118.63 | 0.44 | 54.20 | 76.00 | 0.82 | -63.82 | 89.11 |
| 0727 | 0.81 | -33.14 | 50.01 | 0.83 | 25.61 | 74.26 | 0.84 | -23.53 | 41.76 | 0.86 | 14.82 | 65.21 |
| 0803 | 0.84 | -5.23 | 33.50 | 0.74 | -3.98 | 60.49 | 0.80 | 7.76 | 37.51 | 0.76 | -18.23 | 62.71 |
| 0815 | 0.64 | -23.28 | 47.89 | 0.85 | 10.32 | 54.98 | 0.70 | -14.77 | 39.99 | 0.89 | 0.59 | 45.22 |
| 0822 | 0.31 | 41.50 | 74.81 | 0.61 | -53.60 | 102.12 | 0.40 | 40.63 | 69.94 | 0.65 | -54.17 | 98.97 |
| 0829 | 0.72 | 27.15 | 44.16 | 0.76 | -54.76 | 83.20 | 0.75 | 30.79 | 44.97 | 0.77 | -59.43 | 86.22 |
| 0902 | 0.28 | -52.44 | 83.25 | 0.51 | 32.89 | 76.48 | 0.21 | -45.77 | 75.84 | 0.52 | 24.37 | 71.69 |
| 0913 | 0.08 | -11.45 | 57.50 | 0.61 | -57.38 | 81.83 | 0.06 | -11.89 | 49.63 | 0.54 | -57.78 | 84.58 |
| 0914 | 0.12 | -36.52 | 67.38 | 0.28 | 19.46 | 89.30 | 0.03 | -34.34 | 64.85 | 0.38 | 25.41 | 75.96 |

Variations in landscape characteristics systematically trigger variations in surface variables. Landscape inhomo- geneity can be classified using two conditions: nonlinear vegetation density variations between sub- pixels (e.g., different types of vegetation mixed with each other or with bare soil) and coarse pixels containing different landscapes (e.g., vegetation or bare soil mixed with buildings or water). To evaluate the effects of TSFA, stations with a typical severe heterogeneous surface at EC4, a weak heterogeneous surface at EC11, a typical pixel (called "TP" hereafter) at the boundary of the oasis and bare soil (sample 62, line 102 in the image of study area), and a uniform surface at EC15, were selected to analyze the temperature sharpening results.

EC4 is used as an example because its land cover and sub-pixel variations of temperature were complicated. Table 10 compares the turbulent heat fluxes calculated using the IPUS, TRFA and

TSFA methods. Significant differences were observed between the TSFA and IPUS results and be- tween the TRFA and IPUS results due to the heterogeneity of the surface. The LE calculated using the TSFA method was more consistent with *in situ* measurements than the LE calculated using the

IPUS method because the MBE and RMSE decreased  considerably, the $R^2$ increased, and the accuracy was improved by approximately 40 $W \cdot m^{-2}$. However, the LE calculated by  the

TRFA was more accurate than the LE calculated by  the TSFA, as discussed below.

The H calculated by using the TSFA method was more accurate than the H calculated by using the TRFA and IPUS methods. The  RMSE of the results from the TRFA method was rela- tively close to the  RMSE of the results from the TSFA method because the TRFA method also considers the effects of the heterogeneity of landscapes. In addition, the H values obtained from the TRFA method were always greater than those obtained from the TSFA method. Because the

TSFA turbulent heat flux results are the same as the TRFA turbulent heat flux results for buildings and water bodies in our pixel ET algorithm,  the difference between the TSFA and TRFA results depend on the vegetation and bare soil.  Additionally, the 300 m resolution LST is larger than the LST of the sub-pixels, such as pixels containing vegetation or bare soil. This relationship occurs for two reasons: (1) the coarse pixels contain buildings and result in a larger 300 m resolution LST
and (2) the LSTs were underestimated at EC4 (as shown in Table 3), which would underestimate
the value of $\Delta\widehat{T}_{300}$ in Eq. (3) and, consequently, the sharpening temperature at 30 m and H. Be-
cause the LE was calculated as a residual item in the energy balance equation, the errors of the other
three energy balance components would accumulate in the LE term. At EC4, the $R_n$ was overesti-
mated by approximately 80 W·m$^{-2}$, as discussed in section 4.1, but the scale effect of $R_n$ was not
obvious (see section 4.3.2), and the G was overestimated by approximately 20 W·m$^{-2}$. These results
would lead to low decreased the accuracy of the available energy and overestimated the error by 60
W·m$^{-2}$. As Because the TRFA overestimates H, the underestimation of H in the TSFA would result
in larger overestimation of LE than that estimated by the TRFA. Consequently, the LE calculated
by using the TSFA method is less accurate than the LE calculated by using the TRFA method.

**Table 10.** Comparison of the turbulent heat flux results at EC4

| EC4 | H(W·m$^{-2}$) | | | | LE(W·m$^{-2}$) | | | |
|------|------|------|------|------|------|------|------|------|
| Date | EC | IPUS | TRFA | TSFA | EC | IPUS | TRFA | TSFA |
| 0619 | 150.65 | 105.86 | 154.71 | 142.13 | 278.55 | 402.60 | 344.05 | 357.79 |
| 0630 | 138.32 | 99.91 | 153.53 | 126.88 | 341.98 | 419.83 | 358.12 | 386.07 |
| 0708 | 117.04 | 63.47 | 131.79 | 112.16 | 361.16 | 502.60 | 424.85 | 444.01 |
| 0727 | 136.41 | 4.87 | 85.99 | 72.33 | 306.53 | 543.48 | 452.01 | 467.96 |
| 0803 | 68.97 | 36.51 | 111.73 | 74.76 | 389.63 | 498.21 | 414.67 | 454.23 |
| 0815 | 104.60 | 12.69 | 88.26 | 82.56 | 357.34 | 522.31 | 436.43 | 441.95 |
| 0822 | 125.34 | 85.93 | 120.68 | 93.18 | 318.08 | 415.15 | 370.76 | 400.99 |
| 0829 | 82.93 | 73.06 | 103.84 | 74.76 | 317.68 | 362.04 | 322.77 | 355.16 |
| 0902 | 162.05 | 93.74 | 144.49 | 132.60 | 280.41 | 375.42 | 315.16 | 326.29 |
| 0913 | 119.42 | 151.44 | 157.07 | 130.85 | 263.18 | 234.93 | 222.62 | 249.59 |
| 0914 | 110.02 | 88.24 | 128.37 | 99.33 | 262.33 | 333.82 | 285.04 | 314.91 |

units: W·m$^{-2}$

| | IPUS | | | TRFA | | | TSFA | | |
|------|------|------|------|------|------|------|------|------|------|
| Variable | $R^2$ | MBE | RMSE | $R^2$ | MBE | RMSE | $R^2$ | MBE | RMSE |
| EC4-H | 0.11 | -44.65 | 61.73 | 0.25 | 5.88 | 26.33 | 0.51 | -16.93 | 26.54 |
| EC4-LE | 0.49 | 99.21 | 119.55 | 0.56 | 42.69 | 62.40 | 0.60 | 63.92 | 76.78 |

Fig. 8 shows that the classes and temperatures of 10×10 sub-pixels at 30 m correspond to the
pixels with a resolution of 300 m at the EC tower. In the IPUS upscaling scheme, the 300 m pixels
included buildings, and maize and vegetable crops at the 30 m resolution and were identified as
maize. The canopy height gap between maize and vegetables was large during our study period,
resulting in the overestimation of the canopy height. For more additional details, see the error anal-
ysis in section 4.4. However, because buildings corresponded with $H = 0.6R_n$ in this study, ignor-
ing the contributions of buildings would result in the underestimation of H. Fig. 8(a) shows the
temperature-sharpening results for in the EC4 pixel on August 29. The temperature achieved re-
trieved at a resolution of 300 m scale was 303.49 K. Compared with the *in situ* measurement of
313.24 K, the temperature was underestimated at a resolution of 300 m was underestimated. Even
when substituting the *in situ* temperature into the ET model, the value of H reached 399.60 W·m$^{-2}$
and the LE became 0 W·m$^{-2}$. When substituting the *in situ* temperature into the TRFA method, H
was 396.49 W·m$^{-2}$ and LE was 18.7 W·m$^{-2}$, indicating that the LE was underestimated and the H
was overestimated with large errors. After processing by temperature sharpening, the distribution of the temperature at the 30 m resolution agreed with the classification. Temperature sharpening improved the description of heterogeneity based on the thermodynamic-driven force of the turbulent heat flux. These results apply to the ET model  based on the classification map and high-reso- lution inputs and correspond  to more accurate sensible heat flux  estimates.

[Figure]

**Figure 8.** Distribution of classes and temperatures over (a) EC4, (b) EC15, (c) EC11 and (d) TP on August 29, 2012

The land surface of EC15 was uniform and  consisted of pure pixels covering maize fields. Consequently, the temperature distribution at  30 m resolution was very homogeneous, and the  surface temperature variations were comprised within a range of 1.6 K. Table 11 shows the *in situ* validation results  at EC15, for which the overall accuracy is not high due to the low LST retrieval accuracy on July 8, which is discussed in section 4.4.1. For  homogeneous surfaces, the gaps between IPUS, TRFA and TSFA were not large (within 10 W·m$^{-2}$), and the accuracy did not improve (MBE and RMSE did not  exhibit obvious variations). Statistically sharpening the temperature may increase the uncertainty of the model results for  homogeneous surface; however, this influence could be omitted.

**Table 11.** Comparison of the turbulent heat fluxes results at EC15

| EC15 | H (W·m$^{-2}$) | | | LE (W·m$^{-2}$) | | | |
|---|---|---|---|---|---|---|---|
| Date | EC | IPUS | TRFA | TSFA | EC | IPUS | TRFA | TSFA |
| 0619 | 92.55 | 106.60 | 109.25 | 99.81 | 419.47 | 427.19 | 419.99 | 429.98 |
| 0630 | 42.37 | 43.99 | 45.51 | 44.67 | 551.73 | 527.12 | 525.17 | 526.09 |
| 0708 | 18.34 | 217.53 | 235.48 | 209.90 | 620.95 | 425.71 | 397.49 | 424.86 |
| 0727 | 27.68 | 21.22 | 31.11 | 24.30 | 597.76 | 589.58 | 579.43 | 586.47 |
| 0803 | 2.33 | 33.32 | -0.07 | 0.01 | 592.37 | 565.20 | 601.33 | 601.33 |
| 0815 | 48.81 | 32.31 | 46.28 | 44.62 | 553.74 | 561.92 | 547.48 | 549.11 |
| 0822 | 54.59 | 154.34 | 151.77 | 158.60 | 473.68 | 408.37 | 410.80 | 405.07 |
| 0829 | 9.80 | 94.97 | 95.01 | 90.91 | 473.54 | 399.25 | 398.52 | 402.93 |
| 0913 | 176.96 | 265.62 | 209.65 | 257.81 | 307.72 | 165.40 | 221.68 | 173.58 |
| 0914 | 188.34 | 198.15 | 197.04 | 196.60 | 274.98 | 275.07 | 276.05 | 276.56 |

units: W·m$^{-2}$

| | IPUS | | | TRFA | | | TSFA | | |
|---|---|---|---|---|---|---|---|---|---|
| Variable | $R^2$ | MBE | RMSE | $R^2$ | MBE | RMSE | $R^2$ | MBE | RMSE |
| EC15-H | 0.40 | 40.64 | 74.64 | 0.33 | 45.93 | 80.81 | 0.40 | 40.36 | 72.88 |
| EC15-LE | 0.74 | -52.11 | 83.48 | 0.71 | -48.80 | 82.51 | 0.74 | -49.00 | 81.94 |

The weak heterogeneous land surface at EC11 contained barley, maize and vegetables in a 300 m pixel resolution with a fractional area of 58:41:1 and was classified as barley at the 300 m resolution. The distributions of the classes and temperatures are shown in Fig. 8(c). The pixel belongs to the first conditions of heterogeneity (nonlinear vegetation density variation between sub-pixels). Table 12 shows the *in situ* validation results of EC11.  The improvements in the accuracies of H and LE by temperature resampling or sharpening were not as obvious as the improvements at EC4, which contained total different landscapes (the other inhomogeneous condition).

Theoretically, the LE from the TSFA and TRFA at EC11 should be smaller than  the IPUS LE values in the energy balance system. The height of maize (range from 0.3 to 2 m) was  generally higher than the height of barley (range from 0.9 to 1.1 m) in the study area from June to August. Taller vegetation resulted in  larger surface roughness and smaller aerodynamic resistance, which led to larger H values and smaller LE values, and vice versa (e.g., vegetables with a canopy height of 0.2 m). When using the TSFA and TRFA methods, patch landscapes consisting of different crops, such as maize and vegetables, were considered. Thus, the LE was smaller than the IPUS LE. On June 19, the canopy height of maize was 0.74 m, which was lower than the canopy height of barley (1 m) and indicated that the H values resulted from the TRFA and TSFA methods were less than  H resulted from the IPUS method. Because our validation method considered the influence of source area, the *in situ* turbulent heat flux validation results included the effects of neighboring pixels (i.e., on August 3, the turbulent heat flux values of the
pixel corresponding  to the location of EC11 was only  assigned a weight of 37% in
the source area).
The differences between the TSFA and TRFA methods  were small and resulted from the
LST differences between the 30 m resolution temperature-sharpening results and the LST retrieved
at the 300 m resolution , but these differences were not evident at EC11. For example, on August
29, the temperature range was 1.4 K, as shown in Fig. 8(c). This temperature was even less than the
temperature range at EC15 because the observation system at EC15 was a superstation with a 40 m
tall tower that may cause  large shadow effects and result in a relative large temperature range.
Hence, the temperature sharpening effect is not obvious after aggregating the flux at the 300 m
resolution under dense vegetation canopies. However, temperature sharpening can still decrease the
heterogeneity that results from thermal dynamics.
The excess errors at EC11 was caused by  the relatively low LST accuracy, with
$R^2$, MBE, and RMSE values of 0.42, 1.59 K and 2.98 K, respectively. On August 29, the temperature
retrieved at  300 m scale was 301.6 K, and the  observed ground
temperature was 300.20 K. The LST at the 300 m resolution was slightly overestimated. When the
*in situ* temperature was substituted into the IPUS algorithm, the value of H decreased to 16.06 W·m$^{-}$
$^{2}$ and the LE became 467.43 W·m$^{-2}$. When  the *in situ* temperature was substituted into
the TRFA scheme, the value of H was 22.43 W·m$^{-2}$ and the LE was 461.58 W·m$^{-2}$, which were
similar to the ground observations.

**Table 12.** Comparison of the turbulent heat flux results at EC11

| EC11 | H(W·m$^{-2}$) | | | | LE(W·m$^{-2}$) | | | |
|------|------|------|------|------|------|------|------|------|
| Date | EC | IPUS | TRFA | TSFA | EC | IPUS | TRFA | TSFA |
| 0619 | 33.94 | 173.69 | 158.12 | 158.18 | 531.46 | 391.60 | 407.42 | 407.40 |
| 0630 | 25.03 | 3.29 | 23.12 | 21.37 | 635.22 | 586.37 | 566.48 | 568.28 |
| 0708 | 32.29 | 68.17 | 97.16 | 96.13 | 601.98 | 567.73 | 538.77 | 539.81 |
| 0727 | 21.42 | -1.17 | -1.58 | -3.77 | 587.70 | 618.80 | 619.19 | 621.46 |
| 0803 | 7.01 | 24.85 | 20.34 | 19.52 | 614.28 | 575.03 | 585.29 | 586.16 |
| 0815 | 38.94 | 12.51 | 15.52 | 16.02 | 567.07 | 584.31 | 581.31 | 580.82 |
| 0822 | 69.25 | 73.45 | 83.11 | 84.38 | 516.07 | 483.23 | 473.60 | 472.40 |
| 0829 | 29.77 | 48.21 | 60.9 | 60.81 | 473.22 | 427.92 | 415.32 | 415.45 |
| 0902 | 193.97 | 154.58 | 197.01 | 197.49 | 306.62 | 361.96 | 319.54 | 319.03 |
| 0913 | 288.37 | 168.42 | 176.4 | 177.71 | 160.29 | 216.53 | 208.49 | 207.19 |
| 0914 | 240.33 | 268.91 | 256.29 | 256.40 | 199.52 | 156.00 | 168.63 | 168.55 |

units: W·m$^{-2}$

| | IPUS | | | TRFA | | | TSFA | | |
|------|------|------|------|------|------|------|------|------|------|
| Variable | $R^2$ | MBE | RMSE | $R^2$ | MBE | RMSE | $R^2$ | MBE | RMSE |
| EC11-H | 0.61 | -1.07 | 61.31 | 0.57 | -0.36 | 63.24 | 0.67 | -0.21 | 55.50 |
| EC11-LE | 0.88 | -19.83 | 63.16 | 0.89 | -18.12 | 60.02 | 0.90 | -21.29 | 58.11 |

Another typical pixel located at the boundary of the bare soil and the oasis with no flux meas-
urements was used to evaluate the correction effects of landscapes and temperature sharpening. The
land surface of the TP contained maize, vegetables and bare soil at a fraction of 35:31:34. Table 13
shows that when neither the heterogeneity of the landscape nor the LST are considered, the relative
error of LE  reached 180 W·m$^{-2}$. In addition, if only the LST heterogeneity is not considered, the LE relative error  reached 48 W·m$^{-2}$. This result also reveals that the influences of landscape inhomogeneity are greater than the influences of inhomogeneity on the LST in mixed pixels.

**Table 13.** Comparison of the turbulent heat flux results at TP

| | H (W·m$^{-2}$) | | | LE (W·m$^{-2}$) | | |
|---|---|---|---|---|---|---|
| Date | IPUS | TRFA | TSFA | IPUS | TRFA | TSFA |
| 0619 | 186.31 | 149.73 | 143.98 | 321.04 | 358.22 | 364.79 |
| 0630 | 383.65 | 191.59 | 158.79 | 67.03 | 259.36 | 292.89 |
| 0708 | 498.36 | 240.20 | 204.18 | 0.29 | 259.25 | 293.41 |
| 0727 | 276.79 | 136.06 | 84.01 | 206.52 | 347.64 | 402.23 |
| 0803 | 214.14 | 75.45 | 53.72 | 252.37 | 392.08 | 416.41 |
| 0815 | 214.14 | 98.24 | 72.05 | 252.37 | 368.64 | 393.68 |
| 0822 | 436.48 | 369.28 | 276.70 | 0.00 | 67.79 | 162.80 |
| 0829 | 235.29 | 117.16 | 67.21 | 183.62 | 302.41 | 356.75 |
| 0902 | 423.61 | 212.15 | 180.92 | 0.00 | 211.77 | 241.36 |
| 0913 | 338.00 | 285.04 | 216.26 | 0.00 | 53.62 | 122.58 |
| 0914 | 270.44 | 148.20 | 100.19 | 115.19 | 238.43 | 286.51 |

units: W·m$^{-2}$

| | IPUS | | | TRFA | | |
|---|---|---|---|---|---|---|
| Variable | $R^2$ | MBE | RMSE | $R^2$ | MBE | RMSE |
| TP-H | 0.62 | 174.47 | 185.49 | 0.95 | 42.28 | 48.01 |
| TP-LE | 0.71 | -175.91 | 186.63 | 0.97 | -43.11 | 49.04 |

**4.3.2. Comparison of the TRFA and IPUS methods**

Using data of September 13 as an example, the spatial distributions of the four components of the energy balance calculated by the IPUS and TRFA methods are shown in Fig. 9 and Fig. 10, respectively. TSFA minus IPUS and TSFA minus TRFA, which display the spatial distributions of the scale effect, are shown in Fig. 11. Scatterplots of TSFA versus IPUS and TRFA are shown in Fig. 12.

 A comparison of Fig. 6 with Fig. 9 shows that the spatial distributions of the fluxes greatly changes, except for $R_n$. The TSFA results are synoptically smoother than the IPUS results because the land cover types and temperature distributions in mixed pixels  are not considered in IPUS . For example, the boundary between the oasis and uncultivated land becomes a belt of intermediate G, H and LE because these mixed pixels include uncultivated land and vegetation. However, mixed pixels are classified as the dominant land use type in the parameterization process of IPUS. This result overlooks the contributions of heat flux from complex land use types and overestimates or underestimates the turbulent heat flux by approximately 50 W·m$^{-2}$. Since the TSFA can integrate the effects of these land areas and reveals the relative actual surface conditions, the heat flux results of TSFA vary less dramatically than  those of IPUS, as shown in the figures. The results are similar in the oasis area.

Based on  Fig. 6 and Fig. 10, the TRFA and TSFA methods are similar. Because the TRFA method considers the sub-pixel landscapes that could be  significant source of error in the ET models, the difference between the TSFA and TRFA methods is mainly resulted from the differences between the sharpened LST and retrieved, resampled LST  of sub-pixels at the 30 m resolution. In addition, the bias between the TSFA and TRFA is not as obvious as the bias between the TSFA and IPUS methods, as shown in Fig. 11(c)-(f). Furthermore, Fig. 11(f) shows that the LEs calculated by using the TSFA method  in most oasis areas were slightly greater than the LEs calculated by using the TRFA method, which  yielded values of approximately 20 $W \cdot m^{-2}$.

The quadrangular with a relatively unstable bias shown in Fig. 11(a) is caused by the $L_d$ that was calculated from the MOD05 water vapor product which exists quadrangular even after preprocessing the instrument malfunction gap.  In Fig. 11, the differences of the four energy components of the pure pixels between these three methods are within 5 $W \cdot m^{-2}$, and the mixed pixels have different ranges.

[Figure]

**Figure 9.** Maps of the four energy components, (a) $R_n$, (b) G, (c) H and (d) LE, calculated using the IPUS method on September 13, 2012

[Figure]

**Figure 10.** Maps of the four energy components, (a) $R_n$, (b) G, (c) H and (d) LE, calculated using the TRFA method
on September 13, 2012.

[Figure]

**Figure 11.** Maps of the bias of the energy balance components calculated using the TSFA method minus the IPUS

method: (a) $R_n$, (b) G, (c) H, (d) LE, and TSFA minus TRFA: (e) H and (f) LE.

[Figure]

**Figure 12.** Scatter plots between the TSFA and IPUS results: (a) $R_n$, (b) G, (c) H and (d) LE; and TSFA and TRFA

results: (e) $R_n$, (f) G, (g) H and (h) LE. MBD and RMSD are the mean bias deviation and root mean square deviation between the TSFA and IPUS results, respectively.

Fig. 12 shows the scatter plots between the results from of the TSFA method and the other two methods for all four energy balance components . Fig. 11(a)(e) shows that $R_n$ does not vary much between the three methods  and the scatter is centralized around the 1:1 line. However, regarding the spatial scale effect, the differences in $G$, $H$ and $LE$ calculated by using the IPUS and TSFA methods are obvious. The scatter plots  dispersed at the mixed pixels, and the differences between the TRFA and TSFA results are relatively smaller. When using the TSFA method, the temperature sharpening results can be divided into results that are higher and lower than the LST retrieved at 300 m. Compared  to the LST retrieved at 300 m when using the TRFA method, a higher LST  is counterbalanced by a lower LST when calculating H  using the TSFA. Thus, the effect of temperature heterogeneity is neutralized in this case. This observation is potentially resulted from the temperature-sharpening algorithms because they tend to overestimate the sub-pixel LST for cooler landscapes and underestimate the sub-pixel LST for warmer areas in the image (Kustas et al., 2003).

However, $LE$ is calculated as a residual; thus, the difference of $LE$ is resulted from  $G$ and $H$. When the 300 m mixed pixels contained various types of landcapes, they  were categorized as one type of landscape  in the IPUS  method and  a single temperature value  was used to evaluate the thermal dynamic effects when using the TRFA method. Pixels with highly different $G$, $H$ and $LE$ values are mainly distributed near the mixed pixels, as shown in Fig. 10. An explanation for these deviations is provided below.

The parameterization of $G$ and $H$  are based on the land cover type. For example, for buildings, $G = 0.4R_n$ (Kato and Yamaguchi, 2005) (which is usually greater than the $G$ of vegetation and bare soil deduced from Eq.(9)) and $H = 0.6R_n$, and for water, $G = 0.226R_n$ and $LE = R_n - G$. From the land cover map shown in Fig. 4, four major classes exist in the study area: buildings with a high $H$, uncultivated land with a relatively high $H$, cropland with a relatively low $H$, and water with $H = 0$.

(1) If a pixel contains cropland and buildings and is categorized as cropland, the building area within the pixel is ignored  in the IPUS method. In this case, $G$ and $H$ are underestimated and $LE$ is overestimated. In addition, after considering the landscapes  using the TRFA method, the $LE$ is underestimated and $H$ is overestimated because the pixels contain buildings that are still reflected indistinctly by LST at 300 m because the detailed temperature heterogeneity cannot be represented by the TRFA method. These points are shown in green in Fig. 11. However, if the pixel is categorized as built-up, the building area within a pixel is exaggerated, which causes $G$ and $H$ to be overestimated and $LE$ to be underestimated when using the IPUS method. This situation is similar to  that illustrated by the green  associated with the TRFA results and is shown by the red points in Fig. 11.

(2) At the  boundary of the oasis and uncultivated land, the mixed pixels are divided into cropland,  LE is overestimated, $G$ and $H$ are underestimated in the IPUS method, and vice versa.  LE is also overestimated in the pixels containing water and other types of land cover (generally bare soil in our study area). These pixels are categorized as water and are shown as blue points in Fig. 11. Some of the blue $LE$ points calculated by using the TSFA method are slightly smaller than those calculated  using the TRFA method for pixels containing vegetation. At noon, the temperature of vegetation in those pixels is lower than that of water bodies.

(3) In mixed pixels that contain various crops, such as maize and vegetables,  LE is underestimated if the area of maize within the pixel is overestimated because the canopy height of the maize  is taller than that of vegetables This relationship results in the over-estimation of H when using the IPUS and TRFA methods. In addition, G depends on the FVC values of the crops when using the IPUS method.  Moreover, G depends on $R_n$ when using the TRFA method and is nearly  identical to the values of G obtained  using the TSFA method.

At the study area , we compared the TRFA and IPUS methods to quantify the ability of the TSFA method to simulate the heterogeneities of the land surface on September 13 (see Table 14).  In pure pixels, the LE biases among the IPUS, TRFA and TSFA methods were small. In mixed pixels, the LE bias between the TSFA and IPUS methods varied from 35.36 to 65.66 $W \cdot m^{-2}$, and the bias between the TSFA and TRFA methods varied from 4.41 to 22.53 $W \cdot m^{-2}$. More class types in mixed pixels correspond to larger biases. Table 15 shows the bias of the mixed pixels that contain buildings and bare soil between the three methods.  In mixed pixels with buildings, the IPUS and TRFA methods  generally underestimated  LE and had large bias values compared  to those of the TSFA method.  In mixed pixels without buildings and bare soil, the bias between TRFA (or IPUS) and TSFA was relatively small, which indicates that the landscape and temperature inhomogeneity  were accounted for by the TSFA method. The aforementioned analyses demonstrate that the TSFA method can consider the heterogeneous effects of mixed pixels.

**Table 14.** Comparison of the latent heat flux in pixels containing different numbers of class types

| Number of class types in pixels | IPUS ($W \cdot m^{-2}$) | | | TRFA ($W \cdot m^{-2}$) | | | Pixel number |
|---|---|---|---|---|---|---|---|
| | $R^2$ | MBD | RMSD | $R^2$ | MBD | RMSD | |
| 1 | 1.00 | 0.21 | 0.21 | 1.00 | 0.05 | 0.61 | 11,398 |
| 2 | 0.85 | -7.18 | 35.36 | 1.00 | -0.35 | 4.41 | 8212 |
| 3 | 0.66 | -2.32 | 52.55 | 0.98 | -7.33 | 12.56 | 4762 |
| 4 | 0.49 | 1.88 | 65.66 | 0.96 | -11.56 | 16.55 | 2824 |
| 5 | 0.98 | -30.92 | 62.69 | 0.96 | -16.90 | 22.53 | 4 |

Notes:  The number of class types in mixed pixels  represents the number of classification types that were contained in the pixels. For example, 1 represents the pure pixels, 2 represents mixed pixels containing two land use types, etc. MBD and RMSD are the mean bias deviation and root mean square deviation, respectively, between the TSFA results and the TRFA and IPUS results.

**Table 15.** Comparison of the latent heat fluxes of typical mixed pixels

| Types of mixed pixels | IPUS ($W \cdot m^{-2}$) | | | TRFA ($W \cdot m^{-2}$) | | | Pixel number |
|---|---|---|---|---|---|---|---|
| | $R^2$ | MBD | RMSD | $R^2$ | MBD | RMSD | |
| Mixed pixels containing buildings | 0.58 | -1.02 | 61.94 | 0.97 | -9.64 | 14.66 | 4918 |
| Mixed pixels do not containing buildings | 0.81 | -5.49 | 39.21 | 0.99 | -2.12 | 7.60 | 10,884 |
| Mixed pixels containing bare soil | 0.73 | -1.52 | 49.04 | 0.98 | -5.96 | 11.86 | 9049 |
| Mixed pixels do not containing bare soil | 0.65 | -7.55 | 45.28 | 0.98 | -2.46 | 7.83 | 6753 |

Considering the landscapes and inhomogeneous distribution of LST, the TSFA method ensures that none of the end members (30 m pixel) are ignored or exaggerated. Thus, the distribution of LE calculated using the TSFA method is smoother and more rational than the distributions of LE calculated using the other methods. At the regional scale, the TSFA method describes the heterogeneity of the land surface more precisely. The degree of achievable estimation accuracy is discussed hereafter.

**4.4. Error analysis**

Since LE is calculated as a residual term in the energy balance equations, the sensitivity of H was analyzed first. Land surface variables (including LST, LAI, canopy height, and FVC) and meteorological variables including wind speed, air temperature, air pressure and relative humidity are the major factors for H sensitive analysis. Fig. 13 presents a case of sensitivity analysis results for H. In this case, LST is 303.9 K, and it ranges from 298.4~309.4 K with a step size of 0.5 K, LAI is set to 1.4 and it ranges from 0.14~2.66 with a step size of 0.14, e. Canopy height is 1 m and it ranges from 0.1~1.9 m with a step size of 0.1 m,. Additionally, FVC=0.5, wind speed u=2.48 m·s$^{-1}$, air temperature Ta=297.9 K, air pressure = 97.2 kPa, and RH=40.29%. In addition, the land cover is maize, and the reference H is 230.2 W·m$^{-2}$.

[Figure]

**Figure 13.** Sensitivity analysis of the surface variables for sensible heat flux

The air pressure is stable over a short period and has little effect on the ET results. Although "excess resistance" was calculated from the friction velocity, the meteorological data were provided by ground observations; thus, the meteorological data are relatively accurate. As shown in Fig. 13, LAI, canopy height and LST are sensitive variables.

The parameterization of the momentum roughness length indicates that H LAI is sensitive to H LAI, with decreasing sensitivity when the LAI is greater than 1. When the LAI is less than 1, the momentum roughness length increases as the LAI increases, and the H and turbulent exchange are enhanced. However, when the LAI is greater than 1, the plant canopy could be is regarded as a continuum that is not a sensitive variable to H. Because our study area is dominated by agriculture and the study period was extended from July to September, the crops in the HRB middle stream grew quickly, so thus, the LAI was usually greater than 1. Thus, LST and canopy height are the main sources of error.

**4.4.1. The error of Errors in LST**

As shown in Fig. 13, 1 K LST bias would result in 21% error of H while H is 230.2 W·m$^{-2}$. However, the sensitivity of the LST is unstable and depends on the strength of the turbulence. The strength of the turbulence determines the mass and energy transport and the resistance of heat transfer, which influences the sensitivity of to the LST. A weaker turbulence corresponds to a weaker LST sensitivity, and vice versa.

The A sensitivity analysis of LE induced by LST was also implemented performed. In order to exclude other factors' the influence of other factors, homogeneous stations were chosen stations were chosen with homogeneous landscapes within coarse pixels. These results are shown in Table 16. The LE results obtained from the observed LST are consistent with the *in situ* observations but and have less bias. The LE was overestimated when the LST was underestimated, and vice versa.

Because the magnitude of LE was greater than that of H, the relative error of LE was less than the relative error of H. However, 1 K of LST bias  resulted in an average LE error of 30 W·m$^{-2}$, which is consistent with the sensitivity analysis of H shown in Fig. 13. Specifically, 1 K of LST bias would resulted in an LE biases of 8.7 W·m$^{-2}$ (in desert, SSW) to 84.4 W·m$^{-2}$ (in oasis, EC8), which indicates that the sensitivity of LST is unstable.

**Table 16.** The results of the LST error analyses at the  stations with homogeneous landscapes

| Station | Date | Re-trieved LST (K) | Ob-served LST (K) | LST bias (K) | EC-LE (W·m$^{-2}$) | LE from retrieved LST (W·m$^{-2}$) | LE from observed LST (W·m$^{-2}$) | LE relative error (%) | H relative error (%) |
|---|---|---|---|---|---|---|---|---|---|
| EC8 | 0619 | 304.92 | 301.74 | 3.18 | 415.89 | 321.80 | 399.78 | -22.62 | 68.58 |
| EC7 | 0630 | 302.5 | 299.35 | 3.15 | 611.22 | 453.59 | 557.97 | -25.79 | 886.08 |
| EC10 | 0708 | 303.58 | 300.5 | 3.08 | 617.83 | 504.44 | 549.53 | -18.35 | 390.24 |
| EC15 | 0708 | 303.55 | 300.13 | 3.42 | 620.95 | 425.71 | 603.73 | -31.44 | 450.57 |
| EC7 | 0727 | 298.87 | 300.55 | -1.68 | 577.59 | 643.56 | 566.62 | 11.42 | -132.47 |
| SSW | 0727 | 307.86 | 316.82 | -8.96 | 119.35 | 238.07 | 78.43 | 99.48 | -60.36 |
| EC8 | 0822 | 299.58 | 297.77 | 1.81 | 543.56 | 416.23 | 467.42 | -23.42 | 88.59 |
| EC10 | 0822 | 301.61 | 298.04 | 3.57 | 503.82 | 398.82 | 513.67 | -20.84 | 138.61 |
| EC15 | 0822 | 300.59 | 297.69 | 2.9 | 473.68 | 408.37 | 495.49 | -13.79 | 129.60 |
| EC8 | 0829 | 301.54 | 300.44 | 1.1 | 514.31 | 402.93 | 428.78 | -21.66 | 63.91 |
| EC15 | 0829 | 301.41 | 299.84 | 1.57 | 473.54 | 399.25 | 459.66 | -15.69 | 182.34 |
| SSW | 0902 | 304.9 | 303.42 | 1.48 | 226.88 | 127.96 | 149.83 | -43.60 | 11.36 |

Notes: "LST bias" is calculated as the retrieved LST minus the observed LST; "EC-LE" is the *in situ* latent heat flux;

"LE relative error" is the relative error between the retrieved and observed LST and is expressed as ((LE from retrieved LST)-(LE from observed LST))/(LE from observed LST)×100% and "H relative error" is calculated in the same way.

**4.4.2.**  **Errors in canopy height**

In this paper, canopy height was obtained from a phenophase and classification map. Thus, the accuracy of the canopy height was mainly dependent on the classification accuracy and plant growth state. Even within the same region, the canopy height of a crop can differ due to differences in seeding times and soil attributes, such as soil moisture and fertilization.

The land use type was orchard at EC17 . However, in our land classification map, the land use at EC17 was other crops, which includes vegetables and orchards. Thus, it was difficult to set the canopy height. In our study area, most of the other crops were vegetables (canopy height of 0.2 m), and the height of the orchard was approximately 4 m; thus, a value of 0.2 m would overestimate the LE. The LE estimations with incorrect canopy heights and correct orchard canopy height at EC17 are shown in Table 17. The days of large LST bias were removed, and the bias between the model and ground observations decreased. The excess errors were caused by errors in the LST and land use, such as buildings and maize in the mixed pixels.

**Table 17.** The results of the canopy height error analyses at EC17

| Date | EC-LE (W·m$^{-2}$) | LE from incorrect canopy height (W·m$^{-2}$) | LE-I relative error (%) | LE from correct canopy height (W·m$^{-2}$) | LE-C relative error (%) |
|---|---|---|---|---|---|
| 20120815 | 499.62 | 562.06 | 12.50 | 521.83 | 4.45 |

| 20120822 | 366.27 | 519.01 | 41.70 | 396.54 | 8.26 |
| 20120902 | 377.96 | 471.68 | 24.80 | 336.52 | -10.96 |
| 20120914 | 465.38 | 352.78 | -24.20 | 258.07 | -44.55 |

Notes: "LE-I relative error" is the relative error between the LE from incorrect canopy height and observed LE and is expressed as ((LE from incorrect canopy height)-(EC-LE))/( EC-LE)×100%, "LE-C relative error" is the relative error between the LE from correct canopy height and observed LE and is expressed as ((LE from correct canopy height)-(EC-LE))/( EC-LE)×100%.

Except for the error source discussed  previously, the following sources of error were unavoidable:

(1) Although the remotely sensed turbulent heat flux is instantaneous, the EC data are averaged over time. Thus, the time scales do not match in the validation.

(2) The calibration coefficients of the HJ-1B satellite's CCD and IRS drifts because of  instruments aging.

(3) Geometric correction causes half-pixel bias equal to or less than the deviation of the artificially subjective interpretation.

A one-source model and simplified parameterization schemes  were used in this paper to determining surface roughness lengths and heat transfer coefficients . The one-source model combines soil evaporation and plant transpiration and assumes that SPAC is a one-source continuum . This assumption is reasonable when the surface is densely covered by vegetation but relies on the accuracy of the difference between the LST and air temperature, as previously mentioned. When a one-source model is applied to an area covered by sparse vegetation, such as semi-arid or arid areas, this assumption is irrational.

**5. Discussions**

The TSFA describes the surface heterogeneity much better than the IPUS and TRFA. The IPUS aggregates the land surface variables from 30 m to 300 m, which results in the loss  of land surface details and leads to the scale effects. Although the TRFA uses 30 m information from VNIR bands and partially decreases the heterogeneity, it treats the pivotal variable LST as homogeneous at 300 m resolution, which results in considerable error. In summary, the advantages of the TSFA method  are described as follows:

(1) The temperature sharpening algorithm in TSFA is capable of decreasing the influences of the heterogeneity of the LST, which  is consistent with results from previous studies  (Kustas et al., 2003; Bayala and Rivas, 2014; Mukherjee et al., 2014). As analyzed in section 4.3, the non-consideration of the heterogeneity of LST in mixed pixels is ill-founded and causes errors when estimating ET.

(2) In the one-source energy balance model,  different parameterization schemes were employed for different landscapes. In the IPUS, a single land cover is assigned to a mixed pixel, which results in a large error. However, the TSFA method is used to calculate the surface flux at 30 m and is aggregated to 300 m using the area-weighting method, which considers all of the sub-pixel landscapes and improves the accuracy.

Some problems exist in the temperature-sharpening algorithms. The temperature-downscaling method used in this paper caused "boxy" anomalies in parts of the sharpened temperature fields in large pixels because of the constant residual term, $\Delta\widehat{T}_{300}$, in Eq. (3) within large pixels. This situation also occurred in the temperature-sharpening algorithm proposed by Agam et al. (2007). In addition, our temperature sharpening algorithm tends to overestimate the sub-pixel LST for cooler landscapes and underestimate the sub-pixel LST for warmer areas (Kustas et al., 2003). This inaccurate estimation causes errors that are difficult to evaluate when estimating the turbulent heat flux. For example, the small turbulent heat flux bias between TSFA and TRFA was caused by  a counterbalancing effect as analyzed in section 4.3.1.  Additional temperature sharpening algorithms under heterogeneous surfaces should be evaluated using  real datasets when applied in ET models  (Ha et al., 2011).

The  retrieval methods of land surface variables were validated against  other areas . For example, the albedo algorithm was previously applied to retrieve Global Land Surface Satellite (GLASS) Products (Liang et al., 2014), the LST retrieval algorithm was validated in the Haihe River Basin in northern China (Li et al., 2011a), and the soil heat flux correction algorithm was validated in the GAME-Tibet campaign (Yang and Wang, 2008). Since the surface of the Heihe River Basin is very heterogeneous, additional comparisons of our algorithm in other areas  would be  helpful.

In addition, to correct the discrepancy between remotely sensed radiative surface temperature and aerodynamic temperature at the source of heat transport, a brief and well performed parameterization scheme (under uniformly flat plant surface) of "excess" resistance was used to calculate the aerodynamic resistance of heat transfer (Jiao et al., 2014). Since  our study  is based on mixed pixels,  multiple parameterization methods should be compared to select the optimum method.

Because of the sensitive variables of the one-source energy balance model used in this paper, the accuracies of the LST and canopy height greatly influenced the turbulent heat flux. HJ-1B IRS  has a single-thermal channel, and the single-channel LST-retrieving algorithm may be unstable under wet atmospheric conditions (water vapor contents higher than 3 g/cm$^2$) (Li et al., 2010a), which may create a bottleneck for ET estimations by HJ-1B. The canopy height is a priori knowledge based on phenophase classifications and would influence the accuracy of the surface roughness calculation. Multi-source remote sensing data could be used to improve the accuracy of calibrations and land surface variable estimation. Active microwave and LiDAR data (Colin and Faivre, 2010) could be used to obtain the canopy height, which would decrease the dependence on the accuracy of the classification.

The energy balance closure has a significant influence on the evaluation of the model calculated heat flux results. In our study area, the EC energy balance closure ratio was greater than 0.75 (Liu et al., 2011b). Studies have shown that  not-captured low-frequency eddies (Von Randow et al., 2008), extension of averaging time (Charuchittipan et al., 2014) and lack of an accurate accounting of heat storage terms (Meyers and Hollinger, 2004) are potential reasons for the energy imbalance and so forth. The conserving Bowen ratio and residual closure technique are often used to force energy balance. We chose the residual closure method at last because the conserving Bowen ratio method  yields irrational sensible heat flux due to small or negative Bowen ratios (large LEs due to the "oasis effect") in the oasis-desert system. Energy balance closure was problematic at times for turbulent flux system and tended to be associated with significant discrepancies in LE (Prueger et al., 2005).

Since a footprint model was used in the validation, the footprints discrepancies between *in*

*situ* measurements and remote sensing pixel may cause biases. For example, model validation results were calculated  based on the relative weights of the footprint model and  multiplied by the heat flux results of the coarse pixels  in the source area from the upwind direction. However, the heat fluxes of coarse pixels included the contributions of  non-overlapped sub-pixels within the coarse pixel. These pixels are influenced by the heterogeneity of underlying surface, it would cause uncertainties in the validation.

**6. Conclusion**

The effects of surface heterogeneity in ET estimation have been studied here by employing the IPUS, TRFA and TSFA methods over heterogeneous surface. Compared  to the IPUS and TRFA methods, the TSFA method  exhibits more consistent agreement with *in situ* measurements (energy balance forced by the residual closure method)  based on the footprint validation results.  The IPUS approach does not consider surface heterogeneity at all, which causes significant error in the heat fluxes (i.e., 186 $W\cdot m^{-2}$). The TRFA considers heterogeneity of landscapes besides LST heterogeneity, with a heat flux error (i.e., 49 $W\cdot m^{-2}$) that is less than that of IPUS. However, this error is non-negligible. As a sensitive variable of the ET model, canopy height is mainly determined by classification, and the application of classification at a 30 m resolution can improve the accuracy of the canopy height. Additionally, the sharpened surface temperature at a resolution of 30 m decreases the thermodynamic uncertainty caused by the land surface. The TSFA method can capture the heterogeneities of the land surface and integrate the effects of landscapes in mixed pixels that are neglected at coarse spatial resolutions.

HJ-1B satellite data have advantages because of their high spatiotemporal resolution and free access. Because the satellites are still in operation, the long-term data  are promising for applications  of monitoring energy budgets.

**Acknowledgements**

The authors would like to thank Dr. Franz Meixner of the Max Planck Institute for Biogeochemistry in Mainz and Dr. Albrecht Neftel of the Agroscope Research Station in Zurich for providing the footprint calculation tool. We are grateful to the research team of Professor ShaoMin Liu at Beijing Normal University for providing eddy covariance and automatic meteorological station data. This study was supported by the Special Fund from the Chinese Academy of Sciences (KZZD-EW-TZ-18) and the Chinese Natural Science Foundation Project (grant no. 41371360). The authors thank Bo Zhong at Institute of Remote Sensing and Digital Earth, Chinese Academy of Sciences for his very helpful comments and many corrections. Generous help for revising the paper was provided by the editors and reviewers.

**Appendix**

| Notation | | Application (for calculating) |
|---|---|---|
| 6S radiation transfer mode | Second Simulation of a Satellite Signal in the Solar Spectrum radiation transfer mode | Albedo, $S_d$ |

| | | |
|---|---|---|
| α | Surface broadband albedo | $S_d$, $R_n$ |
| ABT | At-nadir brightness temperature (K) | $L_d$ |
| AMS | Automatic meteorological station | |
| AOD | Aerosol optical depth | $S_d$ |
| BRDF | Bidirectional reflectance distribution function | α |
| CCD | Charge-coupled device | |
| CV | Coefficient of variation | Sharpened LST |
| EC | Eddy covariance | |
| FVC | Fractional vegetation coverage | LSE, G, LAI |
| G | Soil heat flux (W·m$^{-2}$) | |
| G(θ) | G function, Foliage angle distribution | LAI |
| H | Sensible heat flux (W·m$^{-2}$) | |
| HRB | The Heihe River Basin | |
| IPUS | Input parameter upscaling scheme | |
| IRS | Infrared scanner | |
| $L_d$ | Downward atmospheric longwave radiation (W·m$^{-2}$) | $R_n$ |
| LSE/ε | Land surface emissivity | LST |
| $ε_v$/$ε_g$ | The vegetation/ground emissivity | |
| LST/$T_{rad}$ | Land surface temperature/Surface radiation temperature (K) | H |
| MBE/MBD | Mean bias error (deviation) | |
| NCEP | National Centers for Environmental Prediction | LST |
| NDVI/NDVI$_{30}$ | Normalized difference vegetation index | FVC, Sharpened LST |
| NDVI$_{300}$ | 300 m NDVI aggregated from NDVI | Sharpened LST |
| NDVI$_s$/NDVI$_v$ | Normalized difference vegetation index of bare soil/fully covered vegetation | FVC |
| P(θ) | Angular distribution of the canopy gap fraction | LAI |
| $r_a$ | Aerodynamic resistance (s·m$^{-1}$) | H |
| $r_{ex}$ | "Excess" resistance (s·m$^{-1}$) | heat transfer resistance |
| $R_n$ | Net radiation (W·m$^{-2}$) | |
| RMSE/RMSD | Root mean square error (deviation) | |
| $S_d$ | Downward shortwave radiation (W·m$^{-2}$) | $R_n$ |
| SPAC | The soil-plant-atmosphere continuum | |
| SZA | Solar zenith angle | $S_d$ |
| $T_a$ | Air temperature (K) | H |
| $T_{aero}$ | Aerodynamic surface temperature obtained by extrapolating the logarithmic air-temperature profile to the roughness length for heat transport (K) | H |
| TOA | Top of the atmosphere | |
| TOMS | Total ozone mapping spectrometer | $S_d$ |
| TRFA | Temperature resampling and flux aggregation | |
| TSFA | Temperature sharpening and flux aggregation | |
| ULR | Upward longwave radiation (W·m$^{-2}$) | $R_n$ |
| USR | Upward shortwave radiation (W·m$^{-2}$) | $R_n$ |
| VNIR | Visible/near-infrared | |

| VZA/$\theta$ | View zenith angle | | $L_d$, LAI |
|---|---|---|---|

**References**

Agam, N., Kustas,W. P., Anderson, M. C., Li, F., and Colaizzi, P. D.: Utility of thermal s harpening over Texas high plains irrigated agricultural fields, J. Geophys. Res.-Atmos., 112, D1 9110, doi:10.1029/2007JD008407, 2007.

Allen, R., Tasumi, M., and Trezza, R.: Satellite-Based Energy Balance for Mapping Evapo transpiration with Internalized Calibration (METRIC)—Model, J. Irrig. Drain. E.-ASCE, 133, 38 0-394, 10.1061/(ASCE)0733-9437(2007)133:4(380), 2007.

Ambast, S. K., Keshari, A. K., and Gosain, A. K.: An operational model for estimating R egional Evapotranspiration through Surface Energy Partitioning (RESEP), Int. J. Remote Sens., 23, 4917-4930, 10.1080/01431160110114501, 2002.

Bastiaanssen, W. G. M., Menenti, M., Feddes, R. A., and Holtslag, A. A. M.: A remote s ensing surface energy balance algorithm for land (SEBAL). 1. Formulation, J. Hydrol., 212–213 , 198-212, http://dx.doi.org/10.1016/S0022-1694(98)00253-4, 1998.

Bateni, S. M., and Liang, S.: Estimating surface energy fluxes using a dual-source data as similation approach adjoined to the heat diffusion equation, Journal of Geophysical Research: A tmospheres, 117, D17118, 10.1029/2012JD017618, 2012.

Bayala, M. I., and Rivas, R. E.: Enhanced sharpening procedures on edge difference and water stress index basis over heterogeneous landscape of sub-humid region, The Egyptian Journ al of Remote Sensing and Space Science, 17, 17-27, http://dx.doi.org/10.1016/j.ejrs.2014.05.002, 2014.

Bin, L., and Roni, A.: The Impact of Spatial Variability of Land-Surface Characteristics o n Land-Surface Heat Fluxes, Journal of Climate, 7, 527-537, 10.1175/1520-0442(1994)007<0527 :TIOSVO>2.0.CO;2, 1994.

Blyth, E. M., and Harding, R. J.: Application of aggregation models to surface heat flux f rom the Sahelian tiger bush, Agricultural and Forest Meteorology, 72, 213-235, http://dx.doi.org/ 10.1016/0168-1923(94)02164-F, 1995.

Bonan, G. B., Pollard, D., and Thompson, S. L.: Influence of Subgrid-Scale Heterogeneity in Leaf Area Index, Stomatal Resistance, and Soil Moisture on Grid-Scale Land–Atmosphere In teractions, Journal of Climate, 6, 1882-1897, 10.1175/1520-0442(1993)006<1882:IOSSHI>2.0.CO; 2, 1993.

Bonan, G. B., Levis, S., Kergoat, L., and Oleson, K. W.: Landscapes as patches of plant functional types: An integrating concept for climate and ecosystem models, Global Biogeochemi cal Cycles, 16, 5-1-5-23, 10.1029/2000GB001360, 2002.

Cammalleri, C., Anderson, M. C., Gao, F., Hain, C. R., and Kustas, W. P.: A data fusion approach for mapping daily evapotranspiration at field scale, Water Resour. Res., 49, 4672-468 6, 10.1002/wrcr.20349, 2013.

Charuchittipan, D., Babel, W., Mauder, M., Leps, J.-P., and Foken, T.: Extension of the A veraging Time in Eddy-Covariance Measurements and Its Effect on the Energy Balance Closure , Boundary-Layer Meteorology, 152, 303-327, 10.1007/s10546-014-9922-6, 2014.

Chen, J.: An important drawback and the improvement of the evapotranspiration model wi th remote sensing, Chinese Sci. Bull., 6, 454-457, 1988.

Chen, J., Chen, B. Z., Black, T. A., Innes, J. L., Wang, G. Y., Kiely, G., Hirano, T., and Wohlfahrt, G.: Comparison of terrestrial evapotranspiration estimates using the mass transfer an d Penman-Monteith equations in land surface models, J. Geophys. Res.-Biogeo., 118, 1715-173 1, 10.1002/2013jg002446, 2013.

Chen, W., Cao, C., He, Q., Guo, H., Zhang, H., Li, R., Zheng, S., Xu, M., Gao, M., Zha o, J., Li, S., Ni, X., Jia, H., Ji, W., Tian, R., Liu, c., Zhao, Y., and Li, J.: Quantitative estima tion of the shrub canopy LAI from atmosphere-corrected HJ-1 CCD data in Mu Us Sandland, Sci. China Earth Sci., 53, 26-33, 2010.

Choudhury, B. J., Reginato, R. J., and Idso, S. B.: An analysis of infrared temperature ob servations over wheat and calculation of latent heat flux, Agr. Forest Meteorol., 37, 75-88, http ://dx.doi.org/10.1016/0168-1923(86)90029-8, 1986.

Choudhury, B. J., and Monteith, J. L.: A four-layer model for the heat budget of homoge neous land surfaces, Q. J. Roy. Meteor. Soc., 114, 373-398, 10.1002/qj.49711448006, 1988.

Cleugh, H. A., Leuning, R., Mu, Q. Z., and Running, S. W.: Regional evaporation estimat es from flux tower and MODIS satellite data, Remote Sens. Environ., 106, 285-304, http://dx.d oi.org/10.1016/j.rse.2006.07.007, 2007.

Colin, J., and Faivre, R.: Aerodynamic roughness length estimation from very high-resoluti on imaging LIDAR observations over the Heihe basin in China, Hydrol. Earth Syst. Sci., 14, 2661-2669, 10.5194/hess-14-2661-2010, 2010.

Ershadi, A., McCabe, M. F., Evans, J. P., Mariethoz, G., and Kavetski, D.: A Bayesian an alysis of sensible heat flux estimation: Quantifying uncertainty in meteorological forcing to imp rove model prediction, Water Resources Research, 49, 2343-2358, 10.1002/wrcr.20231, 2013a.

Ershadi, A., McCabe, M. F., Evans, J. P., and Walker, J. P.: Effects of spatial aggregation on the multi-scale estimation of evapotranspiration, Remote Sens. Environ., 131, 51-62, http://d x.doi.org/10.1016/j.rse.2012.12.007, 2013b.

Fan, L., Xiao, Q., Wen, J., Liu, Q., Tang, Y., You, D., Wang, H., Gong, Z., and Li, X.: Evaluation of the Airborne CASI/TASI Ts-VI Space Method for Estimating Near-Surface Soil Moisture, Remote Sensing, 7, 3114-3137, 10.3390/rs70303114, 2015.

Fisher, J. B., Tu, K. P., and Baldocchi, D. D.: Global estimates of the land–atmosphere w ater flux based on monthly AVHRR and ISLSCP-II data, validated at 16 FLUXNET sites, Re mote Sens. Environ., 112, 901-919, http://dx.doi.org/10.1016/j.rse.2007.06.025, 2008.

Ha, W., Gowda, P. H., and Howell, T. A.: Downscaling of Land Surface Temperature Ma ps in the Texas High Plains with the TsHARP Method, GIScience & Remote Sensing, 48, 583 -599, 10.2747/1548-1603.48.4.583, 2011.

Ha, W., Gowda, P. H., and Howell, T. A.: A review of downscaling methods for remote sensing-based irrigation management: part I, Irrigation Science, 31, 831-850, 10.1007/s00271-012-0331-7, 2013.

He, L., Chen, J. M., Pisek, J., Schaaf, C. B., and Strahler, A. H.: Global clumping index map derived from the MODIS BRDF product, Remote Sens. Environ., 119, 118-130, http://dx.d oi.org/10.1016/j.rse.2011.12.008, 2012.

Hu, Y., Gao, Y., Wang, J., Ji, G., Shen, Z., Cheng, L., Chen, J., and Li, S.: Some achiev ements in scientific research during HEIFE, Plateau Meteorology, 13, 225-236, 1994.

Hu, Z. L., and Islam, S.: A framework for analyzing and designing scale invariant remote sensing algorithms, Geoscience and Remote Sensing, IEEE Transactions on, 35, 747-755, 10.11 09/36.581996, 1997.

Jia, Z. Z., Liu, S. M., Xu, Z. W., Chen, Y. J., and Zhu, M. J.: Validation of remotely se nsed evapotranspiration over the Hai River Basin, China, Journal of Geophysical Research: At mospheres, 117, D13113, 10.1029/2011JD017037, 2012.

Jiao, J. J, Xin, X. Z., Yu S. S., Zhou, T. and Peng, Z. Q.: Estimation of surface energy balance from HJ-1 satellite data. Journal of Remote Sensing, 18(5), 1048-1058, doi:10.11834/jrs. 20143322, 2014.

Jiang, B., Liang, S., Townshend, J. R., and Dodson, Z. M.: Assessment of the Radiometri c Performance of Chinese HJ-1 Satellite CCD Instruments, IEEE J. Sel. Top. Appl., 6, 840-85 0, 10.1109/JSTARS.2012.2212236, 2013.

Jiang, L., and Islam, S.: A methodology for estimation of surface evapotranspiration over l arge areas using remote sensing observations, Geophys. Res. Lett., 26, 2773-2776, 10.1029/199 9GL006049, 1999.

Jiang, L., and Islam, S.: Estimation of surface evaporation map over Southern Great Plains using remote sensing data, Water Resour. Res., 37, 329-340, 10.1029/2000WR900255, 2001.

Jiang, L., and Islam, S.: An intercomparison of regional latent heat flux estimation using r emote sensing data, Int. J. Remote Sens., 24, 2221-2236, 10.1080/01431160210154821, 2003.

Jin, Y. F., Randerson, J. T., and Goulden, M. L.: Continental-scale net radiation and evapo transpiration estimated using MODIS satellite observations, Remote Sens. Environ., 115, 2302-2 319, http://dx.doi.org/10.1016/j.rse.2011.04.031, 2011.

Kalma, J., McVicar, T., and McCabe, M.: Estimating Land Surface Evaporation: A Review of Methods Using Remotely Sensed Surface Temperature Data, Surveys in Geophysics, 29, 42 1-469, 10.1007/s10712-008-9037-z, 2008.

Kato, S., and Yamaguchi, Y.: Analysis of urban heat-island effect using ASTER and ETM + Data: Separation of anthropogenic heat discharge and natural heat radiation from sensible he at flux, Remote Sensing of Environment, 99, 44-54, http://dx.doi.org/10.1016/j.rse.2005.04.026, 2 005.

Kormann, R., and Meixner, F.: An Analytical Footprint Model For Non-Neutral Stratificati on, Bound.-Lay. Meteorol., 99, 207-224, 10.1023/A:1018991015119, 2001.

Kustas, W. P.: Estimates of Evapotranspiration with a One- and Two-Layer Model of Heat Transfer over Partial Canopy Cover, J. Appl. Meteorol., 29, 704-715, 10.1175/1520-0450(1990) 029<0704:EOEWAO>2.0.CO;2, 1990.

Kustas, W. P., Norman, J. M., Anderson, M. C., and French, A. N.: Estimating subpixel s urface temperatures and energy fluxes from the vegetation index–radiometric temperature relatio nship, Remote Sens. Environ., 85, 429-440, http://dx.doi.org/10.1016/S0034-4257(03)00036-1, 20 03.

Kustas, W. P., Moran, M. S., and Meyers, T. P.: The Bushland Evapotranspiration and Agr icultural Remote Sensing Experiment 2008 (BEAREX08) Special Issue, Adv. Water Resour., 50, 1-3, http://dx.doi.org/10.1016/j.advwatres.2012.11.006, 2012.

Leuning, R., Zhang, Y. Q., Rajaud, A., Cleugh, H., and Tu, K.: A simple surface conduct ance model to estimate regional evaporation using MODIS leaf area index and the Penman-Mo nteith equation, Water Resour. Res., 44, W10419, 10.1029/2007WR006562, 2008.

Li, H., Liu, Q. H., Zhong, B., Du, Y. M., Wang, H. S., and Wang, Q.: A single-channel algorithm for land surface temperature retrieval from HJ-1B/IRS data based on a parametric m odel, Geoscience and Remote Sensing Symposium (IGARSS), 2010 IEEE International, Honolul u, Hawaii, USA, 2448-2451, 2010a.

Li, H., Liu, Q., Jiang, J., Wang, H., and Sun, L.: Validation of the land surface temperatu re derived from HJ-1B/IRS data with ground measurements, Geoscience and Remote Sensing S ymposium (IGARSS), 2011 IEEE International, Vancouver, Canada, 293-296, 2011a.

Li, H., Sun, D., Yu, Y., Wang, H., Liu, Y., Liu, Q., Du, Y., Wang, H., and Cao, B.: Eval uation of the VIIRS and MODIS LST products in an arid area of Northwest China, Remote S ens. Environ., 142, 111-121, http://dx.doi.org/10.1016/j.rse.2013.11.014, 2014.

Li, J., Gu, X., Li, X., Yu, T., Chen, H., and Long, M.: Validation of HJ-1B Thermal Ban d On-board Calibration and Its Sensitivity Analysis, Remote Sensing Information, 1, 3-9, doi:10 .3969/j.issn.1000-3177.2011.01.001, 2011b.

Li, L., Xin, X. Z., Su, G. L., and Liu, Q. H.: Photosynthetically active radiation retrieval based on HJ-1A/B satellite data, Sci. China Earth Sci., 53, 81-91, 10.1007/s11430-010-4142-5, 2010b.

Li, X., Li, X. W., Li, Z. Y., Ma, M. G., Wang, J., Xiao, Q., Liu, Q., Che, T., Chen, E. X., Yan, G. J., Hu, Z. Y., Zhang, L. X., Chu, R. Z., Su, P. X., Liu, Q. H., Liu, S. M., Wang , J. D., Niu, Z., Chen, Y., Jin, R., Wang, W. Z., Ran, Y. H., Xin, X. Z., and Ren, H. Z.: Wa tershed Allied Telemetry Experimental Research, Journal of Geophysical Research: Atmospheres, 114, D22103, 10.1029/2008JD011590, 2009.

Li, X., Cheng, G. D., Liu, S. M., Xiao, Q., Ma, M. G., Jin, R., Che, T., Liu, Q. H., Wa ng, W. Z., Qi, Y., Wen, J. G., Li, H. Y., Zhu, G. F., Guo, J. W., Ran, Y. H., Wang, S. G., Z hu, Z. L., Zhou, J., Hu, X. L., and Xu, Z. W.: Heihe Watershed Allied Telemetry Experimenta l Research (HiWATER): Scientific Objectives and Experimental Design, B. Am. Meteorol. Soc., 94, 1145-1160, 10.1175/BAMS-D-12-00154.1, 2013a.

Li, Z. L., Tang, B. H., Wu, H., Ren, H., Yan, G., Wan, Z., Trigo, I. F., and Sobrino, J. A.: Satellite-derived land surface temperature: Current status and perspectives, Remote Sensing of Environment, 131, 14-37, http://dx.doi.org/10.1016/j.rse.2012.12.008, 2013b.

Liang, S., Stroeve, J., and Box, J. E.: Mapping daily snow/ice shortwave broadband albed o from Moderate Resolution Imaging Spectroradiometer (MODIS): The improved direct retrieval algorithm and validation with Greenland *in situ* measurement, Journal of Geophysical Research : Atmospheres, 110, D10109, 10.1029/2004JD005493, 2005.

Liang, S. L., Zhang, X. T., Xiao, Z. Q., Cheng, J., Liu , Q., and Zhao, X.: Global LAnd Surface Satellite (GLASS) Products: Algorithms, Validation and Analysis, 1 ed., SpringerBriefs in Earth Sciences, Springer International Publishing, 2014.

Liu, Q., Qu, Y., Wang, L. Z., Liu, N. F., and Liang, S. L.: Glass-Global Land Surface Br oadband Albedo Product: Algorithm Theoretical Basis Document. Version, 1, 1-50, College of Global Change and Earth System Science, Beijing Norman University, 2011a.

Liu, S. M., Xu, Z. W., Wang, W. Z., Jia, Z. Z., Zhu, M. J., Bai, J., and Wang, J. M.: A comparison of eddy-covariance and large aperture scintillometer measurements with respect to t he energy balance closure problem, Hydrol. Earth Syst. Sci., 15, 1291-1306, 2011b.

Liu, S., Xu, Z., Song, L., Zhao, Q., Ge, Y., Xu, T., Ma, Y., Zhu, Z., Jia, Z., and Zhang, F.: Upscaling evapotranspiration measurements from multi-site to the satellite pixel scale over h eterogeneous land surfaces, Agricultural and Forest Meteorology, http://dx.doi.org/10.1016/j.agrfor met.2016.04.008, 2016.

Long, D., and Singh, V. P.: A modified surface energy balance algorithm for land (M-SEB AL) based on a trapezoidal framework, Water Resour. Res., 48, W02528, 10.1029/2011WR0106 07, 2012a.

Long, D., and Singh, V. P.: A Two-source Trapezoid Model for Evapotranspiration (TTME) from satellite imagery, Remote Sens. Environ., 121, 370-388, http://dx.doi.org/10.1016/j.rse.201 2.02.015, 2012b.

Maayar, E. M., and Chen, J. M.: Spatial scaling of evapotranspiration as affected by heter ogeneities in vegetation, topography, and soil texture, Remote Sens. Environ., 102, 33-51, http:/ /dx.doi.org/10.1016/j.rse.2006.01.017, 2006.

Mallick, K., Boegh, E., Trebs, I., Alfieri, J. G., Kustas, W. P., Prueger, J. H., Niyogi, D., Das, N., Drewry, D. T., Hoffmann, L., and Jarvis, A. J.: Reintroducing radiometric surface tem perature into the Penman-Monteith formulation, Water Resources Research, 51, 6214-6243, 10.1 002/2014WR016106, 2015.

McCabe, M. F., and Wood, E. F.: Scale influences on the remote estimation of evapotrans piration using multiple satellite sensors, Remote Sensing of Environment, 105, 271-285, 10.101 6/j.rse.2006.07.006, 2006.

Meyers, T. P., and Hollinger, S. E.: An assessment of storage terms in the surface energy balance of maize and soybean, Agricultural and Forest Meteorology, 125, 105-115, http://dx.doi. org/10.1016/j.agrformet.2004.03.001, 2004.

Moran, M. S., Humes, K. S., and Pinter Jr, P. J.: The scaling characteristics of remotely-s ensed variables for sparsely-vegetated heterogeneous landscapes, Journal of Hydrology, 190, 337 -362, http://dx.doi.org/10.1016/S0022-1694(96)03133-2, 1997.

Mu, Q. Z., Zhao, M. S., and Running, S. W.: Improvements to a MODIS global terrestria l evapotranspiration algorithm, Remote Sens. Environ., 115, 1781-1800, http://dx.doi.org/10.1016/ j.rse.2011.02.019, 2011.

Mukherjee, S., Joshi, P. K., and Garg, R. D.: A comparison of different regression models for downscaling Landsat and MODIS land surface temperature images over heterogeneous lan dscape, Advances in Space Research, 54, 655-669, http://dx.doi.org/10.1016/j.asr.2014.04.013, 20 14.

Neftel, A., Spirig, C., and Ammann, C.: Application and test of a simple tool for operatio nal footprint evaluations, Environ. Pollut., 152, 644-652, http://dx.doi.org/10.1016/j.envpol.2007.0 6.062, 2008.

Nilson, T.: A theoretical analysis of the frequency of gaps in plant stands, Agr. Meteorol., 8, 25-38, 1971.

Norman, J. M., Kustas, W. P., and Humes, K. S.: Source approach for estimating soil and vegetation energy fluxes in observations of directional radiometric surface temperature, Agr. Forest Meteorol., 77, 263-293, http://dx.doi.org/10.1016/0168-1923(95)02265-Y, 1995.

Norman, J. M., Anderson, M. C., Kustas, W. P., French, A. N., Mecikalski, J., Torn, R., Diak, G. R., Schmugge, T. J., and Tanner, B. C. W.: Remote sensing of surface energy fluxes at 101-m pixel resolutions, Water Resour. Res., 39, 1221, 10.1029/2002WR001775, 2003.

Paulson, C. A.: The mathematical representation of wind speed and temperature profiles in the unstable atmospheric surface layer, J. Appl. Meteorol., 9, 857-861, 1970.

Prueger, J. H., Hatfield, J. L., Parkin, T. B., Kustas, W. P., Hipps, L. E., Neale, C. M. U. , MacPherson, J. I., Eichinger, W. E., and Cooper, D. I.: Tower and Aircraft Eddy Covariance Measurements of Water Vapor, Energy, and Carbon Dioxide Fluxes during SMACEX, Journal of Hydrometeorology, 6, 954-960, 10.1175/JHM457.1, 2005.

Shuttleworth, W. J., and Wallace, J.: Evaporation from sparse crops‐an energy combination theory, Q. J. Roy. Meteor. Soc., 111, 839-855, 1985.

Song, Y., Wang, J. M., Yang, K., Ma, M. G., Li, X., Zhang, Z. H., and Wang, X. F.: A revised surface resistance parameterisation for estimating latent heat flux from remotely sensed data, Int. J. Appl. Earth Obs., 17, 76-84, http://dx.doi.org/10.1016/j.jag.2011.10.011, 2012.

Su, Z.: The Surface Energy Balance System (SEBS) for estimation of turbulent heat fluxes , Hydrol. Earth Syst. Sci., 6, 85-100, 10.5194/hess-6-85-2002, 2002.

Sun, L., Sun, R., Li, X. W., Chen, H. L., and Zhang, X. F.: Estimating Evapotranspiration using Improved Fractional Vegetation Cover and Land Surface Temperature Space, Journal of Resources and Ecology, 2, 225-231, 10.3969/j.issn.1674-764x.2011.03.005, 2011.

Sun, L., Liang, S. L., Yuan, W. P., and Chen, Z. X.: Improving a Penman–Monteith evapotranspiration model by incorporating soil moisture control on soil evaporation in semiarid areas, International Journal of Digital Earth, 6, 134-156, 10.1080/17538947.2013.783635, 2013.

Valor, E., and Caselles, V.: Mapping land surface emissivity from NDVI: Application to European, African, and South American areas, Remote Sens. Environ., 57, 167-184, http://dx.doi.org/10.1016/0034-4257(96)00039-9, 1996.

Venturini, V., Islam, S., and Rodriguez, L.: Estimation of evaporative fraction and evapotranspiration from MODIS products using a complementary based model, Remote Sens. Environ., 112, 132-141, http://dx.doi.org/10.1016/j.rse.2007.04.014, 2008.

Vermote E F, Tanre D, Deuze J L, et al. Second Simulation of a Satellite Signal in the Solar Spectrum-Vector. 6S User Guide Version 3, 2006.

Von Randow, C., Kruijt, B., Holtslag, A. A. M., and de Oliveira, M. B. L.: Exploring eddy-covariance and large-aperture scintillometer measurements in an Amazonian rain forest, Agricultural and Forest Meteorology, 148, 680-690, http://dx.doi.org/10.1016/j.agrformet.2007.11.011, 2008.

Wang, K. C., and Liang, S. L.: An improved method for estimating global evapotranspiration based on satellite determination of surface net radiation, vegetation index, temperature, and soil moisture, Geoscience and Remote Sensing Symposium, 2008. IGARSS 2008. IEEE International, Boston, Massachusetts, USA, III - 875-III - 878, 2008.

Wang, Q., Wu, C., Li, Q., and Li, J.: Chinese HJ-1A/B satellites and data characteristics, Sci. China Earth Sci., 53, 51-57, 10.1007/s11430-010-4139-0, 2010.

Xin, X., and Liu, Q.: The Two-layer Surface Energy Balance Parameterization Scheme (TSEBPS) for estimation of land surface heat fluxes, Hydrol. Earth Syst. Sci., 14, 491-504, 10.5194/hess-14-491-2010, 2010.

Xin, X., Liu, Y. N., Liu, Q., and Tang, Y.: Spatial-scale error correction methods for regional fluxes retrieval using MODIS data, J. Remote Sens., 16(2), 207-231, 2012.

Xu, T. R., Bateni, S. M., and Liang, S. L.: Estimating Turbulent Heat Fluxes With a Weak-Constraint Data Assimilation Scheme: A Case Study (HiWATER-MUSOEXE), IEEE Geosci. Remote S., 12, 68-72, 10.1109/LGRS.2014.2326180, 2015.

Xu, Z. W., Liu, S. M., Li, X., Shi, S. J., Wang, J. M., Zhu, Z. L., Xu, T. R., Wang, W. Z., and Ma, M. G.: Intercomparison of surface energy flux measurement systems used during the HiWATER‐MUSOEXE, Journal of Geophysical Research: Atmospheres, 118, 13,140-113,157, 10.1002/2013JD020260, 2013.

Yang, K., and Wang, J.: A temperature prediction-correction method for estimating surface soil heat flux from soil temperature and moisture data, Sci. China Ser. D-Earth Sci., 51, 721-729, 10.1007/s11430-008-0036-1, 2008.

Yang, Y. T., and Shang, S. H.: A hybrid dual-source scheme and trapezoid framework–based evapotranspiration model (HTEM) using satellite images: Algorithm and model test, Journal of Geophysical Research: Atmospheres, 118, 2284-2300, 10.1002/jgrd.50259, 2013.

Yao, Y. J., Liang, S. L., Cheng, J., Liu, S. M., Fisher, J. B., Zhang, X. D., Jia, K., Zhao, X., Qin, Q. M., Zhao, B., Han, S., Jie, Zhou, G. S., Zhou, G. Y., Li, Y. L., and Zhao, S. H. : MODIS-driven estimation of terrestrial latent heat flux in China based on a modified Priestley–Taylor algorithm, Agr. Forest Meteorol., 171–172, 187-202, http://dx.doi.org/10.1016/j.agrforme t.2012.11.016, 2013.

Yebra, M., Van Dijk, A., Leuning, R., Huete, A., and Guerschman, J. P.: Evaluation of optical remote sensing to estimate actual evapotranspiration and canopy conductance, Remote Sens. Environ., 129, 250-261, http://dx.doi.org/10.1016/j.rse.2012.11.004, 2013.

Yu, S., Xin, X., and Liu, Q.: Estimation of clear-sky longwave downward radiation from HJ-1B thermal data, Sci. China Earth Sci., 56, 829-842, 10.1007/s11430-012-4507-z, 2013.

Zhang, R., Sun, X., Wang, W., Xu, J., Zhu, Z., and Tian, J.: An operational two-layer remote sensing model to estimate surface flux in regional scale: physical background, Sci. China Ser. D, 34, 200-216, 2005.

Zhang, X., Zhao, X., Liu, G., Kang, Q., and Wu, D.: Radioactive Quality Evaluation and Cross Validation of Data from the HJ-1A/B Satellites' CCD Sensors, Sensors, 13, 8564, doi:10.3390/s130708564, 2013.

Zhong, B., Ma, P., Nie, A., Yang, A., Yao, Y., Lü, W., Zhang, H., and Liu, Q.: Land cover mapping using time series HJ-1/CCD data, Sci. China Earth Sci., 57, 1790-1799, 10.1007/s11430-014-4877-5, 2014a.

Zhong, B., Zhang, Y., Du, T., Yang, A., Lv, W., and Liu, Q.: Cross-Calibration of HJ-1/CCD Over a Desert Site Using Landsat ETM+ Imagery and ASTER GDEM Product, Geoscience and Remote Sensing, IEEE Transactions on, 52, 7247-7263, 10.1109/TGRS.2014.2310233, 2014b.

Zhu, G. F., Su, Y. H., Li, X., Zhang, K., and Li, C. B.: Estimating actual evapotranspiration from an alpine grassland on Qinghai-Tibetan plateau using a two-source model and parameter uncertainty analysis by Bayesian approach, J. Hydrol., 476, 42-51, 10.1016/j.jhydrol.2012.10.006, 2013.